

# Parity and spin CFT with boundaries and defects

**Ingo Runkel[1]⋆, Lóránt Szegedy[2]† and Gérard M. T. Watts[3]‡**

**1** Fachbereich Mathematik, Universität Hamburg,
Bundesstraße 55, 20146 Hamburg, Germany
**2** Faculty of Physics, University of Vienna, Boltzmangasse 5, 1090 Wien, Austria
**3** Department of Mathematics, King's College London, Strand, London WC2R 2LS, UK

⋆ ingo.runkel@uni-hamburg.de , † lorant.szegedy@univie.ac.at , ‡ gerard.watts@kcl.ac.uk

## Abstract

This paper is a follow-up to [82] in which two-dimensional conformal field theories in the presence of spin structures are studied. In the present paper we define four types of CFTs, distinguished by whether they need a spin structure or not in order to be well-defined, and whether their fields have parity or not. The cases of spin dependence without parity, and of parity without the need of a spin structure, have not, to our knowledge, been investigated in detail so far. We analyse these theories by extending the description of CFT correlators via three-dimensional topological field theory developed in [45] to include parity and spin. In each of the four cases, the defining data are a special Frobenius algebra $F$ in a suitable ribbon fusion category, such that the Nakayama automorphism of $F$ is the identity (oriented case) or squares to the identity (spin case). We use the TFT to define correlators in terms of $F$ and we show that these satisfy the relevant factorisation and single-valuedness conditions. We allow for world sheets with boundaries and topological line defects, and we specify the categories of boundary labels and the fusion categories of line defect labels for each of the four types. The construction can be understood in terms of topological line defects as gauging a possibly non-invertible symmetry. We analyse the case of a $\mathbb{Z}_2$-symmetry in some detail and provide examples of all four types of CFT, with Bershadsky-Polyakov models illustrating the two new types.



# 1   Introduction and summary

The earliest and most thoroughly investigated two-dimensional conformal field theories (CFTs) are those where the world sheets are just complex one-dimensional manifolds, and where no further geometric structure or parity grading is needed to define the theory. Such CFTs were the focus of the foundational paper [12] on rational conformal field theory, and of early classification results [17]. These theories are algebraically well-understood via their relation to three-dimensional topological field theory [45].

The second most studied kind of CFTs are those which require a spin structure to be well-defined, and where the state spaces are $\mathbb{Z}_2$-graded by "fermion number". The most notable example is given by massless free fermions. But already for this class of theories a systematic bootstrap formulation in terms of crossing relations for operator product expansion coefficients was not available until recently [82]. In that paper, correlators of spin CFTs are described via topological line defects in a "parity enhancement" of an underlying bosonic theory [76]. A related approach to fermionic CFTs is via gauging suitable $\mathbb{Z}_2$-symmetries [54,57,65] giving a continuum version of the Jordan-Wigner transformation. Minimal model CFTs with half-integer spin fields have also been investigated earlier in [38,77], but without reference to spin structures.

In fact, the paper [82] and the present follow-up paper grew out of the desire to develop a systematic approach to consistency conditions and their solutions for these "fermionic CFTs". However, the term "fermionic" can stand for different properties of the theory: it can refer to the presence of half-integer spin fields, i.e. fields that transform under the double cover of the rotation group $SO(2)$, or it can refer to the statistics, i.e. to parity signs that arise when reordering the fields.

## 1.1   Four types of conformal field theory

To illustrate this, let $z(t)$, $w(t)$ be two paths in the complex plane as follows: consider a correlator where two copies of a field $\phi$ are inserted, one at $z(t)$ and one at $w(t)$, plus possibly other fields whose positions remain fixed. Continue the correlator from $t = 0$ to $t = \pi$, so that the two copies of $\phi$ exchange places, as in Figure 1. In the theories we study in this paper

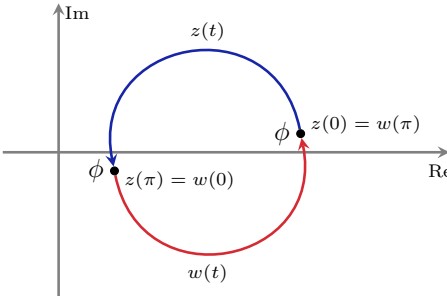

Figure 1: The choice of path to interchange the positions of two identical fields.

there are three sources of signs which can arise in this process,

$$\left\langle \phi(z(t))\phi(w(t))\cdots \right\rangle\big|_{t=0} = (-1)^{M+P+S}\left\langle \phi(z(t))\phi(w(t))\cdots \right\rangle\big|_{t=\pi}. \tag{1}$$

Here, $M$ is the contribution from analytic continuation controlled by the conformal weights of the fields, $P$ gives the contribution of parity, and $S$ arises from the effect the monodromy has on the spin structure. Single-valuedness requires $(-1)^{M+P+S} = 1$.

It turns out that parity and spin-structure dependence can independently be present or not, leading to four types of CFT:

| type of CFT | no parity | parity |
|:---:|:---:|:---:|
| oriented | ① | ② |
| spin | ③ | ④ |

$$\tag{2}$$

Here, "oriented" refers to the fact that no spin structure is needed to obtain well-defined correlators, just the orientation induced by the complex structure on the world sheet. The CFTs mentioned in the first paragraph are of type ①, in this case we have $(-1)^P = 1 = (-1)^S$ for all fields. Those in the second paragraph are of type ④, where $(-1)^P$ and $(-1)^S$ take both values. In fact, the situation is a little more subtle than (2), as we elaborate on in sections 1.2 and 1.3.

In this paper we present consistency conditions on correlators – namely compatibility with gluing and monodromy-freeness with or without spin structure or parity – and a construction of solutions to these conditions for all four types of CFTs. Our construction includes theories with boundaries and topological line defects. We do this by extending the approach via 3d TFTs which was developed for theories of type ① in [45] to the remaining three types. That is, we express the consistency conditions and solutions in terms of a 3d TFT. For type ④, the present paper provides the framework to prove the claims in the prequel [82]; the proofs themselves will be presented in a further part of this series.

Consistency conditions and their solutions for CFTs of types ② and ③ have, to the best of our knowledge, not been systematically studied in the literature so far.

## 1.2 Oriented parity CFT via topological field theory

Let us describe our approach in more detail. The starting point is a rational vertex operator algebra $\mathcal{V}$ and the modular fusion category $\mathcal{C}$ formed by its representations. The category $\mathcal{C}$ defines the Reshetikhin-Turaev TFT used in [36, 37, 40, 45, 47] to describe type ① theories.

To treat theories of type ②, we slightly extend this TFT to include parity. Namely, let $\widehat{\mathcal{C}}$ be the product

$$\widehat{\mathcal{C}} := \mathcal{C} \boxtimes \mathcal{SV}ect, \tag{3}$$

where $\mathcal{SVect}$ is the category of finite-dimensional super-vector spaces. Each simple object of $\mathcal{C}$ comes in two variants in $\widehat{\mathcal{C}}$, one parity even, and one parity odd. For two parity-odd objects, the braiding acquires an additional minus sign. Note that $\widehat{\mathcal{C}}$ itself is not a modular fusion category as it has symmetric centre $\mathcal{SVect}$.

Let $\mathcal{Bord}_3(\widehat{\mathcal{C}})$ denote the category of three-dimensional bordisms with embedded $\widehat{\mathcal{C}}$-decorated ribbon graphs. We consider the TFT

$$\widehat{\mathcal{Z}}_{\mathcal{C}} \colon \mathcal{Bord}_3(\widehat{\mathcal{C}}) \longrightarrow \mathcal{SVect}, \tag{4}$$

which is basically the product of the Reshetikhin-Turaev TFT for $\mathcal{C}$ and the trivial $\mathcal{SVect}$-valued TFT, see Section 4 for details.

Consider a surface $\Sigma$ with marked points $p_1, \ldots, p_n$ labelled by $X_1, \ldots, X_n \in \widehat{\mathcal{C}}$. To this, the TFT $\widehat{\mathcal{Z}}_{\mathcal{C}}$ assigns a vector space, which can be interpreted as the space of conformal blocks for the VOA $\mathcal{V}$ on $\Sigma$, but where the $\mathcal{V}$-representations are now of even or odd parity. This affects the monodromy-behaviour of conformal blocks by including parity signs.

From here on, the construction of CFT correlators is the same as in [45], but we go through it in some detail in Section 5 to stress the effect of including parity.

Let $\Sigma$ be an oriented world sheet, possibly with boundaries and defect lines. Bulk insertions are labelled by a tuple $(U, \bar{V}, \delta, \phi)$ and boundary fields by $(W, \nu, \psi)$, where $U, \bar{V} \in \mathcal{C}$ (not $\widehat{\mathcal{C}}$) give the holomorphic and antiholomorphic representation the bulk field transforms in, $W \in \mathcal{C}$ is the representation for the boundary field, $\delta, \nu \in \{\pm 1\}$ describe the parity of the fields, and $\phi, \psi$ take values in appropriate multiplicity spaces which depend on the theory under consideration.

From $\Sigma$ one constructs the double $\widetilde{\Sigma} = (\Sigma \sqcup \Sigma^{\mathrm{rev}})/\sim$, where $\Sigma^{\mathrm{rev}}$ is an orientation reversed copy of $\Sigma$ and $\sim$ identifies boundary points of $\Sigma$ and $\Sigma^{\mathrm{rev}}$. For example, if $\Sigma$ is a disc, then $\widetilde{\Sigma}$ is a sphere. A bulk insertion on $\Sigma$ splits into two points on $\widetilde{\Sigma}$, one on $\Sigma$ labelled by $U$ and one on $\Sigma^{\mathrm{rev}}$ labelled by $\bar{V}$. Boundary insertions result in a single marked point on $\widetilde{\Sigma}$ labelled by $W$. The first key ingredient of the TFT construction is:

> The correlator for a world sheet $\Sigma$ of an oriented parity CFT is an element of the space of conformal blocks on the double $\widetilde{\Sigma}$, i.e. in $\mathrm{Bl}(\Sigma) := \widehat{\mathcal{Z}}_{\mathcal{C}}(\widetilde{\Sigma})$.

Specifying an oriented parity CFT now amounts to, firstly, giving the multiplicity spaces for bulk fields in terms of $U, \bar{V}$ and the parity $\epsilon$ (which may be zero-dimensional), and ditto for boundary fields. This specifies the field content of the theory. Secondly, one has to give a collection of vectors $\mathrm{Corr}^{\mathrm{or}}(\Sigma) \in \mathrm{Bl}(\Sigma)$. The (bi)linear combinations of conformal blocks described by these vectors are then the correlators of the oriented parity CFT.

The collection $\{\mathrm{Corr}^{\mathrm{or}}(\Sigma)\}_{\Sigma}$ has to satisfy two consistency conditions: it has to be monodromy free, which amounts to mapping class group invariance, and it has to be compatible with gluing of world sheets, see Section 5.4 for details. These consistency conditions can be expressed via the TFT $\widehat{\mathcal{Z}}_{\mathcal{C}}$.

The second key point of the TFT construction is that consistent collections of correlators for CFTs of type ② can be constructed from a suitable algebraic input:

> From a symmetric special Frobenius algebra $B \in \widehat{\mathcal{C}}$ one can construct a consistent collection of correlators $\mathrm{Corr}_B^{\mathrm{or}}(\Sigma) \in \mathrm{Bl}(\Sigma)$. Boundaries of $\Sigma$ are labelled by $B$-modules and defect lines by $B$-$B$-bimodules.

The correlators $\mathrm{Corr}_B^{\mathrm{or}}(\Sigma)$ are described via the TFT $\widehat{\mathcal{Z}}_{\mathcal{C}}$ as follows: From $\Sigma$ one obtains a bordism $M_{\Sigma} \colon \emptyset \longrightarrow \widetilde{\Sigma}$ which contains a $\widehat{\mathcal{C}}$-decorated ribbon graph defined in terms of $B$ and the field insertions, boundaries and line defects on $\Sigma$. As a 3-manifold, $M_{\Sigma} = \Sigma \times [-1, 1]/\sim$,

where $(z, t) \sim (z, -t)$ for all $z \in \partial \Sigma$, $t \in [-1, 1]$. For example, if $\Sigma$ is a disc, then $M_\Sigma$ is a 3-ball. One then sets

$$\mathrm{Corr}_B^{\mathrm{or}}(\Sigma) := \widehat{\mathcal{Z}}_{\mathcal{C}}(M_\Sigma) \ \in \ \mathrm{Bl}(\Sigma). \tag{5}$$

If in this construction one takes $B \in \mathcal{C} \subset \widehat{\mathcal{C}}$, i.e. $B$ is purely parity even, one recovers precisely the construction in [45] of correlators of oriented CFTs without parity, which is type ① in the above table.

A more intuitive way to understand this construction is to think of $M_\Sigma$ as a fattening of the world sheet $\Sigma$, together with a surface defect inserted in place of the embedded copy of $\Sigma$ in $M_\Sigma$. The surface defect determines how the holomorphic and antiholomorphic fields combine to form a consistent collection of correlators. This interpretation of the construction in [45] was given in [64], and a detailed study of surface defects in Reshetikhin-Turaev TFTs can be found in [23, 50]. We will, however, not elaborate more on this point of view in the present paper.

Our first main result is (Theorems 5.4 and 5.6):

**Theorem 1.1.** *Let $B \in \widehat{\mathcal{C}}$ be a symmetric special Frobenius algebra. Then the collection of correlators $\{\mathrm{Corr}_B^{\mathrm{or}}(\Sigma)\}_\Sigma$, where $\Sigma$ runs over oriented world sheets with boundaries and defects, is monodromy free and compatible with gluing.*

One notable difference of oriented CFT with parity to that without parity is that the state spaces are now super-vector spaces, and that the modular invariant torus partition function is obtained as a super-trace. It is therefore a $\mathbb{Z}$-bilinear combination of characters, rather than a $\mathbb{Z}_{\geq 0}$-bilinear combination as in the parity-less case. See Section 5.5 and the example in Section 7.5 for more on this point.

After this discussion we can be more detailed about the relation between CFTs of types ① and ②. Firstly, one should distinguish whether one is working with just the bulk CFT without boundaries and defects, or with a CFT on surfaces with boundaries and defects. Let us denote these as type ①$_{\mathrm{bulk}}$ and ①$_{\mathrm{bnd\&def}}$, respectively, for $i = 1, 2$. We have the inclusions

$$①_{\mathrm{bulk}} \subset ②_{\mathrm{bulk}}, \qquad \text{and} \qquad ①_{\mathrm{bnd\&def}} \subset ②_{\mathrm{bnd\&def}}. \tag{6}$$

For the first inclusion one considers a bulk theory without parity as a bulk theory with parity which happens to involve only even parity fields, and similarly for the second inclusion for all bulk, boundary and defect fields. In terms of Theorem 1.1 this happens if $B \in \widehat{\mathcal{C}}$ is actually contained in $\mathcal{C}$, and if one only considers $B$-modules and $B$-bimodules contained in $\mathcal{C}$, rather than in $\widehat{\mathcal{C}}$.

We say that a CFT is *strictly of type* ② if it is not in the image of the inclusions in (6). In this sense, the table in (2) is about bulk theories without boundaries and defects which are strictly of the given type.

By restricting to surfaces without boundaries and defects, a CFT of type ①$_{\mathrm{bnd\&def}}$ becomes a theory of type ①$_{\mathrm{bulk}}$. However, the latter may be in the image of the first inclusion in (6), i.e. a CFT of type ②$_{\mathrm{bnd\&def}}$ can restrict to ①$_{\mathrm{bulk}}$. This happens if $B \in \mathcal{C}$ but one considers $B$-modules and $B$-bimodules contained in $\widehat{\mathcal{C}}$.

Turning this around, one can always *enhance a CFT of type* ①$_{\mathrm{bnd\&def}}$ *to a CFT of type* ②$_{\mathrm{bnd\&def}}$ by adding parity to all boundaries and defects, resulting in a doubling of boundary and defect conditions. In terms of Theorem 1.1, this means that $B \in \mathcal{C}$, but rather than considering only modules and bimodules in $\mathcal{C}$ (which gives ①$_{\mathrm{bnd\&def}}$) one allows modules and bimodules in $\widehat{\mathcal{C}}$ (which gives ②$_{\mathrm{bnd\&def}}$). The enhanced theory is strictly of type ②$_{\mathrm{bnd\&def}}$ (there are boundary and defect fields of either parity), but it restricts to a theory of type ①$_{\mathrm{bulk}}$.

This will actually be an important intermediate step for example when construction the free fermion CFT from the Ising CFT later on.

### 1.3 Spin CFT via oriented CFT with defects

Here we discuss the situation where the world sheet is equipped with a spin structure, that is, with a double cover of the oriented frame bundle such that the fibre over each point of the world sheet is connected. This results in CFTs of types ③ and ④. We will only treat the case with parity in detail. The case without parity can be obtained from this by choosing purely even input data, analogous to the oriented case above.

One can define correlators for spin CFTs with parity directly in terms of the TFT $\widehat{\mathcal{Z}}_{\mathcal{C}}$ in a construction that resembles that for oriented CFTs with parity. However, we find it convenient to take a slightly different route. Namely, we define correlators of spin CFTs in terms of correlators of oriented CFTs equipped with a specific choice of defect network. This is also the approach taken in [76, 82].

Let $\underline{\Sigma}$ be a world sheet with spin structure, and possibly with boundaries and line defects, and denote by $\Sigma$ the underlying oriented 2-manifold. Our main technical tool will be the combinatorial model for spin structures developed in [75, 80, 85] and reviewed in Section 2. The spin structure on $\underline{\Sigma}$ is encoded by choosing a decomposition $P_\Sigma$ of $\Sigma$ into polygons, and assigning an index $s_e \in \{0, 1\}$ to each edge $e$ of the decomposition, subject to an admissibility condition at each vertex. These indices encode the spin structure, and we write $P_\Sigma(\underline{\Sigma})$ for this combinatorial presentation of the spin structure of $\underline{\Sigma}$.

The spin structure on $\underline{\Sigma}$ is not required to extend to insertion points of bulk fields (it can always be extended to boundary insertions). We distinguish two types of bulk insertions: those where the spin structure does nonetheless extend are called Neveu-Schwarz (NS) insertions, and those where it does not extend are Ramond (R) insertions.

The input for the construction of correlators is a special Frobenius algebra $F \in \widehat{\mathcal{C}}$ whose Nakayama automorphism $N_F$ squares to the identity,

$$N_F^2 = \mathrm{id}_F .\tag{7}$$

Detailed definitions are given in Section 5.1. Here we just note that the Nakayama automorphism measures the failure of the invariant pairing on $F$ to be symmetric. In particular, $F$ is symmetric iff $N_F = \mathrm{id}_F$. Symmetric algebras therefore satisfy (7), but it turns out that the resulting correlators are independent of the spin structure of $\underline{\Sigma}$.

In the spin CFT constructed from $F$, boundaries on $\underline{\Sigma}$ are labelled by $\mathbb{Z}_2$-equivariant $F$-modules, and defects by $\mathbb{Z}_2$-equivariant $F$-$F$-bimodules, see Section 6.1. The action of $\mathbb{Z}_2$ is defined by twisting the $F$-action with $N_F$.

From $\underline{\Sigma}$ we will now construct an oriented world sheet with boundaries and defects,

$$W_{\mathrm{or}}(\underline{\Sigma}; F),\tag{8}$$

for the oriented parity CFT defined by the trivial symmetric special Frobenius algebra $B = \mathbf{1} \in \widehat{\mathcal{C}}$. Note that for $B = \mathbf{1}$, the boundary and defect labels are just objects in $\widehat{\mathcal{C}}$.

The underlying oriented 2-manifold of $W_{\mathrm{or}}(\underline{\Sigma}; F)$ is $\Sigma$, the underlying manifold for $\underline{\Sigma}$. Let $P_\Sigma(\underline{\Sigma})$ be a combinatorial presentation of the spin structure of $\underline{\Sigma}$. Then $W_{\mathrm{or}}(\underline{\Sigma}; F)$ is obtained by placing a defect graph dual to the polygonal decomposition on $\Sigma$, where the defects are labelled by $F$ and the vertices are built from products and coproducts of $F$ in a specific way. The $F$-defect lines that cross an edge of $P_\Sigma$ are equipped with a power of $N_F$ depending on the index $s_e$ of that edge. The precise description of this deliberately vague formulation is given in Sections 6.2 and 6.3.

We now define the correlator of the spin CFT as

$$\mathrm{Corr}_F^{\mathrm{spin}}(\underline{\Sigma}) := \mathrm{Corr}_{B=\mathbf{1}}^{\mathrm{or}}(W_{\mathrm{or}}(\underline{\Sigma}; F)).\tag{9}$$

Our second main result (Theorems 6.9 and 6.11) is:

**Theorem 1.2.** *Let $F \in \widehat{\mathcal{C}}$ be a special Frobenius algebra with $N_F^2 = \mathrm{id}_F$. Then the collection of correlators $\{\mathrm{Corr}_F^{\mathrm{spin}}(\Sigma)\}_{\underline{\Sigma}}$ is monodromy free and compatible with gluing.*

In the situation considered in (1), the monodromy does affect the spin structure on the surface, and so monodromy freeness is now the statement that $(-1)^{M+P+S} = 1$, where both $(-1)^S$ and $(-1)^P$ can be $+1$ or $-1$.

We can summarise this discussion as follows:

> For $F \in \widehat{\mathcal{C}}$ a special Frobenius algebra with $N_F^2 = \mathrm{id}_F$, the family $\{\mathrm{Corr}_F^{\mathrm{spin}}(\Sigma)\}_{\underline{\Sigma}}$, defined via $F$-defect networks in the $B = \mathbf{1}$ oriented CFT, provides a consistent collection of correlators on spin surfaces with boundaries and defects. Boundaries are labelled by $\mathbb{Z}_2$-equivariant $F$-modules and defect lines by $\mathbb{Z}_2$-equivariant $F$-$F$-bimodules.

In other words, we start from the diagonal theory of type $\textcircled{1}_{\mathrm{bnd\&def}}$ (we consider $B = \mathbf{1}$ in $\mathcal{C}$). Its conformal boundary conditions and topological defects are labelled by objects in $\mathcal{C}$. We then enhance it to type $\textcircled{2}_{\mathrm{bnd\&def}}$ as described in the end of the previous section (by allowing $B$-modules in $\widehat{\mathcal{C}}$). This does not affect the bulk theory, which is still of type $\textcircled{1}_{\mathrm{bulk}}$, but boundary conditions and defects are now labelled by objects in $\widehat{\mathcal{C}}$. Among the defects of the latter theory, we look for a defect $F$ (i.e. an object $\widehat{\mathcal{C}}$) which can be equipped with the structure of a special Frobenius algebra such that $(N_F)^2 = \mathrm{id}_F$. Thinking of $F$ as a possibly non-invertible symmetry, we can gauge $F$. In our setting, this amounts to inserting a network of $F$-defects as described above – we will come back to this point of view on gauging in Section 1.5 below. Since in general $F$ is not symmetric but only satisfies $(N_F)^2 = \mathrm{id}_F$, this is a priori not well-defined on oriented manifolds but is on world sheets with a spin structure.[1]

The resulting theory is in general of type $\textcircled{4}_{\mathrm{bnd\&def}}$. If $F$ happens to be contained in $\mathcal{C}$, the bulk theory does only involve fields of even parity, and one can restrict oneself to boundaries and defects labelled by $F$-modules and bimodules in $\mathcal{C}$, rather than $\widehat{\mathcal{C}}$. The restricted theory is of type $\textcircled{3}_{\mathrm{bnd\&def}}$. This is the spin-version of the inclusion (6), i.e.

$$\textcircled{3}_{\mathrm{bulk}} \subset \textcircled{4}_{\mathrm{bulk}}, \qquad \text{and} \qquad \textcircled{3}_{\mathrm{bnd\&def}} \subset \textcircled{4}_{\mathrm{bnd\&def}}. \tag{10}$$

In the context of bulk CFTs on spin world sheets, there are the additional inclusions

$$\textcircled{1}_{\mathrm{bulk}} \subset \textcircled{3}_{\mathrm{bulk}}, \qquad \text{and} \qquad \textcircled{2}_{\mathrm{bulk}} \subset \textcircled{4}_{\mathrm{bulk}}. \tag{11}$$

This happens if the bulk CFT turns out to be independent of the spin structure, i.e. if there is an underlying CFT defined on oriented world sheets, and the correlators on spin world sheets are obtained by first forgetting the spin structure and then evaluating the oriented theory. In our setting this happens if $F$ is symmetric, i.e. if $(N_F)^2 = \mathrm{id}$ is satisfied because we already have[1] $N_F = \mathrm{id}$. There are corresponding inclusions for types $\textcircled{i}_{\mathrm{bnd\&def}}$, but we will skip a more detailed discussion.

As an example of the construction of spin CFT correlators, in Section 6.5 we treat the torus with its four possible spin structures. We illustrate how $F$ determines the coefficients in the bilinear combination of characters giving the torus partition function in each of the four cases.

## 1.4 Examples from a self-dual invertible object

It turns out that one can already obtain examples of CFTs which are strictly of type $\textcircled{i}_{\mathrm{bulk}}$ for each of $i = 1, 2, 3, 4$ by considering algebras of the form $A_+ = \mathbf{1} \oplus G \in \mathcal{C}$ and $A_- = \mathbf{1} \oplus \Pi G \in \widehat{\mathcal{C}}$,

---

[1]To obtain an oriented theory, it is actually enough for $N_F$ to be an inner automorphism (see Remark 6.7). The condition $(N_F)^2 = \mathrm{id}$ can be analogously weakened. For a Morita-invariant formulation of this condition in the context of fully extended topological field theory, see [24].

where $G \in \mathcal{C}$ satisfies $G \otimes G \cong \mathbf{1}$ and $\Pi G \in \widehat{\mathcal{C}}$ denotes the parity-odd copy of $G$. The objects $A_\pm$ each carry an up-to-isomorphism unique structure of a special Frobenius algebra whose Nakayama automorphism squares to the identity, or is already equal to the identity. They can thus serve as an input for the constructions presented in Sections 1.2 and 1.3. We consider the trivial case $G \cong \mathbf{1}$ first, and then turn to $G \not\cong \mathbf{1}$.

### 1.4.1  The case $G \cong \mathbf{1}$

The case of $A_- = \mathbf{1} \oplus \Pi\mathbf{1}$ is instructive, both for its simplicity and for the fact that, when applied to the trivial $c = 0$ CFT, it gives the Arf-invariant CFT, or rather, 2d TFT of type ④. In that example $\mathcal{C} = \mathcal{V}ect$, $\widehat{\mathcal{C}} = \mathcal{SV}ect$, and $A_-$ is the Clifford algebra in one odd generator, see Section 7.1 for more details.

The case $A_+ = \mathbf{1} \oplus \mathbf{1}$ simply results in two copies of the initial theory and we do not consider it further.

### 1.4.2  The case $G \not\cong \mathbf{1}$

Let now $G \in \mathcal{C}$ satisfy $G \not\cong \mathbf{1}$ and $G \otimes G \cong \mathbf{1}$. This implies that $G$ is invertible and self-dual, or, in other words, that $G$ is an order two simple current. The quantum dimension of $G$ necessarily satisfies $\dim(G)^2 = 1$. Let $\theta_G$ be the eigenvalue of the ribbon twist on $G$. It is related to the conformal weight $h$ via $\theta_G = \exp(2\pi i h)$. We require that $h \in \frac{1}{2}\mathbb{Z}$, i.e. that

$$\theta_G \in \{\pm 1\}. \tag{12}$$

For each of the eight possible choices of $\dim G$, $\nu$, $\theta_G$, the algebra $A_\nu$ can serve as an input to determine a consistent collection of correlators. If $N_{A_\nu} = \mathrm{id}_{A_\nu}$, i.e. if $A_\nu$ is symmetric, one obtains a theory of type ① ($\nu = 1$) or type ② ($\nu = -1$). If $N_{A_\nu} \neq \mathrm{id}_{A_\nu}$ one correspondingly gets a theory of type ③ ($\nu = 1$) or ④ ($\nu = -1$).

The dimension and twist multiply under taking products in the following sense. Let $\mathcal{C}$ and $\mathcal{C}'$ be modular fusion categories, obtained as representation categories of rational VOAs $\mathcal{V}$ and $\mathcal{V}'$, respectively. Suppose there are order two simple currents $G \in \mathcal{C}$ and $G' \in \mathcal{C}'$. Then $\mathcal{C} \boxtimes \mathcal{C}'$ is the representation category for the product VOA $\mathcal{V} \otimes_{\mathbb{C}} \mathcal{V}'$, and it contains the order two simple currents $G \boxtimes \mathbf{1}'$, $\mathbf{1} \boxtimes G'$, and $G \boxtimes G' =: \mathbb{G}$. Then

$$\dim(\mathbb{G}) = \dim(G)\dim(G'), \qquad \theta_{\mathbb{G}} = \theta_G \theta_{G'}. \tag{13}$$

The parity is also multiplicative, and so the eight cases for the triple $(\dim G, \nu, \theta_G)$ form a $(\mathbb{Z}_2)^{\times 3}$.

We collect the case $G \cong \mathbf{1}$ and the eight cases for $G \not\cong \mathbf{1}$ in Table 1. Let us explain the columns in turn. The first four columns are clear. The meaning of the remaining columns is as follows:

- The column "$A_\nu$ sym." states whether the algebra $A_\nu$ is symmetric, and as just explained that corresponds to the type being ①/② or ③/④.

- The column "$A_\nu$ com." states whether $A_\nu$ is commutative in $\widehat{\mathcal{C}}$ or not. Commutative algebras define extensions of the VOA $\mathcal{V}$ [19,56].

- In the column "$G$ hol." we consider the bulk state space of the theory defined by $A_\nu$. If this space contains a field that transforms in the holomorphic / antiholomorphic representation $(G, \mathbf{1})$, i.e. if the theory contains the elements of $G$ as holomorphic fields, we mark a "$\checkmark$", and we indicate the parity $\pm$ of those fields, and, in the spin case, whether they are in the NS or R sector. It turns out that a holomorphic copy of $G$ exists iff an antiholomorphic copy of $G$ exists.

Table 1: Properties of the CFT defined by $A_\nu$ in dependence on the dimension and twist of the invertible self-dual object $G$ and on the parity $\nu$ of the second summand of $A_\nu$.

| $G$ | $\dim G$ | $\nu$ | $\theta_G$ | type | $A_\nu$ sym. | $A_\nu$ com. | $G$ hol. | example |
|---|---|---|---|---|---|---|---|---|
| $\mathbf{1}$ | $1$ | $-1$ | $1$ | ④ | $\times$ | $\times$ | $\checkmark$ [$-$,R] | Arf invariant, $c = 0$ |
| $\not\cong \mathbf{1}$ | $1$ | $1$ | $1$ | ① | $\checkmark$ | $\checkmark$ | $\checkmark$ [$+$] | Potts $c = \frac{4}{5}$ |
| | $1$ | $1$ | $-1$ | ① | $\checkmark$ | $\times$ | $\times$ | Ising $c = \frac{1}{2}$ |
| | $1$ | $-1$ | $1$ | ④ | $\times$ | $\times$ | $\checkmark$ [$-$,R] | fermionic tetra-Ising $c = \frac{4}{5}$ |
| | $1$ | $-1$ | $-1$ | ④ | $\times$ | $\checkmark$ | $\checkmark$ [$-$,NS] | free fermion $c = \frac{1}{2}$ |
| | $-1$ | $1$ | $1$ | ③ | $\times$ | $\times$ | $\checkmark$ [$+$,R] | (spin BP)$\times$(Ising) $c = \frac{9}{10}$ |
| | $-1$ | $1$ | $-1$ | ③ | $\times$ | $\checkmark$ | $\checkmark$ [$+$,NS] | spin BP $c = \frac{2}{5}$ |
| | $-1$ | $-1$ | $1$ | ② | $\checkmark$ | $\checkmark$ | $\checkmark$ [$-$] | (parity BP)$\times$(Ising) $c = \frac{9}{10}$ |
| | $-1$ | $-1$ | $-1$ | ② | $\checkmark$ | $\times$ | $\times$ | parity BP $c = \frac{2}{5}$ |

- The last column lists an example of the corresponding theory. These are discussed in more detail in Section 7. BP stands for Bershadsky-Polyakov. The table also contains two product theories which illustrate (13).

In particular, we obtain explicit examples of CFTs of type ② and ③ in terms of Bershadsky-Polyakov vertex operator algebras (Section 7.5), which to the best of our knowledge are new.

For spin theories, the torus partition function with R-R spin structure is itself modular invariant. For the free fermion, this particular torus partition function is zero, and for supersymmetric models it is constant, but for the fermionic tetra-critical Ising model it is neither zero nor constant. We compute the corresponding $\mathbb{Z}$-bilinear combination of $c = \frac{4}{5}$ Virasoro minimal model characters in Section 7.4, and find that it is given by the difference of the $A$- and $D$-type modular invariant, i.e. by tetra-critical Ising minus three-state Potts. This agrees with [57], where this model was first considered.

## 1.5 Relation to gauging topological symmetries

A very useful point of view on the construction of consistent collections of CFT correlators is to understand it as a generalised orbifold or, equivalently, as a gauging of topological symmetries, or as an internal state sum.

Orbifolds of CFTs were first considered for finite groups [26, 31], and amount to adding twisted sectors to the theory and then passing to suitably invariant states. This procedure has a conceptually important reinterpretation as gauging a discrete symmetry of the original theory [32], which also provides a connection to state sum constructions.

The study of topological line defects in CFT [43, 45, 79] allows one to understand discrete symmetries of CFTs as invertible line defects. Topological line defects are typically not invertible, and it is fruitful to think of the collection of all topological defects as an extension of the notion of symmetry of a field theory. We will refer to such not-necessarily-invertible defects as a "topological symmetry" of a given theory. For rational CFTs, and in fact for 2d QFTs in general, these form a pivotal monoidal category, or a bicategory with adjoints in the case where one includes topological interfaces [28, 43, 88].

The idea to orbifold a given theory by a not-necessarily-invertible topological symmetry was first formulated in [44] in the context of rational conformal field theory as a reinterpretation of the TFT construction of CFT correlators on oriented world sheets given in [45]. This

procedure was called a "generalised orbifold", but recently is more often referred to as "gauging a topological symmetry".

For oriented topological field theories in arbitrary dimension, taking generalised orbifolds / gauging topological symmetries is discussed in [22]. For Reshetikhin-Turaev TFTs there is a completeness result analogous to the one above [20, 72]: any two such TFTs based on modular fusion categories in the same Witt class are obtained from one another by gauging a topological symmetry.

The idea to use gauging of topological symmetry to relate oriented and spin topological field theories was developed in [2, 9, 53]. In two-dimensional CFTs it was used in [76, 82] in the guise of defect networks. In [54, 57, 65, 66] the gauging of $\mathbb{Z}_2$-symmetries is used to link oriented and spin CFTs. The examples considered in Section 1.4 can be understood as such a $\mathbb{Z}_2$-gauging. We compare the results of our construction with those in [57] in Section 7.4.4.

As described at the end of Section 1.3, the construction of spin CFTs in this paper can be reformulated in the context of gauging as follows: Starting from a diagonal CFT of type ①$_{\text{bnd\&def}}$, we pass to the theory with parity enhanced defects of type ②$_{\text{bnd\&def}}$ and gauge a suitable topological symmetry $F$ with the property that the gauging-procedure needs a spin structure on the world sheet to be well-defined. This results in a spin theory of type ④$_{\text{bnd\&def}}$.

An example which makes use of gauging a non-invertible symmetry is the exceptional spin CFT for $su(2)$ at level 10. The starting point is the usual WZW model for $su(2)$ at level 10, by which we mean the oriented CFT with diagonal modular invariant. After parity enhancement one can identify a non-invertible topological defect $F$ whose gauging produces a spin CFT, which we call "exceptional" as the sum over spin structures produces the $E_6$-type modular invariant oriented $su(2)_{10}$ WZW CFT. The relevant defect decomposes as $F = \underline{0} + \underline{6} + \Pi\underline{4} + \Pi\underline{10}$, where the underlined numbers are Dynkin-labels and $\Pi$ refers to the parity shift. The summands $\underline{6}$ and $\Pi\underline{4}$ describe non-invertible topological defects. However, we will not discuss this model in the present paper.

Boundaries and topological defects in spin theories were studied using the folding trick in [73], from the point of view of classifying algebras in [82], via gauging of a $\mathbb{Z}_2$-symmetry in [13, 34, 51], via the Cardy consistency condition in [61], and in relation to super-fusion categories in [16].

To conclude the introduction, let us discuss in which sense we expect our construction to be complete, and how it can generalise beyond rational CFTs.

Given a rational VOA $\mathcal{V}$, one can ask what are all the oriented and spin bulk theories which contain a holomorphic and an anti-holomorphic copy of $\mathcal{V}$ in the space of bulk fields, which have a unique vacuum (in the NS-sector in the spin case), and which have non-degenerate two-point functions. Let us refer to the collection of such theories as $\mathcal{T}(\mathcal{V})$. We expect that all theories in $\mathcal{T}(\mathcal{V})$ can be obtained by our construction for an appropriate choice of Frobenius algebra in $\widehat{\mathcal{C}}$, generalising the existence and uniqueness result for type ① theories in [40, 41, 63]. What is more, it is known from [44] that any two type ① CFTs in $\mathcal{T}(\mathcal{V})$ are generalised orbifolds of one another (i.e. can be obtained from each other by gauging). We expect the same to hold for any two CFTs $C_1, C_2 \in \mathcal{T}(\mathcal{V})$, even if one is spin and the other oriented, or if one involves parity and the other does not. That is, there should exist a topological defect $F$ in $C_1$ so that the generalised orbifold (or gauging) by it produces $C_2$. Interfaces between spin and oriented theories have already been studied in [52, 73, 82].

The construction we present in this paper is not restricted to rational CFTs. Indeed, the same gauging procedure works in any 2d QFT, conformal or not, for which one can find a topological defect $F$ and topological junctions for product, coproduct, unit and counit, such that it forms a special Frobenius algebra with $N_F = \text{id}$ or $(N_F)^2 = \text{id}$ in the pivotal monoidal category of topological defects of the QFT (as defined in [28, Sec. 2.4]). The massive Ising CFT and the massive free fermion would be an example of this.

**Organisation of this paper**

In Section 2, we review the combinatorial description of spin structures and give our conventions for bordisms and world sheets with boundaries and defect lines. We discuss $r$-spin structures in general, rather than just 2-spin structures, as having $r \in \mathbb{Z}_{\geq 0}$ as a parameter does not add any difficulty and makes the description more transparent.

Section 3 gives a short overview of the relation between modular fusion categories and Reshetikhin-Turaev type 3d TFTs on the one side, and representations of rational VOAs and their conformal blocks on the other side. This section prepares the ground and sets the notation for the parity-extension in the next section.

In Section 4 we show how to include parity in the 3d TFT. On the TFT side we describe the product of the TFT in Section 3 with the trivial $\mathcal{SV}ect$-valued TFT. On the VOA side we explain how to interpret the state spaces of the parity-enhanced TFT in terms of spaces of conformal blocks, where insertion points in addition have even or odd parity.

Sections 5 and 6 contain the main new constructions presented in this paper. In Section 5 we extend the TFT description of CFT correlators with boundaries and topological line defects to oriented CFTs with parity: correlators are given as elements in the corresponding spaces of conformal blocks. Both are given by evaluating the TFT from Section 4 on suitable three-manifolds and surfaces.

Section 6 treats spin CFTs with parity. Here, the world sheets are equipped with a spin structure, and they can have boundaries and line defects. The approach we take is to express correlators on world sheets with spin structure in terms of correlators for the oriented CFT introduced in Section 5, which now feature a defect network that encodes the spin structure.

In Section 7 we present a number of explicit examples which illustrate the abstract constructions. This includes in particular the Bershadsky-Polyakov models which provide examples for spin CFTs without parity, as well as for oriented parity CFTs, both of which have not been systematically studied before.

Finally, in the appendix we collect several proofs and detailed computations we omitted in the main text.

## 2 Bordisms with boundaries, defects, and spin structures

In this section we collect the geometric structures on two-dimensional bordisms that we will need to define the oriented and spin CFTs with boundaries and defects. We start with the geometric and combinatorial description of spin structures, and then turn to boundaries and line defects.

In the description of spin structures we will be slightly more general than needed later. Namely, we review $r$-spin structures while later we only work with 2-spin structures. The reason is that $r$-spin structures are not more complicated to describe and having $r$ as a parameter removes the degeneracy $1 = -1 \in \mathbb{Z}_2$ one has in the 2-spin case. After this review section, spin structures only appear again in Sections 6 and 7.

### 2.1 Geometric description of open-closed spin surfaces

Here we describe open-closed bordisms with parametrised boundaries, as well as including spin structures and line defects.

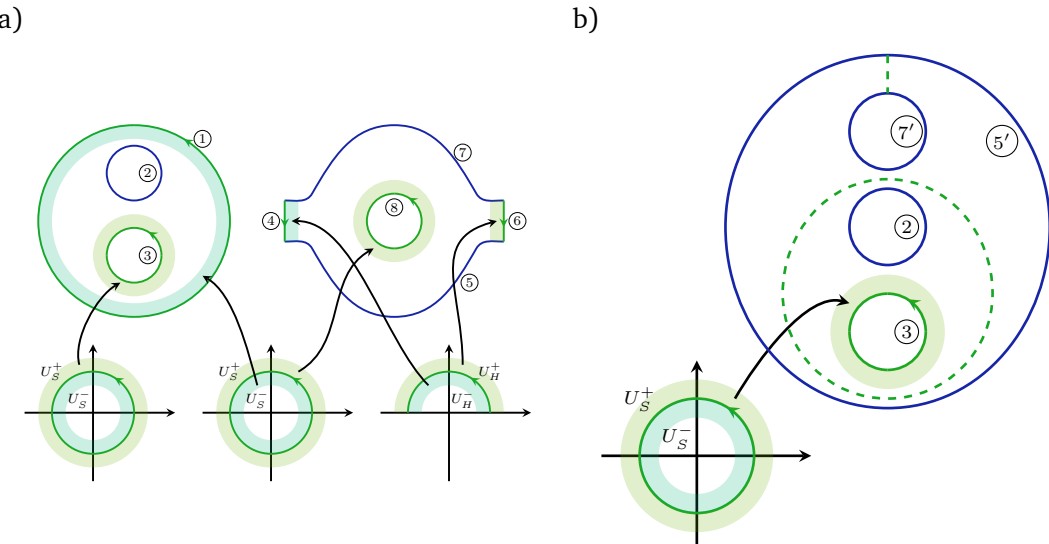

Figure 2: a) An open-closed bordism $\Sigma$. The boundary $\partial\Sigma$ decomposes as follows: $\partial^f\Sigma$ consists of the components labelled $2, 5, 7$, $\partial^c_{in}\Sigma$ consists of $3, 8$, $\partial^c_{out}\Sigma$ of 1, $\partial^o_{in}\Sigma$ of 6, $\partial^o_{out}\Sigma$ of 4. b) The bordism obtained by gluing components 1 and 8, as well as 4 and 6. The location of the now erased gluing boundary is shown as dashed lines.

## Open-closed bordisms

By a *surface* we mean an oriented smooth 2d manifold, possibly with boundaries and corners. For a 2d manifold with corners, each point has a coordinate chart to some open subset of the upper right closed quadrant $\mathbb{R}^2_{\geq 0} = \{(x, y) \in \mathbb{R}^2 | x, y \geq 0\}$, see e.g. [70, Sec. 3] for more details. The coordinate chart changing maps for points on the boundary or for corner points are required to extend to diffeomorphisms between open subset of $\mathbb{R}^2$, even if initially defined just on open subsets of $\mathbb{R}^2_{\geq 0}$. This allows one to assign a (two-dimensional) tangent space to boundary points and corners of $\Sigma$.

An *open-closed bordism* is a compact surface $\Sigma$, where parts of the boundary are parametrised, allowing bordisms to be glued together. While this is standard, see e.g. [84, Sec. 3.1] and Figure 2, we need some details to describe the spin case later. Boundary components and their neighbourhood will be parametrised by certain subsets of $\mathbb{R}^2$ and of the closed upper half plane $\overline{\mathbb{H}} = \mathbb{R} \times \mathbb{R}_{\geq 0}$:

- $U_S \subset \mathbb{R}^2$ denotes an open neighbourhood of the unit circle, and $U_S^+$ is the intersection of such a neighbourhood with $\{p \in \mathbb{R}^2 | |p| \geq 1\}$, i.e. $\mathbb{R}^2$ minus the open unit disc, while $U_S^-$ stands for the intersection with the closed unit disc.

- $U_H \subset \overline{\mathbb{H}}$ and $U_H^\pm$ are the same as above, but in addition intersected with $\overline{\mathbb{H}}$. Note that $U_H^\pm$ contain corners.

We will take $U_S$, $U_S^\pm$, etc, to denote neighbourhoods of the type described above, not a fixed choice made once and for all.

The boundary $\partial\Sigma$ (which includes the corners) is decomposed into disjoint subsets as

$$\partial\Sigma = \partial^g\Sigma \cup \partial^f\Sigma, \qquad (14)$$

where $\partial^g\Sigma$ is referred to as the *gluing boundary* and $\partial^f\Sigma$ as the *free boundary*. The free boundary consists of circles and closed intervals in $\partial\Sigma$, where the endpoints of the intervals are required to be corners. The gluing boundary is further decomposed into an in- and outgoing and open/closed part:

$$\partial^g\Sigma = \partial^c_{\text{in}}\Sigma \cup \partial^c_{\text{out}}\Sigma \cup \partial^o_{\text{in}}\Sigma \cup \partial^o_{\text{out}}\Sigma. \tag{15}$$

The closed gluing boundary consists of circular components of $\partial\Sigma$, and the open gluing boundary of open intervals between two corners in $\partial\Sigma$. A connected component of $\partial\Sigma$ without corners can either be a closed gluing boundary or a free boundary. If a connected component of $\partial\Sigma$ has corners, its further decomposition has to alternate between free and open gluing boundaries at the corners.

Each component $B$ of the gluing boundary $\partial^g\Sigma$ is equipped with a parametrising map, cf. Figure 2:

- *in-going closed gluing boundary $B \subset \partial^c_{\text{in}}\Sigma$*: A smooth map $\varphi : U^+_S \longrightarrow \Sigma$ whose image is open in $\Sigma$ and which is a diffeomorphism onto its image. It follows that $\varphi$ maps $S^1 \subset \mathbb{R}^2$ to a boundary component of $\Sigma$, and we require this boundary component to be $B$.

- *out-going closed gluing boundary $B \subset \partial^c_{\text{out}}\Sigma$*: A map $\varphi : U^-_S \longrightarrow \Sigma$ with the same properties as above.

- *in-going open gluing boundary $B \subset \partial^o_{\text{in}}\Sigma$*: A map $\varphi : U^+_H \longrightarrow \Sigma$ with the same properties as above. It follows that $\varphi$ maps the closed upper half circle to part of the boundary of $\Sigma$, and the two endpoints to corners of $\Sigma$. We require $B$ to be the image of the open upper half circle.

- *out-going open gluing boundary $B \subset \partial^o_{\text{out}}\Sigma$*: A map $\varphi : U^-_H \longrightarrow \Sigma$ as above.

When it is clear from the context, in the following we will say e.g. "closed boundary" or "gluing boundary" instead of "closed boundary component" or "gluing boundary component" for brevity.

A *diffeomorphism of open-closed bordisms* is an orientation preserving diffeomorphism of the underlying surface compatible with the (germs of the) boundary parametrisations.

Given a (not necessarily connected) open-closed world sheet, one can use the parametrising maps to glue an in-going to an out-going closed boundary, and ditto for open boundaries. See again Figure 2 for an example.

**Spin structures on surfaces**

We now review the definition of $r$-spin structures on surfaces, where $r \in \mathbb{Z}_{\geq 0}$. The case $r = 2$ is the usual spin case, and the case $r = 0$ is equivalent to considering a framing on the surface, while for $r = 1$ one just recovers the underlying oriented surface. More details can be found in e.g. [69, 74, 86].

Let us first discuss the simpler case of a surface $\Sigma$ which is equipped with a metric. Denote by $F^{SO}_\Sigma \longrightarrow \Sigma$ the bundle of oriented orthonormal frames in the tangent space of $\Sigma$. The group $SO_2$ acts freely and transitively on the fibres of $F^{SO}_\Sigma$, i.e. $F^{SO}_\Sigma$ is an $SO_2$-principal bundle. The frame bundle is well-defined also over the boundary and over corner points of $\Sigma$ due to the requirement that changes of charts extend smoothly to a neighbourhood of the boundary of $\mathbb{R}^2_{\geq 0} \subset \mathbb{R}^2$.

Write $\widetilde{SO_2}^r$ for the $r$-fold cover of $SO_2$ if $r > 0$, and for the universal cover if $r = 0$, and denote the covering map by

$$p^r_{SO} : \widetilde{SO_2}^r \longrightarrow SO_2. \tag{16}$$

For $r > 0$, $\widetilde{SO_2}^r$ is isomorphic to $SO_2$, but for $r = 0$ it is not. An explicit description of $\widetilde{SO_2}^r$ valid for all $r$ is

$$p_{SO}^r : \mathbb{R}/r\mathbb{Z} \longrightarrow SO_2, \quad x \longmapsto e^{2\pi i x}. \tag{17}$$

An *r-spin structure* on $\Sigma$ is an $\widetilde{SO_2}^r$-principal bundle $P \longrightarrow \Sigma$ together with a bundle map $p : P \longrightarrow F_\Sigma^{SO}$ which intertwines the $\widetilde{SO_2}^r$ and $SO_2$ actions. Some properties of $r$-spin structures are:

- Being a principal bundle, the fibre of $P$ over a point $z \in \Sigma$ looks like $\widetilde{SO_2}^r$, in particular it is connected.

- The fibre of $P$ above a point in the frame bundle $F_\Sigma^{SO}$ (via the bundle map $p$) consists of $r$ points (for $r > 0$), respectively of infinitely many points (for $r = 0$). They are in bijection with the kernel $\mathbb{Z}_r := \mathbb{Z}/r\mathbb{Z}$ of the covering map $p_{SO}^r$.

- A closed path in the frame bundle $F_\Sigma^{SO}$ defines a holonomy in $\mathbb{Z}_r$ by lifting the path to the spin bundle $P$. For example, the path obtained by acting on a given frame by $x \longmapsto e^{2\pi i x}$, $x \in [0,1]$, lifts to a non-closed (for $r \neq 1$) path in $P$ with holonomy $1 \in \mathbb{Z}_r$.

If $\Sigma$ does not carry a metric, one can no longer speak of orthonormal frames, but only of oriented frames. The oriented frame bundle $F_\Sigma \longrightarrow \Sigma$ is a principal bundle for $GL_2^+$, the group of linear endomorphisms of positive determinant. As in the metric case, one considers its $r$-fold cover (for $r > 0$), respectively the universal cover (for $r = 0$),

$$p_{GL}^r : \widetilde{GL_2}^r \longrightarrow GL_2^+. \tag{18}$$

For $r = 2$ an explicit realisation of $\widetilde{GL_2}^r$ is given in [75, Sec. 2.2]. One now simply replaces $SO$ by $GL$ in the above discussion:

**Definition 2.1.** An *r-spin structure on* $\Sigma$ is an $\widetilde{GL_2}^r$-principal bundle $P$ over $\Sigma$ together with a bundle map $p : P \longrightarrow F_\Sigma$ which intertwines the $\widetilde{GL_2}^r$ and $GL_2^+$ actions.

We will call a surface with $r$-spin structure an *r-spin surface* and denote it by $\mathbb{\Sigma} = (\Sigma, P, p)$. An *isomorphism of r-spin surfaces* $F$ from $\mathbb{\Sigma} = (\Sigma, P, p)$ to $\mathbb{\Sigma}' = (\Sigma', P', p')$ is a map $F : P \longrightarrow P'$ of $\widetilde{GL_2}^r$-principal bundles with underlying diffeomorphism $f : \Sigma \longrightarrow \Sigma'$ such that

$$\begin{array}{ccc}
P & \xrightarrow{\;F\;} & P' \\
{\scriptstyle p}\downarrow & & \downarrow{\scriptstyle p'} \\
F_\Sigma & \xrightarrow{\;df_*\;} & F_{\Sigma'} \\
\downarrow & & \downarrow \\
\Sigma & \xrightarrow{\;f\;} & \Sigma'
\end{array} \tag{19}$$

commutes. Here, $df_*$ is the map induced by $f$ on the frame bundle, acting by $df$ on each vector in a given frame. An *isomorphism of r-spin structures* on a given surface is an isomorphism of $r$-spin surfaces whose underlying diffeomorphism is the identity.

Since $GL_2^+$ retracts to $SO_2$, dropping the metric does not change the number of isomorphism classes of $r$-spin structures on a given surface.

Given a diffeomorphism $f : \Sigma \longrightarrow \Sigma'$ as above, we automatically get a bundle isomorphism $df_* : F_\Sigma \longrightarrow F_{\Sigma'}$ as in (19). We can use this to pull back the bundle $p' : P' \longrightarrow F_{\Sigma'}$ to $F_\Sigma$, resulting in a spin structure on $\Sigma$. Thus, if $\mathbb{\Sigma}' = (\Sigma', P', p')$ is an $r$-spin surface, $\Sigma$ is a surface,

and $f : \Sigma \longrightarrow \Sigma'$ is a diffeomorphism, we obtain a *pull-back r-spin structure* on $\Sigma$, which we denote by $f^*P'$ with resulting $r$-spin surface

$$(\Sigma, f^*P', f^*p'). \tag{20}$$

By construction, $(\Sigma, f^*P', f^*p')$ and $\mathbb{\Sigma}'$ are isomorphic as $r$-spin surfaces.

**Spin structures via holonomies**

We have already noted that given an $r$-spin surface $\mathbb{\Sigma}$, a closed path in the frame bundle $F_\Sigma$ defines a $\mathbb{Z}_r$-valued holonomy by lifting the path from $F_\Sigma$ to the $r$-spin bundle $P$. In fact, these holonomies determine the isomorphism class of the $r$-spin structure. More formally, isomorphism classes of $r$-spin structures on $\Sigma$ are in 1-1 correspondence with elements $H^1(F_\Sigma, \mathbb{Z}_r)$ which assign the value 1 to each path in $F_\Sigma$ given by rotating a frame once around itself in anticlockwise direction while keeping the point in $\Sigma$ fixed, see e.g. [83, Sec. 2].

It turns out that one can also assign a holonomy to a closed path in the surface $\Sigma$, rather than in $F_\Sigma$, provided the path is smooth with nowhere vanishing derivative. Namely, let $\gamma : S^1 \longrightarrow \Sigma$ be a smooth curve such that $\frac{d}{dt}\gamma(t) \neq 0$ for all $t$. Pick a smooth lift $\widehat{\gamma} : S^1 \longrightarrow F_\Sigma$ of $\gamma$ to the frame bundle, such that the first vector of the frame at a point $z = \gamma(t)$ is the derivative $\frac{d}{dt}\gamma(t)$, and the second vector is chosen to produce an oriented frame at $z$. All such lifts $\widehat{\gamma}$ are homotopic, and the further lift from $F_\Sigma$ to $P$ defines the holonomy

$$\zeta(\gamma) \in \mathbb{Z}_r. \tag{21}$$

Again, knowing the holonomies for a generating set of smooth closed curves determines the $r$-spin structure on $\Sigma$ up to isomorphism. The possible holonomies are constrained by the condition that a path $\gamma$ bounding a disc and running anticlockwise must have holonomy $\zeta(\gamma) = 1 \in \mathbb{Z}_r$. For example, by considering $\gamma$ to be the equator on a sphere, this implies that a sphere can be equipped with an $r$-spin structure only if $1 = -1 \mod r$, i.e. for $r \in \{1, 2\}$.

**Open-closed spin bordisms**

To describe parametrised boundaries for spin surfaces, we need to equip the open sets $U_S$, $U_H$ in the description of open-closed bordisms with an $r$-spin structure. The two cases behave differently:

- *Closed gluing boundary:* $r$-spin structures on the punctured complex plane $\mathbb{C}^\times$ are characterised by their holonomy along the unit circle. We pick a standard $r$-spin surface $\mathbb{C}^y$ with underlying surface $\mathbb{C}^\times$ for each $y \in \mathbb{Z}_r$. We use the explicit representatives from [74, Sec. 3.4], for which the holonomy along the unit circle $S^1$ with anticlockwise orientation is actually

$$\zeta(S^1) = 1 - y. \tag{22}$$

  The minus sign is just a convention which is natural from the point of view of the explicit construction. The shift by 1, however, is convenient as the resulting $\mathbb{Z}_r$ grading by $y$ of bulk fields in the CFT will be respected by the OPE (see Figure 27 below, though a more detailed discussion of OPEs will only be given in a follow-up paper). Geometrically, this is the observation that $y = 0$ corresponds to the unique spin structure which extends from $\mathbb{C}^\times$ to $\mathbb{C}$, and on the level of $r$-spin TFT in 2d this follows from [85, Sec. 5.1].[2]

  By $U_S^y$ we denote open neighbourhoods of $S^1 \subset \mathbb{C}^\times$, together with the $r$-spin structure obtained by restricting that of $\mathbb{C}^y$.

---

[2]In [85] grading by holonomy is used, and for the pair of pants with two ingoing and one outgoing closed gluing boundary, the holonomy at the outgoing circle is obtained by adding those of the ingoing circles and subtracting 1. Thus the holonomy grading is not preserved by this bordism.

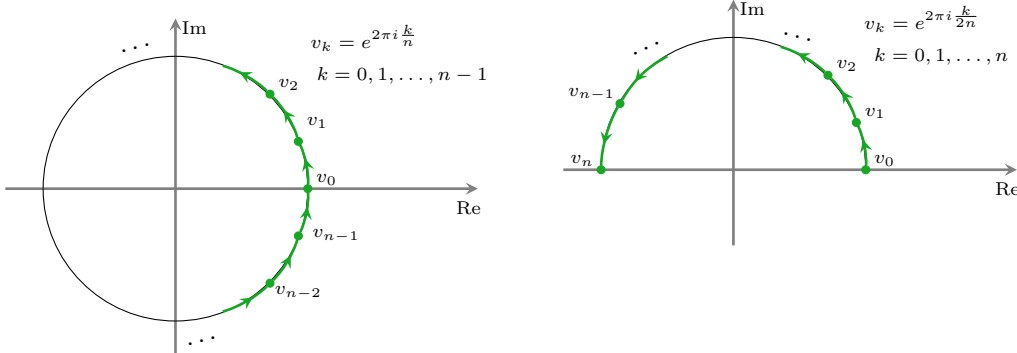

Figure 3: Standard position of $n$ vertices (resp. $n + 1$ vertices) and $n$ edges on the unit circle (resp. unit half-circle).

- *Open gluing boundary:* There is a unique $r$-spin structure on $\overline{\mathbb{H}}$ (including the origin or not does not make a difference). We write $\overline{\mathbb{H}}$ for the $r$-spin surface obtained by restricting $\mathbb{C}^0$ to $\overline{\mathbb{H}}$, and we denote by $U_H$ open neighbourhoods of the closed unit half-circle with the $r$-spin structure obtained by restricting that of $\overline{\mathbb{H}}$.

An *open-closed $r$-spin bordism* $\Sigma$ is now defined in exactly the same way as in the non-spin case above, except for two modifications: Firstly, all parametrising maps are now $r$-spin maps, and secondly, for in/out-going gluing boundaries the parametrising maps $\varphi$ have domain $U_S^{y,\pm}$ for a $y$ determined by the $r$-spin structure on $\Sigma$ by (22) via the holonomy $\zeta(\varphi(S))$. Here, $S$ is a rescaled version of the unit circle that lies in the relevant domain $U_S^{\pm}$. A closed gluing boundary whose parametrising map has domain $U_S^{y,\pm}$ will be called of

$$\text{type } y \in \mathbb{Z}_r\,. \tag{23}$$

As in the non-spin case, for open-closed spin bordisms in- and out-going boundaries can be glued using the parametrisation by $r$-spin maps. In the closed case, this requires the in- and out-going boundaries to be of the same type $y$.

The $r$-spin structure on an open-closed bordism can again be characterised up to isomorphism by its holonomies $\zeta$, provided we also include smooth arcs between gluing boundaries rather than just smooth closed curves. We will skip the detailed description of how holonomies for such arcs are defined geometrically, as they will not be used in this paper, and refer to [24, Sec. 3.3.4] for details.

## 2.2 Combinatorial description of spin structures

In this section we recall the combinatorial description of $r$-spin structures developed in [74, 75, 80, 85]. The idea is to decompose the surface into polygons, glued along their edges. Each polygon is contractible and hence allows for a unique-up-to-isomorphism $r$-spin structure. The data of the global $r$-spin structure is then encoded in the transition functions across the edges, given by values in $\mathbb{Z}_r$ that are called *edge indices* below. Not all collections of edge-indices are allowed, but only those where the $r$-spin structure extends to the vertices of the polygonal decomposition, giving linear conditions on the edge indices. The easiest way to link the geometric and combinatorial point of view is by computing holonomies.

**Marked polygonal decompositions**

Let $\Sigma$ be an open-closed world sheet (not already equipped with an $r$-spin structure). By a *polygonal decomposition* $T_\Sigma$ of $\Sigma$ we mean writing the surface as a disjoint union of embedded

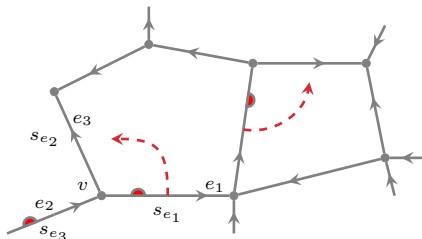

Figure 4: Example of a marked polygonal decomposition. The red half-dots indicate chosen edges for the polygons, the red dotted arrows indicate the anticlockwise orientation around the vertices. Vertex $v$ is used in Figure 5.

polygons with identified edges and vertices as prescribed by the embedding. Technically this is a PLCW decomposition [62], see also [80] for a discussion in the present context.

At gluing boundaries we require that the edges and vertices of $T_\Sigma$ are given by the images of those on the standard (half)circle with $n$ edges (see Figure 3) under the corresponding parametrising map. This will ensure a consistent gluing procedure below. The number of edges covering a gluing boundary may vary from boundary to boundary.[3]

Let $V$ denote the set of vertices, $E$ the set of edges and $F$ the set of faces, i.e. the polygons. Let $E^f \subset E$ denote the edges that lie on the free boundary $\partial^f \Sigma$ of $\Sigma$.

A *marking of a polygonal decomposition* $T_\Sigma$ consists of

- an orientation $o$ of each each edge in $E \setminus E^f$,

- a choice of edge $m$ for each polygon (before gluing the edges of the polygons together to give the decomposition of $\Sigma$),

- for each edge $e \in E \setminus E^f$ an *edge index* $s_e \in \mathbb{Z}_r$.

For edges on the gluing boundary we require that their orientation agrees with those of the edges on the standard (half)circle under the parametrising maps. An example of a marking is shown in Figure 4. We write

$$T_\Sigma(o, m, s), \tag{24}$$

for a marked polygonal decomposition of $\Sigma$. To see why in the second bullet point we demand to choose the edge of a polygon before gluing, consider a torus decomposed into one square. One has to select one of the four edges of the square, but after gluing there are only two edges (and one vertex).

For a given decomposition $T_\Sigma$ with orientations $o$ and choice of marked edges $m$ to give rise to an $r$-spin structure on $\Sigma$, the edge indices have to satisfy a consistency condition. To describe this condition, we will need some notation.

Firstly, we split each edge $e \in E \setminus E^f$ into two half-edges. For a vertex $v$, we write $H_v$ for the set of half-edges starting or ending at $v$, and $D_v \subset H_v$ for the subset of half edges which come from the marked edge of the polygon immediately anticlockwise of that half-edge.[4] We denote the induced edge index of a half-edge $h$ as $s_h$. This is illustrated in Figure 5.

---

[3]In the combinatorial model in [80,85] only one edge is allowed on each gluing boundary. For the treatment of bordisms with defects it is helpful to allow several edges on a given gluing boundary, and we use this slightly more flexible combinatorial model. The relation to [80,85] is explained in Appendix A.1.3.

[4]To be precise, we consider the polygon before gluing the edges into the decomposition of $\Sigma$. In the example of the torus above, before gluing, the polygon is a square and there is only one half-edge of the marked edge for which the square lies anticlockwise of the half-edge and only this half-edge contributes to $D_v$.

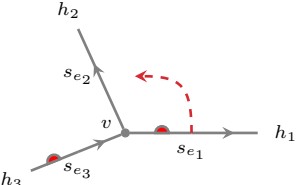

Figure 5: Illustration of the half-edges around the vertex $v$ showing which contribute to $D_v$. Here one has $H_v = \{h_1, h_2, h_3\}$, $|H_v| = 3$, $D_v = \{h_1\}$ $|D_v| = 1$, $\widehat{s}_{h_1} = s_{e_1}$, $\widehat{s}_{h_2} = s_{e_2}$, $\widehat{s}_{h_3} = -1 - s_{e_3}$. The consistency condition is $s_{h_1} + s_{h_2} - 1 - s_{h_3} \equiv 1 - 3 + 1 \pmod{r}$.

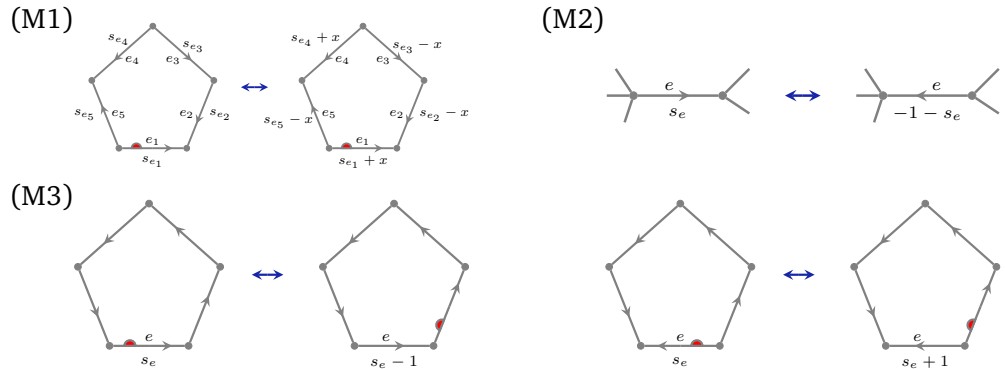

Figure 6: Moves (M1)–(M3) changing the marking of a fixed polygonal decomposition.

Given a half edge $h \in H_v$, we write

$$\widehat{s_h} = \begin{cases} s_h, & \text{if the half edge is leaving } v, \\ -1 - s_h, & \text{if the half edge is entering } v. \end{cases} \tag{25}$$

We call a vertex *constrained* if it is an interior vertex, or if it lies on a gluing boundary and is not the image of $v_0$ in case of a closed gluing boundary (cf. Figure 3), and not the image of $v_0$ or of $v_n$ in case of an open gluing boundary. Let $V^c \subset V$ be the subset of constrained vertices. In other words, the complement $V \setminus V^c$ consists of vertices on the free boundary and of the image of $v_0$ on each closed gluing boundary.

We call a marking $(o, m, s)$ on $T_\Sigma$ *admissible* if for every vertex $v \in V^c$ we have[5]

$$\sum_{h \in H_v} \widehat{s_h} \equiv |D_v| - |H_v| + 1 \pmod{r}. \tag{26}$$

There is no condition for unconstrained vertices, i.e. vertices that lie in $V \setminus V^c$. However, we will later prescribe a fixed spin structure near closed gluing boundaries, which leads to a condition similar to (26) for each $v \in V \setminus V^c$ on a closed gluing boundary, see (30) below.

Not all admissible markings $(o, m, s)$ on $T_\Sigma$ describe distinct $r$-spin structures on $\Sigma$. The redundancy is most easily described by fixing $o, m$ and by considering the following move on edge indices (Figure 6):

---

[5] To obtain an $r$-spin structure on $\Sigma$ it is enough to impose the admissibility condition on inner vertices only. Imposing the condition also for the boundary vertices in $V^c$, as we do here, simplifies the gluing prescription below. In the case of a closed gluing boundary it will also result in the simple expression (28) for the holonomy along a curve parallel to the boundary.

(M4)                   (M5)

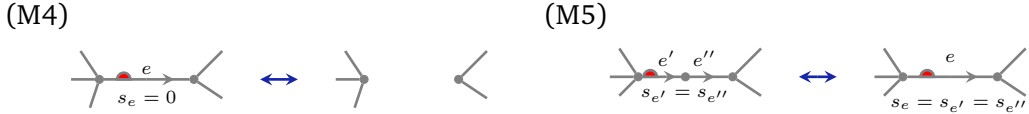

Figure 7: Moves (M4) and (M5) changing a marked polygonal decomposition.

(M1) For a given polygon $p$ and $x \in \mathbb{Z}_r$, shift the edge index $s_e$ of an edge $e \in E \setminus E^{\mathrm{f}}$ on the boundary of $p$ by $+x$ if it is oriented clockwise with respect to the orientation of $p$ and by $-x$ if it is oriented anticlockwise. If $p$ lies on both sides of $e$ (i.e. $e$ arises by gluing two edges of $p$ together), $s_e$ is not changed.

Geometrically, this amounts to acting with $x \in \mathbb{Z}_r$ on the fibres of the spin structure $P$ seen as a $\mathbb{Z}_r$-principal bundle over the frame bundle $F_\Sigma$, both restricted to the polygon $p$.

The move (M1) preserves admissibility and we call two admissible assignments $s, s'$ of edge signs *equivalent* if they are related by a sequence of (M1)-moves.

**Theorem 2.2.** *Let $\Sigma$ be an open-closed bordism, $T_\Sigma$ a polygonal decomposition, and $o, m$ choices of edge orientations and chosen edges. Then equivalence classes of admissible edge indices are in bijection with isomorphism classes of $r$-spin structures on $\Sigma$.*

We explain in Appendix A.1.1 how this follows from similar results in [74, 80, 85].

Given an $r$-spin structure $P$ on $\Sigma$, by the above theorem there exists an admissible marking on $T_\Sigma$ which encodes $P$. We write

$$T_\Sigma(P), \tag{27}$$

for such a choice of marking.

The move (M1) does not change the isomorphism class of the $r$-spin structure defined by the combinatorial data. There are four more moves which change one or more of $T_\Sigma$, $o$, $m$ and $s$ without changing the $r$-spin structure [80, Thm. 2,13, Prop. 2.18] and [85, Prop. 3.1.9, Prop. 3.1.11]:

(M2) Change the edge orientation for an inner edge $e$ and change the edge index $s_e$ to $-1 - s_e$ (Figure 6).

(M3) Shift the marked edge of a face anticlockwise. Change the edge index by $\pm 1$ depending on the edge orientation, unless the marked edge is on the free boundary in which case it does not carry an orientation or an edge index (Figure 6).

(M4) Remove an inner edge $e$ which is adjacent to two distinct polygons. We require that $e$ is the chosen edge of exactly one of the two polygons, and that $e$ has orientation and edge index as in Figure 7.

(M5) Remove a 2-valent vertex whose two adjacent edges are distinct. If the vertex is on the free boundary, there are no additional conditions. If the the vertex is an inner vertex, the orientations, chosen edges and edge indices have to be as shown in Figure 7. On a gluing boundary, no vertices can be removed or added.

For the moves (M3)–(M5) it is understood that the inverse moves are included as well. The move (M2) is self-inverse. Note that (M1) is redundant because it can be obtained by by iterating (M3). We include it anyway because it is the only move that operates solely on the edge indices and thereby simplifies the formulation of Theorem 2.2.

Let $\Sigma, \Sigma'$ be surfaces, and let $T_{\Sigma'}(o', m', s')$ be a polygonal decomposition of $\Sigma'$ with admissible marking. Given a diffeomorphism $f : \Sigma \longrightarrow \Sigma'$, we can pull back the decomposition $T_{\Sigma'}$ and the marking $(o', m', s')$ along $f$. We write $f^* T_{\Sigma'}(o', m', s')$ for this pull-back. Let now $P'$ be the $r$-spin structure on $\Sigma'$ described by $T_{\Sigma'}(o', m', s')$. By construction, the pull-back marked decomposition $f^* T_{\Sigma'}(o', m', s')$ describes the pull-back $r$-spin structure $f^* P'$ from (20). This can be seen by noting that holonomies do not change under pullback. In the case that $\Sigma = \Sigma'$, we arrive at the following statement:

**Proposition 2.3.** *Let $P$ be an $r$-spin structure on $\Sigma$, $T_\Sigma$ a polygonal decomposition, and $f : \Sigma \longrightarrow \Sigma$ be a diffeomorphism. Then the combinatorial presentation $T_\Sigma(f^* P)$ of the pull-back $r$-spin structure $f^* P$ on the polygonal decomposition $P_\Sigma$ of $\Sigma$ can be obtained from the pull-back decomposition $f^* T_\Sigma(P)$ by a sequence of moves (M1)-(M5).*

**Combinatorial model and holonomies**

We described in Section 2.1 how an $r$-spin structure is determined by its holonomies up to isomorphism. Since a marked polygonal decomposition $T_\Sigma(o, m, s)$ determines an $r$-spin structure, it is possible to compute the holonomies in terms of the combinatorial data. The procedure is not complicated but slightly technical, and is described in detail in [80, Sec. 2.4] and reviewed in Appendix A.1.2. Two important instances are:

- Consider a small circular path running anticlockwise around an interior vertex $v$. In the notation used in (26), the holonomy is given by

$$|H_v| - |D_v| + \sum_{h \in H_v} \widehat{s_h} \in \mathbb{Z}_r \,. \tag{28}$$

  The condition that this holonomy is 1 amounts to the condition for the spin structure to extend from a punctured disc to the whole disc. Note that this is also the admissibility condition (26), explaining its geometric meaning.

- Consider a circular path running parallel to a closed gluing boundary oriented in the same way as the image of the unit circle under the parametrising map. Let $v$ be the unique unconstrained vertex on that gluing boundary (i.e. the image of $v_0$ in Figure 3). It is shown in Appendix A.1.2 that the holonomy is given by

$$\delta_v \Big( |H_v| - |D_v| - 1 + \sum_{h \in H_v} \widehat{s_h} \Big), \tag{29}$$

  where $\delta_v = +1$ if $v \in \partial^c_{in} \Sigma$ and $\delta_v = -1$ if $v \in \partial^c_{out} \Sigma$. Recall from (23) the definition of the type $y \in \mathbb{Z}_r$ of a closed gluing boundary. By (22), the holonomy is related to the type via

$$(\text{hol. in (29)}) \; = \; 1 - y \,, \tag{30}$$

  which can also be thought of as a constraint on the edge labels if the type is fixed.

**Examples of $r$-spin surfaces**

Let us consider two examples of polygonal decompositions, where in both cases we use a single polygon.

The first example is a 3-holed sphere with two in-going closed gluing boundaries of types $x, y \in \mathbb{Z}_r$ and an out-going closed gluing boundary of type $z$. The decomposition is shown in

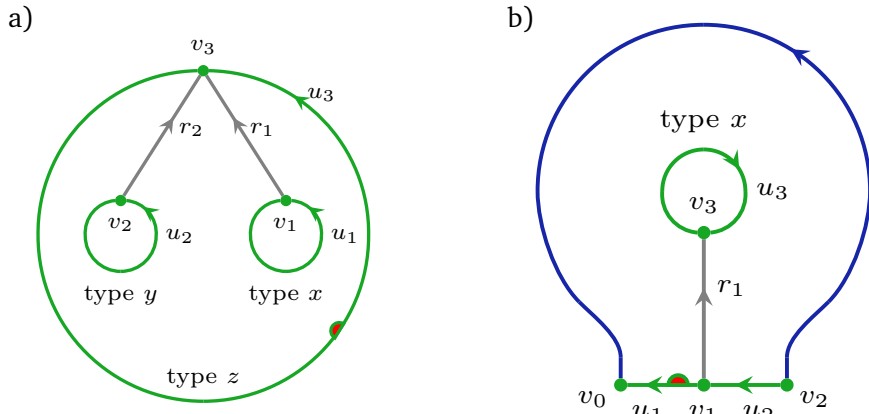

Figure 8: A marked polygonal decomposition for a) an $r$-spin sphere with two incoming and one outgoing closed gluing boundary, and b) an $r$-spin disc with one ingoing open and one outgoing closed gluing boundary, as well as a free boundary interval. In this example, the open gluing boundary is covered by two edges.

Figure 8 a). The admissibility conditions (30) at the three vertices are:

$$v_1: \underbrace{|H_{v_1}|}_{=3} - \underbrace{|D_{v_1}|}_{=0} - 1 + \underbrace{\sum_{h \in H_{v_1}} \widehat{s_h}}_{=r_1-1} = 1 - x \qquad \Longleftrightarrow \qquad r_1 = -x,$$

$$v_2: 3 - 0 - 1 + r_2 - 1 = 1 - y \qquad \Longleftrightarrow \qquad r_2 = -y,$$

$$v_3: -(4 - 1 - 1 + (-1 - r_1) + (-1 - r_2) - 1) = 1 - z \qquad \Longleftrightarrow \qquad z = x + y. \tag{31}$$

Note that there exists an $r$-spin structure if and only if $z = x + y$. In this case, by Theorem 2.2 the isomorphism classes of $r$-spin structures are parametrised by $u_1, u_2, u_3 \in \mathbb{Z}_r$ up to the move (M1). E.g. we can use (M1) to set $u_3 = 0$, so that the isomorphism classes of $r$-spin structures are parametrised by $u_1, u_2 \in \mathbb{Z}_r$.

The second example is an annulus with one in-going open gluing boundary and one outgoing closed gluing boundary of type $x$, see Figure 8 b). The constrained vertices are $v_1$ and $v_3$, and the conditions are:

$$\text{(30) at } v_3: -(3 - 0 - 1 + (-1 - r_1) - 1) = 1 - x \qquad \Longleftrightarrow \qquad r_1 = 1 - x,$$

$$\text{(26) at } v_1: u_1 + (-1 - u_2) + r_1 = 0 - 3 + 1 \qquad \Longleftrightarrow \qquad u_1 = u_2 + x - 2. \tag{32}$$

There thus exist $r$-spin structures for every $x \in \mathbb{Z}_r$. Using (M1) we can set $u_3 = 0$, so that the isomorphism classes of $r$-spin structures are parametrised by $u_1 \in \mathbb{Z}_r$.

**Gluing of marked polygonal decompositions**

Consider a (not necessarily connected) open-closed bordism $\Sigma$ with admissible marked polygonal decomposition $T_\Sigma(o, m, s)$. Let $\Sigma'$ be the open-closed bordism obtained by either identifying an out-going and an in-going open gluing boundary, or an out-going and an in-going closed gluing boundary of the same type in the sense of (30). The identification is defined via the boundary parametrisation maps by the standard (half) circle (recall Figure 3), so that in particular the in- and outgoing gluing boundaries must contain the same number of edges and vertices.

On $\Sigma'$ we obtain the induced admissible polygonal decomposition as follows. Denote the edges on the in- and outgoing boundary that are glued together by $e_i^{\text{in}}$ and $e_i^{\text{out}}$, $i = 1, \ldots, n$.

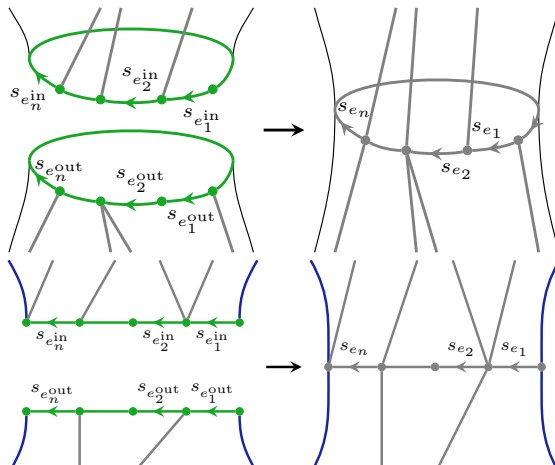

Figure 9: Gluing cell decompositions along closed and open boundaries. The new edge labels are $s_{e_i} := s_{e_i^{\text{in}}} + s_{e_i^{\text{out}}}$.

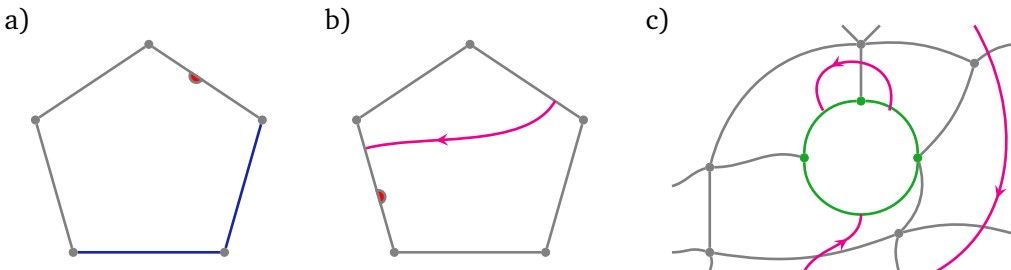

Figure 10: a) Plaquette with some edges on the free boundary of $\Sigma$. The edge following the last edge on the free boundary in anticlockwise direction is the marked edge, the orientation of the edges can be arbitrary (edges on the free boundary carry no orientation). b) Plaquette with defect arc. The defect has to leave the plaquette at the marked edge, the orientation of the edges can again be arbitrary. c) Example of an allowed polygonal decomposition near a closed gluing boundary in the presence of defects. The marking is not shown.

By construction, the orientations of $e_i^{\text{in}}$ and $e_i^{\text{out}}$ agree when compared via the parametrising maps, and we keep this orientation for the glued edges $e_i$ (see Figure 9). The edge index of $e_i$ is given by

$$s_{e_i} := s_{e_i^{\text{in}}} + s_{e_i^{\text{out}}}\,, \tag{33}$$

and the rest of the marked decomposition is not affected by the gluing. Denote the resulting admissible marked decomposition by $T_{\Sigma'}(o', m', s')$.

It is shown in Appendix A.1.3 that (33) does indeed produce an admissible marking for $T_{\Sigma'}$ and that the $r$-spin structure on $\Sigma'$ defined by $T_{\Sigma'}(o', m', s')$ agrees with the one obtained by gluing boundaries of open-closed $r$-spin bordisms via their boundary parametrisation maps as described in Section 2.1.

## 2.3 Open-closed spin bordisms with defects

Let $\Sigma$ be an open-closed bordism. A *defect line* on $\Sigma$ is an embedded loop or an embedded arc, whose endpoints lie on open or closed gluing boundaries, but not on the free boundary. Ac-

cordingly, an *open-closed bordisms with defects* is an open-closed bordism with a finite collection of pairwise disjoint line defects. To avoid corners when gluing, we demand that arcs end on the gluing boundary orthogonally when mapped to the complex plane via the parametrising map.

If $\Sigma$ carries in addition an $r$-spin structure, then we obtain an *open-closed r-spin bordisms with defects*. Note that we do not consider separate $r$-spin structures on individual patches obtained by removing the line defects from $\Sigma$, but rather an $r$-spin structure on the entire surface $\Sigma$. This can be understood as a restriction on the type of defects we consider in this paper.

If the $r$-spin structure is encoded by an admissible marked polygonal decomposition $T_\Sigma(o, m, s)$, this means that the line defects do not affect the admissibility conditions at the vertices, and that the polygonal decomposition can be deformed freely across the line defects. Nonetheless, the description of CFT correlators in Section 6 becomes simpler if we impose the following constraints on $T_\Sigma(o, m, s)$ (see Figure 10):

- No vertex of $T_\Sigma$ lies on a defect line. Defect lines intersect the edges of $T_\Sigma$ transversally.

- A polygon can either intersect a defect line, or have edges on the free boundary, or none of these, but not both.

- If a polygon intersects the free boundary, it does so in a single vertex or in a sequence of consecutive edges, and not all of its edges lie on the free boundary. The edge following the last edge on the free boundary in anticlockwise direction is the marked edge of the polygon (Figure 10 a).

- If a polygon intersects a defect line, it does so in a single arc. The two endpoints of the arc lie on distinct edges of the polygon (before identification), and the defect arc leaves the polygon at its marked edge (Figure 10 b).

- A closed gluing boundary that does not contain an endpoint of a defect line is covered by a single edge. A closed gluing boundary that does contain endpoints of defect lines has as many edges as endpoints.

- An open gluing boundary that does not contain an endpoint of a defect line is covered by two edges. An open gluing boundary that does contain endpoints of defect lines has two edges more than it has defect endpoints, with no defect endpoint lying on the edges touching the free boundary.

One can account for the restrictions on polygons touching the free boundary in terms of those for polygons intersecting defect lines by thinking of the free boundary as accompanied by a parallel line defect.

Subject to these requirements, the moves (M1)–(M5) can be applied in the same way, and one can convince oneself that they relate any two polygonal decompositions satisfying the requirements. Some examples are shown in Figure 11.

The gluing procedure is the same as in the case without defects. We illustrate this in Figure 12.

## 2.4 World sheets and bordisms

When describing CFT correlators in terms of conformal blocks below, we will use world sheets, i.e. surfaces with marked points for field insertions, rather than surfaces with gluing boundaries carrying open or closed states.

In more detail, a *world sheet $\Sigma$ with boundaries and defects*, or just *world sheet* for short, consists of

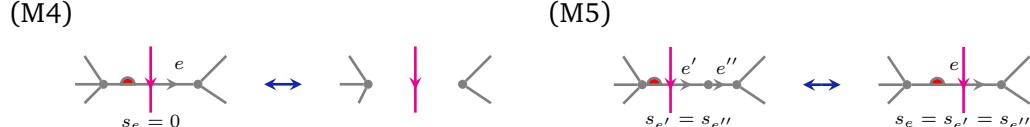

Figure 11: Examples of moves (M4) and (M5) in the presence of defects. The marking changes in the same way as without defects (cf. Figure 7).

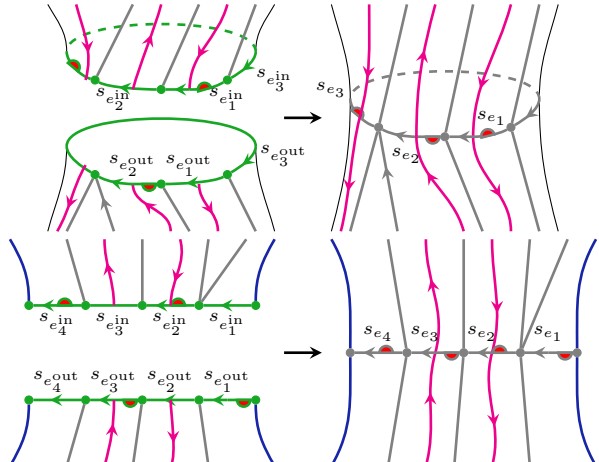

Figure 12: Gluing cell decompositions along closed and open boundary components with defects. In the closed case we have three defect line end points on the gluing boundary and three corresponding edges. In the open case two defect line end points lie on the gluing boundary and we have two corresponding edges in addition to the two edges for the endpoints of the free boundary. The new edge labels are $s_{e_i} := s_{e_i^{\mathrm{in}}} + s_{e_i^{\mathrm{out}}}$.

- a surface with possibly non-empty boundary

- an ordered, finite set of marked points, possibly on the boundary,

- a tangent vector at each marked point,

- a partition of the marked points into in-going and out-going marked points,

- a finite set of embedded oriented loops in the interior of $\Sigma$, and a finite set of embedded oriented arcs which intersect the boundary at most at their endpoints.

For the tangent vectors and for the embedded arcs and loops we require the following conditions:

- We take the boundary of the surface to be oriented by the inward pointing normal, i.e. as the real axis on the upper half plane. For a marked point on the boundary, the tangent vector has to be parallel to the boundary and to point in the direction given by the orientation of the boundary.

- The embedded loops are mutually non-intersecting and are disjoint from the marked points. The embedded arcs have endpoints at marked points, and do not intersect loops. They can intersect other arcs only at their endpoints.

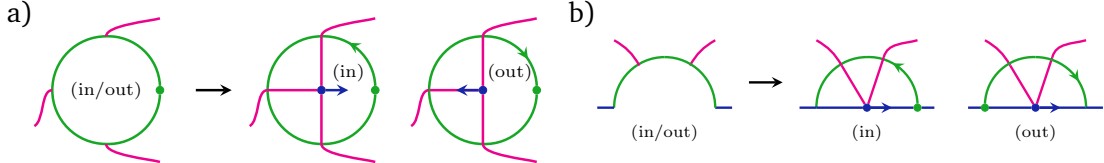

Figure 13: Turning an open-closed bordism $\Sigma_b$ into a world sheet $\Sigma$ by gluing in (half)discs with a marked point at zero. The green dot on the (half)circle gives the image of $1 \in \mathbb{C}$ under the parametrising map. In our convention, the tangent vector points towards 1 for ingoing gluing boundaries and away from 1 for outgoing ones.

We will refer to a marked point with the corresponding tangent vector as an *(ingoing or outgoing) insertion point*. The embedded loops and arcs are the *defect lines*.

Accordingly, a *diffeomorphism of world sheets* is an orientation preserving diffeomorphism of the underlying surfaces which preserves the in- and outgoing marked points, the tangent vectors at the marked points, and the embedded loops and arcs together with their orientation.

One can turn an open-closed bordism $\Sigma_b$ into a world sheet $\Sigma$ by gluing in unit discs and unit half-discs via the parametrising maps. The unit disc has zero as an insertion point with tangent vector point along the positive real axis (and along the negative real axis in case of an outgoing closed gluing boundary only). For out-going boundaries one needs to compose the parametrising map with $z \longmapsto 1/z$ (closed case) or $z \longmapsto -1/z$ (open case) first. The insertion points keep the in/outgoing label of the gluing boundary they replace. If there are defect lines ending on a gluing boundary of the bordism, these are extended as straight radial arcs to the marked point at zero. This is illustrated in Figure 13.

The same procedure works in the presence of $r$-spin structures. Starting from an $r$-spin bordism $\mathbb{\Sigma}_b$ one obtains an *$r$-spin world sheet* $\mathbb{\Sigma}$. Here it is understood that for closed gluing boundaries the $r$-spin structure in general only extends to the punctured disc, i.e. the disc one glues in minus the marked point. If the boundary is ingoing of type $y$, the $r$-spin structure on the punctured disc is the restriction of $\mathbb{C}^y$, and if it is outgoing that of $\mathbb{C}^{2-y}$. Thus the $r$-spin structure extends to the whole disc iff the closed gluing boundary is of type $y = 0$ (ingoing) or $y = 2$ (outgoing).

The difference between in- and outgoing $r$-spin structure on the punctured disc arises from the precomposition with $z \longmapsto 1/z$. This is most easily understood in terms of holonomy: by (22), a anticlockwise unit circle $S^1$ on $\mathbb{C}^y$ produces holonomy $\zeta(S^1) = 1 - y$; under the map $z \longmapsto 1/z$, the curve changes direction, so that the holonomy becomes $y - 1 = 1 - (2 - y)$.

## 3 Reshetikhin-Turaev TFT and conformal blocks

The holomorphic fields of a CFT form a vertex operator algebra, as do the anti-holomorphic fields. We will consider CFTs where the holomorphic and the anti-holomorphic fields both contain a given VOA $\mathcal{V}$, and where this VOA is rational in the sense that its category of representations $\mathcal{C} = \mathrm{Rep}(\mathcal{V})$ is a modular fusion category. Via the Reshetikhin-Turaev construction, $\mathcal{C}$ defines a 3d TFT which (conjecturally) encodes the spaces of conformal blocks of $\mathcal{V}$ as well as their monodromy and gluing properties.

In this section we briefly review these connections to the extend needed in the following.

### 3.1 Modular categories

A modular fusion category $\mathcal{C}$ is finitely semisimple, and it is equipped with a tensor product, duals, a non-degenerate braiding, and a ribbon twist. In particular, $\mathcal{C}$ is a ribbon category, and we will use the standard graphical calculus to represent morphisms in $\mathcal{C}$. For the conventions and definitions stated below, we do in fact not need the braiding and the ribbon twist, but we prefer to stay in the framework of ribbon categories to avoid changing the setting too often. For more details we refer to [33].

An object $U \in \mathcal{C}$ is said to have a *left dual* if there is an object $U^*$ together with evaluation and coevaluation morphisms $\mathrm{ev}_U \colon U^* \otimes U \longrightarrow \mathbf{1}$ and $\mathrm{coev}_U \colon \mathbf{1} \longrightarrow U \otimes U^*$ satisfying the zigzag identities, see [33, Sec. 2.10]. For a *right dual object* $^*U$, the order of the tensor product in the (co)evaluation morphisms is reversed. We use the following graphical notation:

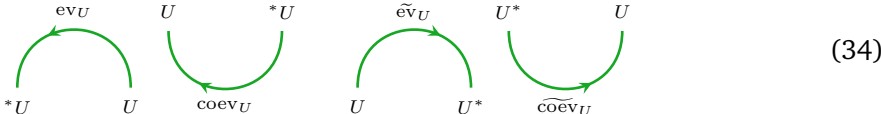

$$ \tag{34} $$

In particular, we read our diagrams from bottom to top.

In a ribbon category, we can always take $^*U = U^*$, and below we will write $U^*$ for both duals.

We write $\mathcal{C}(U,V)$ for the space of morphisms from $U$ to $V$ in $\mathcal{C}$, as a shorthand for $\mathrm{Hom}_{\mathcal{C}}(U,V)$. Out of evaluation and coevaluation one can form a morphism in $\mathcal{C}(\mathbf{1}, \mathbf{1})$, which we identify with $\mathbb{C}$ via $\lambda \longmapsto \lambda \, \mathrm{id}_{\mathbf{1}} \in \mathcal{C}(\mathbf{1}, \mathbf{1})$. The resulting number is the *quantum dimension*, or *dimension* for short, of $U$:

$$ \dim(U) := \qquad = \qquad \tag{35} $$

The fact that the two expressions for $\dim(U)$ coincide is a property of ribbon categories (and more generally of spherical categories). For ribbon categories one can see this by thinking of both sides as ribbons in $\mathbb{R}^3$ (drawn here flat in the paper plane) and deforming one into the other.

The collection of invariants of the Hopf link coloured by (representatives of isomorphism classes of) simple objects $U, V \in \mathcal{C}$ is called the s-matrix,

$$ \mathsf{s}_{U,V} = \mathrm{Tr}_{U \otimes V}(\sigma_{V,U} \circ \sigma_{U,V}) = \qquad = \qquad \tag{36} $$

where $\sigma_{U,V} \colon U \otimes V \longrightarrow V \otimes U$ denotes the braiding of $\mathcal{C}$. For a modular fusion category, the matrix $\mathsf{s}$ with entries $\mathsf{s}_{U,V}$ is non-degenerate.

### 3.2 Conformal blocks and 3d TFT

Let $\mathcal{V}$ be a rational VOA such that $\mathcal{C} = \mathrm{Rep}(\mathcal{V})$ is a a modular fusion category (see [58] for the precise conditions and the proof). In this section we will review a further relation between $\mathcal{V}$ and $\mathcal{C}$, namely that the spaces of conformal blocks obtained from $\mathcal{V}$ agree with the state spaces of the Reshetikhin-Turaev (RT) TFT defined by $\mathcal{C}$. Standard references for this section

are [10, 35, 68, 71, 87]. A proof of factorisation for spaces of conformal blocks has recently been given in [27].

The main aim of this and the next section is to explain why the TFT considerations in the later chapters are indeed the ones relevant to describe CFT correlators and their properties. The construction of parity CFTs and spin CFTs presented in Sections 5 and 6 will be given in the TFT setting and will not make direct use of VOAs.

The RT TFT $\mathcal{Z}_{\mathcal{C}}^{\mathrm{RT}}$ is a symmetric monoidal functor from the category $\mathcal{B}ord_3(\mathcal{C})$ of $\mathcal{C}$-extended surfaces and 3-bordisms with embedded $\mathcal{C}$-coloured ribbon graphs to the category of vector spaces $\mathcal{V}ect$. A $\mathcal{C}$-*extended surface* is a closed surface with a finite ordered set of marked points, where each marked point $p$ consists of a tuple $\boldsymbol{p} = (p, U, v, \delta)$, where $p \in \Sigma$, $U \in \mathcal{C}$, $v$ is a tangent vector at $p$, $\delta \in \{\pm\}$, and which is equipped with a Lagrangian subspace $\lambda \in H_1(\Sigma, \mathbb{R})$. The Lagrangian subspace, as well as an integer assigned to each bordism, is required to compensate a gluing anomaly, we refer to [87, Sec. IV.9] for details. We will mostly not mention this data explicitly in the following.

There is a direct relation between the Hom-spaces of $\mathcal{C}$ and the state spaces of the RT TFT, i.e. the vector spaces that $\mathcal{Z}_{\mathcal{C}}^{\mathrm{RT}}$ assigns to $\mathcal{C}$-extended surfaces. To describe it, let $\mathcal{I}$ be the set labelling isomorphism classes of simple objects in $\mathcal{C}$, and let

$$S_i, \quad i \in \mathcal{I}, \tag{37}$$

be a choice of representatives. We write $L = \bigoplus_{i \in \mathcal{I}} S_i \otimes S_i^*$ for the coend in $\mathcal{C}$. Let $\Sigma$ be a $\mathcal{C}$-extended surface of genus $g$ with marked points $\boldsymbol{p}_i = (p_i, U_i, v_i, \delta_i = -)$, $i = 1, \ldots, n$, and consider the handle body

$$H_f = \tag{38}$$

where $f \in \mathcal{C}(U_1 \otimes \cdots \otimes U_n, L^{\otimes g})$, $P = \bigoplus_{i \in \mathcal{I}} S_i$ and $\iota \colon L \longrightarrow P \otimes P^*$ is the diagonal embedding. We take $H_f$ to be a bordism $\emptyset \longrightarrow \Sigma$ in $\mathcal{B}ord_3(\mathcal{C})$. Applying the TFT functor gives a linear map

$$\mathbb{C} \xrightarrow{\mathcal{Z}_{\mathcal{C}}^{\mathrm{RT}}(H_f)} \mathcal{Z}_{\mathcal{C}}^{\mathrm{RT}}(\Sigma). \tag{39}$$

Evaluating at $1 \in \mathbb{C}$ gives an element in the state space $\mathcal{Z}_{\mathcal{C}}^{\mathrm{RT}}(\Sigma)$. By construction of the RT TFT, the map

$$\mathcal{C}(U_1 \otimes \cdots \otimes U_n, L^{\otimes g}) \longrightarrow \mathcal{Z}_{\mathcal{C}}^{\mathrm{RT}}(\Sigma), \qquad f \longmapsto \mathcal{Z}_{\mathcal{C}}^{\mathrm{RT}}(H_f)(1), \tag{40}$$

is an isomorphism of vector spaces. Changing $\delta_i = -$ to $\delta_i = +$ amounts to replacing $U_i$ by the dual $U_i^*$.

Next we outline the relation between state spaces of the RT TFT and spaces of conformal blocks. Let $\Sigma^c$ be a $\mathcal{C}$-extended surface equipped with a complex structure, and with a holomorphic local coordinate $\varphi$ for each insertion $p$ such that $\varphi(0) = p$ and $\varphi'(0) = v$. The VOA $\mathcal{V}$ assigns to $\Sigma^c$ the *space of conformal blocks* $\beta_{\mathcal{V}}(\Sigma^c)$. This is the subspace

$$\beta_{\mathcal{V}}(\Sigma^c) \subset \mathrm{Hom}_{\mathbb{C}}(U_1 \otimes_{\mathbb{C}} \cdots \otimes_{\mathbb{C}} U_n, \mathbb{C}), \tag{41}$$

of the linear maps $U_1 \otimes_{\mathbb{C}} \cdots \otimes_{\mathbb{C}} U_n \longrightarrow \mathbb{C}$ whose elements satisfy the Ward identities determined by $\mathcal{V}$.

The subspace $\beta_\mathcal{V}(\Sigma^c)$ depends on the moduli of $\Sigma^c$, that is on the insertion points, local coordinates and complex structure, and one can combine the $\beta_\mathcal{V}(\Sigma^c)$ into a vector bundle over the corresponding moduli space. This bundle is equipped with a projectively flat connection, so that a path in moduli space can be lifted to a path in the bundle of conformal blocks, and up to an overall scalar, the endpoint of the lift depends only on the homotopy class of $\gamma$.

By forgetting the complex structure and the local coordinates of $\Sigma^c$ one recovers the underlying $\mathcal{C}$-extended surface $\Sigma$. The spaces of conformal blocks for $\mathcal{V}$ conjecturally[6] agree with the state spaces of the RT TFT for $\mathcal{C} = \mathrm{Rep}(\mathcal{V})$ in the sense that there is a linear isomorphism

$$F(\Sigma^c) : \beta_\mathcal{V}(\Sigma^c) \xrightarrow{\sim} \mathcal{Z}_\mathcal{C}^{\mathrm{RT}}(\Sigma) \tag{42}$$

(more precisely, one has an equivalence of modular functors, but we will not go into this). One way to make the above isomorphism explicit is to first use that conformal blocks on a sphere with insertions at points 1, 0, and $\infty$ labelled by $(U, -)$, $(V, -)$ and $(W, +)$, respectively, are intertwining operators of type $\binom{W}{U\,V}$ for the $\mathcal{V}$-modules $U, V, W$. By construction, these define the tensor product on $\mathcal{C}$, i.e. are canonically identified with $\mathcal{C}(U \otimes V, W)$. Then one can use factorisation of conformal blocks to reduce more complicated surfaces to this case.

## 3.3 Compatibility with transport

Here we briefly recall the relation between paths in the fine moduli space of complex structures and families of complex curves with base given by an interval. We then consider families of $\mathcal{C}$-extended surfaces with complex structure to formulate the compatibility of the isomorphism (42) with transport along paths.

Let $\Sigma_0$ be a surface without complex structure and without marked points. The fine moduli space of complex structures, or Teichmüller space, is defined as

$$\mathcal{T}(\Sigma_0) := \{(\Sigma^c, \phi : \Sigma_0 \longrightarrow \Sigma^c)\} / \sim, \tag{43}$$

where $\Sigma^c$ is a surface with complex structure, $\phi$ is an orientation preserving diffeomorphism and the equivalence relation is defined as follows: $(\Sigma^c, \phi) \sim (\Sigma'^c, \phi')$ if there is a biholomorphic map $\psi : \Sigma^c \longrightarrow \Sigma'^c$ such that $\psi \circ \phi$ and $\phi'$ are homotopic.

Let $\gamma : [0, 1] \longrightarrow \mathcal{T}(\Sigma_0)$ be a path in the fine moduli space from $\gamma(0) = \Sigma^c$ to $\gamma(1) = \widetilde{\Sigma}^c$. One can turn $\gamma$ into a bordisms $E_\gamma : \Sigma \longrightarrow \widetilde{\Sigma}$ as follows. Consider the family of complex curves $E_\gamma \longrightarrow [0, 1]$ obtained by pulling back the universal curve over $\mathcal{T}(\Sigma_0)$ (where over each point is the corresponding complex curve) along $\gamma$. Thus the fibre at $t \in [0, 1]$ is $\gamma(t)$. The total space $E_\gamma$ can also be thought of as a bordism between the fibres over 0 and 1. Note that every family of complex surfaces over $[0, 1]$ defines a path in $\mathcal{T}(\Sigma_0)$, so that pulling back along the path reproduces the original family [14].

In the following we will use the notions path and family interchangeably, and we will denote the family obtained from a path $\gamma$ by $E_\gamma$.

Let $E_\gamma$ be a family of $\mathcal{C}$-extended surfaces with complex structure and with local coordinates around the marked points. Write $\Sigma^c := \gamma(0)$ and $\widetilde{\Sigma}^c := \gamma(1)$, so that when forgetting the complex structure on the fibres, $E_\gamma$ defines a bordism $\Sigma \longrightarrow \widetilde{\Sigma}$. The isomorphism (42) is compatible with transport along $E_\gamma$ in the following sense: parallel transport via the projectively flat connection on the bundle of conformal blocks gives a linear isomorphism $T_\gamma : \beta_\mathcal{V}(\Sigma^c) \longrightarrow \beta_\mathcal{V}(\widetilde{\Sigma}^c)$. On the other hand, the bordisms $E_\gamma : \Sigma \longrightarrow \widetilde{\Sigma}$ defines a linear isomorphism $\mathcal{Z}_\mathcal{C}^{\mathrm{RT}}(E_\gamma) : \mathcal{Z}_\mathcal{C}^{\mathrm{RT}}(\Sigma) \longrightarrow \mathcal{Z}_\mathcal{C}^{\mathrm{RT}}(\widetilde{\Sigma})$. The compatibility relation (again conjectural) is

$$F(\widetilde{\Sigma}^c) \circ T_\gamma \ \propto \ \mathcal{Z}_\mathcal{C}^{\mathrm{RT}}(E_\gamma) \circ F(\Sigma^c), \tag{44}$$

---

[6]For certain examples this is known, see e.g. [4], and the results in [27] might allow to show this in general (for rational $\mathcal{V}$).

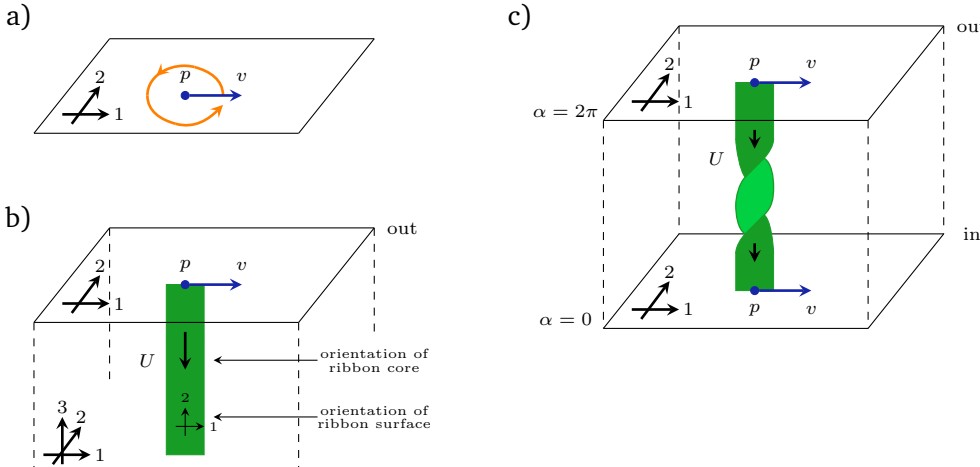

Figure 14: a) Closed path $\gamma$ in the moduli space given by rotating the local coordinate once by $2\pi$. The fibres of the corresponding family $E_\gamma$ differ only in the angle $\alpha$ of the tangent vector at $v$. b) The orientation conventions for an outgoing boundary component of a bordisms, and for how the tangent vector $v$ and the orientation of core and surface of a ribbon attached to the marked point $(p, U, v, -)$ are related. c) The bordism $E_\gamma \colon \Sigma(0) \longrightarrow \Sigma(2\pi) = \Sigma(0)$ defined by the path $\gamma$.

where the two sides may differ by a multiplicative constant.

Finally, we need to say how both sides of (42) behave under changing the order of the $n$ marked points on $\Sigma^c$ by some permutation $\pi \in S_n$. Note that the labels $U, v, \delta$ of a given marked point $p$ do not change, just its position in the total order on the marked points.

Let $\Sigma_\pi^c$ be the surface with the new ordering of its insertion points. For an element $b \in \beta_{\mathcal{V}}(\Sigma^c)$, this amounts to precomposing $b$ with the corresponding permutation of factors in the tensor product $U_1 \otimes_{\mathbb{C}} \cdots \otimes_{\mathbb{C}} U_n$ (cf. (41)), i.e. with the permutation built from the symmetric braiding in vector spaces. Let $T_\pi \colon \beta_{\mathcal{V}}(\Sigma^c) \longrightarrow \beta_{\mathcal{V}}(\Sigma_\pi^c)$ be the resulting linear map.

For $\mathcal{Z}_{\mathcal{C}}^{\mathrm{RT}}(\Sigma)$, one simply considers the cylinder $E_\pi := \Sigma \times [0, 1]$ with vertical ribbons and changes the ordering of marked points in the target object, so that $E_\pi$ becomes a bordism $\Sigma \longrightarrow \Sigma_\pi$. Then

$$F(\Sigma_\pi^c) \circ T_\pi = \mathcal{Z}_{\mathcal{C}}^{\mathrm{RT}}(E_\pi) \circ F(\Sigma^c). \tag{45}$$

Let us illustrate compatibility with transport in three examples: rotating the local coordinate frame by $2\pi$, moving one insertion point around another, and executing a Dehn-twist on a torus. Some more examples can be found in [47, Sec. 5].[7]

**Rotating the coordinate frame**

Consider the case where $\Sigma^c(\alpha)$ is the Riemann sphere $\mathbb{C} \cup \{\infty\}$ with an insertion $\boldsymbol{p} = (p = 0, U, v = e^{i\alpha}, \delta = -)$ and local coordinate $\varphi(z) = e^{i\alpha}z$, and other insertions elsewhere, see Figure 14 a). Suppose that $U$ is an irreducible $\mathcal{V}$-module of lowest conformal weight $h_U$. For the family $E_\gamma$ given by taking $\alpha$ from 0 to $2\pi$ in $\Sigma^c(\alpha)$, $T_\gamma$ acts by precomposing with $e^{2\pi i L_0} = e^{2\pi i h_U}$: for $b \in \beta_{\mathcal{V}}(\Sigma^c)$

$$T_\gamma b = b \circ e^{2\pi i L_0}|_U = e^{2\pi i h_U} b, \tag{46}$$

---

[7]When comparing results, one has to take into account the slightly different orientation conventions used in [47] and the fact that there the ribbon twist is taken to be $e^{-2\pi i h_U}$.

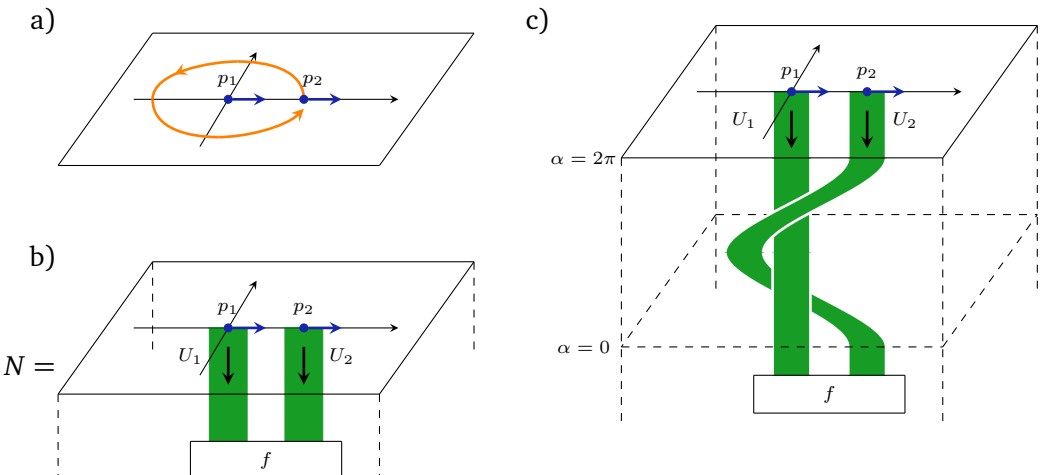

Figure 15: a) Closed path $\gamma$ in the moduli space given by moving $p_2$ around $p_1$, each point describes a fibre in the family $E_\gamma$. b) Bordism $N : \emptyset \longrightarrow \Sigma(0)$ given by a solid 3-ball (only part of which is shown) and embedded ribbon graph. c) The composition $E_\gamma \circ N$. The dotted line $\alpha = 0$ indicates where the boundary between $E_\gamma$ and $N$ was before composition.

where in the second expression it is understood that $e^{2\pi i L_0}|_U$ only acts on the tensor factor $U$ and is extended as the identity to the other factors.

In Figure 14 b) we give our conventions for the relative orientations of three-manifold, an out-going boundary component, and ribbons ending on it. The family $E_\gamma$ thought of as a bordism is shown in Figure 14 c). Applying the TFT functor gives $\mathcal{Z}_\mathcal{C}^{\mathrm{RT}}(E_\gamma) = \theta_{U^*} \mathrm{id}_{U_*}$, where $\theta_{U^*} = \theta_U = e^{2\pi i h_U}$ is the ribbon twist on the simple object $U \in \mathcal{C}$. Thus (44) holds in this case (with proportionality constant 1).

**Taking one field insertion around another**

We again take $\Sigma^c(\alpha)$ to be the Riemann sphere $\mathbb{C} \cup \{\infty\}$, but this time with exactly two insertions $\boldsymbol{p_1} = (0, U_1, 1, -)$ and $\boldsymbol{p_2} = (e^{i\alpha}, U_2, 1, -)$, see Figure 15 a). We assume that $U_1$ and $U_2$ are irreducible and satisfy $U_1 \cong U_2^*$, and that both have the same lowest conformal weight $h$. For $b \in \beta_\mathcal{V}(\Sigma^c)$, and $v_1 \in U_1$, $v_2 \in U_2$ lowest weight vectors, the dependence on the insertion points $p_1, p_2$ is $b(v_1, v_2) = (\mathrm{const})(p_2 - p_1)^{-2h}$. From this one reads off the effect of $T_\gamma$ as

$$T_\gamma b = e^{-4\pi i h} b \, . \tag{47}$$

In Figure 15 b) we show part of a bordism $N : \emptyset \longrightarrow \Sigma(0) = \Sigma(2\pi)$, which is a solid 3-ball with embedded ribbon graph as shown. The state space $\mathcal{Z}_\mathcal{C}^{\mathrm{RT}}(\Sigma(0))$ is one-dimensional, and $b = \mathcal{Z}_\mathcal{C}^{\mathrm{RT}}(N)$ provides a basis (for some non-zero choice of $f$). Figure 15 c) shows the composition $E_\gamma \circ N : \emptyset \longrightarrow \Sigma$. By deforming the ribbon graph one can convince oneself that indeed

$$\mathcal{Z}_\mathcal{C}^{\mathrm{RT}}(E_\gamma)(b) = \theta_{U_1}^{-2} b \, , \tag{48}$$

in agreement with (47).

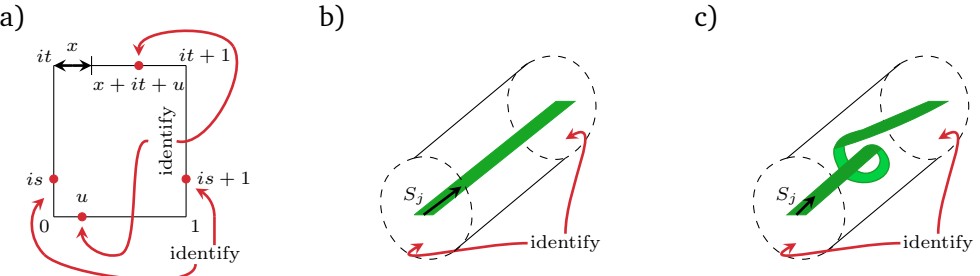

Figure 16: a) Closed path in the moduli space given by taking the shift $x$ in the identification from 0 to 1. b) Solid torus $N_j : \emptyset \longrightarrow \Sigma(0)$. c) The composition $E_\gamma \circ N_j$.

**Dehn twist on the torus**

Let $\Sigma^c(x)$ be the rectangle in $\mathbb{C}$ with corners $0, 1, it, it + 1$ for some $t > 0$, and identify edges parallel to the imaginary axis via $is \sim 1 + is$, and the edges parallel to the real axis with a twist $s \sim it + x + s$ for some $x \in [0, 1]$, see Figure 16 a). One checks that $\Sigma^c(x)$ is biholomorphic to the torus obtained by quotienting $\mathbb{C}$ by $\mathbb{Z} + \tau\mathbb{Z}$ with $\tau = x + it$ (via the map sending $z \in \Sigma^c(x)$ to its class in the quotient). We will need at least one insertion point on $\Sigma^c$, which we take to be 0, and we take it to be labelled by the vacuum module $\mathcal{V}$. We take $\gamma$ to be the path given by $\Sigma^c(x)$ with $x \in [0, 1]$.

For $S_j$, $j \in \mathcal{I}$ one of the chosen irreducible $\mathcal{V}$-modules, let $\chi_j(w, \tau) = \mathrm{tr}_{S_j}\left(w_0 \exp(2\pi i \tau(L_0 - \frac{c}{24})\right)$ be the torus conformal block with one insertion of an element $w \in \mathcal{V}$ (only its zero mode contributes) and where $\tau = it$. Then $\beta_\mathcal{V}(\Sigma^c(0))$ has basis $\{\chi_j(-, \tau)\}_{j \in \mathcal{I}}$ [91], and

$$(T_\gamma \chi_j)(w, it) = \chi_j(w, it + 1) = e^{2\pi i(h_j - \frac{c}{24})}\chi_j(w, it). \tag{49}$$

On the TFT side, $\chi_j(w, \tau)$ is represented by a solid torus $N_j : \emptyset \longrightarrow \Sigma(0)$ with an $S_j$-labelled ribbon at its centre, see Figure 16 b) (vacuum insertions are not visible in the TFT presentation). We write $v_j = \mathcal{Z}_\mathcal{C}^{\mathrm{RT}}(N_j)$ for the corresponding element in the state space $\mathcal{Z}_\mathcal{C}^{\mathrm{RT}}(\Sigma(0))$. The composition $E_\gamma \circ N_j$ is shown in Figure 16 c). It follows that

$$\mathcal{Z}_\mathcal{C}^{\mathrm{RT}}(M_\gamma)(v_j) = \theta_{S_j} v_j. \tag{50}$$

Thus (44) is satisfied, but this time with a non-trivial proportionality constant.

# 4 3d TFTs with values in super-vector spaces

The most basic example of a (2-)spin CFT, namely that of single free fermion, requires us to distinguish between even and odd fields, and to include parity signs when they are re-ordered. Mathematically this can be described by working with super-vector spaces and their parity-dependent braiding. In this section we define the corresponding generalisation of RT TFT we will need for this. We start by stating our conventions for super-vector spaces, then define the trivial TFT valued in super-vector spaces, as well as its product with a RT TFT. Finally we relate this product TFT to conformal blocks of purely even vertex operator super-algebras, or, in other words, VOAs whose representations are considered in super-vector spaces.

## 4.1 Super-vector spaces

In this section we give our conventions for the category $\mathcal{SV}ect$ of finite dimensional super-vector spaces.

The objects of $\mathcal{SV}ect$ are finite dimensional $\mathbb{Z}_2$-graded vector spaces over $\mathbb{C}$, and the morphisms are degree preserving (i.e. even) linear maps. We write $X = X^0 \oplus X^1 \in \mathcal{SV}ect$ for the degree 0 (even) and degree 1 (odd) components of the super-vector space $X$. There are two simple objects up to isomorphism, namely the even and the odd 1-dimensional vector spaces

$$K^+ := \mathbb{C}^{1|0}, \quad \text{and} \quad K^- := \mathbb{C}^{0|1}. \tag{51}$$

The category $\mathcal{SV}ect$ is equipped with an involution $\Pi$ called *parity shift*. The parity shift functor simply exchanges the parity on a super-vector space,

$$\Pi(X)^0 = X^1, \quad \Pi(X)^1 = X^0. \tag{52}$$

On morphisms, $\Pi$ acts as the identity.

$\mathcal{SV}ect$ is a braided monoidal category via the graded tensor product (with monoidal unit $K^+$) and braiding

$$\sigma_{X,Y} : X \otimes Y \longrightarrow Y \otimes X, \quad x \otimes y \longmapsto (-1)^{|x| \cdot |y|} \, y \otimes x, \tag{53}$$

for $X, Y \in \mathcal{SV}ect$ and $x, y$ homogeneous of degree $|x|, |y| \in \mathbb{Z}_2$. This braiding is symmetric, so that $\mathcal{SV}ect$ is a symmetric monoidal category.

For the ribbon structure we fix the left and right duals of $X \in \mathcal{SV}ect$ to be the dual vector space

$$^*X = X^* = \text{Hom}_{\mathbb{C}}(X, \mathbb{C}), \quad (X^*)^i = \text{Hom}_{\mathbb{C}}(X^i, \mathbb{C}) \quad (i \in \mathbb{Z}_2), \tag{54}$$

of not necessarily degree preserving linear maps, so that linear maps $X^i \longrightarrow \mathbb{C}$ have $\mathbb{Z}_2$-degree $i$. The left duality morphisms are the same as for vector spaces,

$$\begin{aligned} \text{ev}_X : X^* \otimes X &\longrightarrow K^+, & \text{coev}_X : K^+ &\longrightarrow X \otimes X^*, \\ \xi \otimes x &\longmapsto \xi(x), & 1 &\longmapsto \textstyle\sum_{j=1}^{\dim_{\mathbb{C}}(X)} e_j \otimes \varphi_j, \end{aligned} \tag{55}$$

where $(e_j, \varphi_j)_{j=1,\dots,\dim_{\mathbb{C}}(X)}$ is a dual basis pair for $X$ and $X^*$. The right duality morphisms differ from those in vector spaces by a parity sign coming from the symmetric braiding,

$$\widetilde{\text{ev}}_X = \text{ev}_X \circ \sigma_{X,X^*} : X \otimes X^* \longrightarrow K^+, \quad \widetilde{\text{coev}}_X = \sigma_{X,X^*} \circ \text{coev}_X : K^+ \longrightarrow X^* \otimes X. \tag{56}$$

The ribbon twist can now be computed from the duals and the braiding to be

$$\theta_X = \text{id}_X : X \longrightarrow X. \tag{57}$$

Altogether, $\mathcal{SV}ect$ is a ribbon category with trivial ribbon twist and symmetric braiding.

The quantum dimension of a super-vector space $X \in \mathcal{SV}ect$ (cf. (35)) is what is usually called the super-dimension $\text{sdim}_{\mathbb{C}}(X)$,

$$\dim(X) = \text{sdim}_{\mathbb{C}}(X) = \dim_{\mathbb{C}}(X^0) - \dim_{\mathbb{C}}(X^1). \tag{58}$$

For the simple objects we have

$$\dim(K^+) = +1, \quad \text{and} \quad \dim(K^-) = -1. \tag{59}$$

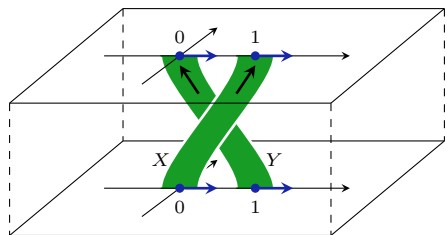

Figure 17: Underlying 3-manifold with embedded ribbon graph for the bordisms $M : \Sigma_1 \longrightarrow \Sigma_2$ and $M' : \Sigma_1 \longrightarrow \Sigma'_2$.

## 4.2 The trivial 3d TFT with values in $\mathcal{SVect}$

The *trivial 3d TFT with values in $\mathcal{SVect}$* is a symmetric monoidal functor SV from the category $\mathcal{Bord}_3(\mathcal{SVect})$ of $\mathcal{SVect}$-extended surfaces and 3-bordisms with $\mathcal{SVect}$-coloured ribbon graphs to $\mathcal{SVect}$:

$$\mathsf{SV} : \mathcal{Bord}_3(\mathcal{SVect}) \longrightarrow \mathcal{SVect}. \tag{60}$$

Here $\mathcal{SVect}$ plays two different roles: on the one hand, it is used as a ribbon category to label marked points and ribbon graphs. On the other hand, it is used as a target category for the functor, and in this case we only use its symmetric monoidal structure. The definition of SV is as follows:

- *On objects:* The value of SV on an extended surface $\Sigma$ with ordered marked points $(p_i, X_i, v_i, \delta_i)$, $i = 1, \dots, n$ is

$$\mathsf{SV}(\Sigma) := X_1^{\delta_1} \otimes \cdots \otimes X_n^{\delta_n}, \tag{61}$$

  where $X_i^+ = X_i$ and $X_i^- = X_i^*$. Note that this just depends on the ordered set of marked points, and not on the underlying surface $\Sigma$.

- *On morphisms:* Let $\Sigma$ and $\Sigma'$ be extended surfaces with marked points $(p_i, X_i, v_i, \delta_i)$, $i = 1, \dots, n$ and $(q_j, Y_j, w_j, v_j)$, $j = 1, \dots, m$, respectively. Let $M : \Sigma \longrightarrow \Sigma'$ be a bordism with embedded $\mathcal{SVect}$-coloured ribbon graph $\Gamma$. If one just retains the combinatorial data of $\Gamma$ and forgets the surrounding manifold and the framing of the ribbons, one obtains an even linear map

$$\widetilde{\Gamma} : X_1^{\delta_1} \otimes \cdots \otimes X_n^{\delta_n} \longrightarrow Y_1^{v_1} \otimes \cdots \otimes Y_m^{v_m}. \tag{62}$$

  This is well-defined since as a ribbon category, $\mathcal{SVect}$ is symmetric and has trivial twist. We set

$$\mathsf{SV}(M) := \widetilde{\Gamma}. \tag{63}$$

For example, if $M$ is a bordism $\emptyset \longrightarrow \emptyset$, i.e. a closed 3-manifold, and if $M$ has empty ribbon graph $\Gamma = \emptyset$, then $\mathsf{SV}(M) = 1$. If the same $M$ contains a ribbon graph $\Gamma'$ consisting of loops labelled $X_1, \dots, X_n$, then $\mathsf{SV}(M) = \prod_{i=1}^n \mathrm{sdim}_{\mathbb{C}}(X_i)$, independent of $M$ and how the loops are linked or framed.

As another example, consider three extended surfaces $\Sigma_1, \Sigma_2, \Sigma'_2$, which all have underlying surface $\mathbb{C} \cup \{\infty\}$ but which differ in their marked points:

- $\Sigma_1$: $p_1 = (0, X, +)$ and $p_2 = (1, Y, +)$,

- $\Sigma_2$: $q_1 = (0, Y, +)$ and $q_2 = (1, X, +)$,

- $\Sigma'_2$: $q'_2 = (0, Y, +)$ and $q'_1 = (1, X, +)$.

In each case, the index $i = 1, 2$ gives the order of the two points, so that $\Sigma_2$ and $\Sigma_2'$ only differ in the ordering of the two marked points. We have

$$\mathrm{SV}(\Sigma_1) = X \otimes Y, \quad \mathrm{SV}(\Sigma_2) = Y \otimes X, \quad \mathrm{SV}(\Sigma_2') = X \otimes Y. \tag{64}$$

Let $M \colon \Sigma_1 \longrightarrow \Sigma_2$ and $M' \colon \Sigma_1 \longrightarrow \Sigma_2'$ be two bordisms which have the same underlying 3-manifold $(\mathbb{C} \cup \{\infty\}) \times [0, 1]$, and which have the same embedded ribbon graph as shown in Figure 17, so that they only differ in their target object. Then

$$\mathrm{SV}(M) = \left[ X \otimes Y \xrightarrow{\sigma_{X,Y}} Y \otimes X \right], \quad \mathrm{SV}(M') = \left[ X \otimes Y \xrightarrow{\mathrm{id}} X \otimes Y \right]. \tag{65}$$

Thus, even though the bordism in Figure 17 looks like a crossing, whether or not it gets mapped to the symmetric braiding in $\mathcal{SV}ect$ depends on the ordering of the marked points.

A related example is to take $\Sigma$ with marked points $\boldsymbol{p_i} = (p_i, X_i, +)$, $i = 1, \dots, n$ and $\Sigma'$ with the same marked points but ordered differently, $\boldsymbol{p_i'} = \boldsymbol{p_{\pi(i)}}$ for some permutation $\pi \in S_n$. Then we can consider the cylinder $M = \Sigma \times [0, 1]$ with vertical ribbons as a bordism $M \colon \Sigma \longrightarrow \Sigma'$. In this case, $\mathrm{SV}(M) \colon X_1 \otimes \cdots \otimes X_n \longrightarrow X_{\pi(1)} \otimes \cdots \otimes X_{\pi(n)}$ implements the permutation of factors with parity signs.

**Remark 4.1.** The above construction is not specific to three dimensions. One can in the same way define a TFT on $n$-dimensional bordisms with embedded $\mathcal{SV}ect$ labelled ribbon graphs which takes values in $\mathcal{SV}ect$ for any $n \geq 1$. However, we will only need the three-dimensional case in this paper.

### 4.3 Reshetikhin-Turaev TFT with values in $\mathcal{SV}ect$

Let $\mathcal{C}$ be a modular fusion category and set

$$\widehat{\mathcal{C}} := \mathcal{C} \boxtimes \mathcal{SV}ect. \tag{66}$$

The product '$\boxtimes$' denotes the Deligne product of abelian categories, see [33, Sec. 1.11]. Since $\mathcal{C}$ is semisimple, the objects of $\mathcal{C} \boxtimes \mathcal{SV}ect$ are simply sums of products $U \boxtimes M$, $U \in \mathcal{C}$, $M \in \mathcal{SV}ect$, and its morphisms are the corresponding direct sums of tensor products (over $\mathbb{C}$) of the Hom-spaces of $\mathcal{C}$ and $\mathcal{SV}ect$.

In particular, if $\{S_i\}_{i \in \mathcal{I}}$ denotes a choice of representatives of the isomorphism classes of simple objects of $\mathcal{C}$ as in (37), then the simple objects of $\widehat{\mathcal{C}}$ are $\{S_i \boxtimes K^+, S_i \boxtimes K^-\}_{i \in \mathcal{I}}$.

The category $\widehat{\mathcal{C}}$ has symmetric centre $\mathcal{SV}ect$ and hence is not modular and cannot be used directly as input for the Reshetikhin-Turaev construction.[8] But we can still use $\widehat{\mathcal{C}}$ to decorate bordisms, and – as we now describe – one can still use it to obtain a TFT with values in $\mathcal{SV}ect$, i.e. a symmetric monoidal functor

$$\widehat{\mathcal{Z}}_{\mathcal{C}} \colon \mathcal{B}ord_3(\widehat{\mathcal{C}}) \longrightarrow \mathcal{SV}ect. \tag{67}$$

We define $\widehat{\mathcal{Z}}_{\mathcal{C}}$ to be the product of the RT TFT $\mathcal{Z}_{\mathcal{C}}^{\mathrm{RT}} \colon \mathcal{B}ord_3(\mathcal{C}) \longrightarrow \mathcal{V}ect$ and the trivial $\mathcal{SV}ect$ valued TFT $\mathrm{SV}$. In more detail, the value of $\widehat{\mathcal{Z}}_{\mathcal{C}}$ on objects and morphisms is as follows:

- *On objects:* Let $\Sigma$ be a $\widehat{\mathcal{C}}$ extended surface with marked points $(p_i, U_i \boxtimes X_i, v_i, \delta_i)$, $i = 1, \dots, n$, i.e. all objects labels have product form. In this case we get a $\mathcal{C}$-extended

---

[8]Braided fusion categories whose symmetric centre is $\mathcal{SV}ect$ are called *slightly-degenerate*, see e.g. [30] for properties of such categories. If one uses a different spherical structure on $\mathcal{SV}ect$ to the one used here, namely where the quantum dimensions are all positive (and hence the twist is $-1$ on $K^-$), slightly degenerate categories are called *super-modular* in [8], and $\widehat{\mathcal{C}}$ would be an example of a slit super-modular category.

surface $\Sigma'$ and a $\mathsf{SV}$-extended surface $\Sigma''$ by forgetting the second, respectively the first factor. Then

$$
\begin{aligned}
\widehat{\mathcal{Z}}_{\mathcal{C}}(\Sigma) &:= \mathcal{Z}_{\mathcal{C}}^{\mathrm{RT}}(\Sigma') \otimes \mathsf{SV}(\Sigma'') \\
&\overset{(*)}{\cong} \mathcal{C}(U_1^{-\delta_1} \otimes \cdots \otimes U_n^{-\delta_n}, L^{\otimes g}) \otimes X_1^{\delta_1} \otimes \cdots \otimes X_n^{\delta_n},
\end{aligned}
\tag{68}
$$

where for $(*)$ we in addition assume $\Sigma$ to be connected and of genus $g$, and where we consider the Hom-space $\mathcal{C}(\cdots)$ as a purely even super-vector space. In $(*)$ we furthermore used the isomorphism (40), for which the marked points have $\delta_i = -$ and which is the reason for the relative signs. The definition of $\widehat{\mathcal{Z}}_{\mathcal{C}}$ for marked points labelled by general objects from $\widehat{\mathcal{C}}$ is by linear extension via direct sums.

- *On morphisms:* For morphisms we proceed analogously. Let $M: \Sigma_1 \longrightarrow \Sigma_2$ be a bordism where all labels of objects and coupons are of factorised form. One then obtains bordisms $M': \Sigma_1' \longrightarrow \Sigma_2'$ and $M'': \Sigma_1'' \longrightarrow \Sigma_2''$ in $\mathcal{B}ord_3(\mathcal{C})$ and $\mathcal{B}ord_3(\mathcal{SV}ect)$, respectively. We set

$$
\widehat{\mathcal{Z}}_{\mathcal{C}}(M) := \mathcal{Z}_{\mathcal{C}}^{\mathrm{RT}}(M') \otimes \mathsf{SV}(M'').
\tag{69}
$$

For general morphisms one extends linearly.

For a connected $\widehat{\mathcal{C}}$-extended surface $\Sigma_g$ of genus $g$ with marked points $(p_i, Y_i, v_i, -)$ for $Y_i \in \widehat{\mathcal{C}}$ $(i = 1, \ldots, n)$ we can rewrite (68) as

$$
\widehat{\mathcal{Z}}_{\mathcal{C}}(\Sigma_g) = \bigoplus_{\epsilon \in \{\pm\}} \widehat{\mathcal{C}}(Y_1 \otimes \cdots \otimes Y_n, L^g \boxtimes K^\epsilon) \otimes K^\epsilon.
\tag{70}
$$

Later we will need the value of $\widehat{\mathcal{Z}}_{\mathcal{C}}$ on $S^3$ without embedded ribbon graph, which is given by

$$
\widehat{\mathcal{Z}}_{\mathcal{C}}(S^3) = \mathcal{Z}_{\mathcal{C}}^{\mathrm{RT}}(S^3) \cdot \mathsf{SV}(S^3) = \mathcal{D}_{\mathcal{C}} \cdot 1,
\tag{71}
$$

where

$$
\mathcal{D}_{\mathcal{C}} = \sqrt{\mathrm{Dim}(\mathcal{C})}, \quad \mathrm{Dim}(\mathcal{C}) = \sum_{i \in \mathcal{I}} \dim(S_i)^2,
\tag{72}
$$

is a fixed choice of square root of the global dimension $\mathrm{Dim}(\mathcal{C})$ of $\mathcal{C}$.

A related construction of a TFT with values in the symmetric centre of the ribbon category one starts from can be found in [67]. That construction works for general symmetric centres, not just $\mathcal{SV}ect$, but its formulation does not include bordisms with embedded ribbon graphs.

## 4.4 Relation to conformal blocks of vertex operator super algebras

The VOAs $\mathcal{V}$ that occurred in Section 3 were "bosonic" in the sense that they were objects in the category $\mathcal{V}ect_\infty$ of possibly infinite dimensional vector spaces. If $\mathcal{V}$ is in addition equipped with a $\mathbb{Z}_2$-grading (and with even structure maps and suitable parity signs in its defining conditions), it is called a *vertex operator super-algebra* (VOSA),[9] see e.g. [19,89] for more details. In this case we have $\mathcal{V} \in \mathcal{SV}ect_\infty$, the category of possibly infinite dimensional super-vector spaces.

For a VOSA $\mathcal{V} \in \mathcal{SV}ect_\infty$, we consider $\mathbb{Z}_2$-graded representations, i.e. $\mathcal{V}$-modules are also objects in $\mathcal{SV}ect_\infty$ (again with corresponding parity signs and even structure maps). We write $\mathrm{Rep}_{\mathcal{S}}(\mathcal{V})$ instead of $\mathrm{Rep}(\mathcal{V})$ to stress this point. The morphisms in $\mathrm{Rep}_{\mathcal{S}}(\mathcal{V})$ are parity-even linear maps that intertwine the $\mathcal{V}$-actions.

---

[9]This is not to be confused with vertex algebras that contain a copy of some supersymmetric extension of the Virasoro algebra. Though the latter would in particular also be $\mathbb{Z}_2$-graded.

### $\widehat{\mathcal{C}}$ as representations of a purely even VOSA

A VOSA $\mathcal{W}$ can be split into its parity even and parity odd subspace $\mathcal{W} = \mathcal{W}_0 \oplus \mathcal{W}_1$. Then $\mathcal{W}_0$ is a bosonic VOA and $\mathcal{W}_1$ is a $\mathcal{W}_0$-module.

Conversely, we can take a bosonic rational VOA $\mathcal{V}$ and understand it as a VOSA $\mathcal{V} \in \mathcal{SV}ect_\infty$ concentrated in even degree, $\mathcal{V} = \mathcal{V}_0$. Then its representation category in super-vector spaces satisfies

$$\mathrm{Rep}_{\mathcal{S}}(\mathcal{V}) \;=\; \mathrm{Rep}(\mathcal{V}) \boxtimes \mathcal{SV}ect. \tag{73}$$

If we set $\mathcal{C} = \mathrm{Rep}(\mathcal{V})$ as before, then $\mathcal{C}$ is a modular fusion category, and

$$\widehat{\mathcal{C}} = \mathcal{C} \boxtimes \mathcal{SV}ect = \mathrm{Rep}_{\mathcal{S}}(\mathcal{V}). \tag{74}$$

Thus the category $\widehat{\mathcal{C}}$ from (66) describes the representation theory of a bosonic rational VOA $\mathcal{V}$ in super-vector spaces.

### State spaces of $\widehat{\mathcal{Z}}_{\mathcal{C}}$ as conformal blocks of a purely even VOSA

Recall from in Section 3.2 the discussion of the relation between the state spaces of $\mathcal{Z}_{\mathcal{C}}^{\mathrm{RT}}$ and the spaces $\beta_{\mathcal{V}}(\Sigma^c)$ of conformal blocks for $\mathcal{V}$ seen as a bosonic VOA.

When considering $\mathcal{V}$ as a purely even VOSA, its representation category is given by $\widehat{\mathcal{C}}$, so that the representations acquire a $\mathbb{Z}_2$-grading by parity. Accordingly the space $\mathrm{Hom}_{\mathbb{C}}(U_1 \otimes_{\mathbb{C}} \cdots \otimes_{\mathbb{C}} U_n, \mathbb{C})$ of all linear maps in (41) is now $\mathbb{Z}_2$-graded, too. Since $\mathcal{V}$ is purely even, the defining conditions of the subspace $\beta_{\mathcal{V}}(\Sigma^c)$ are not sensitive to parity. In more detail, if $\widehat{\Sigma}^c$ is a $\widehat{\mathcal{C}}$-extended surface with complex structure and marked points labelled by $\mathcal{V}$-modules $M_i \boxtimes X_i$, with $M_i \in \mathcal{C}$ and $X_i \in \mathcal{SV}ect$, we obtain a $\mathcal{C}$-extended surface $\Sigma^c$, where the marked points are labelled only by $M_i$. Then

$$\beta_{\mathcal{V}}(\widehat{\Sigma}^c) = \beta_{\mathcal{V}}(\Sigma^c) \otimes X_1 \otimes \cdots \otimes X_n. \tag{75}$$

Comparing this to (42) and (68), we see that $\beta_{\mathcal{V}}(\widehat{\Sigma}^c)$ is isomorphic to $\widehat{\mathcal{Z}}_{\mathcal{C}}(\widehat{\Sigma})$ (with $\widehat{\Sigma}$ obtained from $\widehat{\Sigma}^c$ by forgetting the complex structure).

The transport maps in (44) do not change the order of the marked points and hence only act on the first tensor factor in (68) and in (75). Thus the proportionality in (44) remains valid. Changing the order of points leads to the same parity factor on both sides of (45), as in both cases the same permutation $\pi$ is now expressed via the braiding in $\mathcal{SV}ect$ instead of $\mathcal{V}ect$. Hence, the identity (45) remains valid as well.

### Example: conformal two-point blocks on the Riemann sphere

Let $\mathcal{V}$ be the VOA of the $c = \frac{1}{2}$ Ising CFT, i.e. the unique unitary simple VOA at $c = \frac{1}{2}$, and let $M_\epsilon$ be the irreducible $\mathcal{V}$-module with lowest conformal weight $h_\epsilon = \frac{1}{2}$ (see Section 7.3 for a more detailed discussion of the Ising CFT).

Let $\psi \in M_\epsilon$ be the lowest weight vector in $M_\epsilon$. The space of conformal blocks on $\mathbb{C} \cup \{\infty\}$ with insertions of $M_\epsilon$ at $z$ and $w$ is one-dimensional. An element $b$ depends on the insertion points as (via parallel transport)

$$b(z,w)(\psi,\psi) = (z-w)^{-1}, \tag{76}$$

for an appropriate normalisation of $\psi$. Suppose the insertion points are ordered such that $z$ is first and $w$ is second. Consider the operation of exchanging the points $z$ and $w$ by moving them along half-circles on $\mathbb{C}$. Call the resulting path in moduli space $\gamma$. This is not yet a closed path, as the ordering of the insertion points is different: now $w$ is first and $z$ is second. After

reversing the order via the transposition $\pi \in S_2$ one arrives back at the original surface. The effect of the two operations on $b$ is

$$(T_\pi T_\gamma b)(z,w)(\psi,\psi) = -(z-w)^{-1} = -b(z,w). \tag{77}$$

In this sense, $b$ is not single valued under exchange of the insertion points. Now consider $\mathcal{V}$ as a purely even VOSA and take $\psi \in \Pi(M_\epsilon)$, the parity shifted version of $M_\epsilon$, cf. (52). Then $T_\pi$ produces an addition parity sign and one now has

$$(T_\pi T_\gamma b)(z,w)(\psi,\psi) = +(z-w)^{-1} = b(z,w). \tag{78}$$

Thus after shifting the parity of the representations labelling the insertion points, the combined operation of exchanging points and reordering produces trivial monodromy.

Let us see how the same effect arises in the TFT description, i.e. on the right hand side of (44) and (45). In the bosonic case this is similar to Figure 15, but with only a half-turn rather than a full turn. The transport bordism $M_\gamma$ is shown in Figure 17 (with $z = 0$, $w = 1$, and one has to take $X = Y = M_\epsilon$). We consider the composition $E = E_\pi \circ E_\gamma$ of the change-of-ordering bordism $E_\pi$ with the transport bordism $E_\gamma$. The bordism $E$ is an endomorphism $\Sigma \longrightarrow \Sigma$ of the $\mathcal{C}$-extended surface $\Sigma = \mathbb{C} \cup \{\infty\}$ with marked points 0 and 1, in this order. The state space $\mathcal{Z}_\mathcal{C}^{\mathrm{RT}}(\Sigma)$ is one-dimensional and spanned by $v = \mathcal{Z}_\mathcal{C}^{\mathrm{RT}}(N)$ with $N$ as in Figure 15 b) (with $U_1 = U_2 = M_\epsilon$). One finds $\mathcal{Z}_\mathcal{C}^{\mathrm{RT}}(E)(v) = e^{-\pi i h_\epsilon} v = -v$, as expected.

Next consider the TFT $\widehat{\mathcal{Z}}_\mathcal{C}$ and the Riemann sphere with insertions of $\Pi(M_\epsilon) = M_\epsilon \boxtimes K^- \in \widehat{\mathcal{C}}$. In this case, for the composition $E = E_\pi \circ E_\gamma$ one finds

$$\widehat{\mathcal{Z}}_\mathcal{C}(E) \stackrel{(69)}{=} \mathcal{Z}_\mathcal{C}^{\mathrm{RT}}(E') \otimes \mathsf{SV}(E'') = (-\mathrm{id}) \otimes (-\mathrm{id}) = \mathrm{id}, \tag{79}$$

where $\mathcal{Z}_\mathcal{C}^{\mathrm{RT}}(E') = -\mathrm{id}$ was just computed above, and $\mathsf{SV}(E'') = -\mathrm{id}$ is precisely the first example in (65). This result agrees with (78).

## 5 Oriented CFT with parity signs

In this section we present the construction of rational conformal field theory on oriented world sheets with boundaries and defects via three-dimensional topological field theory as developed in [36, 37, 40, 45, 47]. We do this in some detail since we will use the $\mathcal{SVect}$-valued TFT and the $\mathcal{SVect}$-valued conformal blocks described in the previous section. This entails that the requirement of singe-valuedness under monodromy now involves parity signs, which has not been treated in the above works.

We fix a modular fusion category $\mathcal{C}$ and will work with the product $\widehat{\mathcal{C}} = \mathcal{C} \boxtimes \mathcal{SVect}$ as in (66). We think of this as in (74): $\widehat{\mathcal{C}} = \mathrm{Rep}_S(\mathcal{V})$, where $\mathcal{V}$ is a bosonic rational VOA and one considers its representations in super-vector spaces.

### 5.1 Algebras and modules

Here we recall some algebraic background on algebras and modules that we will need. This could be presented in an arbitrary pivotal tensor category, but to avoid changing the setting too often, we work in $\widehat{\mathcal{C}}$.

Throughout this paper, we will often implicitly use the embeddings $\mathcal{C} \hookrightarrow \widehat{\mathcal{C}}$ and $\mathcal{SVect} \hookrightarrow \widehat{\mathcal{C}}$. For example, given an object $U \in \mathcal{C}$ and a super-vector space $M \in \mathcal{SVect}$, the product $U \otimes M$ stands for $(U \boxtimes K^+) \otimes (\mathbf{1} \boxtimes M) = U \boxtimes M$.

A *Frobenius algebra* is an object $A \in \widehat{\mathcal{C}}$ together with a multiplication $\mu$, unit $\eta$, comultiplication $\Delta$, and counit $\epsilon$, such that the comultiplication is a bimodule map $A \longrightarrow A \otimes A$, see [45, 48] for more details. We use the graphical notation

$$
\mu = \text{(diagram)} , \qquad \eta = \text{(diagram)} , \qquad \Delta = \text{(diagram)} , \qquad \epsilon = \text{(diagram)} . \tag{80}
$$

A Frobenius algebra is called *simple*, if it is simple as a bimodule over itself.

A *morphism of (Frobenius) algebras* is a morphism which commutes with the (co)product and (co)unit. An important role will be played in the following by the *Nakayama automorphism of A*, which is indeed an isomorphism $N_A : A \longrightarrow A$ of Frobenius algebras, see e.g. [49]. Explicitly, $N_A$ and its inverse are given by[10]

$$
N_A = \text{(diagram)} , \qquad N_A^{-1} = \text{(diagram)} , \qquad N_A^n = \text{(diagram)}\, n , \qquad \text{(diagram)}\, 1 = \text{(diagram)} , \tag{81}
$$

where the last two equalities give our notation for powers of Nakayama automorphisms.

We say that $A$ is *normalised special* (or just *special* for short) if $\mu \circ \Delta = \mathrm{id}_A$ and if $\varepsilon \circ \eta \neq 0$. We call $A$ *symmetric* if $N_A = \mathrm{id}_A$. If $A$ is symmetric and special, then one can check that $\varepsilon \circ \eta = \dim(A)$, and so for symmetric special Frobenius algebras one necessarily has $\dim(A) \neq 0$. This implies that a special Frobenius algebra $A$ with $\dim(A) = 0$ is never symmetric, we will see an example of this in Section 6.1.

For left and right $A$-modules $X$ and $Y$ we denote the actions by $\rho_l : A \otimes X \longrightarrow X$ and $\rho_r : Y \otimes A \longrightarrow Y$, respectively, and we use the graphical notation

$$
\rho_l = \text{(diagram)} , \qquad \rho_r = \text{(diagram)} . \tag{82}
$$

A morphism of left (right) modules is a morphism commuting with the corresponding action. We denote the category of left (right) $A$-modules by ${}_A\widehat{\mathcal{C}}$ (resp. $\widehat{\mathcal{C}}_A$). An $A_1$-$A_2$-bimodule is a left $A_1$- and right $A_2$-module with commuting actions. We denote the category of $A_1$-$A_2$-bimodules by ${}_{A_1}\widehat{\mathcal{C}}_{A_2}$.

Given an $A_1$-$A_2$-bimodule $X$ and an $A_2$-$A_3$-bimodule $Y$, one can consider their tensor product over $A_2$, written as $X \otimes_{A_2} Y$. If $A_2$ is special, the tensor product can be conveniently described as the image of an idempotent $p$ on $X \otimes Y$, namely

$$
p = \text{(diagram)} = \text{(diagram with } \iota \text{ and } \pi) \tag{83}
$$

where we have also introduced the embedding and projection maps for the image of $p$:

$$
\pi : X \otimes Y \longrightarrow X \otimes_{A_2} Y , \quad \iota : X \otimes_{A_2} Y \longrightarrow X \otimes Y , \quad \pi \circ \iota = \mathrm{id}_{X \otimes_{A_2} Y} , \quad p = \iota \circ \pi . \tag{84}
$$

---

[10]We follow the convention of [74, 80]. In [21, 49] the Nakayama automorphism is the inverse of (81).

The tensor product $X \otimes_{A_2} Y$ is an $A_1$-$A_3$-bimodule via the action

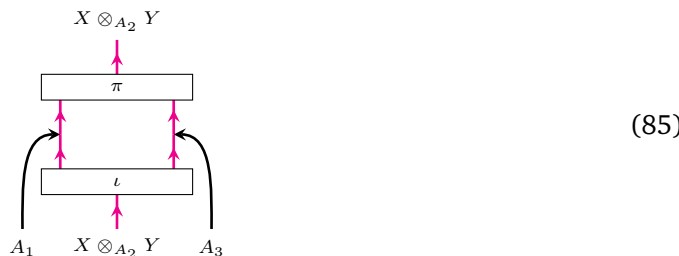

$$(85)$$

The same construction applies to iterated tensor products $X_1 \otimes_{A_2} X_2 \otimes_{A_3} \cdots \otimes_{A_n} X_n$.

Finally we turn to the definition of duals, or rather adjoints, which relate bimodules in $_{A_1}\widehat{\mathcal{C}}_{A_2}$ and $_{A_2}\widehat{\mathcal{C}}_{A_1}$. As explained e.g. in [21, Sec. 4.3], in case that $A_1$ and $A_2$ are not symmetric, the Nakayama automorphisms enters the definition of adjoints. Namely, let $X \in {}_{A_1}\widehat{\mathcal{C}}_{A_2}$. To start with, we turn $X^*$ (the dual in $\widehat{\mathcal{C}}$) into an $A_2$-$A_1$-bimodule via the actions

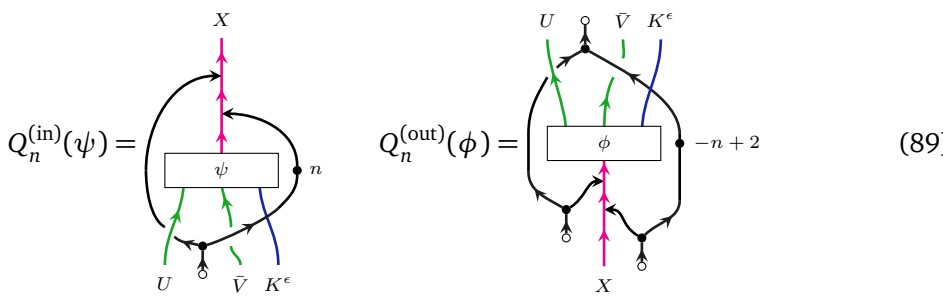

$$(86)$$

Then we define

$$X^{\dagger} := (X^*)_{N_{A_1}^{-1}} . \qquad (87)$$

The twist of the right action is necessary for the evaluation and coevaluation maps to induce the counit $X \otimes_{A_2} X^{\dagger} \longrightarrow A_1$ and unit $A_2 \longrightarrow X^{\dagger} \otimes_{A_1} X$ of the adjunction, we refer to [21, Prop. 4.7] for details. In general, this adjunction is not two-sided, but there are two situations important here, where also the corresponding maps $X^{\dagger} \otimes_{A_1} X \longrightarrow A_2$ and $A_1 \longrightarrow X \otimes_{A_2} X^{\dagger}$ exist:

- If $A_1$ and $A_2$ are symmetric, i.e. if $N_{A_1} = \mathrm{id}_{A_1}$ and $N_{A_2} = \mathrm{id}_{A_2}$, then $X^{\dagger} = X^*$ is a two-sided adjoint – this is the situation relevant in Section 5.3.

- For equivariant bimodules one can define these maps by suitably modifying the (co)evaluation maps of $\widehat{\mathcal{C}}$ – this will be done in Section 6.1.

Finally, we introduce two idempotents, $Q_n^{(\mathrm{in})}$ and $Q_n^{(\mathrm{out})}$ which will be used below (in both the oriented and spin case, with appropriate choices for the algebra $A$) to describe the labels of insertion points in the interior of a world sheet. Let $X \in {}_A\widehat{\mathcal{C}}_A$, $U, \bar{V} \in \mathcal{C}$ (not $\widehat{\mathcal{C}}$), $\epsilon \in \{\pm 1\}$ and $n \in \mathbb{Z}$. The idempotents act on the following Hom-spaces,

$$Q_n^{(\mathrm{in})} \text{ on } \widehat{\mathcal{C}}(U \otimes \bar{V} \otimes K^\epsilon, X), \qquad Q_n^{(\mathrm{out})} \text{ on } \widehat{\mathcal{C}}(X, U \otimes \bar{V} \otimes K^\epsilon), \qquad (88)$$

where $K^{\pm}$ was defined in (51). The action is given by, for $\psi : U \otimes \bar{V} \otimes K^\epsilon \longrightarrow X$ and $\phi : X \longrightarrow U \otimes \bar{V} \otimes K^\epsilon$

$$Q_n^{(\mathrm{in})}(\psi) = \qquad Q_n^{(\mathrm{out})}(\phi) = \qquad (89)$$

A short computation shows that these are indeed idempotents. We denote their images by, for $t \in \{\mathrm{in}, \mathrm{out}\}$,

$$\mathcal{M}_{n,\epsilon}^{(t)}(U, \bar{V}; X) := \mathrm{im}\, Q_n^{(t)}, \tag{90}$$

so that

$$\mathcal{M}_{n,\epsilon}^{(\mathrm{in})}(U, \bar{V}; X) \subset \widehat{\mathcal{C}}(U \otimes \bar{V} \otimes K^\epsilon, X), \qquad \mathcal{M}_{n,\epsilon}^{(\mathrm{out})}(U, \bar{V}; X) \subset \widehat{\mathcal{C}}(X, U \otimes \bar{V} \otimes K^\epsilon). \tag{91}$$

These subspaces will later play the role of multiplicity spaces for bulk field insertions. They are dual to each other via the trace pairing:

**Lemma 5.1.** *The pairing* $\langle\,,\,\rangle_{\mathrm{bulk}} \colon \mathcal{M}_{n,\epsilon}^{(\mathrm{out})}(U, \bar{V}; X) \times \mathcal{M}_{n,\epsilon}^{(\mathrm{in})}(U, \bar{V}; X) \longrightarrow \mathbb{C}$, *where*

$$\langle f, g \rangle_{\mathrm{bulk}} := \frac{1}{\widehat{\mathcal{Z}}_{\mathcal{C}}(S^3)\,\epsilon\,\dim(U)\dim(\bar{V})}\, \mathrm{tr}_{U \otimes \bar{V} \otimes K^\epsilon}(f \circ g), \tag{92}$$

*is non-degenerate.*

The extra normalisation factors are conventional but make the conditions on correlators in Section 5.4 look simpler. The invariant $\widehat{\mathcal{Z}}_{\mathcal{C}}(S^3)$ was given in (71).

*Proof.* Denote by $\langle\,,\,\rangle$ the extension of the pairing $\langle\,,\,\rangle_{\mathrm{bulk}}$ to the entire Hom-space. That is, $\langle f, g \rangle$ is given by the same trace-formula as in (92), but now arbitrary $f \in \widehat{\mathcal{C}}(U \otimes \bar{V} \otimes K^\epsilon, X)$ and $g \in \widehat{\mathcal{C}}(X, U \otimes \bar{V} \otimes K^\epsilon)$ are allowed. Since the trace-pairing is non-degenerate in a pivotal fusion category, so is $\langle\,,\,\rangle$.

A slightly more lengthy but straightforward computation with string diagrams gives

$$\langle f, Q_n^{(\mathrm{in})}(g) \rangle = \langle Q_n^{(\mathrm{out})}(f), g \rangle. \tag{93}$$

A standard argument now shows that the restriction $\langle\,,\,\rangle_{\mathrm{bulk}}$ is also non-degenerate: Suppose that $g \in \mathcal{M}_{n,\epsilon}^{(\mathrm{in})}(U, \bar{V}; X)$. By non-degeneracy there is an $f \in \widehat{\mathcal{C}}(U \otimes \bar{V} \otimes K^\epsilon, X)$ such that $\langle f, g \rangle \neq 0$. Using (93) and that $Q_n^{(\mathrm{in})}$ is an idempotent, one concludes that also $\langle Q_n^{(\mathrm{out})}(f), g \rangle$ is non-zero. $\qquad\square$

We will also need a pairing between multiplicity spaces of boundary field insertions. These will be given directly by Hom-spaces: for $M, N \in {}_A\widehat{\mathcal{C}}$, $W \in \mathcal{C}$ and $X \in {}_A\widehat{\mathcal{C}}_A$, we define

$$\langle\,,\,\rangle_{\mathrm{bnd}} \colon \widehat{\mathcal{C}}(M^* \otimes_A X \otimes_A N, W \otimes K^\epsilon) \times \widehat{\mathcal{C}}(W \otimes K^\epsilon, M^* \otimes_A X \otimes_A N) \longrightarrow \mathbb{C},$$

$$\langle f, g \rangle_{\mathrm{bnd}} = \frac{1}{\epsilon\,\dim(W)}\, \mathrm{tr}_{W \otimes K^\epsilon}(f \circ g). \tag{94}$$

This pairing is just the trace-pairing of $\widehat{\mathcal{C}}$ and is hence non-degenerate.

## 5.2 Oriented CFT without boundaries and defects

In this section we recall how to assign correlators to oriented world sheets (without spin structure). We follow [45, 47] except that we use the $\mathcal{SVect}$ valued TFT $\widehat{\mathcal{Z}}_{\mathcal{C}}$ from Section 4.3.

We fix a symmetric special Frobenius algebra $B$, i.e.

$$B \in \widehat{\mathcal{C}}, \quad N_B = \mathrm{id}_B. \tag{95}$$

This determines which particular full CFT we will describe in terms of the conformal blocks for $\mathcal{V}$.

**Decorated world sheets**

By a *decorated world sheet* we mean a world sheet $\Sigma$ as in Section 2.4 together with additional decorations. In this section we treat the case without boundaries or defects, and so the only decorations are bulk insertion points. A bulk insertion point is a marked point $(p, v, t)$ with $v$ a tangent vector at $p$ and $t \in \{\text{in}, \text{out}\}$, labelled in addition by a tuple

$$(U, \bar{V}, \epsilon, \phi), \tag{96}$$

where

- $U, \bar{V} \in \mathcal{C}$ give the holomorphic and antiholomorphic representation of the bulk insertion,

- $\epsilon \in \{\pm\}$ gives the $\mathbb{Z}_2$-parity, and

- $\phi \in \mathcal{M}^{(t)}_{n=0,\epsilon}(U, \bar{V}; B)$, where $t$ is the in/out label of the marked point, parametrises the multiplicity space of bulk fields of type $(U, \bar{V}, \epsilon)$, see (97) below.

Since we have $N_B = \text{id}$, the space $\mathcal{M}^{(t)}_{n,\epsilon}(U, \bar{V}; B)$ does not depend on $n$, and in this sense the choice $n = 0$ made above is arbitrary.

**Space of bulk fields**

If the labels $U$ and $\bar{V}$ of a bulk field are chosen to be irreducible, they describe a summand $U \otimes_{\mathbb{C}} \bar{V} \otimes_{\mathbb{C}} K^\epsilon$ in the space of bulk fields, which carries an action of the holomorphic and of the antiholomorphic copy of $\mathcal{V}$ on $U$ and $\bar{V}$, respectively, and which has parity $\epsilon$. The multiplicity of $U \otimes_{\mathbb{C}} \bar{V} \otimes_{\mathbb{C}} K^\epsilon$ in the space of in/outgoing bulk fields is given by $\dim_{\mathbb{C}} \mathcal{M}^{(\text{in/out})}_{n=0,\epsilon}(U, \bar{V}; B)$.

To describe the space of bulk fields in purely categorical terms, we need to introduce the category $\mathcal{C} \boxtimes \mathcal{C}^{\text{rev}} \boxtimes \mathcal{SV}ect$. Here, $\mathcal{C}^{\text{rev}}$ is the same tensor category as $\mathcal{C}$, but with inverse braiding and twist, and describes the representations of the antiholomorphic copy of $\mathcal{V}$. The product '$\boxtimes$' denotes the Deligne product as in (66).[11]

The *space of in- and outgoing bulk fields of the oriented CFT* is given by

$$\mathcal{H}^{(t)}_{\text{bulk}}(B) = \bigoplus_{i,j \in \mathcal{I}, \epsilon \in \{\pm 1\}} S_i \boxtimes \bar{S}_j \boxtimes K^\epsilon \otimes \mathcal{M}^{(t)}_{0,\epsilon}(S_i, \bar{S}_j; B) \in \mathcal{C} \boxtimes \mathcal{C}^{\text{rev}} \boxtimes \mathcal{SV}ect, \tag{97}$$

where $t \in \{\text{in}, \text{out}\}$, and $\mathcal{I}$ is the index set for simple objects in $\mathcal{C}$ as in (37). When writing the tensor product with the $\mathbb{C}$-vector space $\mathcal{M}^{(t)}_{0,\epsilon}(U, \bar{V}; B)$, we are using the natural embedding of $\mathcal{V}ect$ in $\mathcal{C} \boxtimes \mathcal{C}^{\text{rev}} \boxtimes \mathcal{SV}ect$ as multiples of the tensor unit. In other words, $\mathcal{M}^{(t)}_{0,\epsilon}(S_i, \bar{S}_j; B)$ is the multiplicity space of the object $S_i \boxtimes \bar{S}_j \boxtimes K^\epsilon$ in $\mathcal{H}^{(t)}_{\text{bulk}}(B)$.

**Connecting manifold**

In the TFT approach, CFT correlators are described as elements in the appropriate TFT state space. To obtain an actual function of field insertions which depends on insertion points and complex structure moduli, one needs to use the relation between TFT state spaces and conformal blocks as outlined in Sections 3.2 and 4.4.

To a decorated world sheet $\Sigma$ we assign a $\hat{\mathcal{C}}$-extended surface $\widetilde{\Sigma}$, the *double of* $\Sigma$, as follows. First, let $\Sigma^{\text{rev}}$ be the same surface as $\Sigma$ but with opposite orientation. The surface underlying $\widetilde{\Sigma}$ is simply the disjoint union of $\Sigma$ and $\Sigma^{\text{rev}}$:[12]

$$\widetilde{\Sigma} = \Sigma \sqcup \Sigma^{\text{rev}}. \tag{98}$$

---

[11]In terms of the relative Deligne product one can also write $\mathcal{C} \boxtimes \mathcal{C}^{\text{rev}} \boxtimes \mathcal{SV}ect = \hat{\mathcal{C}} \boxtimes_{\mathcal{SV}ect} \hat{\mathcal{C}}^{\text{rev}}$, but we will not use that formulation.

[12]Recall that here we assume that $\Sigma$ has empty boundary. In the presence of a boundary this prescription has to be modified, see Section 5.3 below.

Each marked point $p$ of $\Sigma$ produces two points $p_+ \in \Sigma$, $p_- \in \Sigma^{\text{rev}}$ on $\widetilde{\Sigma}$. If $p$ has tangent vector $v$ and is decorated with the objects $(U, \bar{V}, \epsilon, \phi)$, then the two marked points on $\widetilde{\Sigma}$ are

$$(p_+, U, v, \delta), \quad (p_-, \bar{V} \otimes K^\epsilon, v, \delta). \tag{99}$$

Here, $\delta = -$ if $p$ is an ingoing insertion point, and $\delta = +$ if $p$ is outgoing. The ordering of the marked points on $\widetilde{\Sigma}$ is such that if $p_1 < p_2 < \ldots$ on $\Sigma$, then $p_{1,+} < p_{1,-} < p_{2,+} < p_{2,-} < \ldots$ on $\widetilde{\Sigma}$. Attaching the parity factor $K^\epsilon$ to $p_-$ instead of $p_+$ is a convention, see Remark 5.2 below.

To complete the definition of the double $\widetilde{\Sigma}$ as a $\mathcal{C}$-extended surface, we need to give a Lagrangian subspace of $\lambda \subset H_1(\widetilde{\Sigma}, \mathbb{R})$. We will do this after introducing the connecting manifold.

The *connecting manifold of* $\Sigma$ is the three-manifold[12]

$$M_\Sigma := \Sigma \times [-1, 1]. \tag{100}$$

Its boundary is $\partial M_\Sigma = \widetilde{\Sigma}$, and we consider $M_\Sigma$ as a bordism

$$M_\Sigma : \emptyset \longrightarrow \widetilde{\Sigma}. \tag{101}$$

The Lagrangian subspace $\lambda$ of $\widetilde{\Sigma}$ is given by the kernel of the homomorphism $H_1(\widetilde{\Sigma}, \mathbb{R}) \longrightarrow H_1(M_\Sigma, \mathbb{R})$ induced by the inclusion $\widetilde{\Sigma} = \partial M_\Sigma \subset M_\Sigma$ (see [40, App. B] for more details).

To complete the definition of the connecting manifold, we need to describe the ribbon graph in $M_\Sigma$ in terms of the decorated world sheet $\Sigma$. Let $\Sigma_0 := \Sigma \times \{0\}$ be the embedding of the world sheet in the "middle" of the connecting manifold. The ribbon graph in $M_\Sigma$ consists of

1. vertical ribbons (in the $[-1, 1]$-direction) connecting the marked points of $\widetilde{\Sigma}$ to $\Sigma_0$

2. a ribbon graph embedded in $\Sigma_0$, such that the 2-orientation of all ribbons embedded in $\Sigma_0$ agrees with the 2-orientation of $\Sigma_0$.[13]

In part 1 of the ribbon graph, for an insertion point $p \in \Sigma$ labelled by $(U, \bar{V}, \epsilon, \phi)$, the ribbon graph in a neighbourhood of the vertical line $\{p\} \times [-1, 1]$ in $M_\Sigma$ looks as in Figure 18, depending on whether the insertion point is ingoing or outgoing.

For part 2 of the ribbon graph, place a network of $B$-ribbons in $\Sigma_0$ with three-valent vertices labelled by product and coproduct of $B$ as appropriate for the orientations. The $B$-ribbons of the field insertions are connected to this network, and the network is such that the components of $\Sigma_0$ minus the $B$-ribbons are all homeomorphic to discs. This latter point can be achieved for example by choosing the graph dual to a triangulation of $\Sigma$. Proposition 5.3 below states that the value of the TFT does not depend on the particular choice of $B$-ribbon graph.

**Correlators**

The correlator for a decorated world sheet $\Sigma$ will be an element in the space of conformal blocks for the double $\widetilde{\Sigma}$, i.e. in the TFT state space

$$\text{Bl}(\Sigma) := \widehat{\mathcal{Z}}_\mathcal{C}(\widetilde{\Sigma}). \tag{102}$$

We already assumed that $\Sigma$ has empty boundary, so that $\widetilde{\Sigma} = \Sigma \sqcup \Sigma^{\text{rev}}$, with marked points as described above. If in addition $\Sigma$ is connected and of genus $g$, by monoidality of $\widehat{\mathcal{Z}}_\mathcal{C}$ and by

---

[13]We use different orientation conventions for the ribbon graph in the connecting manifold as compared to [45,47]. For example, the conventions described in [47, Sec. 3.1] are such that the 2-orientation of the embedded ribbon graph is opposite to that of $\Sigma_0$.

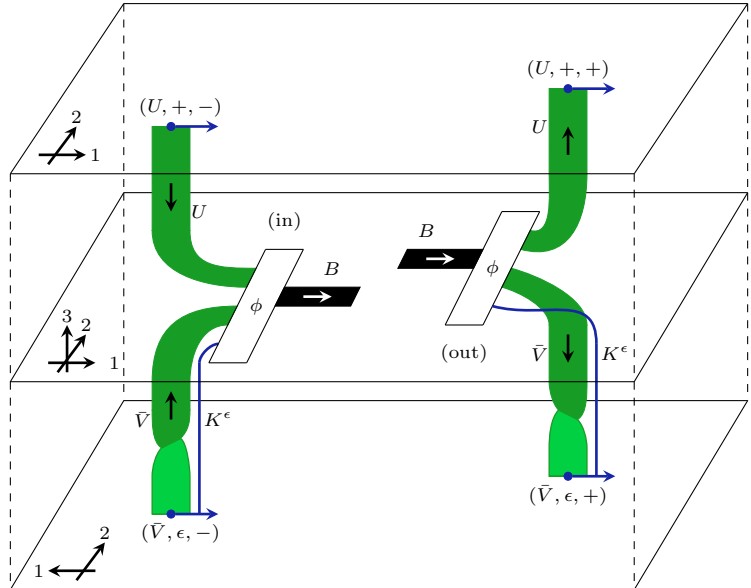

Figure 18: Ribbon graph in the connecting manifold near a bulk insertion point labelled $(U, \bar{V}, \epsilon, \phi)$, both for an ingoing and an outgoing marked point. Observe the opposite half twists on the ribbons for in- and outgoing insertions.

(68), we have explicitly

$$
\begin{aligned}
\mathrm{Bl}(\Sigma) &= \widehat{\mathcal{Z}}_{\mathcal{C}}(\Sigma \sqcup \Sigma^{\mathrm{rev}}) \\
&= \mathcal{Z}_{\mathcal{C}}^{\mathrm{RT}}(\Sigma') \otimes_{\mathbb{C}} \mathcal{Z}_{\mathcal{C}}^{\mathrm{RT}}(\Sigma^{\mathrm{rev}\prime}) \otimes_{\mathbb{C}} \mathsf{SV}((\Sigma \cup \Sigma^{\mathrm{rev}})'') \\
&\cong \mathcal{C}(U_1^{-\delta_1} \otimes \cdots \otimes U_n^{-\delta_n}, L^{\otimes g}) \otimes_{\mathbb{C}} \mathcal{C}^{\mathrm{rev}}(\bar{V}_1^{-\delta_1} \otimes \cdots \otimes \bar{V}_n^{-\delta_n}, L^{\otimes g}) \otimes_{\mathbb{C}} K^{\epsilon_1} \otimes_{\mathbb{C}} \cdots \otimes_{\mathbb{C}} K^{\epsilon_n},
\end{aligned}
\tag{103}
$$

where we identified $K^{\pm} = (K^{\pm})^*$.

In terms of the VOA $\mathcal{V}$, according to (40) and (42), the factor $\mathcal{C}(\dots)$ is interpreted as the space of holomorphic conformal blocks on $\Sigma$, the factor $\mathcal{C}^{\mathrm{rev}}(\dots)$ as the space of anti-holomorphic conformal blocks, and the product of $K^{\epsilon_i}$ collects the various parities of the insertion points.

The correlator $\mathrm{Corr}_B^{\mathrm{or}}(\Sigma)$ for the world sheet $\Sigma$ is a bilinear combination of holomorphic and anti-holomorphic conformal blocks, and so an element of $\mathrm{Bl}(\Sigma)$. The relevant element is defined in terms of the TFT $\widehat{\mathcal{Z}}_{\mathcal{C}}$ and the connecting manifold $M_{\Sigma}$ via

$$
\mathrm{Corr}_B^{\mathrm{or}}(\Sigma) := \widehat{\mathcal{Z}}_{\mathcal{C}}(M_{\Sigma}) \in \mathrm{Bl}(\Sigma).
\tag{104}
$$

We stress that even though the space of blocks $\mathrm{Bl}(\Sigma)$ is a super-vector space which may include odd components, the element $\mathrm{Corr}_B^{\mathrm{or}}(\Sigma)$ is always purely even, since $\widehat{\mathcal{Z}}_{\mathcal{C}}(M_{\Sigma}) \colon K^+ \longrightarrow \mathrm{Bl}(\Sigma)$ is a morphism in $\mathcal{SVect}$, i.e. an even linear map.

**Remark 5.2.** Placing the parity factor $K^{\epsilon}$ at the point $p_-$ on $\Sigma^{\mathrm{rev}}$ instead of with $p_+$ on $\Sigma$ is a convention. Note that either choice produces the same space $\mathrm{Bl}(\Sigma)$ in (103) because the SV-factor in $\widehat{\mathcal{Z}}_{\mathcal{C}}$ only depends on the ordering of the marked points and not on their positions, and placing $K^{\epsilon}$ with $p_+$ or $p_-$ does not change the order. Similarly, both choices will produce the same vector $\mathrm{Corr}_B^{\mathrm{or}}(\Sigma)$ in $\mathrm{Bl}(\Sigma)$.

As the notation suggests, the element $\mathrm{Corr}_B^{\mathrm{or}}(\Sigma)$ only depends on the decorated world sheet $\Sigma$:

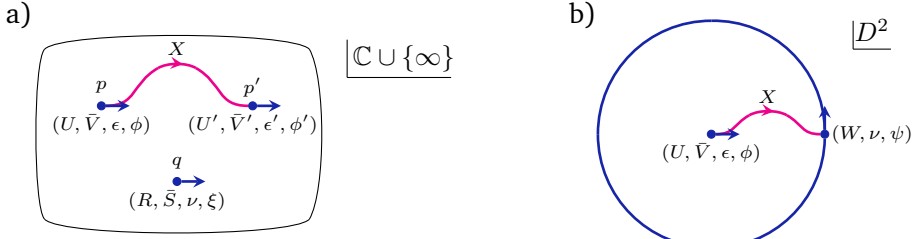

Figure 19: a) Riemann sphere with three insertions (only the part near the insertions is shown), where $p$, $q$ are ingoing and $p'$ is outgoing. Here, $\xi \in \mathcal{M}_{0,\nu}^{(\mathrm{in})}(R, \bar{S}; B)$, $\phi \in \mathcal{M}_{0,\epsilon}^{(\mathrm{in})}(U, \bar{V}; X)$, and $\phi' \in \mathcal{M}_{0,\epsilon'}^{(\mathrm{out})}(U', \bar{V}'; X)$. b) Disc with one ingoing defect field insertion in the bulk and one ingoing defect field inserted on the boundary. Here, $\phi \in \mathcal{M}_{0,\epsilon}^{(\mathrm{in})}(U, \bar{V}; X)$, and $\psi \in \mathcal{C}(W \boxtimes K^\nu, M^* \otimes_B X^* \otimes_B N)$.

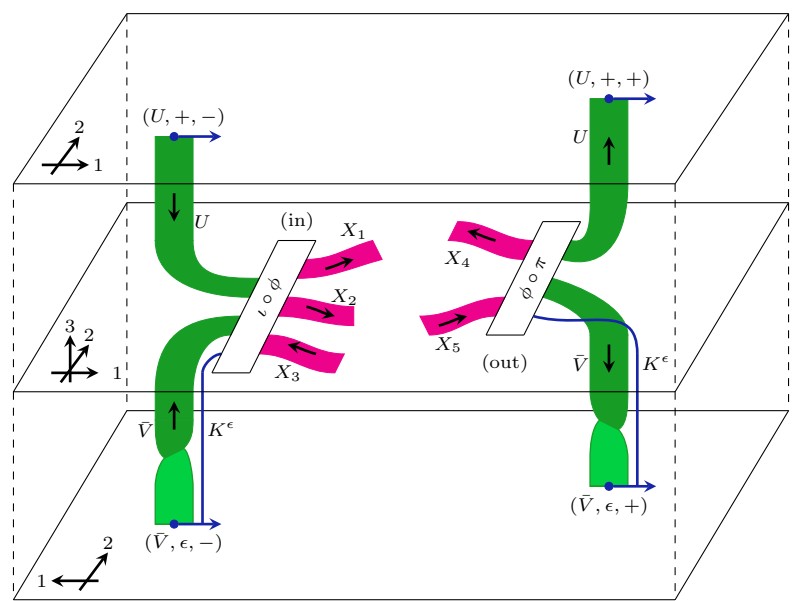

Figure 20: Connecting manifold near bulk insertions with attached defect lines.

**Proposition 5.3.** $\mathrm{Corr}_B^{\mathrm{or}}(\Sigma)$ *does not depend on the choice of $B$-ribbon graph in part 2 of the construction of $M_\Sigma$.*

The proof is the same as in the non-$\mathcal{SV}ect$-valued case in [40,45,47], see in particular [40, App. B.3] for details.

The correlators $\mathrm{Corr}_B^{\mathrm{or}}(\Sigma)$ satisfy the required compatibility conditions with respect to transport and gluing. We discuss this in detail in Section 5.4.

## 5.3 Including boundaries and defects

We now extend the construction of correlators as elements of TFT state spaces to world sheets with boundaries and defects. The idea is the same as above, but now involves more notation.

**Decorated world sheets**

Let $\Sigma$ be a world sheet as in Section 2.4, possibly with boundaries and defects. A decoration of $\Sigma$ consists of the following data (see Figure 19 for two examples):

- A connected component of the boundary minus the boundary insertion points gets labelled by a left $B$-module in $\widehat{\mathcal{C}}$. The $B$-module describes the *boundary condition* for the given stretch of boundary.

- A connected component of the defect network minus field insertions gets labelled by a $B$-$B$-bimodule in $\widehat{\mathcal{C}}$ describing the *defect condition*.

- A boundary insertion which separates boundary conditions $M$ and $N$, in this order along the orientation of the boundary, and which is not the endpoint of a defect line, is labelled by

$$(W, \epsilon, \psi). \tag{105}$$

Here $W \in \mathcal{C}$, $\epsilon \in \{\pm 1\}$ is the parity, and $\psi \in \widehat{\mathcal{C}}(W \otimes K^\epsilon, M^* \otimes_B N)$ (ingoing insertion) or $\psi \in \widehat{\mathcal{C}}(M^* \otimes_B N, W \otimes K^\epsilon)$ (outgoing insertion).

- Consider a boundary insertion between boundary conditions $M, N$ on which defect lines with defect conditions $X_1, \ldots, X_m$ start or end. Let $\delta_i = +$ if $X_i$ is pointing away from the insertion, and $\delta_i = -$ otherwise. Set

$$X = X_1^{\delta_i} \otimes_B \cdots \otimes_B X_m^{\delta_m} \in {}_B\widehat{\mathcal{C}}_B, \quad \text{where} \quad X_i^+ = X_i \text{ and } X_i^- = X_i^*. \tag{106}$$

The boundary insertion is again labelled by (105), but now $\psi \in \widehat{\mathcal{C}}(W \otimes K^\epsilon, M^* \otimes_B X \otimes_B N)$ (ingoing), respectively $\psi \in \widehat{\mathcal{C}}(M^* \otimes_B X \otimes_B N, W \otimes K^\epsilon)$ (outgoing).

- The label of a bulk insertion which is not connected to any line defects has already been described in (96).

- Consider a bulk insertion on which defect lines with defect conditions $X_1, \ldots, X_m$ start or end, and let $X$ be as in (106). The insertion is again labelled as in (96), but now $\phi \in \mathcal{M}_{n=0,\epsilon}^{(t)}(U, \bar{V}; X)$, with $t = \text{in}$ (ingoing insertion) and $t = \text{out}$ otherwise, and where the multiplicity spaces are given in (90).

**Spaces of boundary and defect fields**

From the above description of field labels, we can read off the spaces of boundary and bulk fields with attached defect lines. Namely, for boundary insertions we have, with $X$ as in (106)

$$\mathcal{H}_{M,N}^{(\text{in})}(X) = \bigoplus_{i \in \mathcal{I}, \epsilon \in \{\pm 1\}} S_i \boxtimes K^\epsilon \otimes \widehat{\mathcal{C}}(S_i \otimes K^\epsilon, M^* \otimes_B X \otimes_B N) \in \widehat{\mathcal{C}},$$

$$\mathcal{H}_{M,N}^{(\text{out})}(X) = \bigoplus_{i \in \mathcal{I}, \epsilon \in \{\pm 1\}} S_i \boxtimes K^\epsilon \otimes \widehat{\mathcal{C}}(M^* \otimes_B X \otimes_B N, S_i \otimes K^\epsilon) \in \widehat{\mathcal{C}}. \tag{107}$$

For bulk insertions with attached defect lines we get, with $t \in \{\text{in}, \text{out}\}$,

$$\mathcal{H}_{\text{bulk}}^{(t)}(X) = \bigoplus_{i,j \in \mathcal{I}, \epsilon \in \{\pm 1\}} S_i \boxtimes \bar{S}_j \boxtimes K^\epsilon \otimes \mathcal{M}_{0,\epsilon}^{(t)}(U, \bar{V}; X) \in \mathcal{C} \boxtimes \mathcal{C}^{\text{rev}} \boxtimes \mathcal{SVect}. \tag{108}$$

The algebra $B$, seen as a bimodule over itself, labels the trivial defect. And indeed, setting $X = B$ in the above state space gives the space of bulk insertions without attached line defects from (97).

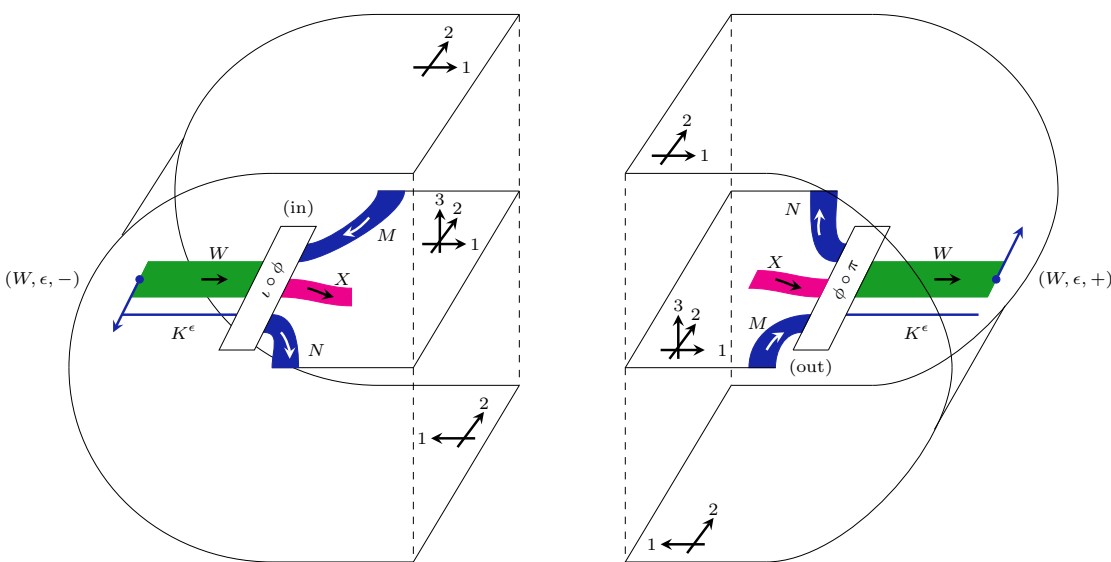

Figure 21: Connecting manifold near boundary insertion.

**Connecting manifold**

In the presence of boundaries we need to extend the definition of the double of a decorated world sheet and of the connecting manifold. Namely, given a decorated world sheet $\Sigma$, the double $\widetilde{\Sigma}$ is a $\widehat{\mathcal{C}}$-extended surface whose underlying surface is obtained by gluing $\Sigma$ and $\Sigma^{\mathrm{rev}}$ together along their boundary:

$$\widetilde{\Sigma} = \Sigma \sqcup \Sigma^{\mathrm{rev}}/(x \sim x^{\mathrm{rev}}, \ x \in \partial\Sigma). \tag{109}$$

Note that $\widetilde{\Sigma}$ has empty boundary.

As before, a bulk insertion in $\Sigma$ (with or without attached defects) labelled as in (96) gives rise to two marked points on $\widetilde{\Sigma}$ labelled as in (99).

Each marked point $p$ on the boundary of $\Sigma$ gives a single marked point $p$ in $\widetilde{\Sigma}$. Since the tangent vector $v$ at $p$ is parallel to the boundary, it makes sense to take the same tangent vector in $\widetilde{\Sigma}$. If the label of $p$ on the decorated world sheet is as in (105), the marked point on $\widetilde{\Sigma}$ is labelled by

$$(p, W \otimes K^{\epsilon}, v, \delta), \tag{110}$$

where as in (99), $\delta = -$ if $p$ is an ingoing insertion point, and $\delta = +$ if $p$ is outgoing.

The definition of the connecting manifold is extended from (100) to

$$M_{\Sigma} := \Sigma \times [-1, 1]/\sim, \qquad (x, t) \sim (x, -t), \quad x \in \partial\Sigma. \tag{111}$$

Its boundary is $\partial M_{\Sigma} = \widetilde{\Sigma}$. The construction of the ribbon graph in $M_{\Sigma}$ has two parts as in Section 5.2.

For part 1, the relevant vertical parts of the ribbon graph for in/outgoing bulk and boundary insertions, possibly with defects, are shown in Figures 20 and 21.

For part 2, the only modification is that the network of $B$-lines also attaches to the (bi)modules labelling boundaries and defects via the corresponding $B$-action. The network of $B$-lines has to be such that each connected component of the world sheet minus all $B$-lines and all defect lines is contractible.

As an example, for the two decorated world sheets in Figure 19 the connecting manifolds are shown in Figures 22 and 23.

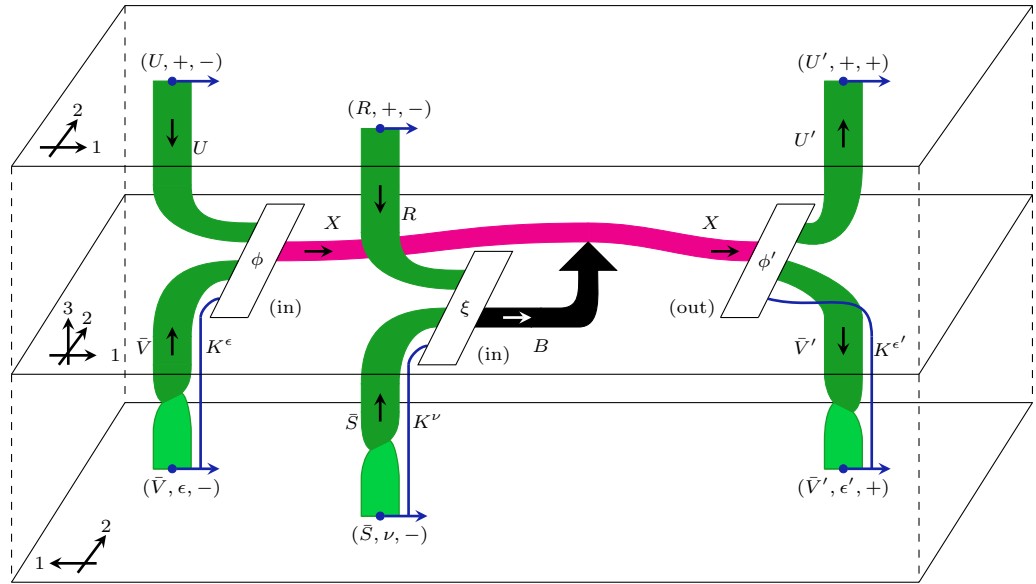

Figure 22: Connecting manifold for the world sheet shown in Figure 19 a).

**Correlators**

For a decorated world sheet $\Sigma$ with boundaries and defects, the space of conformal blocks is still defined as in (102), namely $\mathrm{Bl}(\Sigma) = \widehat{\mathcal{Z}}_{\mathcal{C}}(\widetilde{\Sigma})$, and correlators are given as in (104):

$$\mathrm{Corr}^{\mathrm{or}}_B(\Sigma) = \widehat{\mathcal{Z}}_{\mathcal{C}}(M_{\Sigma}) \in \mathrm{Bl}(\Sigma), \tag{112}$$

and as before, these are always purely even elements of $\mathrm{Bl}(\Sigma)$. Proposition 5.3 applies equally to the present situation with boundaries and defects. The proof is as in [37, 40] and we omit the details.

We now turn to the description of the compatibility conditions with respect to transport and gluing and give the proof that the collection of correlators $\{\mathrm{Corr}^{\mathrm{or}}_B(\Sigma)\}_{\Sigma}$ satisfies these conditions.

### 5.4 Consistency of oriented correlators in the presence of parity

From the input data of a symmetric special Frobenius algebra $B \in \widehat{\mathcal{C}}$, the above construction produces a family of vectors $\{\mathrm{Corr}^{\mathrm{or}}_B(\Sigma)\}_{\Sigma}$, where $\Sigma$ runs over decorated world sheets and $\mathrm{Corr}^{\mathrm{or}}_B(\Sigma) \in \mathrm{Bl}(\Sigma)$.

For such a collection of correlators to be *consistent*, we require two conditions, which we elaborate in the following:

(T) *(Transport)* If two decorated world sheets $\Sigma$ and $\Sigma'$ are joined by a path $\gamma$ in decorated world sheets, then $\mathrm{Corr}^{\mathrm{or}}_B(\Sigma')$ agrees with the transport of $\mathrm{Corr}^{\mathrm{or}}_B(\Sigma)$ along $\gamma$.

(G) *(Gluing)* If $\Sigma'$ is obtained from $\Sigma$ by gluing after removing from $\Sigma$ a disc around an ingoing and an out-going field insertion, then $\mathrm{Corr}^{\mathrm{or}}_B(\Sigma')$ is obtained from $\mathrm{Corr}^{\mathrm{or}}_B(\Sigma)$ by summing over intermediate labels.

We now describe in detail how these conditions are expressed in terms of the TFT $\widehat{\mathcal{Z}}_{\mathcal{C}}$. This closely follows [37, 40], up to some modifications due to the inclusion of parity in our setting.

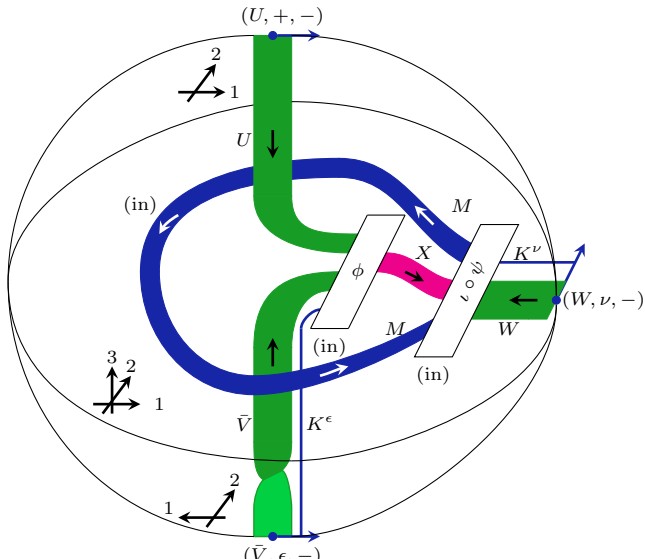

Figure 23: Connecting manifold for the world sheet shown in Figure 19 b).

**Transport**

Recall from Section 3.3 that we will use paths and families interchangeably. To formulate the transport condition we will use the language of families.

Let $E_\gamma$ be a family of world sheets over $[0,1]$ with fibre $\gamma(t)$ for $t \in [0,1]$ and $\Sigma = \gamma(0)$, $\Sigma' = \gamma(1)$. For each $t \in [0,1]$ consider the double $\widetilde{\gamma}(t)$ of $\gamma(t)$ as in (109). This defines a family $E_{\widetilde{\gamma}}$ of $\widehat{\mathcal{C}}$-extended surfaces, as well as the corresponding transport bordism as in (44),

$$\widetilde{\Sigma} = \widetilde{\gamma}(0), \quad \widetilde{\Sigma}' = \widetilde{\gamma}(1), \quad E_{\widetilde{\gamma}} \colon \widetilde{\Sigma} \longrightarrow \widetilde{\Sigma}'. \tag{113}$$

The compatibility with transport can now be stated as follows:

**Theorem 5.4.** *Let $B \in \widehat{\mathcal{C}}$ be a symmetric special Frobenius algebra, and let $E_\gamma$ be a family of decorated world sheets from $\Sigma$ to $\Sigma'$. The correlators for $B$ satisfy*

$$\widehat{\mathcal{Z}}_{\mathcal{C}}(E_{\widetilde{\gamma}})(\mathrm{Corr}_B^{\mathrm{or}}(\Sigma)) = \mathrm{Corr}_B^{\mathrm{or}}(\Sigma'). \tag{114}$$

*Proof.* Since $E_\gamma$ is a family over a contractible space it is diffeomorphic to the trivial family $\Sigma \times [0,1]$ (together with a corresponding diffeomorphism $\Sigma' \longrightarrow \Sigma$ for the fibre $\gamma(1)$). The bordism $E_{\widetilde{\gamma}} \circ M_\Sigma$ is therefore diffeomorphic to $M_{\Sigma'}$, possibly up to the choice of network of $B$-ribbon graph, and so by Proposition 5.3, evaluating $\widehat{\mathcal{Z}}_{\mathcal{C}}$ yields the same linear map in either case. $\square$

Let $f \colon \Sigma \longrightarrow \Sigma'$ be an orientation preserving diffeomorphism of decorated world sheets. This induces a diffeomorphism $\widetilde{f} \colon \widetilde{\Sigma} \longrightarrow \widetilde{\Sigma}'$ between the corresponding doubles. Let $E_{\widetilde{f}}$ be the mapping cylinder for $\widetilde{f}$. Then $E_{\widetilde{f}}$ is a bordism from $\widetilde{\Sigma}$ to $\widetilde{\Sigma}'$.

**Corollary 5.5.** *We have*

$$\widehat{\mathcal{Z}}_{\mathcal{C}}(E_{\widetilde{f}})(\mathrm{Corr}_B^{\mathrm{or}}(\Sigma)) = \mathrm{Corr}_B^{\mathrm{or}}(\Sigma'). \tag{115}$$

*Proof.* The mapping cylinder $E_{\widetilde{f}}$ is a special case of a family of decorated world sheets from $\Sigma$ to $\Sigma'$. This reduces the above statement to Theorem 5.4. $\square$

**Gluing**

The proof of consistency with gluing will require more notation and will be more involved than that of consistency with transport.

To start with, we need to choose basis / dual basis pairs in the multiplicity spaces of bulk and boundary fields. Namely, for a given $X \in {}_B\widehat{\mathcal{C}}_B$ and $M, N \in {}_B\widehat{\mathcal{C}}$:

- *(bulk)* Let $U, \bar{V} \in \mathcal{C}$ and $\epsilon \in \{\pm 1\}$. Choose bases

$$\{\alpha\} \subset \mathcal{M}_{0,\epsilon}^{(\text{in})}(U, \bar{V}; X), \quad \text{and} \quad \{\bar{\beta}\} \subset \mathcal{M}_{0,\epsilon}^{(\text{out})}(U, \bar{V}; X), \tag{116}$$

  cf. (90), which are dual to each other with respect to the pairing (92): $\langle \bar{\beta}, \alpha \rangle_{\text{bulk}} = \delta_{\alpha,\beta}$.

- *(boundary)* Let $W \in \mathcal{C}$ and $\epsilon \in \{\pm 1\}$. Choose bases

$$\{\alpha\} \subset \widehat{\mathcal{C}}(W \otimes K^\epsilon, M^* \otimes_B X \otimes_B N), \quad \text{and} \quad \{\bar{\beta}\} \subset \widehat{\mathcal{C}}(M^* \otimes_B X \otimes_B N, W \otimes K^\epsilon), \tag{117}$$

  which are dual to each other with respect to the pairing in (94): $\langle \bar{\beta}, \alpha \rangle_{\text{bnd}} = \delta_{\alpha,\beta}$.

Let $\Sigma$ be a decorated world sheet for $B$ and let $\Sigma_b$ be the underlying bordism. Let $b_{\text{in}} \in \partial_{\text{in}}^t \Sigma_b$ and $b_{\text{out}} \in \partial_{\text{out}}^t \Sigma_b$, $t \in \{\text{c}, \text{o}\}$ be two open or two closed gluing boundaries, cf. (15). We require that they are parametrised by the same (half-)annulus with defects, so that they can be consistently glued.

If $b_{\text{in/out}}$ are closed gluing boundaries, we will write

$$\Sigma(U, \bar{V}, \epsilon, \alpha, \bar{\beta}), \tag{118}$$

for the world sheet $\Sigma$ where the puncture of the disc glued to $b_{\text{in}}$ is labelled by $(U, \bar{V}, \epsilon, \alpha)$ and that for $b_{\text{out}}$ by $(U, \bar{V}, \epsilon, \bar{\beta})$, with $\alpha, \bar{\beta}$ the basis vectors introduced in (116). If $b_{\text{in/out}}$ are open gluing boundaries, we will correspondingly write

$$\Sigma(W, \epsilon, \alpha, \bar{\beta}), \tag{119}$$

for the world sheet $\Sigma$ where the punctures of the glued half-disc are labelled $(W, \epsilon, \alpha)$ and $(W, \epsilon, \bar{\beta})$ in terms of the basis vectors in (117).

Let $\Sigma_b'$ be the bordism obtained by gluing $b_{\text{in}}$ to $b_{\text{out}}$, and let $\Sigma'$ be the corresponding decorated world sheet, which has the same decoration as $\Sigma$ away from the two (half-)discs which got omitted in the gluing.

The final ingredient we need for the gluing condition is the bordism of the TFT which implements the gluing of marked points on the $\widehat{\mathcal{C}}$-extended surfaces given by the doubles of the corresponding world sheets:

- *(open gluing boundary)* The open gluing bordism

$$G_{W,\epsilon} \colon \widetilde{\Sigma}(W, \epsilon, \alpha, \bar{\beta}) \longrightarrow \widetilde{\Sigma}', \tag{120}$$

  is defined as follows. Let $S$ be the circle in the doubled surface $\widetilde{\Sigma}'$ which is the preimage of the glued half-circles $b_{\text{in}} \sim b_{\text{out}}$ under the projection $\widetilde{\Sigma}' \longrightarrow \Sigma'$ from the double to the world sheet. Let $A_S$ be a small closed annular neighbourhood of $S$.

  Abbreviate $\widetilde{\Sigma} = \widetilde{\Sigma}(W, \epsilon, \alpha, \bar{\beta})$. Denote by $p$ and $q$ the punctures on the half-discs glued to $b_{\text{in}}$ and $b_{\text{out}}$, respectively. Let $D_p$ and $D_q$ be the two closed discs in $\widetilde{\Sigma}$ which are the preimages of the half-discs containing $p$ and $q$ under the projection $\widetilde{\Sigma} \longrightarrow \Sigma$. We obtain the identity[14]

$$\widetilde{\Sigma}' \setminus \text{Inn}(A_S) = \widetilde{\Sigma} \setminus \text{Inn}(D_p \sqcup D_q). \tag{121}$$

---

[14]Strictly speaking, we have to enlarge $D_p$ and $D_q$ slightly by the corresponding halves of $A_S$ in $\widetilde{\Sigma}$ for the identity to hold. We use the same notation for these enlarged closed discs.

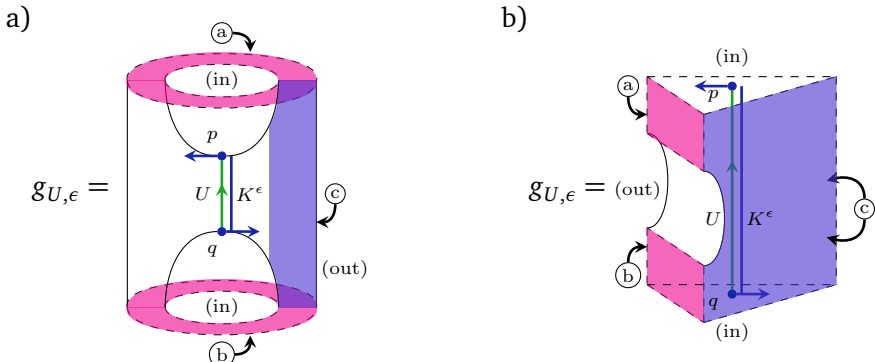

Figure 24: a) The three-manifold with corners $g_{U,\epsilon}$ used in the description of the gluing bordisms. b) Wedge presentation of the same manifold: we identify the faces marked by ©.

In the presentations a) and b) of $g_{U,\epsilon}$ the parts of the boundary marked ⓐ agree in both presentations, as do those marked ⓑ. The two sides © that are identified in presentation b) are also shown in presentation a).

Finally, $G_{W,\epsilon}$ is defined to be $\left(\widetilde{\Sigma}' \backslash \mathrm{Inn}(A_S)\right) \times [0,1]$ glued to the 3-manifold (with corners) $g_{W,\epsilon}$ shown in Figure 24 along $\partial(A_S) \times [0,1]$.

- *(closed gluing boundary)* The closed gluing bordism is denoted by

$$G_{U,\bar{V},\epsilon} : \widetilde{\Sigma}(U,\bar{V},\epsilon,\alpha,\bar{\beta}) \longrightarrow \widetilde{\Sigma}'. \tag{122}$$

To define it, let $S_+$ and $S_-$ be the two circles in the doubled surface $\widetilde{\Sigma}'$ which are the preimage of the glued circles $b_{\mathrm{in}} \sim b_{\mathrm{out}}$ under the projection $\widetilde{\Sigma}' \longrightarrow \Sigma'$. Let $A_{S_+}$ and $A_{S_-}$ be small closed annular neighbourhoods. Analogously to the open case, there are four closed discs $D_{p_\pm}$ and $D_{q_\pm}$ in the double $\widetilde{\Sigma} := \widetilde{\Sigma}(U,\bar{V},\epsilon,\alpha,\bar{\beta})$, so that

$$\widetilde{\Sigma}' \backslash \mathrm{Inn}(A_{S_+} \sqcup A_{S_-}) = \widetilde{\Sigma} \backslash \mathrm{Inn}(D_{p_+} \sqcup D_{p_-} \sqcup D_{q_+} \sqcup D_{q_-}). \tag{123}$$

Then $G_{U,\bar{V},\epsilon}$ is defined to be $\left(\widetilde{\Sigma}' \backslash \mathrm{Inn}(A_{S_+} \sqcup A_{S_-})\right) \times [0,1]$ with $g_{U,+}$ glued along $\partial(A_{S_+}) \times [0,1]$ and $g_{\bar{V},\epsilon}$ glued along $\partial(A_{S_-}) \times [0,1]$.

The compatibility with gluing can now be stated as follows:[15]

**Theorem 5.6.** *Let $B \in \widehat{\mathcal{C}}$ be a symmetric special Frobenius algebra. Then:*

- *(closed gluing) Let $\Sigma'$ be obtained by gluing from $\Sigma(U,\bar{V},\epsilon,\alpha,\bar{\beta})$ as above. The correlators for $B$ satisfy*

$$\mathrm{Corr}_B^{\mathrm{or}}(\Sigma') = \sum_{U,\bar{V},\epsilon,\alpha} \widehat{\mathcal{Z}}_{\mathcal{C}}(G_{U,\bar{V},\epsilon})(\mathrm{Corr}_B^{\mathrm{or}}(\Sigma(U,\bar{V},\epsilon,\alpha,\bar{\alpha}))), \tag{124}$$

*where $U, \bar{V}$ run over simples of $\mathcal{C}$, $\epsilon \in \{\pm\}$ and $\alpha$ runs over the basis (116).*

---

[15]In [37, 40] there is an extra term in the gluing condition arising from a two-point correlator. Since we work with in- and outgoing field insertions, we obtain the simpler gluing relation stated in Theorem 5.6.

- *(open gluing)* Let $\Sigma'$ be obtained by gluing from $\Sigma(W, \epsilon, \alpha, \bar{\beta})$ as above. The correlators for $B$ satisfy

$$\mathrm{Corr}_B^{\mathrm{or}}(\Sigma') = \sum_{W,\epsilon,\alpha} \widehat{\mathcal{Z}}_{\mathcal{C}}(G_{W,\epsilon})(\mathrm{Corr}_B^{\mathrm{or}}(\Sigma(W,\epsilon,\alpha,\bar{\alpha}))), \tag{125}$$

  where $W$ runs over simples of $\mathcal{C}$, $\epsilon \in \{\pm\}$ and $\alpha$ runs over the basis (117).

The proof of this theorem is given in Appendix A.3.

## 5.5 Example: torus partition function

The aim of this section is to show that the partition function of the oriented parity CFT defined by a symmetric special Frobenius algebra $B \in \widehat{\mathcal{C}}$ is the super-trace over the (in-going, say) space of bulk fields as given in (97).

Let $T^2$ be a 2-torus without insertion points, thought of as a $\widehat{\mathcal{C}}$-extended surface. By (68), the corresponding TFT state space is

$$\widehat{\mathcal{Z}}_{\mathcal{C}}(T^2) = \mathcal{Z}_{\mathcal{C}}^{\mathrm{RT}}(T^2) \otimes \mathsf{SV}(T^2) \cong \mathcal{C}(\mathbf{1}, L). \tag{126}$$

A basis of this space can be given by solid tori with embedded ribbons. Namely, for $X \in \widehat{\mathcal{C}}$ let $N_X$ be a solid torus with an embedded $X$-labelled ribbon, see Figure 16 b). From the definition of $\widehat{\mathcal{Z}}_{\mathcal{C}}$, one checks that

$$\widehat{\mathcal{Z}}_{\mathcal{C}}(N_{X \otimes K^\epsilon}) = \epsilon \, \widehat{\mathcal{Z}}_{\mathcal{C}}(N_X) \, \in \, \widehat{\mathcal{Z}}_{\mathcal{C}}(T^2). \tag{127}$$

Thus, labelling the central ribbon by the simple objects $S_i \otimes K^+$ and $S_i \otimes K^-$ of $\widehat{\mathcal{C}}$, with $i \in \mathcal{I}$, produces linearly dependent elements in $\widehat{\mathcal{Z}}_{\mathcal{C}}(T^2)$ which differ by a sign. In particular, in giving a basis we can restrict ourselves to simples of the form $S_i = S_i \otimes K^+$, namely we choose

$$\chi_i := \widehat{\mathcal{Z}}_{\mathcal{C}}(N_{S_i}), \quad i \in \mathcal{I}. \tag{128}$$

Let us now think of $T^2$ as a world sheet without field insertions. By (102), the relevant space of blocks is

$$\mathrm{Bl}(T^2) = \widehat{\mathcal{Z}}_{\mathcal{C}}(T^2 \sqcup (T^2)^{\mathrm{rev}}) = \widehat{\mathcal{Z}}_{\mathcal{C}}(T^2) \otimes_{\mathbb{C}} \widehat{\mathcal{Z}}_{\mathcal{C}}((T^2)^{\mathrm{rev}}), \tag{129}$$

which has basis $\left\{\chi_i \otimes \overline{\chi}_j\right\}_{i,j \in \mathcal{I}}$. The overline is to indicate that the boundary $T^2$ is taken with reverse orientation. We would like to compute the coefficients $Z(B)_{ij}$ in

$$\mathrm{Corr}_B^{\mathrm{or}}(T^2) = \sum_{i,j \in \mathcal{I}} Z(B)_{ij} \, \chi_i \otimes_{\mathbb{C}} \overline{\chi}_j \, \in \, \mathrm{Bl}(T^2). \tag{130}$$

The result is:

**Proposition 5.7.** *For $B \in \widehat{\mathcal{C}}$ a symmetric special Frobenius algebra, the coefficients $Z(B)_{ij}$ in* (130) *are given in terms of the multiplicity spaces in* (97) *as*

$$Z(B)_{ij} = \dim \mathcal{M}_{0,+}^{(\mathrm{in})}(S_i, \bar{S}_j; B) - \dim \mathcal{M}_{0,-}^{(\mathrm{in})}(S_i, \bar{S}_j; B). \tag{131}$$

*Proof.* The computation works analogously to [45, Sec. 5.3], but we will go through it in detail to show where the additional parity signs appear.

The basis $\{\chi_i^*\}$ dual to the $\chi_i$ consists of solid tori $T^2 \longrightarrow \emptyset$ with embedded $S_i$-ribbon running in the opposite direction. This gives

$$Z(B)_{ij} = (\chi_i^* \otimes_{\mathbb{C}} \overline{\chi}_j^*) \circ \mathrm{Corr}_B^{\mathrm{or}}(T^2) = \widehat{\mathcal{Z}}_{\mathcal{C}}(C_{i,j}), \tag{132}$$

where $C_{i,j}$ is the three-manifold $S^2 \times S^1$ with embedded ribbon graph as follows:

$$C_{i,j} = \qquad\qquad\qquad\qquad\qquad\qquad\qquad\qquad \text{(133)}$$

Next we expand the identity on $S_i^* \otimes B \otimes S_j^*$ into a sum over simple objects $S_k \otimes K^\epsilon$ of $\widehat{\mathcal{C}}$ by choosing a basis $\{\alpha_{k,\epsilon}\}$ in $\widehat{\mathcal{C}}(S_k \otimes K^\epsilon, S_i^* \otimes B \otimes S_j^*)$ and a corresponding dual basis $\{\bar{\alpha}_{k,\epsilon}\}$ in $\widehat{\mathcal{C}}(S_i^* \otimes B \otimes S_j^*, S_k \otimes K^\epsilon)$. This results in the equality

$$\mathrm{id}_{S_i^* \otimes B \otimes S_j^*} = \sum_{k,\epsilon,\alpha} \alpha_{k,\epsilon} \circ \bar{\alpha}_{k,\epsilon}. \qquad\qquad \text{(134)}$$

Note that here we have to include the sum over $\epsilon$ as $S_k \otimes K^+$ and $S_k \otimes K^-$ are distinct simple objects of $\widehat{\mathcal{C}}$. Inserting the above identity into $C_{i,j}$ gives

$$\widehat{\mathcal{Z}}_{\mathcal{C}}(C_{i,j}) = \sum_{k,\epsilon,\alpha} \widehat{\mathcal{Z}}_{\mathcal{C}}(C_{i,j}^{k,\epsilon,\alpha}), \qquad\qquad \text{(135)}$$

where

$$C_{i,j}^{k,\epsilon,\alpha} = \qquad\qquad\qquad\qquad\qquad\qquad\qquad\qquad \text{(136)}$$

Since $S_k \otimes K^\epsilon$ is simple, there are numbers $\lambda_{k,\epsilon}$ such that

$$\sum_{\alpha} \bar{\alpha}_{k,\epsilon} \circ p \circ \alpha_{k,\epsilon} = \lambda_{k,\epsilon} \cdot \mathrm{id}_{S_k \otimes K^\epsilon}, \qquad\qquad \text{(137)}$$

where $p$ is the endomorphism on $S_i^* \otimes B \otimes S_j^*$ given by

$$p = \qquad\qquad\qquad\qquad\qquad\qquad\qquad\qquad \text{(138)}$$

It is easy to check that $p$ is an idempotent: $p \circ p = p$.

Let $C(X)$ be the manifold $S^2 \times S^1$ with a single ribbon labelled $X$ running along the $S^1$-direction. Then (135) simplifies to

$$\widehat{\mathcal{Z}}_{\mathcal{C}}(C_{i,j}) = \sum_{k,\epsilon} \lambda_{k,\epsilon} \widehat{\mathcal{Z}}_{\mathcal{C}}(C(S_k \otimes K^\epsilon)). \qquad\qquad \text{(139)}$$

The TFT state space on $S^2$ with a single marked point labelled by $S_k \otimes K^\epsilon$ is $K^\epsilon$ for $S_k = \mathbf{1}$ and zero otherwise (cf. (68)). From this, the invariant for $C(S_k \otimes K^\epsilon)$ is obtained as a trace, and one gets (cf. (69)),

$$\widehat{\mathcal{Z}}_{\mathcal{C}}(C(S_k \otimes K^\epsilon)) = \mathcal{Z}_{\mathcal{C}}^{\mathrm{RT}}(C(S_k)) \cdot \mathrm{SV}(C(K^\epsilon)) = \delta_{k,\mathbf{1}} \cdot \epsilon. \qquad\qquad \text{(140)}$$

Putting all this together, we arrive at

$$Z(B)_{ij} = \sum_{\epsilon} \lambda_{\mathbf{1},\epsilon} \cdot \epsilon \,. \tag{141}$$

It remains to show that $\lambda_{\mathbf{1},\epsilon} = \dim \mathcal{M}_{0,\epsilon}^{(\mathrm{in})}(S_i, \bar{S}_j; B)$. To see this, note that $P_\epsilon := p \circ (-)$ acts as an idempotent on $\widehat{\mathcal{C}}(K^\epsilon, S_i^* \otimes B \otimes S_j^*)$. Choosing the basis $\{\alpha_{\mathbf{1},\epsilon}\}$ to consist of eigenvectors shows that $\lambda_{\mathbf{1},\epsilon} = \dim \mathrm{im}(P_\epsilon)$.

In Proposition 6.12 below we will show in a more general setting that there is a linear isomorphism $f : \widehat{\mathcal{C}}(S_i \otimes S_j \otimes K^\epsilon, B) \longrightarrow \widehat{\mathcal{C}}(K^\epsilon, S_i^* \otimes B \otimes S_j^*)$ which satisfies $f \circ Q_0^{(\mathrm{in})} = P_\epsilon \circ f$ with respect to the bulk field projectors given in (89). This completes the proof. $\qquad\square$

**Remark 5.8.** 1. Suppose $\widehat{\mathcal{C}} = \mathrm{Rep}_{\mathcal{S}}(\mathcal{V})$ for a bosonic rational VOA $\mathcal{V}$ as in Section 4.4. As argued in Section 3.3, $\chi_i$ from (128) does represent the character of the $\mathcal{V}$-module $S_i$ (or, more generally, the corresponding torus one-point block with an arbitrary insertion of $\mathcal{V}$). In this situation, the torus amplitude $\mathrm{Corr}_B^{\mathrm{or}}(T^2)$ in (130) does indeed represent the super-trace of $q^{L_0 - c/24}\bar{q}^{\bar{L}_0 - c/24}$ over $\mathcal{H}_{\mathrm{bulk}}^{(\mathrm{in})}$ (possibly with an additional insertion from $\mathcal{V} \otimes_{\mathbb{C}} \overline{\mathcal{V}}$).

2. It was shown in Corollary 5.5 that the correlators $\mathrm{Corr}_B^{\mathrm{or}}(\Sigma)$ are mapping class group invariants. Thus, in particular, the bilinear combination of characters in (130) is $SL(2,\mathbb{Z})$-invariant. This would in general not be true if one took the usual trace over $\mathcal{H}_{\mathrm{bulk}}^{(\mathrm{in})}$ rather than the super-trace. (The usual trace amounts to adding dimensions in (131) rather than subtracting them.) We will see an example of this in Section 7.5 below.

# 6 Spin CFT with parity signs

In this section we combine the ingredients set up so far to describe spin CFTs, that is, CFTs defined on spin surfaces, and which may or may not include parity. While in Section 2 we discussed $r$-spin structures in general, here we restrict ourselves to the case $r = 2$, and we will just say "spin" instead of "2-spin". Furthermore, we will work in the setting with parity, as it is easy to recover the parity-less situation by restricting to the parity-even sector.

We start in Section 6.1 by introducing the $\mathbb{Z}_2$-equivariant modules and bimodules needed to describe boundary conditions and defects of the spin CFT. In Sections 6.2 and 6.3 we give the correlators of the spin CFT in terms of those of an underlying oriented CFT with an appropriate network of additional line defects. The proof that this prescription is monodromy-free and consistent with gluing is given in Section 6.4. As an example, in Section 6.5 we give the relation between (super-)traces over state spaces and the correlators on the torus for the four possible spin structures.

As in Section 5 we fix a modular fusion category $\mathcal{C}$ and set $\widehat{\mathcal{C}} = \mathcal{C} \boxtimes \mathcal{SVect}$.

## 6.1 $\mathbb{Z}_2$-equivariant modules and bimodules

Let $F_1, F_2 \in \widehat{\mathcal{C}}$ be Frobenius algebras with $N_{F_i}^2 = \mathrm{id}_{F_i}$. Consider the endofunctor on $_{F_1}\widehat{\mathcal{C}}_{F_2}$ which sends an $F_1$-$F_2$-bimodule $X$ to the twisted bimodule $_{N_{F_1}}X_{N_{F_2}}$. This defines a strict $\mathbb{Z}_2$-action on $_{F_1}\widehat{\mathcal{C}}_{F_2}$. We denote the category of $\mathbb{Z}_2$-equivariant objects by

$$\mathcal{D}(F_1, F_2) := \left(_{F_1}\widehat{\mathcal{C}}_{F_2}\right)^{\mathbb{Z}_2} \,. \tag{142}$$

In the following we will give an explicit description of $\mathcal{D}(F_1, F_2)$. For more details on equivariantisation see e.g. [33, Sec. 2.7].

A $\mathbb{Z}_2$-*equivariant $F_1$-$F_2$-bimodule* is an $F_1$-$F_2$-bimodule $X \in \widehat{\mathcal{C}}$ together with an involution

$$N_X = \ \ : X \longrightarrow X, \quad N_X^2 = \mathrm{id}_X, \tag{143}$$

which satisfies the following compatibility condition with the $F_i$-actions:

$$\tag{144}$$

As before we use the following notation to denote powers of $N_X$:

$$N_X^m = \ \ \tag{145}$$

A *morphism of $\mathbb{Z}_2$-equivariant bimodules* $X, Y \in \mathcal{D}(F_1, F_2)$ is a morphism $f : X \longrightarrow Y$ which is a morphism of $F_1$-$F_2$-bimodules and which commutes with the $\mathbb{Z}_2$-action:

$$f \circ N_X = N_Y \circ f. \tag{146}$$

Let $F \in \widehat{\mathcal{C}}$ be a Frobenius algebra with $N_F^2 = \mathrm{id}_F$. A $\mathbb{Z}_2$-*equivariant left $F$-module* is a $\mathbb{Z}_2$-equivariant $F$-**1**-bimodule, we write

$$\mathcal{B}(F) := \mathcal{D}(F, \mathbf{1}). \tag{147}$$

Let $F_i$ ($i = 1, 2, 3$) be special Frobenius algebras in $\widehat{\mathcal{C}}$ with $N_{F_i}^2 = 1$. The tensor product of $X \in \mathcal{D}(F_3, F_2)$ and $Y \in \mathcal{D}(F_2, F_1)$ over $F_2$ is defined as in Section 5.1, with $\mathbb{Z}_2$-action given by

$$N_{X \otimes_{F_2} Y} := N_X \otimes_{F_2} N_Y. \tag{148}$$

In terms of the projection and embedding maps for the relative tensor product in (84), this reads

$$N_{X \otimes_{F_2} Y} = \ \ = \ \ \tag{149}$$

For $F_1 = F_2 =: F$, an example of a $\mathbb{Z}_2$-equivariant bimodule is $F$, seen as a bimodule over itself, with $\mathbb{Z}_2$-action given by $N_F$. This furnishes the tensor unit in $\mathcal{D}(F, F)$.

**Duals**

Let $Y \in \mathcal{D}(F_1, F_2)$ and recall the definition of $Y^\dagger$ from (87). Together with the $\mathbb{Z}_2$-action

$$N_{Y^\dagger} = \ \ = \ \ = (N_Y)^*, \tag{150}$$

$Y^\dagger$ becomes an object in $\mathcal{D}(F_2, F_1)$. In fact it turns out that in the $\mathbb{Z}_2$-equivariant setting, this is a two-sided adjoint. To describe the various adjunction maps, first consider the following maps in $\widehat{\mathcal{C}}$:

$$\mathrm{ev}'_Y = \mathrm{ev}_Y \circ (\mathrm{id}_{Y^\dagger} \otimes N_Y^{-1}) \colon Y^\dagger \otimes Y \longrightarrow \mathbf{1}, \quad \mathrm{coev}'_Y = (N_Y \otimes \mathrm{id}_{Y^\dagger}) \circ \mathrm{coev}_Y \colon \mathbf{1} \longrightarrow Y \otimes Y^\dagger,$$

$$\widetilde{\mathrm{ev}}'_Y = \widetilde{\mathrm{ev}}_Y \colon Y \otimes Y^\dagger \longrightarrow \mathbf{1}, \qquad\qquad \widetilde{\mathrm{coev}}'_Y = \widetilde{\mathrm{coev}}_Y \colon \mathbf{1} \longrightarrow Y^\dagger \otimes Y. \tag{151}$$

One can easily check that the idempotent $p$ in (83) for the tensor product $Y \otimes_{F_2} Y^\dagger$ or $Y^\dagger \otimes_{F_1} Y$ can be omitted against the four maps in (151), for example $\widetilde{\mathrm{ev}}'_Y \circ p = \widetilde{\mathrm{ev}}'_Y$. In particular, $\mathrm{ev}'_Y$ and $\widetilde{\mathrm{ev}}'_Y$ are balanced maps and descend to the tensor product over $F_2$. Using the projection and embedding maps from (84), one further checks that the maps

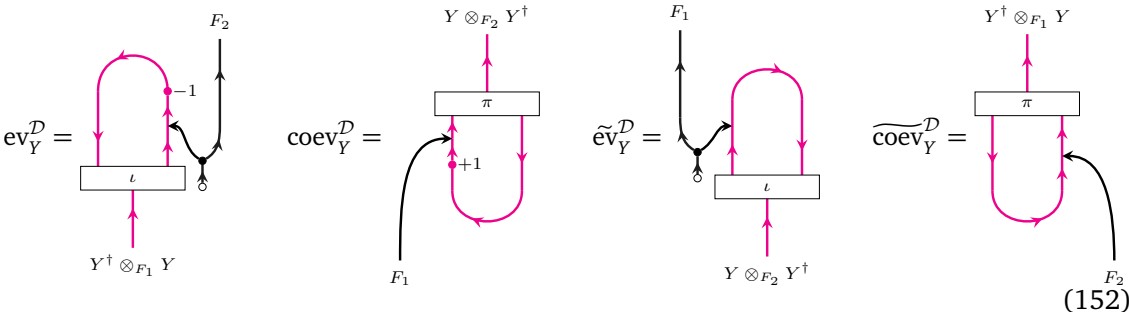

$$\tag{152}$$

are bimodule maps and are compatible with the $\mathbb{Z}_2$-action, i.e. are morphisms in $\mathcal{D}(F_1, F_1)$, respectively $\mathcal{D}(F_2, F_2)$.

We define the dimension function $D$ of a $\mathbb{Z}_2$-equivariant bimodule $Y$ to be

$$\mathrm{D}(Y) := \mathrm{ev}'_Y \circ \widetilde{\mathrm{coev}}'_Y = \phantom{xxxx}_{-1} \overset{(*)}{=} {}_{+1}\phantom{xxxx} = \widetilde{\mathrm{ev}}'_Y \circ \mathrm{coev}'_Y, \tag{153}$$

where in $(*)$ we used that $N_F = N_F^{-1}$ and that $\widehat{\mathcal{C}}$ is spherical. If we apply this definition to $F \in \mathcal{D}(F, F)$ with $F \in \widehat{\mathcal{C}}$ special Frobenius and $N_F^2 = \mathrm{id}_F$, we get

$$\mathrm{D}(F) = \epsilon_F \circ \eta_F \neq 0. \tag{154}$$

This is immediate from taking the trace in $\widehat{\mathcal{C}}$ of the definition of $N_F$ in (81) (and the convention that "special" means "normalised special", cf. Section 5.1).

In case $F$ is simple, the function D in (153) is related to the quantum dimension in $\mathcal{D}(F, F)$ via

$$\dim_{\mathcal{D}(F,F)}(Y) = \frac{D(Y)}{D(F)}. \tag{155}$$

To see this, note that since $F$ is simple, the composition of evaluation and coevaluation in $\mathcal{D}(F, F)$ is a multiple of $\mathrm{id}_F$, which in turn can be computed by inserting both sides in $\epsilon_F \circ (-) \circ \eta_F$.

It follows from (155) and semisimplicity of $\mathcal{D}(F, F)$ that $\mathcal{D}(F, F)$ is spherical.

**Example 6.1.** Consider the case that $\mathcal{C} = \mathcal{V}ect$, so that $\widehat{\mathcal{C}} = \mathcal{SV}ect$. The Clifford algebra $C\ell_1 = \mathbb{C}^{1|1} \in \mathcal{SV}ect$ is generated by one odd element $\theta$ satisfying $\theta^2 = 1$. $C\ell_1$ is a Frobenius algebra with counit and coproduct given by

$$\varepsilon(1) = 2, \quad \varepsilon(\theta) = 0, \quad \Delta(1) = \tfrac{1}{2}(1 \otimes 1 + \theta \otimes \theta), \quad \Delta(\theta) = \tfrac{1}{2}(1 \otimes \theta + \theta \otimes 1), \tag{156}$$

and with these choices $C\ell_1$ is also special. The Nakayama automorphism is given by

$$N(1) = 1, \quad N(\theta) = -\theta. \tag{157}$$

With this, it is immediate that

$$D(C\ell_1) = 2, \tag{158}$$

while the super-dimension (58) is $\dim(C\ell_1) = 0$.

## 6.2 Spin CFT without defects and boundaries

Following [76], in this section, we convert a world sheet with spin structure into an oriented world sheet containing a defect network using the combinatorial model for spin structures reviewed in Section 2. The spin CFT is then defined by evaluating this defect correlator in the oriented CFT. This procedure applies in the same way to theories with and without parity.

We fix a special Frobenius algebra $F \in \widehat{\mathcal{C}}$ with $N_F^2 = \mathrm{id}_F$.

We will describe the construction for spin CFTs and oriented CFTs with parity. The setting without parity is recovered by choosing $F \in \mathcal{C}$ and only using parity label "+" below.

**Decorated spin world sheets**

Let $\Sigma_b = (\Sigma_b, \sigma)$ be a spin bordism with no defects or free boundaries, and hence with only closed gluing boundaries. Let $\Sigma$ be the spin world sheet with ordered in/outgoing marked points obtained by gluing in discs as in Figure 13 a). Then $\Sigma$ becomes a *decorated spin world sheet* if each marked point $p$ is labelled by a tuple

$$(U, \bar{V}, \epsilon, y, \phi), \tag{159}$$

where

- $U, \bar{V} \in \mathcal{C}$ give the holomorphic and antiholomorphic representation of the bulk insertion,

- $\epsilon \in \{\pm\}$ gives the $\mathbb{Z}_2$-parity,

- $y \in \mathbb{Z}_2$ is the type of the gluing boundary and is related to the holonomy of the spin structure around the insertion via (22),

- $\phi \in \mathcal{M}_{y,\epsilon}^{(t)}(U, \bar{V}; F)$, with $t = \mathrm{in}$ if the insertion point is ingoing, and $t = \mathrm{out}$ otherwise (recall (90)). The morphism $\phi$ parametrises the multiplicity space of bulk fields of type $(U, \bar{V}, \epsilon, y)$, see (161) below.

The relation between the type of an insertion and being Neveu-Schwarz (NS) or Ramond (R) is as follows:

| type | name |
|:---:|:---:|
| 0 | NS |
| 1 | R |

$$\tag{160}$$

**Space of bulk fields in the NS and R sector**

As in the oriented case, the multiplicity spaces allow one to read off the space of bulk fields. Namely, the *space of in- and outgoing bulk fields of type $y$ of the spin CFT* is given by

$$\mathcal{H}_{\mathrm{bulk}}^{y,(t)}(F) = \bigoplus_{i,j \in \mathcal{I}, \epsilon \in \{\pm 1\}} S_i \boxtimes \bar{S}_j \boxtimes K^\epsilon \otimes \mathcal{M}_{y,\epsilon}^{(t)}(S_i, \bar{S}_j; F) \in \mathcal{C} \boxtimes \mathcal{C}^{\mathrm{rev}} \boxtimes \mathcal{SV}ect, \tag{161}$$

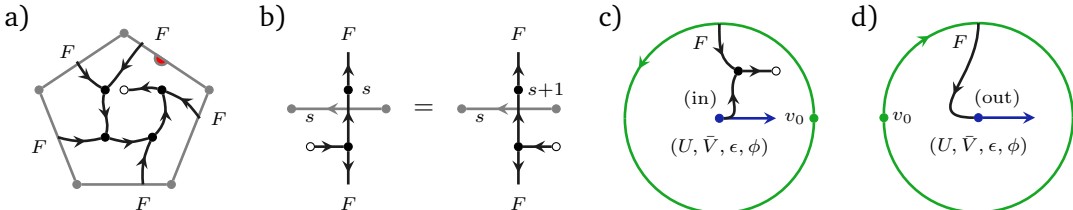

Figure 25: Building blocks of the spin defect network: a) defect graph inside a plaquette, b) defect graph near an edge. c) and d) give the defect graphs to be placed on the discs glued to in- and out-going closed gluing boundaries. Note the position of $v_0$ in d): the disc is rotated by 180° relative to the disc in c).

where $t \in \{\text{in}, \text{out}\}$, and $\mathcal{I}$ is the index set for simple objects in $\mathcal{C}$ as in (37). If we also split the state space according to even/odd parity, we obtain (restricting to the ingoing case)

$$
\mathcal{H}^{\text{NS,even}}_{\text{bulk}}(F) = \bigoplus_{i,j \in \mathcal{I}} S_i \boxtimes \bar{S}_j \boxtimes K^+ \otimes \mathcal{M}^{(\text{in})}_{0,+}(S_i, \bar{S}_j; F),
$$

$$
\mathcal{H}^{\text{NS,odd}}_{\text{bulk}}(F) = \bigoplus_{i,j \in \mathcal{I}} S_i \boxtimes \bar{S}_j \boxtimes K^- \otimes \mathcal{M}^{(\text{in})}_{0,-}(S_i, \bar{S}_j; F),
$$

$$
\mathcal{H}^{\text{R,even}}_{\text{bulk}}(F) = \bigoplus_{i,j \in \mathcal{I}} S_i \boxtimes \bar{S}_j \boxtimes K^+ \otimes \mathcal{M}^{(\text{in})}_{1,+}(S_i, \bar{S}_j; F),
$$

$$
\mathcal{H}^{\text{R,odd}}_{\text{bulk}}(F) = \bigoplus_{i,j \in \mathcal{I}} S_i \boxtimes \bar{S}_j \boxtimes K^- \otimes \mathcal{M}^{(\text{in})}_{1,-}(S_i, \bar{S}_j; F). \tag{162}
$$

**Translation into an oriented world sheet with $F$-defects**

Let $\Sigma$ be a decorated spin world sheet (without boundaries and defects) as above. From $\Sigma$ we will construct an oriented world sheet with defects

$$
W_{\text{or}}(\Sigma; F), \tag{163}
$$

for the symmetric special Frobenius algebra $B = \mathbf{1} \in \widehat{\mathcal{C}}$. Note that defects in the oriented CFT defined by $B = \mathbf{1}$ as in Section 5 are described simply by objects in $\widehat{\mathcal{C}}$. In particular, $F$ itself defines a defect for the oriented $B = \mathbf{1}$ theory.

Let $\Sigma_b = (\Sigma_b, \sigma)$ be the spin bordisms from which $\Sigma$ was defined. Let $T_{\Sigma_b}(\sigma)$ be the combinatorial presentation of the spin structure $\sigma$ by a marked polygonal decomposition. The defect graph on $W_{\text{or}}(\Sigma; F)$ is obtained from $T_{\Sigma_b}(\sigma)$ as follows:

- For each plaquette of $T_{\Sigma_b}(\sigma)$ place the $F$-defect graph shown in Figure 25 a) on $W_{\text{or}}(\Sigma; F)$, and for each edge the graph shown in Figure 25 b).

- The marked points in $W_{\text{or}}(\Sigma; F)$ are as for $\Sigma$, except that one forgets the type,

$$
(U, \bar{V}, \epsilon, y, \phi) \quad \rightsquigarrow \quad (U, \bar{V}, \epsilon, \phi), \tag{164}
$$

  which is of the form (96). To see that $\phi$ is indeed a label for the oriented theory, note that for $B = \mathbf{1}$, the inclusions in (91) (with $X = F$) are actually equalities.

- For each disc glued to the bordism $\Sigma_b$ to obtain the world sheet $\Sigma$ place the $F$-defect graph shown in Figure 25 c) and d), depending on whether the gluing boundary was in- or outgoing.

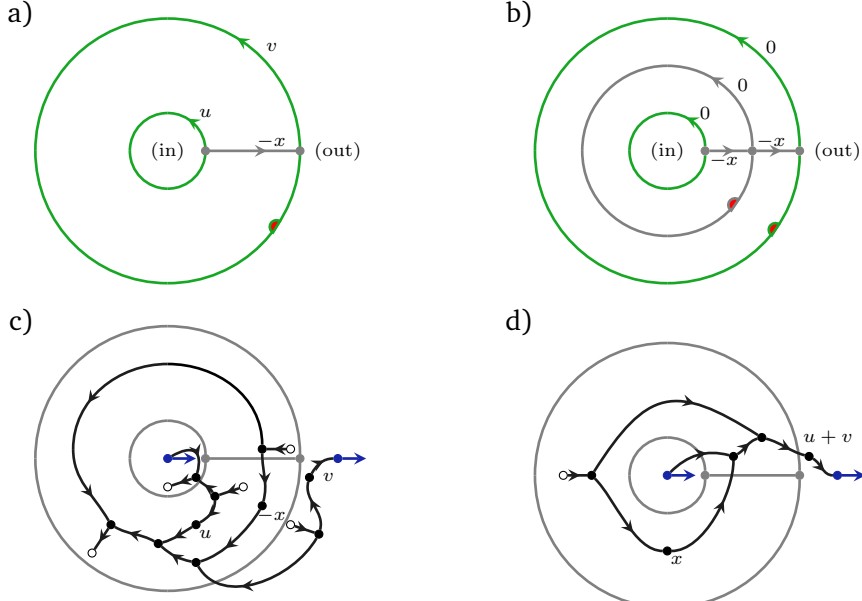

Figure 26: a) A spin cylinder over $S_x^1$ with $x \in \mathbb{Z}_2$ and $u, v \in \mathbb{Z}_2$. b) Two spin cylinders with $u = v = 0$ glued together. This can be simplified to the cylinder in a) with $u = v = 0$. c) The defect network on the world sheet corresponding to a) obtained by gluing discs with field insertions. d) Simplification of the defect network which still gives the same correlator as c).

Two examples of spin bordisms and associated decorated world sheets in the oriented defect CFT for $B = \mathbf{1}$ are shown in Figures 26 and 27.

- *Figure 26:* Part a) shows a cylinder with one in-going and one out-going closed gluing boundary of type $x$. The corresponding defect graph is shown in part c). Below it will be shown that the rules in Figure 28 can be used to simplify defect graphs, and in the present case the result is shown in part d). For $u = v = 0$ the cylinder is the identity in the bordism category. This can be seen in an example in part b) where two such cylinders have been glued together. One can remove the middle loop using (M4) and then the middle vertex by (M5), resulting in Part a) with $u = v = 0$. Indeed, Part d) for $u = v = 0$ is just the projector $Q_x^{(\text{in})}$ which acts as the identity on a bulk insertion of type $x$ labelled with $\phi \in \mathcal{M}_{x,\epsilon}^{(in)}(U, \bar{V}; F)$ as in (159).

- *Figure 27:* Part a) shows a pair of pants with spin structure. This particular bordism and the resulting world sheet determine the OPE of bulk fields in the spin CFT in terms of those of defect fields in the oriented CFT. Note that the resulting $F$-defect graph can be deformed to agree with the one used in [82, Eq. 2.5].

**Remark 6.2.** Unless we say otherwise, when rearranging the defect networks obtained from the combinatorial spin structure, we will not make use of the fact that the edge indices take values in $\mathbb{Z}_2$, but instead treat them as valued in $\mathbb{Z}$. For example, in Figures 26 and 32 we distinguish between $x$ and $-x$. The reason for this is two-fold. One the one hand it makes the computations easier to follow, and on the other hand this already sets the stage for the discussion of $r$-spin CFTs to which we hope to return in the future.

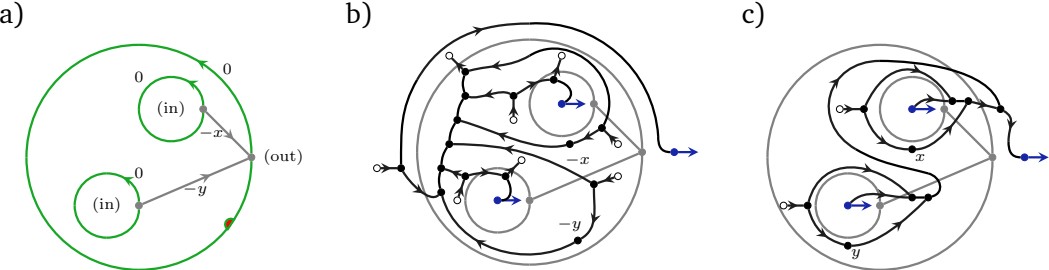

Figure 27: a) A spin pair of pants with one outgoing and two incoming closed gluing boundaries (of types $x$ and $y$, respectively) and combinatorial description of the spin structure. b) The corresponding defect network. c) Simplified network which gives the same correlator.

**Correlators of the spin CFT**

Let $\Sigma$ be a decorated spin world sheet and let $W_{\mathrm{or}}(\Sigma; F)$ be the corresponding world sheet for the oriented CFT with $B = \mathbf{1}$. The space of conformal blocks for $\Sigma$ and the correlators $\mathrm{Corr}_F^{\mathrm{spin}}(\Sigma)$ of the spin CFT for $F$ are defined in terms of those of the oriented theory for $B = \mathbf{1}$ given in (103) and (104) as

$$\mathrm{Bl}(\Sigma) := \mathrm{Bl}(W_{\mathrm{or}}(\Sigma; F)),$$
$$\mathrm{Corr}_F^{\mathrm{spin}}(\Sigma) := \mathrm{Corr}_{B=\mathbf{1}}^{\mathrm{or}}(W_{\mathrm{or}}(\Sigma; F)) \in \mathrm{Bl}(\Sigma). \tag{165}$$

We will show in Proposition 6.5 that $\mathrm{Corr}_F^{\mathrm{spin}}(\Sigma)$ is independent of the choice of $T_{\Sigma_b}(\sigma)$, and in Section 6.4 that it is compatible with transport and gluing.

**Manipulating $F$-defects in the oriented CFT**

The network of $F$-defects in $W_{\mathrm{or}}(\Sigma; F)$ can usually be simplified further. A collection of useful identities when working with networks of $F$-defects is given in Figure 28. The identities given in [82, Fig. 1] agree with those in Figure 28. Figures 26 d) and 27 c) provide examples of how the rules in Figure 28 can be used to simplify $F$-networks. That these rules indeed hold is the content of the next proposition.

**Proposition 6.3.** *Let $\Sigma$ and $\Sigma'$ be oriented world sheets with $F$-labelled defect networks which only differ locally in one place in the way shown in Figure 28. Then $\mathrm{Corr}_{B=\mathbf{1}}^{\mathrm{or}}(\Sigma) = \mathrm{Corr}_{B=\mathbf{1}}^{\mathrm{or}}(\Sigma')$.*

*Proof.* By the definition of $\mathrm{Corr}_{B=\mathbf{1}}^{\mathrm{or}}(\Sigma)$ in (112) we have to show $\widehat{\mathcal{Z}}_{\mathcal{C}}(M_\Sigma) = \widehat{\mathcal{Z}}_{\mathcal{C}}(M_{\Sigma'})$. The resulting identities for ribbon graphs in the connecting manifold follow immediately from the defining properties of the special Frobenius algebra $F$. Let us nonetheless briefly comment on two identities in part e) of Figure 28:

- The first identity in e) follows since the ribbon graph in $M_\Sigma$ near the field insertion will be precisely the projector $Q_y^{(\mathrm{in})}$ from (89), and by construction (see (159)) the field $\phi$ lies in $\mathcal{M}_{x,\epsilon}^{(\mathrm{in})}(U, \bar{V}; F) = \mathrm{im}\, Q_x^{(\mathrm{in})}$, cf. (90).

- The second equality follows by inserting the projector $Q_x^{(\mathrm{in})}$ via first identity in e) and using the other rules listed to unwind the $F$-line.

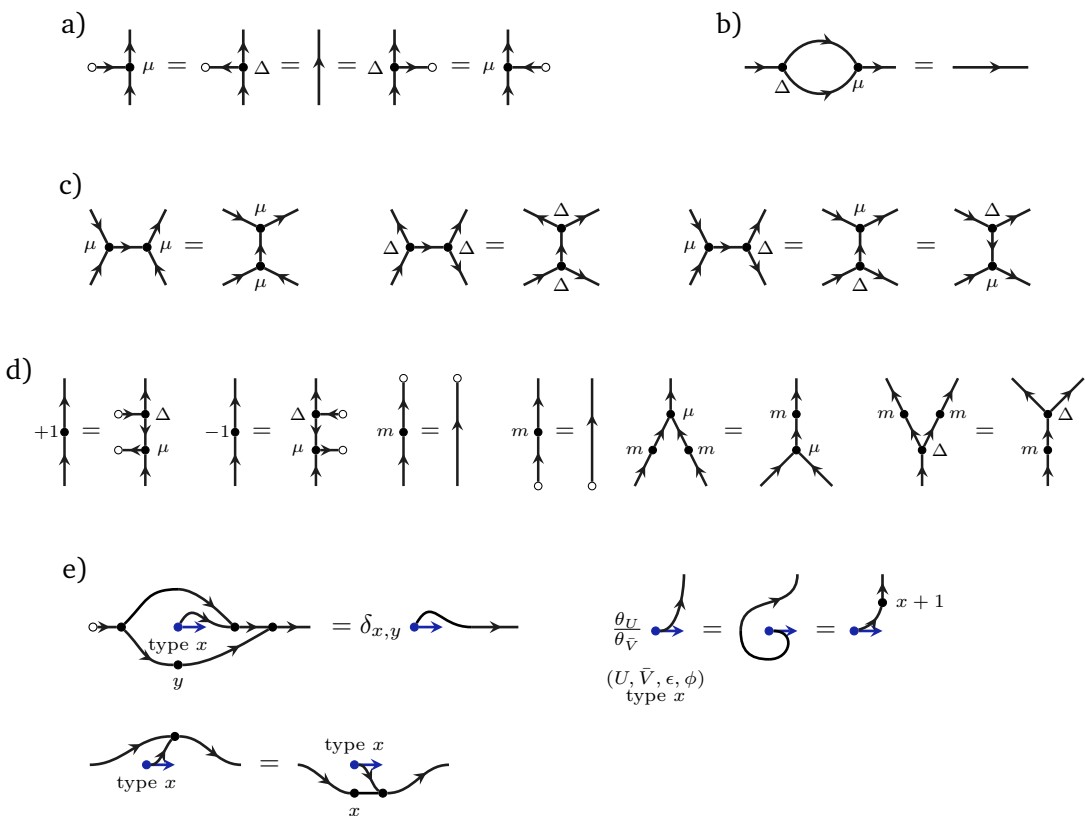

Figure 28: Local modification of a network of $F$-defects that leave the correlator $\mathrm{Corr}^{\mathrm{or}}_{B=1}$ unchanged: a) properties of the unit and counit, b) bubble move, c) associativity conditions, d) properties of the Nakayama automorphism, e) moves involving ingoing bulk field insertions.

- In the third identity in e), the first equality uses two ways to express the $2\pi$-rotation of the tangent vector in the connecting manifold:

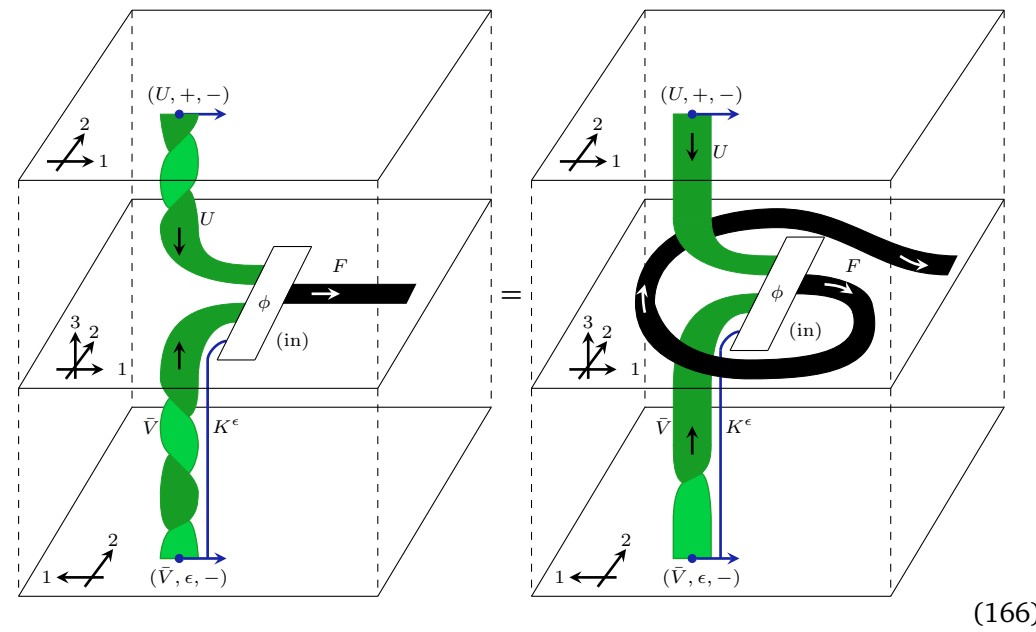

$$(166)$$

□

**Remark 6.4.** When using the associativity moves in Figure 28 e) one has to be careful that one does not accidentally replace any of the following configurations by any of the others as they are inequivalent:

$$\text{(167)}$$

Indeed, if $N_F \neq$ id, i.e. if $F$ is not symmetric, the four local configurations shown above are pairwise distinct. This can be seen by inserting unit and counit in appropriate places. For example, if in the first two diagrams, one attaches a unit to the ingoing top $F$-line and a counit to the outgoing top $F$-line, the first diagram becomes $\text{id}_F$ and the second $N_F^{-1}$.

### 6.3   Including boundaries and defects

Here we extend the definition of spin CFT in terms of the oriented defect CFT to include boundaries and defects. The procedure is the same, but involves extra notation.

**Decorated spin world sheets**

As before, let $\Sigma_b = (\Sigma_b, \sigma)$ be a spin bordism, but now possibly with defects and boundaries. Let $\Sigma$ be the corresponding spin world sheet obtained by gluing in discs or half-discs as in Figure 13. Then $\Sigma$ becomes a *decorated spin world sheet* as follows:

- *(Boundary condition)* A connected component of the boundary minus the boundary insertion points gets labelled by a $\mathbb{Z}_2$-equivariant left $F$-module, i.e. by an object in $\mathcal{B}(F)$, cf. (147).

- *(Defect condition)* A connected component of the defect network minus field insertions gets labelled by a $\mathbb{Z}_2$-equivariant $F$-$F$-bimodule, i.e. by an object in $\mathcal{D}(F, F)$, cf. (142).

- Consider a boundary insertion between boundary conditions $M, N$, and on which defect lines with defect conditions $X_1, \dots, X_m$ start or end (with $m = 0$ corresponding to no attached defects). Let $\delta_i = +$ if $X_i$ is pointing away from the insertion, and $\delta_i = -$ otherwise. Set

$$X = X_1^{\delta_i} \otimes_F \cdots \otimes_F X_m^{\delta_m} \in \mathcal{D}(F, F), \quad \text{where} \quad X_i^+ = X_i, \text{ and } X_i^- = X_i^\dagger. \quad \text{(168)}$$

  The boundary insertion is labelled by

$$(W, \epsilon, \psi). \quad \text{(169)}$$

  Here, $W \in \mathcal{C}$, $\epsilon \in \{\pm 1\}$ is the parity, and $\psi \in \widehat{\mathcal{C}}(W \otimes K^\epsilon, M^\dagger \otimes_F X \otimes_F N)$ (ingoing), respectively $\psi \in \widehat{\mathcal{C}}(M^\dagger \otimes_F X \otimes_F N, W \otimes K^\epsilon)$ (outgoing).

- The label of a bulk insertion which is not connected to any line defects has already been described in (159).

- Consider a bulk insertion on which defect lines with defect conditions $X_1, \dots, X_m$ start or end, and let $X$ be as in (168). The insertion is again labelled as in (159), but now $\phi \in \mathcal{M}_{y,\epsilon}^{(t)}(U, \bar{V}; X)$, where $t \in \{\text{in, out}\}$ and the multiplicity spaces are as given in (90).

After introducing the defect graph obtained from a combinatorial spin structure below, we can comment on why the dagger-dual $X^\dagger$ or $M^\dagger$ from (152) is the relevant dual for defect labels $X$ and boundary labels $M$ attached to components pointing towards the field insertion, rather than, say, the dual $X^*$ or $M^*$ in $\widehat{\mathcal{C}}$, cf. Example 6.6.

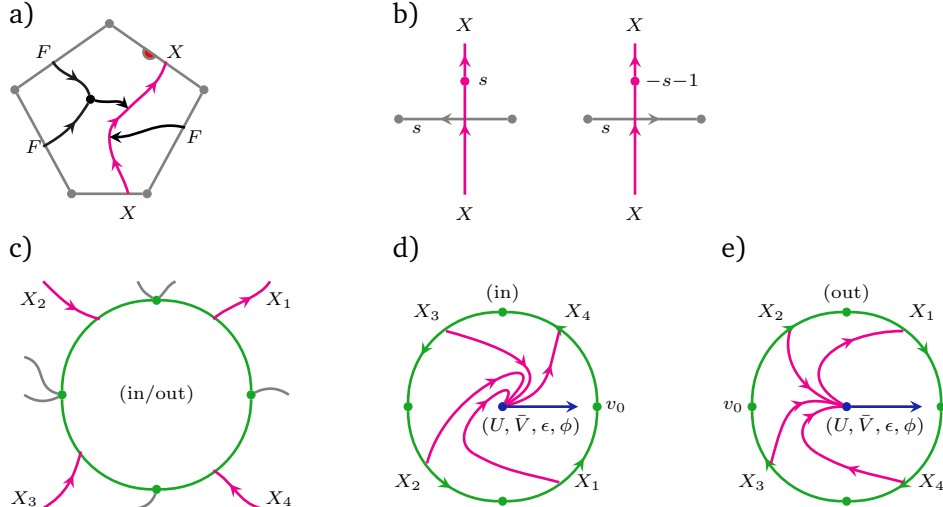

Figure 29: Details of the defect network in $W_{or}(\Sigma;F)$ presenting the combinatorial spin structure for a spin world sheet with defect lines and closed gluing boundaries.

**Spaces of boundary and defect fields**

For the space of boundary fields we get the same as in (107), but with '$\otimes_F$' in place of '$\otimes_B$'. For the bulk defect insertions we have, with $X$ as in (168),

$$\mathcal{H}_{bulk}^{y,(t)}(X) = \bigoplus_{i,j\in\mathcal{I},\epsilon\in\{\pm1\}} S_i \boxtimes \bar{S}_j \boxtimes K^\epsilon \otimes \mathcal{M}_{y,\epsilon}^{(t)}(S_i,\bar{S}_j;X) \in \mathcal{C}\boxtimes\mathcal{C}^{rev}\boxtimes\mathcal{SV}ect. \qquad (170)$$

As in the oriented case, setting $X = F$ in the above state space recovers the space of bulk insertions without attached line defects as in (161).

**Translation into an oriented world sheet with $F$-defects**

Let $\Sigma$ be a decorated spin world sheet, possibly with boundaries and defects. From $\Sigma$ we will construct an oriented world sheet with defects $W_{or}(\Sigma;F)$ for the symmetric special Frobenius algebra $B = \mathbf{1}\in\hat{\mathcal{C}}$.

Let $\Sigma_b = (\Sigma_b,\sigma)$ be the spin bordisms from which $\Sigma$ was defined. Let $T_{\Sigma_b}(\sigma)$ be the combinatorial presentation of the spin structure $\sigma$ by a marked polygonal decomposition, subject to the constraints arising from the presence of defects and boundaries described in Section 2.3.

The defect graph on $W_{or}(\Sigma;F)$ is obtained from $T_{\Sigma_b}(\sigma)$ as follows:

- For plaquettes of $T_{\Sigma_b}(\sigma)$ not containing defects or boundaries, and for insertions in the interior of $\Sigma$ without attached line defects, the defect graph in $W_{or}(\Sigma;F)$ is as in Section 6.2.

- For each plaquette or edge intersecting the defect network of $\Sigma$, place the defect graph shown in Figure 29 a) or b) on $W_{or}(\Sigma;F)$.

- Consider a closed gluing boundary in $\Sigma_b$ with attached defect lines as shown in Figure 29 c). On the disc glued to this boundary in order to obtain $\Sigma$, place the defect graph shown in Figure 29 d) and e).

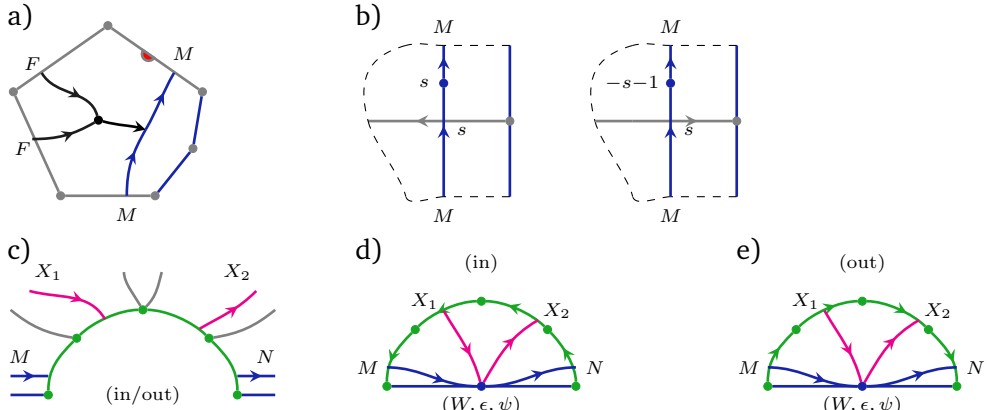

Figure 30: Details of the defect network in $W_{\text{or}}(\Sigma; F)$ for a spin world sheet with free boundaries and open gluing boundaries.

- For each plaquette intersecting the free boundary of $\Sigma$, place the defect graph shown in Figure 30 a), and for each edge starting or ending on the free boundary of $\Sigma$ the graph in Figure 30 b). Recall from Section 2.3 that there may be several edges lying on the boundary, as long as they are in one sequence and the plaquette has at least one non-boundary edge. E.g. in Figure 30 a) there are two boundary edges.

- Consider an open gluing boundary in $\Sigma_b$, possibly with attached defect lines as shown in Figure 30 c) (the case $n = 0$ amounts to no attached defect lines). On the half-disc glued to this gluing boundary, place the defect graph shown in Figure 30 d) and e).

**Correlators of the spin CFT with defects and boundaries**

Let $\Sigma$ be a decorated spin world sheet and let $W_{\text{or}}(\Sigma; F)$ be the corresponding world sheet for the oriented CFT with $B = \mathbf{1}$. The space of conformal blocks for $\Sigma$ and the correlators $\text{Corr}_F^{\text{spin}}(\Sigma)$ of the spin CFT for $F$ are defined in terms of those of the oriented theory for $B = \mathbf{1}$ by the same expression as in (165), in particular

$$\text{Corr}_F^{\text{spin}}(\Sigma) := \text{Corr}_{B=\mathbf{1}}^{\text{or}}(W_{\text{or}}(\Sigma; F)). \tag{171}$$

The construction of $W_{\text{or}}(\Sigma; F)$ involved choosing $T_{\Sigma_b}$ to encode the spin structure. The following proposition states that $\text{Corr}_F^{\text{spin}}(\Sigma)$ does not depend on that choice. The proof will be given in Appendix A.2.

**Proposition 6.5.** *Let $\Sigma$ and $\Sigma'$ be choices for $W_{\text{or}}(\Sigma; F)$ which differ in the marked polygonal decomposition used to encode the spin structure. Then*

$$\text{Corr}_{B=\mathbf{1}}^{\text{or}}(\Sigma) = \text{Corr}_{B=\mathbf{1}}^{\text{or}}(\Sigma'). \tag{172}$$

We will show in the next section that $\text{Corr}_F^{\text{spin}}(\Sigma)$ is compatible with transport and gluing.

**Example 6.6.** 1. As a first example, let us explain why the dagger-dual is used in the multiplicity space for defects or boundary components pointing into a field insertion. Namely, one can use the combinatorial presentation of the spin structure in terms of line defects to produce projectors on the multiplicity space, and one finds in this way the projector for the dagger-dual. We illustrate this in the simplest example, namely a boundary insertion without line defects, in Figure 31.

a)  b)

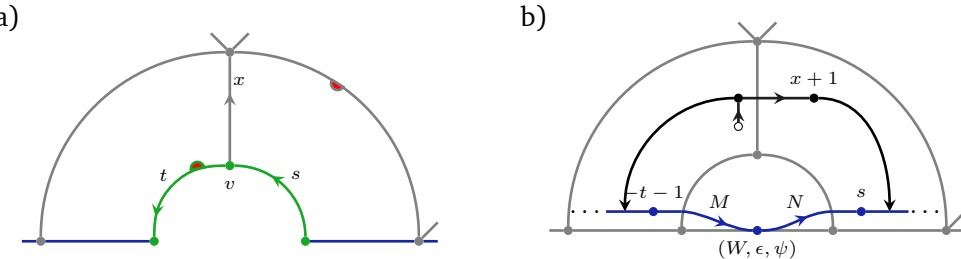

Figure 31: a) Example of a polygonal decomposition near a boundary field insertion on $\Sigma$. The admissibility condition at $v$ is $-s-1+x+t = 0-3+1$, i.e. $x = s-t-1$. b) The corresponding defect network on $W_{\mathrm{or}}(\Sigma; F)$. Taking the action past the isomorphisms $N_N^s$ and $N_M^{-t-1}$ and substituting the value for $x$ produces precisely the projector onto the tensor product $M^\dagger \otimes_F N$, cf. (83) and (87).

2. As an example involving defects, in Figure 32, a contractible defect loop labelled by $X \in \mathcal{D}(F,F)$ is considered. We see that for simple $F$ an $X$-labelled defect loop is equal to the quantum dimension of $X$ in $\mathcal{D}(F,F)$, cf. (155) (and not to the quantum dimension of $X$ in $\widehat{\mathcal{C}}$).

**Remark 6.7.** 1. If $N_F = \mathrm{id}_F$, the special Frobenius algebra $F$ still is a valid choice to define correlators $\mathrm{Corr}_F^{\mathrm{spin}}(\Sigma)$. However, in this case, the value of $\mathrm{Corr}_F^{\mathrm{spin}}(\Sigma)$ is independent of the spin structure, e.g. the NS- and R-state spaces in (162) are the same.

However, conversely, $N_F \neq \mathrm{id}_F$ does not guarantee that the theory detects spin structures. Namely, $N_F$ could be an inner automorphism of $F$, see [75, Sec. 4.10] for an example in 2d TFT.

A more invariant formulation would be in terms of the Serre-automorphism which is implemented by the twisted bimodule $F_{N_F}$. This is isomorphic to $F$ as a bimodule iff $N_F$ is inner. See [24, Sec. 4.1] for more on Serre automorphism and on fully extended $r$-spin TFTs.

2. The formalism presented here also allows one to treat defects between spin CFTs and oriented CFTs. This can be done by treating the interface-defect between the two theories as a boundary as far as the spin structure is concerned (i.e. no edge labels for edges lying entirely in parts of the world sheet which support the oriented CFT). If the algebras $B$ and $F$ describe the oriented and spin CFTs, respectively, then the defects are decorated by bimodules in $\mathcal{D}(B,F)$ or $\mathcal{D}(F,B)$, depending on the defect-orientation. Note that these are still $\mathbb{Z}_2$-equivariant bimodules, but since $N_B = \mathrm{id}_B$, for $X \in \mathcal{D}(B,F)$ (or $X \in \mathcal{D}(F,B)$) the involution $N_X$ is now an intertwiner for the $B$-action.

Let $X \in \mathcal{D}(F,F)$ and write $\mathcal{M}_{y,\epsilon}^{(\mathrm{in})}(U,\bar{V};X)^{\mathrm{inv}}$ for the subspace of $\mathcal{M}_{y,\epsilon}^{(\mathrm{in})}(U,\bar{V};X)$ on which composing with $N_X$ acts as the identity, i.e. for the invariant subspace with respect to the $\mathbb{Z}_2$-action generated by $N_X \circ (-)$. It is not hard to show that

$$\mathcal{M}_{y,\epsilon}^{(\mathrm{in})}(\mathbf{1},\bar{\mathbf{1}};X^\dagger \otimes_F X)^{\mathrm{inv}} \;\cong\; \mathrm{Hom}_{\mathcal{D}(F,F)}(X \otimes K^\epsilon, X_{N^y}). \tag{173}$$

Here, the right action of $F$ on $X \otimes K^\epsilon$ is given by first taking $F$ past $K^\epsilon$ with the symmetric braiding, and the $\mathbb{Z}_2$-equivariant structure is simply $N_X \otimes \mathrm{id}_{K^\epsilon}$.

The space $\mathcal{M}_{y,\epsilon}^{(\mathrm{in})}(\mathbf{1},\bar{\mathbf{1}};X^\dagger \otimes_F X)$ describes the multiplicities of weight $(0,0)$ fields of parity $\epsilon$ in sector $y$ on the defect $X$. Equation (173) shows that we can read off the multiplicity of

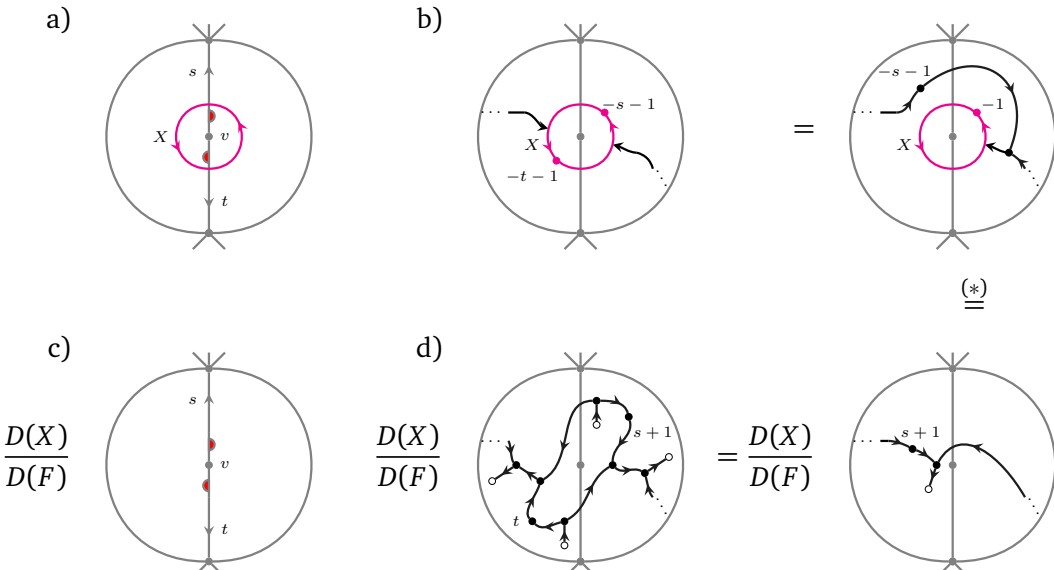

Figure 32: a) Part of a spin world sheet $\Sigma$ with combinatorial spin structure near a vertex $v$ surrounded by a defect loop. The admissibility condition (26) at the vertex $v$ is $s + t = 0 - 2 + 1$. b) The corresponding defect network on the oriented world sheet $W_{or}(\Sigma; F)$ and a reformulation. c) The same local part of $\Sigma$, but without the defect loop and (the resulting correlator is) multiplied by the factor $D(X)/D(F)$. d) Corresponding defect network and simplification. The equality $(*)$ holds under the assumption that $F$ is simple, see (155).

such fields which are in addition $\mathbb{Z}_2$-invariant from an appropriate Hom-space in the pivotal fusion category $\mathcal{D}(F, F)$ labelling the line defects.

Suppose now that $X$ is simple in $\mathcal{D}(F, F)$. Then also $X \otimes K^\epsilon$ and $X_{N^y}$ are simple. Thus in each of the four cases $\epsilon \in \{\pm 1\}$ and $y \in \{0, 1\}$, the spaces in (173) are either zero– or one-dimensional. For $\epsilon = +$ (parity even) and $y = 0$ (NS-sector), the right hand side of (173) is $\mathrm{Hom}_{\mathcal{D}(F,F)}(X, X)$, which is always one-dimensional for simple $X$.

Some of the simple defects may allow for parity-odd weight $(0, 0)$ fields. Of particular interest is the case $\epsilon = -$ and $y = 0$, i.e. the space $\mathrm{Hom}_{\mathcal{D}(F,F)}(X \otimes K^\epsilon, X)$. Denote its dimension by $d$. Then the algebra of $\mathbb{Z}_2$-invariant NS-sector weight $(0, 0)$ fields on $X$ is isomorphic to

- $\mathbb{C}$ if $d = 0$, and

- $C\ell_1$, the Clifford algebra in one odd generator, if $d = 1$.

We stress that this analysis applies to theories of type ② and of type ④ alike, but for type ② one can restrict to $y = 0$.

**Remark 6.8.** 1. In the setting of fermionic topological phases of matter, it was observed in [2] that there are two kinds of simple objects in the relevant categories (called super-pivotal categories there), namely those with endomorphism algebra $\mathbb{C}$ and those with $C\ell_1$. The former were called m-type, and the latter q-type.

We expect that in the way indicated above, the pivotal fusion category $\mathcal{D}(F, F)$ already contains the information necessary to enrich it to a super-pivotal fusion category. A more detailed study of $\mathcal{D}(F, F)$ will be presented in a future publication.

2. In the context of boundary conditions for the fermionic Ising model, i.e. the massless free fermion, the fact that some elementary boundary conditions carry odd weight zero fields was observed in [25,55]. In [82] topological defects in spin CFTs were described by representations of a classifying algebra, and it was noted that they come in two classes, depending on whether they carry a fermionic weight-zero field or not. In [16], these topological defects are called m-type and q-type, respectively, and their categorical properties have been studied.

## 6.4 Consistency of spin correlators

Given a special Frobenius algebra $F \in \widehat{\mathcal{C}}$ with $N_F^2 = $ id, the above construction produces a family of vectors $\{\mathrm{Corr}_F^{\mathrm{spin}}(\Sigma)\}_{\underline{\Sigma}}$. In this section we will verify the consistency conditions (T) and (G) from Section 5.4 for this family of spin correlators. However, rather than directly using the definition in terms of 3d TFT as we did in Section 5.4, here we reduce the statement to the oriented case with defects, for which (F) and (G) have already been shown.

**Transport**

Let $\gamma$ be a path in decorated spin world sheets with defects and boundaries. For the spin structure we use the combinatorial model from Section 2.3. That is, the underlying bordism is equipped with an admissibly marked polygonal decomposition which varies continuously along the path in the sense that the path descends to a path in the space of pairs $(\Sigma_b, T_{\Sigma_b}(o, m, s))$, where $\Sigma_b$ is an open-closed bordism with defects, and $T_{\Sigma_b}(o, m, s)$ is an admissibly marked polygonal decomposition.

In other words, we consider a family $E_\gamma$ whose fibre $\gamma(t)$ over a point $t \in [0,1]$ is a decorated spin world sheet with combinatorial description of its spin structure. By taking the double of $W_{\mathrm{or}}(\gamma(t); F)$ at each fibre, we obtain a family $E_{\widetilde{\gamma}}$ of $\widehat{\mathcal{C}}$-extended surfaces which we think of as a bordism from the double of $W_{\mathrm{or}}(\gamma(0); F)$ to the double of $W_{\mathrm{or}}(\gamma(1); F)$.

In terms of these ingredients, consistency with transport amounts to the following statement:

**Theorem 6.9.** *Let $F \in \widehat{\mathcal{C}}$ be a special Frobenius algebra with $N_F^2 = $ id, and let $E_\gamma$ be a family of decorated spin world sheets from $\underline{\Sigma}$ to $\underline{\Sigma}'$. The correlators for $F$ satisfy*

$$\widehat{\mathcal{Z}}_{\mathcal{C}}(E_{\widetilde{\gamma}})(\mathrm{Corr}_F^{\mathrm{spin}}(\underline{\Sigma})) = \mathrm{Corr}_F^{\mathrm{spin}}(\underline{\Sigma}'). \tag{174}$$

*Proof.* From the definition in (171) we see that we need to show $\mathrm{Corr}_{B=\mathbf{1}}^{\mathrm{or}}(\Sigma) = \mathrm{Corr}_{B=\mathbf{1}}^{\mathrm{or}}(\Sigma')$, where $\Sigma = W_{\mathrm{or}}(\gamma(0); F)$ and $\Sigma' = W_{\mathrm{or}}(\gamma(1); F)$. But this follows from applying Theorem 5.4 to the path $W_{\mathrm{or}}(\gamma(t); F)$. $\qquad\square$

As in Corollary 5.5 in the oriented case, compatibility with transport implies mapping class group(oid) invariance: Let $f : \underline{\Sigma} \longrightarrow \underline{\Sigma}'$ be an isomorphism of decorated spin world sheets. This induces a diffeomorphism $\widetilde{f} : \widetilde{\Sigma} \longrightarrow \widetilde{\Sigma}'$ between the corresponding doubles of $W_{\mathrm{or}}(\underline{\Sigma}; F)$ and $W_{\mathrm{or}}(\underline{\Sigma}'; F)$, respectively. Note that $\widetilde{\Sigma}$ and $\widetilde{\Sigma}'$ do not depend on the choice of polygonal decomposition. Let $E_{\widetilde{f}}$ be the mapping cylinder for $\widetilde{f}$.

**Corollary 6.10.** *We have*

$$\widehat{\mathcal{Z}}_{\mathcal{C}}(E_{\widetilde{f}})(\mathrm{Corr}_F^{\mathrm{spin}}(\underline{\Sigma})) = \mathrm{Corr}_F^{\mathrm{spin}}(\underline{\Sigma}'). \tag{175}$$

**Gluing**

As in the oriented case, we need to choose basis / dual basis pairs in the multiplicity spaces. For boundary fields we can take the choice made in (117) (with $F$ in place of $B$). For bulk

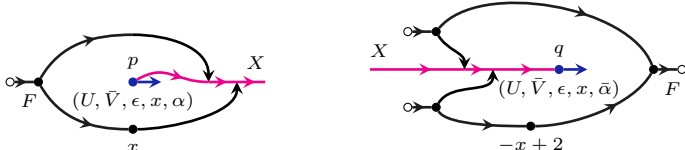

Figure 33: Surface $\Sigma(U, \bar{V}, \epsilon, x, \alpha, \bar{\alpha})$ with $F$-defect network at the in- and outgoing insertion points $p$ and $q$.

fields we need to take into account their type $x \in \{0, 1\}$, namely we choose bases

$$\{\alpha\} \subset \mathcal{M}^{(\text{in})}_{x,\epsilon}(U, \bar{V}; X), \quad \text{and} \quad \{\bar{\beta}\} \subset \mathcal{M}^{(\text{out})}_{x,\epsilon}(U, \bar{V}; X), \tag{176}$$

which are dual to each other with respect to the pairing (92).

Let $\Sigma$ be a decorated spin world sheet for $F$ and $\Sigma_b$ the underlying open-closed spin bordism with defects. Let $b_{\text{in}} \in \partial^t_{\text{in}} \Sigma_b$ and $b_{\text{out}} \in \partial^t_{\text{out}} \Sigma_b$, $t \in \{\text{c, o}\}$ be two open or two closed gluing boundaries. We require that they are parametrised by the same (half-)annulus with defects, and that in the closed case they are of the same type $x$, cf. (160). These conditions ensure that they can be consistently glued.

As in the oriented case, we write

$$\text{closed:} \ \Sigma(U, \bar{V}, \epsilon, \alpha, \bar{\beta}), \quad \text{open:} \ \Sigma(W, \epsilon, \alpha, \bar{\beta}), \tag{177}$$

for the world sheet $\Sigma$ where the puncture of the (half-)disc glued to $b_{\text{in/out}}$ is labelled as in the oriented case in Section 5.4, with $\alpha, \bar{\beta}$ the basis vectors introduced above. Let $\Sigma'_b$ be the bordism obtained by gluing $b_{\text{in}}$ to $b_{\text{out}}$, and let $\Sigma'$ be the corresponding decorated world sheet, which has the same decoration as $\Sigma$ away from the two (half-)discs which got omitted in the gluing.

Recall the gluing bordisms $G_{W,\epsilon}$ and $G_{U,\bar{V},\epsilon}$ from (120) and (122). The compatibility with gluing is the following statement:

**Theorem 6.11.** *Let $F \in \widehat{\mathcal{C}}$ be a special Frobenius algebra with $N_F^2 = \text{id}$. Then:*

- *(closed gluing) Let $\Sigma'$ be obtained by gluing from $\Sigma(U, \bar{V}, \epsilon, \alpha, \bar{\beta})$ as above. The correlators for $F$ satisfy*

$$\text{Corr}_F^{\text{spin}}(\Sigma') = \sum_{U, \bar{V}, \epsilon, \alpha} \widehat{\mathcal{Z}}_{\mathcal{C}}(G_{U,\bar{V},\epsilon})\left( \text{Corr}_F^{\text{spin}}(\Sigma(U, \bar{V}, \epsilon, \alpha, \bar{\alpha})) \right), \tag{178}$$

*where $\alpha$ runs over the basis in (176) for the type $x$ of the boundary components $b_{\text{in/out}}$.*

- *(open gluing) Let $\Sigma'$ be obtained by gluing from $\Sigma(W, \epsilon, \alpha, \bar{\beta})$ as above. The correlators for $B$ satisfy*

$$\text{Corr}_F^{\text{spin}}(\Sigma') = \sum_{W, \epsilon, \alpha} \widehat{\mathcal{Z}}_{\mathcal{C}}(G_{W,\epsilon})\left( \text{Corr}_F^{\text{spin}}(\Sigma(W, \epsilon, \alpha, \bar{\alpha})) \right). \tag{179}$$

*Proof.* The proof in the open case is almost identical to the proof in the oriented case. The only difference is the presence of Nakayama automorphisms of the $F$-modules due to the edge indices on the gluing boundary. By the gluing construction (33) the Nakayama automorphisms simply compose after gluing.

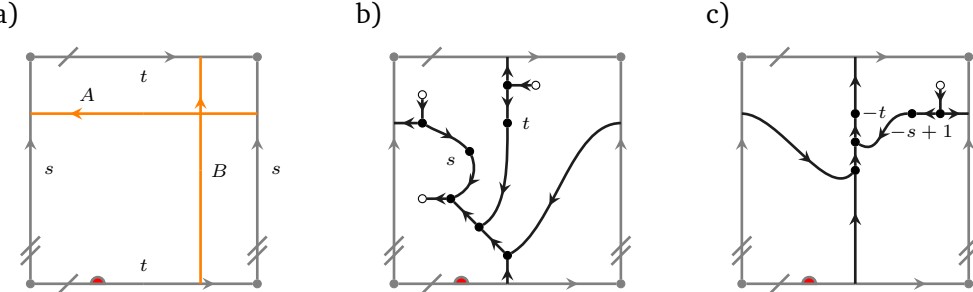

Figure 34: a) Marked cell decomposition of a torus $\mathbb{T}^2_{s,t}$ with spin structure described by the holonomies $s, t \in \mathbb{Z}_2$ along the loops $A$ and $B$ respectively. b) Defect network for the spin torus $\mathbb{T}^2_{s,t}$. c) Simplified defect network for the spin torus $\mathbb{T}^2_{s,t}$.

Let us consider the closed case. By Theorem 5.6 we have

$$
\begin{aligned}
\operatorname{Corr}_F^{\mathrm{spin}}(\Sigma') &\overset{\mathrm{def.}}{=} \operatorname{Corr}_{B=\mathbf{1}}^{\mathrm{or}}(W_{\mathrm{or}}(\Sigma; F)) \\
&\overset{\mathrm{Thm.\,5.6}}{=} \sum_{U,\bar{V},\epsilon,\alpha} \widehat{\mathcal{Z}}_{\mathcal{C}}(G_{U,\bar{V},\epsilon})\big( \operatorname{Corr}_B^{\mathrm{or}}(W_{\mathrm{or}}(\Sigma(U,\bar{V},\epsilon,x,\alpha,\bar{\alpha}))) \big),
\end{aligned}
\tag{180}
$$

where the sum runs over a basis of $\widehat{\mathcal{C}}(U \otimes \bar{V} \boxtimes K^\epsilon, X)$ i.e. the multiplicity space for the oriented $B = \mathbf{1}$ theory. Observe that whenever $\alpha \notin \mathcal{M}_{x,\epsilon}^{(\mathrm{in})}(U, \bar{V}; X)$ the corresponding summand is 0, as the $F$-defect network on $\Sigma$ near the insertion points contains the projector $Q_{x,\epsilon}^{(\mathrm{in/out})}$, as shown in Figure 33. We skip the details on how the projectors arise, the computation is analogous to that in Figure 27. Therefore the sum restricts to a basis of the multiplicity space for the $F$ theory and we have

$$
\begin{aligned}
\operatorname{Corr}_F^{\mathrm{spin}}(\Sigma') &= \sum_{\substack{U,\bar{V},\epsilon,\alpha \\ \alpha \in \mathcal{M}}} \widehat{\mathcal{Z}}_{\mathcal{C}}(G_{U,\bar{V},\epsilon})\big( \operatorname{Corr}_B^{\mathrm{or}}(W_{\mathrm{or}}(\Sigma(U,\bar{V},\epsilon,x,\alpha,\bar{\alpha}))) \big) \\
&\overset{\mathrm{def.}}{=} \sum_{U,\bar{V},\epsilon,\alpha} \widehat{\mathcal{Z}}_{\mathcal{C}}(G_{U,\bar{V},\epsilon})\big( \operatorname{Corr}_F^{\mathrm{spin}}(\Sigma(U,\bar{V},\epsilon,x,\alpha,\bar{\alpha})) \big).
\end{aligned}
\tag{181}
$$

$\square$

## 6.5 Example: Torus partition functions for the four spin structures

Consider the spin tori $\mathbb{T}^2_{s,t}$ in Figure 34 a), where $s, t \in \mathbb{Z}_2$ denote the holonomies $s, t \in \mathbb{Z}_2$ along the loops $A$ and $B$: $\zeta(A) = s$, $\zeta(B) = t$. The action of the mapping class group on the set $\{\mathbb{T}^2_{s,t}\}_{s,t \in \mathbb{Z}_2}$ is as follows [86, Lem. 3.3.1]:

$$
\begin{aligned}
T = T_A &: \mathbb{T}^2_{s,t} \longrightarrow \mathbb{T}^2_{s,t+s}, \\
S = T_B T_A T_B &: \mathbb{T}^2_{s,t} \longrightarrow \mathbb{T}^2_{t,s},
\end{aligned}
\tag{182}
$$

where $T_A$ and $T_B$ are Dehn twists along the loops $A$ and $B$ (here we explicitly use $s = -s$, $t = -t$, cf. Remark 6.2). Note that $T_A$ appears instead of $T_A^{-1}$ due to the non-standard orientation of $A$.

The correlator for $\mathbb{T}^2_{s,t}$ is an element of $\mathrm{Bl}(T^2)$. We use the basis $\{\chi_i \otimes \overline{\chi}_j\}_{i,j \in \mathcal{I}}$ for $\mathrm{Bl}(T^2)$ as in Section 5.5, and we would like to compute the coefficients in the expansion

$$
\operatorname{Corr}_F^{\mathrm{spin}}(\mathbb{T}^2_{s,t}) = \sum_{i,j \in \mathcal{I}} Z(F; s, t)_{ij}\, \chi_i \otimes \overline{\chi}_j \in \mathrm{Bl}(T^2).
\tag{183}
$$

The result is expressed in terms of traces over multiplicity spaces of the linear map given by precomposing with an appropriate power of the Nakayama automorphism:

**Proposition 6.12.** *For $F \in \widehat{\mathcal{C}}$ a special Frobenius algebra with $N_F^2 = \text{id}$, the coefficients $Z(F; s, t)_{ij}$ in (183) are given in terms of the multiplicity spaces in (161) as*

$$Z(F; s, t)_{ij} = \text{tr}_{\mathcal{M}_{1-s,+}^{(\text{in})}(S_i, \bar{S}_j; F)}\big(N_F^{-t} \circ (-)\big) - \text{tr}_{\mathcal{M}_{1-s,-}^{(\text{in})}(S_i, \bar{S}_j; F)}\big(N_F^{-t} \circ (-)\big). \tag{184}$$

The proof is very similar to that of Proposition 5.7, so we will be more brief here, except for when we give the missing details on the isomorphism $f$ promised at the end of that proof.

*Proof.* The coefficients in (183) can be expressed as

$$Z(F; s, t)_{ij} = (\chi_i^* \otimes_{\mathbb{C}} \overline{\chi}_j^*) \circ \text{Corr}_B^{\text{or}}(T^2) = \widehat{\mathcal{Z}}_{\mathcal{C}}(C(s, t)_{i,j}). \tag{185}$$

Rather than giving $C(s, t)_{i,j}$, which is analogous to (133), we directly expand the identity as in (134) and give the manifold corresponding to (136). Namely,

$$\widehat{\mathcal{Z}}_{\mathcal{C}}(C(s, t)_{i,j}) = \sum_{\epsilon, \alpha} \widehat{\mathcal{Z}}_{\mathcal{C}}(C(s, t)_{i,j}^{\epsilon, \alpha}), \tag{186}$$

where

$$C(s,t)_{i,j}^{\epsilon,\alpha} = \qquad\qquad\qquad\qquad\qquad\qquad \tag{187}$$

and where we already used that in the sum over $S_k \otimes K^\epsilon$, only $\mathbf{1} \otimes K^\epsilon$ contributes. The $F$-ribbons and powers of the Nakayama automorphism are prescribed by the defect network in Figure 34 c).

Next, set $H := \widehat{\mathcal{C}}(K^\epsilon, S_i^* \otimes F \otimes S_j^*)$ and define the linear maps $P_x, M_y : H \longrightarrow H$ as

$$P_x(\phi) = \qquad\qquad\qquad M_y(\phi) = \qquad\qquad\qquad \tag{188}$$

Then from (186) we get

$$\widehat{\mathcal{Z}}_{\mathcal{C}}(C(s, t)_{i,j}) = \text{tr}_H(M_{-t} P_{1-s}). \tag{189}$$

In order to compute the trace, we will relate $P_x$ to $Q_x^{(\text{in})}$ from (89). Define the isomorphism $f : \widehat{\mathcal{C}}(S_i \otimes S_j \otimes K^\epsilon, F) \longrightarrow \widehat{\mathcal{C}}(K^\epsilon, S_i^* \otimes F \otimes S_j^*)$ via

$$f(\psi) = \qquad\qquad\qquad\qquad\qquad\qquad \tag{190}$$

We claim that

$$f(Q_x^{(\text{in})}(\psi)) = P_x(f(\psi)), \quad f(N^y \circ \psi) = M_y(f(\psi)). \tag{191}$$

The second equality is clear. For the first equality, one just writes out the left hand side and deforms the resulting string diagram:

$$f(Q_x^{(\text{in})}(\psi)) = \quad \text{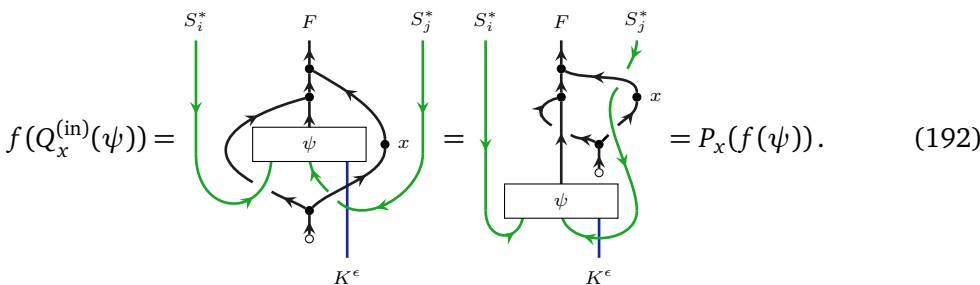} \quad = P_x(f(\psi)). \tag{192}$$

Via the isomorphism $f$, the trace over $H$ in (189) can instead be written as the trace of $\psi \longmapsto N_F^{-t} \circ Q_{s-1}^{(\text{in})}(\psi)$ over $\widehat{\mathcal{C}}(S_i \otimes S_j \otimes K^\epsilon, F)$. Since $Q_x^{(\text{in})}$ is an idempotent which commutes with $N_F \circ (-)$, we can instead trace $N_F^{-t} \circ (-)$ over its image. This proves (184).

Note that for $N_F = \text{id}$, $P_x$ in (188) agrees with $p \circ (-)$ where $p$ is taken from (138), and so the above argument also provides the missing step at the end of the proof of Proposition 5.7.

□

# 7 Examples

In this section we apply the formalism for oriented and spin CFT with or without parity to the simplest non-trivial case, namely when the algebra is a direct sum of $\mathbf{1}$ and an order-2 simple current in $\widehat{\mathcal{C}}$. For $G \in \mathcal{C}$ an invertible simple object with $G \otimes G \cong \mathbf{1}$, we consider the special Frobenius algebras with underlying objects

$$A_+ = \mathbf{1} \oplus G, \quad A_- = \mathbf{1} \oplus \Pi G. \tag{193}$$

In the case $G \cong \mathbf{1}$, only $\Pi\mathbf{1}$ has order two and we only consider $A_-$. The results of this section have already been summarised in Table 1.

## 7.1 The trivial CFT and the Arf invariant

This example was already considered in [75] where it is Example 1. We now present this example in terms of the constructions in this paper.

### 7.1.1 Chiral data

Take $\mathcal{V}$ to the be the irreducible Virasoro VOA at $c = 0$. The stress tensor of this VOA is zero, and the VOA is spanned by a single state $|0\rangle$ of conformal weight $h = 0$: $\mathcal{V} = \mathbb{C}|0\rangle$. There is a single irreducible representation of $\mathcal{V}$ of conformal weight $h = 0$, namely $\mathcal{V}$ itself. The category $\mathcal{C} = \text{Rep}(\mathcal{V})$ is just the category $\mathcal{V}ect$ and has a single simple object $\mathbf{1}$. The CFTs constructed in this example will actually be 2d TFTs.

### 7.1.2 Field content for B – the trivial 2d TFT

The category $\widehat{\mathcal{C}}$ is given by $\mathcal{V}ect \boxtimes \mathcal{S}\mathcal{V}ect \cong \mathcal{S}\mathcal{V}ect$ and thus has two simple objects,

$$\mathbf{1} = \mathbf{1} \boxtimes K^+, \quad \Pi\mathbf{1} = \mathbf{1} \boxtimes K^-.$$

To construct the oriented CFT (with parity), we take the symmetric special Frobenius algebra $B = \mathbf{1}$ and the space of bulk fields can be computed from (89) (remembering that $N_B = \mathrm{id}$ and so the idempotents are independent of $n$) and is simply

$$\mathcal{H}^{(in)}_{\mathrm{bulk}}(\mathbf{1}) = \mathbf{1} \boxtimes \bar{\mathbf{1}} \boxtimes K^+ \,.$$

The only parity-odd field in the bulk lives at the end of the $\Pi\mathbf{1}$ defect, calculated from (89) with $X = \Pi\mathbf{1}$ and notation as in (108)

$$\mathcal{H}^{(in)}_{\mathrm{bulk}}(\Pi\mathbf{1}) = \mathbf{1} \boxtimes \bar{\mathbf{1}} \boxtimes K^- \,.$$

Altogether, this gives a CFT of type $\textcircled{2}_{\mathrm{bnd\&def}}$, which becomes $\textcircled{1}_{\mathrm{bulk}}$ if one only considers world sheets without boundaries and defects.

### 7.1.3 Field content and partition functions for $F$ – the Arf invariant

For the algebra $F \in \widehat{\mathcal{C}}$, we take $F = A_- = \mathbf{1} \oplus \Pi\mathbf{1} = C\ell_1$, the Clifford algebra in one odd generator. One checks that $N_F = \mathrm{id}_{\mathbf{1}} - \mathrm{id}_{\Pi\mathbf{1}}$ is the parity involution, and so the algebra $F = A_-$ defines a CFT of the following type:

| type of CFT | no parity | parity |
|:---:|:---:|:---:|
| oriented | | |
| spin | | $F = \mathbf{1} \oplus \Pi\mathbf{1}$ |

(194)

Since $N_F$ is no longer just the identity, the fields split into Neveu–Schwarz and Ramond sectors, with the following bulk field content:

$$\begin{aligned}
\mathcal{H}^{\mathrm{NS,even}}_{\mathrm{bulk}}(F) &= \mathbf{1} \boxtimes \bar{\mathbf{1}} \boxtimes K^+ \,, & \mathcal{H}^{\mathrm{NS,odd}}_{\mathrm{bulk}}(F) &= 0 \,, \\
\mathcal{H}^{\mathrm{R,even}}_{\mathrm{bulk}}(F) &= 0 \,, & \mathcal{H}^{\mathrm{R,odd}}_{\mathrm{bulk}}(F) &= \mathbf{1} \boxtimes \bar{\mathbf{1}} \boxtimes K^- \,,
\end{aligned}$$

(195)

i.e. there are two bulk fields, both of conformal weight 0: a parity even field in the NS sector and a parity odd field in the Ramond sector. The resulting theory is of type $\textcircled{4}_{\mathrm{bulk}}$, or of type $\textcircled{4}_{\mathrm{bnd\&def}}$ if one includes boundaries and defects.

The $\tau$-independent partition functions in the four sectors are

$$Z^{\mathrm{NS,NS}}_F = 1 \,, \qquad Z^{\mathrm{NS,R}}_F = 1 \,, \qquad Z^{\mathrm{R,NS}}_F = 1 \,, \qquad Z^{\mathrm{R,R}}_F = -1 \,.$$

(196)

This agrees with Example 1 in [75], where it is also verified that this 2d TFT computes the Arf-invariant of a spin structure.

## 7.2 Computing multiplicity spaces for $G \not\cong \mathbf{1}$

Let $G \in \mathcal{C}$ be an invertible simple object with $G \not\cong \mathbf{1}$, $G \otimes G \cong \mathbf{1}$. Write $\dim G \in \{\pm 1\}$ for its quantum dimension and $\theta_G \in \{\pm 1, \pm i\}$ for its twist eigenvalue (see Appendix A.4 for why these are the only possible values).

For a rational VOA $\mathcal{V}$ and $\mathcal{C} = \mathrm{Rep}\,\mathcal{V}$, $\theta_G$ and $\dim G$ can be expressed via the lowest conformal weight $h_G$ of the representation $G$ and the S-matrix describing the modular transformation of characters as

$$\theta_G = e^{2\pi i h_G} \,, \quad \dim G = \frac{S_{G,\mathbf{1}}}{S_{\mathbf{1},\mathbf{1}}} \,.$$

(197)

For $\nu \in \{\pm 1\}$ the object $A_\nu$ in (193) carries a special Frobenius algebra structure iff $\theta_G \in \{\pm 1\}$, in which case this structure is unique up to isomorphism and its Nakayama automorphism satisfies (Lemma A.22):

- $\dim G = \nu$: $A_\nu$ is symmetric, and so $N_{A_\nu} = \mathrm{id}_{A_\nu}$,

- $\dim G \neq \nu$: $A_\nu$ is not symmetric, i.e. $N_{A_\nu} \neq \mathrm{id}_{A_\nu}$, and we have $N_{A_\nu}^2 = \mathrm{id}_{A_\nu}$.

The classification of the CFT defined by $\mathcal{V}$ and $A_\nu$ as in (2) is:

| type of CFT | no parity | parity |
|---|---|---|
| oriented | $\nu = +1$ , $\dim G = +1$ | $\nu = -1$ , $\dim G = -1$ |
| spin | $\nu = +1$ , $\dim G = -1$ | $\nu = -1$ , $\dim G = +1$ |

$$(198)$$

Note that this is independent of the twist $\theta_G$. In unitary theories one necessarily has $\dim(G) > 0$, so that for these examples, the off-diagonal entries (where the bulk is strictly of type ② or ③) will be non-unitary. The diagonal entries may be either unitary or non-unitary, depending on $\mathcal{V}$.

In the examples below we would like to compute the bulk state space(s) and torus partition function(s) for each of the eight possibilities for $(\dim G, \nu, \theta_G)$ as in Table 1. For this, we need the so-called monodromy charge $q_U$ of a simple object $U \in \mathcal{C}$,

$$q_U = \frac{\theta_{G \otimes U}}{\theta_G \, \theta_U} = \frac{s_{U,G}}{\dim(U)\dim(G)} = \frac{S_{U,G} \, S_{11}}{S_{1,U} \, S_{1,G}}, \qquad (199)$$

see (A.54) and Lemma A.23. In the second equality we used that the Hopf link invariant $s_{U,V}$ from (36) can be expressed in terms of the modular S-matrix as $s_{UV} = S_{UV}/S_{11}$.

The multiplicity spaces in (90) are given by (see Lemma A.25)

$$\mathcal{M}_{x,\epsilon}(S_i, \bar{S}_j; A_\nu) = M_1 \oplus M_G, \qquad (200)$$

where

$$M_1 = \begin{cases} \mathcal{C}(S_i \otimes \bar{S}_j, 1), & \text{if } \epsilon = + \text{ and } q_i = (\nu \dim(G))^x, \\ \{0\}, & \text{otherwise,} \end{cases}$$
$$M_G = \begin{cases} \mathcal{C}(S_i \otimes \bar{S}_j, G), & \text{if } \epsilon = \nu \text{ and } q_i = (\nu \dim(G))^{x+1} \theta_G, \\ \{0\}, & \text{otherwise.} \end{cases} \qquad (201)$$

For $A_\nu$ symmetric, the choice of $x$ does not matter and one can take $x = 0$. For $A_\nu$ not symmetric, the choice of $x$ gives the sector of the bulk field (NS/R) as in (160).

Let us make some general observations on CFTs obtained from algebras of the form $A_\nu$

**Remark 7.1.** 1. One can ask when the state space of the CFT defined by $A_\nu$ contains holomorphic or antiholomorphic fields in the representation $G$. From inspecting (201) and using $q_G = 1$ (Lemma A.23), one concludes that the following three statements are equivalent:

(a) $(G \boxtimes \bar{1}) \otimes K^\epsilon$ is a direct summand in the state space $\mathcal{H}_{\text{bulk}}^{(\text{in})}(A_\nu)$ (for $N_{A_\nu} = \mathrm{id}$) or $\bigoplus_{x \in \mathbb{Z}_2} \mathcal{H}_{\text{bulk}}^{x,(\text{in})}(A_\nu)$ (for $N_{A_\nu} \neq \mathrm{id}$), cf. (97) and (161).

(b) $(1 \boxtimes \bar{G}) \otimes K^\epsilon$ is a direct summand in $\mathcal{H}_{\text{bulk}}^{(\text{in})}(A_\nu)$ or $\bigoplus_{x \in \mathbb{Z}_2} \mathcal{H}_{\text{bulk}}^{x,(\text{in})}(A_\nu)$.

(c) One has $\epsilon = \nu$ and $(\nu \dim G)^{x+1} = \theta_G$ for some $x \in \mathbb{Z}_2$.

In the oriented case we have $N_{A_\nu} = \mathrm{id}$, and so $\nu \dim G = 1$. Hence an (anti)holomorphic $G$ is contained in the state space iff $\theta_G = 1$, i.e. iff $G$ has integer conformal weight.

In the spin case, $\nu \dim G = -1$, and so there is *always* an (anti)holomorphic copy $G$ in the state space. It is in the NS-sector ($x = 0$) if $\theta_G = -1$ and in the R-sector ($x = 1$) if $\theta_G = 1$.

This explains the column "$G$ hol." in Table 1.

2. $A_\nu$ is commutative iff $\theta_G = \nu \dim G$ (Lemma A.22). This explains the column "$A_\nu$ com." in Table 1. For a rational VOA $\mathcal{V}$ and $\widehat{\mathcal{C}} = \mathrm{Rep}_{\mathcal{S}}(\mathcal{V})$ (cf. (74)), it follows that $A_\nu$ with $\nu = \theta_G \dim G$ is again a VOA ($\nu = 1$) or a VOSA ($\nu = -1$), see [56, Thm. 3.6], as well as [18, Thm. 3.9] and [19, Thm. 4.2]. We will use this in the examples below to also present the results from the point of view of the extended VO(S)A $A_\nu$.

## 7.3 Ising CFT and free fermions

### 7.3.1 Chiral data

Take $\mathcal{V}$ to be the irreducible Virasoro VOA at $c = \frac{1}{2}$. This describes the chiral symmetry of the minimal model $M(3,4)$. The irreducible representations of $\mathcal{V}$ and their lowest conformal weight $h$ and monodromy charge $q$ are given by:

$$
\begin{array}{c|c|c|c}
\text{name} & \mathbf{1} & \sigma & \epsilon \\
\hline
h & 0 & \frac{1}{16} & \frac{1}{2} \\
\hline
q & 1 & -1 & 1
\end{array}
\tag{202}
$$

We take $G = \epsilon$, so that

$$
\dim(G) = 1, \quad \theta_G = -1. \tag{203}
$$

By (198), the algebras $B = A_+$ and $F = A_-$ define CFTs of the following types:

$$
\begin{array}{c|c|c}
\text{type of CFT} & \text{no parity} & \text{parity} \\
\hline
\text{oriented} & B = \mathbf{1} \oplus \epsilon & \\
\hline
\text{spin} & & F = \mathbf{1} \oplus \Pi\epsilon
\end{array}
\tag{204}
$$

The algebra $B$ is Morita-equivalent to $\mathbf{1}$, but using $B = \mathbf{1} \oplus \epsilon$ exhibits more clearly the difference that the parity-shift of $\epsilon$ makes. By Remark 7.1 (2.), the algebra $F$ is commutative and in fact a VOSA, the free fermion VOSA. It has three irreducible (twisted) modules: $I = F$, the tensor unit, $\Pi I$, the parity shifted version where the ground state has odd parity, and the twisted module $R_t = \sigma \oplus \Pi\sigma$.

To express the S-matrix in terms of these representations of the VOSA $F$, we introduce the following combinations of characters of the Virasoro VOA $\mathcal{V}$:

$$
\chi_I(\tau) = \chi_1(\tau) + \chi_\epsilon(\tau), \quad \widetilde{\chi}_I(\tau) = \chi_1(\tau) - \chi_\epsilon(\tau), \quad \chi_{R_t}(\tau) = 2\chi_\sigma(\tau). \tag{205}
$$

Here, $\chi_I$ is the usual trace of $q^{L_0 - c/24}$ over $F$, $\widetilde{\chi}_I$ is the trace with an additional insertion of the automorphism $N_F$, which amounts to taking the super-trace. The corresponding expressions for $\Pi I$ just produce an overall minus sign. The twisted trace $\widetilde{\chi}_{R_t}(\tau)$ is identically zero. In this basis, and in the order $\{\chi_I, \widetilde{\chi}_I, \chi_{R_t}\}$ as above, the S- and T-matrices read

$$
\mathsf{S} = \begin{pmatrix} 1 & 0 & 0 \\ 0 & 0 & 1/\sqrt{2} \\ 0 & \sqrt{2} & 0 \end{pmatrix}, \qquad \mathsf{T} = e^{-\pi i/24} \begin{pmatrix} 0 & 1 & 0 \\ 1 & 0 & 0 \\ 0 & 0 & e^{\pi i/8} \end{pmatrix}. \tag{206}
$$

We will see a similar block structure for the S-matrix in the other examples. Note, however, that the T-matrix is no longer diagonal.

### 7.3.2 Field content and partition function for $B$ – critical Ising CFT

By (97) we only need to consider $\mathcal{M}_{x,\epsilon}$ for $x = 0$. Since $\nu = +$, from (200) we see that all spaces $\mathcal{M}_{0,-}$ are zero. As expected, combining (97) and (200) we find

$$\mathcal{H}_{\text{bulk}}(B) = \mathbf{1} \boxtimes \bar{\mathbf{1}} \oplus \epsilon \boxtimes \bar{\epsilon} \oplus \sigma \boxtimes \bar{\sigma}. \tag{207}$$

Accordingly, by (131) the vectors $\text{Corr}_B^{\text{or}}(T^2) \in \text{Bl}(T^2)$ are given by

$$\text{Corr}_B^{\text{or}}(T^2) = \chi_{\mathbf{1}} \otimes_{\mathbb{C}} \overline{\chi}_{\mathbf{1}} + \chi_{\epsilon} \otimes_{\mathbb{C}} \overline{\chi}_{\epsilon} + \chi_{\sigma} \otimes_{\mathbb{C}} \overline{\chi}_{\sigma}. \tag{208}$$

To turn these into partition functions which depend on the modular parameter $\tau$, one has to replace the vectors $\chi_i \in \widehat{\mathcal{Z}}_{\mathcal{C}}(T^2)$ from (128) by the $\tau$-dependent characters,

$$
\begin{aligned}
Z_B(\tau) &= \left|\chi_{\mathbf{1}}(\tau)\right|^2 + \left|\chi_{\epsilon}(\tau)\right|^2 + \left|\chi_{\sigma}(\tau)\right|^2 \\
&= \tfrac{1}{2}\left|\chi_I(\tau)\right|^2 + \tfrac{1}{2}\left|\widetilde{\chi}_I(\tau)\right|^2 + \tfrac{1}{4}\left|\chi_{R_t}(\tau)\right|^2.
\end{aligned} \tag{209}
$$

This is indeed the modular invariant partition function of the critical Ising CFT, as expected.

### 7.3.3 Field content and partition functions for $F$ – free fermion CFT

The various sectors of the state spaces as in (162) can computed from the general formula in (200),

$$
\begin{aligned}
\mathcal{H}_{\text{bulk}}^{\text{NS,even}}(F) &= \mathbf{1} \boxtimes \bar{\mathbf{1}} \boxtimes K^+ \oplus \epsilon \boxtimes \bar{\epsilon} \boxtimes K^+, \\
\mathcal{H}_{\text{bulk}}^{\text{NS,odd}}(F) &= \epsilon \boxtimes \bar{\mathbf{1}} \boxtimes K^- \oplus \mathbf{1} \boxtimes \bar{\epsilon} \boxtimes K^-, \\
\mathcal{H}_{\text{bulk}}^{\text{R,even}}(F) &= \sigma \boxtimes \bar{\sigma} \boxtimes K^+, \\
\mathcal{H}_{\text{bulk}}^{\text{R,odd}}(F) &= \sigma \boxtimes \bar{\sigma} \boxtimes K^-,
\end{aligned} \tag{210}
$$

which is of course the expected result for the free fermion state spaces, see e.g. [29, §10.3].

The Nakayama automorphism $N_F$ acts as id on the summand $\mathbf{1}$ of $F$ and as $-$id on the summand $G \boxtimes K^-$. The precomposition with $N_F$ is thus equal to the identity on the multiplicity spaces $M_{\mathbf{1}}$ in (200) and to minus the identity on $M_G$. One can now use (184) to read off $\text{Corr}_F^{\text{spin}}$ for the four spin structures on the torus to be (recall that $\mathbb{T}_{s,t}^2$ is labelled by monodromy)

$$
\begin{array}{lll}
\text{NS,NS} & \text{Corr}_F^{\text{spin}}(\mathbb{T}_{1,1}^2) = \chi_{\mathbf{1}} \otimes_{\mathbb{C}} \overline{\chi}_{\mathbf{1}} + \chi_{\epsilon} \otimes_{\mathbb{C}} \overline{\chi}_{\epsilon} + \chi_{\epsilon} \otimes_{\mathbb{C}} \overline{\chi}_{\mathbf{1}} + \chi_{\mathbf{1}} \otimes_{\mathbb{C}} \overline{\chi}_{\epsilon}, \\
\text{NS,R} & \text{Corr}_F^{\text{spin}}(\mathbb{T}_{1,0}^2) = \chi_{\mathbf{1}} \otimes_{\mathbb{C}} \overline{\chi}_{\mathbf{1}} + \chi_{\epsilon} \otimes_{\mathbb{C}} \overline{\chi}_{\epsilon} - \chi_{\epsilon} \otimes_{\mathbb{C}} \overline{\chi}_{\mathbf{1}} - \chi_{\mathbf{1}} \otimes_{\mathbb{C}} \overline{\chi}_{\epsilon}, \\
\text{R,NS} & \text{Corr}_F^{\text{spin}}(\mathbb{T}_{0,1}^2) = 2 \chi_{\sigma} \otimes_{\mathbb{C}} \overline{\chi}_{\sigma}, \\
\text{R,R} & \text{Corr}_F^{\text{spin}}(\mathbb{T}_{0,0}^2) = 0.
\end{array} \tag{211}
$$

The corresponding $\tau$-dependent partition functions in the four sectors are

$$
\begin{aligned}
Z_F^{\text{NS,NS}}(\tau) &= \left|\chi_{\mathbf{1}}(\tau) + \chi_{\epsilon}(\tau)\right|^2 = |\chi_I(\tau)|^2, \\
Z_F^{\text{NS,R}}(\tau) &= \left|\chi_{\mathbf{1}}(\tau) - \chi_{\epsilon}(\tau)\right|^2 = |\widetilde{\chi}_I(\tau)|^2, \\
Z_F^{\text{R,NS}}(\tau) &= 2\left|\chi_{\sigma}(\tau)\right|^2 = \tfrac{1}{2}|\chi_{R_t}(\tau)|^2, \\
Z_F^{\text{R,R}}(\tau) &= 0.
\end{aligned} \tag{212}
$$

Using (206) one can now see explicitly that these bilinear combinations of characters are compatible with the modular properties of spin tori listed in (182).

Table 2: Kac-table of the minimal model $\mathcal{M}_{5,6}$ of central charge $c = \frac{4}{5}$, and the names we use for the distinct representations in the Kac-table.

| $r = 4$ | | | | | |
|---|---|---|---|---|---|
| $r = 3$ | $\mathbf{1}_\varphi$ | $u_\varphi$ | $f_\varphi$ | $v_\varphi$ | $w_\varphi$ |
| $r = 2$ | | | | | |
| $r = 1$ | $\mathbf{1}$ | $u$ | $f$ | $v$ | $w$ |
| | $s = 1$ | $s = 2$ | $s = 3$ | $s = 4$ | $s = 5$ |

## 7.4 Three-state Potts and fermionic tetra-critical Ising CFTs

### 7.4.1 Chiral data

Take $\mathcal{V}$ to be the irreducible Virasoro VOA at $c = \frac{4}{5}$ describing the chiral symmetry of the diagonal minimal model $M(5,6)$. The irreducible representations are (see Table 2 on how these correspond to entries in the Kac-table):

$$
\begin{array}{c|ccccccccc}
\text{name} & \mathbf{1} & u & f & v & w & \mathbf{1}_\varphi & u_\varphi & f_\varphi & v_\varphi & w_\varphi \\
\hline
h & 0 & \frac{1}{8} & \frac{2}{3} & \frac{13}{8} & 3 & \frac{7}{5} & \frac{21}{40} & \frac{1}{15} & \frac{1}{40} & \frac{2}{5} \\
\hline
q & 1 & -1 & 1 & -1 & 1 & 1 & -1 & 1 & -1 & 1
\end{array}
\tag{213}
$$

We take $G = w$, so that $\dim(G) = 1$, $\theta_G = 1$, and by (198) we obtain CFTs of the following types:

$$
\begin{array}{c|c|c}
\text{type of CFT} & \text{no parity} & \text{parity} \\
\hline
\text{oriented} & B = \mathbf{1} \oplus w & \\
\hline
\text{spin} & & F = \mathbf{1} \oplus \Pi w
\end{array}
\tag{214}
$$

The oriented CFT obtained from $B$ is the critical three-state Potts model, the $D$-type modular invariant at that central charge. The spectrum of the spin CFT defined by $F$ was first given in [57]. In Table 1 we refer to it as the fermionic tetra-critical Ising model, and we explain the name after discussing its field content in Section 7.4.3.

Of the two algebras $B$ and $F$, only $B$ is commutative (Remark 7.1), and the resulting VOA is the chiral symmetry of the first non-trivial member in the series of $W_3$-minimal models due to [90]. As a VOA, $B$ is generated by the Virasoro VOA $\mathbf{1} \subset B$ and by the primary field $W(z)$ of weight 3 in $w \subset B$. The VOA $B$ has six untwisted modules and two twisted modules:

$$
\begin{array}{c||c|c|c|c|c|c||c|c}
 & \multicolumn{6}{c||}{\text{untwisted}} & \multicolumn{2}{c}{\text{twisted}} \\
\hline
\text{name} & I & S & S^* & I_\varphi & S_\varphi & S^*_\varphi & U_t & U_{\varphi,t} \\
\hline
h & 0 & \frac{2}{3} & \frac{2}{3} & \frac{2}{5} & \frac{1}{15} & \frac{1}{15} & \frac{1}{8} & \frac{1}{40} \\
\hline
\text{decomp.} & \mathbf{1} \oplus w & f & f & \mathbf{1}_\varphi \oplus w_\varphi & f_\varphi & f_\varphi & u \oplus v & u_\varphi \oplus v_\varphi
\end{array}
\tag{215}
$$

The representations $S$ and $S^*$ have the same underlying Virasoro representation $f$, but the action of $W(z)$ differs by a sign. Ditto for $S_\varphi$ and $S^*_\varphi$.

This implies that $S$ and $S^*$ have the same restricted character (given by the trace over $q^{L_0 - c/24}$). To determine the S-matrix one needs to work with characters including an insertion from the VOA $B$. These are also called unspecialised characters or torus one-point blocks (with an insertion from the vacuum sector). In this case an insertion of the zero mode $W_0$ of the

field $W(z)$ is enough to lift the degeneracy. A corresponding computation for Bershadsky-Polyakov models is done in more detail in Section 7.5, here we just note that traces over $W_0$ can be found in [59, Sec. 3], and the S-matrix of untwisted representations is given explicitly in [15, Sec. 4.2].

We group the functions needed to give the S-matrix into three groups:

$$\chi_x : \text{untwisted char. of untwisted reps.} \qquad x \in \{I, I_\varphi, S, S_\varphi, S^*, S^*_\varphi\},$$

$$\widetilde{\chi}_x : \text{twisted char. of untwisted reps.} \qquad x \in \{I, I_\varphi\},$$

$$\chi_{x_t} : \text{untwisted char. of twisted reps.} \qquad x_t \in \{U_t, U_{\varphi,t}\},$$

$$\widetilde{\chi}_{x_t} : \text{twisted char. of twisted reps.} \qquad x_t \in \{U_t, U_{\varphi,t}\}. \qquad (216)$$

Here, $\chi_x$ etc. denote the unspecialised characters. The twisted traces are obtained by inserting a $\mathbb{Z}_2$-automorphism (in more detail: by working with $\mathbb{Z}_2$-equivariant modules with respect to the $\mathbb{Z}_2$-automorphism of $B$), for example $\widetilde{\chi}_I = \chi_1 - \chi_w$ (in terms of specialised characters). For $S, S^*, S_\varphi, S^*_\varphi$ there is no such $\mathbb{Z}_2$-automorphism, hence they are not included.

The S-matrix then has the following block decomposition

$$S = \begin{pmatrix}
 & \chi & \widetilde{\chi} & \chi_t & \widetilde{\chi}_t \\
\hline
\chi & C & 0 & 0 & 0 \\
\widetilde{\chi} & 0 & 0 & D & 0 \\
\chi_t & 0 & D & 0 & 0 \\
\widetilde{\chi}_t & 0 & 0 & 0 & -D
\end{pmatrix}, \qquad (217)$$

with the individual blocks given by, in the order of vectors as stated in (216),

$$C = \frac{1}{\sqrt{3}} \begin{pmatrix} D & D & D \\ D & \omega D & \omega^2 D \\ D & \omega^2 D & \omega D \end{pmatrix}, \quad D = \frac{2}{\sqrt{5}} \begin{pmatrix} \sin\frac{\pi}{5} & \sin\frac{2\pi}{5} \\ \sin\frac{2\pi}{5} & -\sin\frac{\pi}{5} \end{pmatrix}, \quad \omega = e^{2\pi i/3}. \qquad (218)$$

### 7.4.2 Field content and partition function for $B$ – critical Potts model

Combining (97) and (200) we find

$$\mathcal{H}_{\text{bulk}}(B) = (1+w) \boxtimes (\bar{1}+\bar{w}) \oplus 2f \boxtimes \bar{f} \oplus (1_\varphi + w_\varphi) \boxtimes (\bar{1}_\varphi + \bar{w}_\varphi) \oplus 2f_\varphi \boxtimes \bar{f}_\varphi. \qquad (219)$$

By (131) we find

$$\text{Corr}_B^{\text{or}}(T^2) = (\chi_1 + \chi_w) \otimes_{\mathbb{C}} (\overline{\chi}_1 + \overline{\chi}_w) + 2\chi_f \otimes_{\mathbb{C}} \overline{\chi}_f$$
$$+ (\chi_{1_\varphi} + \chi_{w_\varphi}) \otimes_{\mathbb{C}} (\overline{\chi}_{1_\varphi} + \overline{\chi}_{w_\varphi}) + 2\chi_{f_\varphi} \otimes_{\mathbb{C}} \overline{\chi}_{f_\varphi}. \qquad (220)$$

As is well-known, rewriting this as a $\tau$-dependent partition function and in terms of characters for the $W_3$-model gives the diagonal theory:

$$Z_B(\tau) = |\chi_I(\tau)|^2 + |\chi_S(\tau)|^2 + |\chi_{S^*}(\tau)|^2 + |\chi_{I_\varphi}(\tau)|^2 + |\chi_{S_\varphi}(\tau)|^2 + |\chi_{S^*_\varphi}(\tau)|^2. \qquad (221)$$

### 7.4.3 Field content and partition functions for $F$ – fermionic tetra-Ising model

The various sectors of the state spaces as in (162) can be computed from the general formula in (200),

$$\mathcal{H}_{\text{bulk}}^{\text{NS,even}}(F) = 1 \boxtimes \bar{1} \boxtimes K^+ \oplus w \boxtimes \bar{w} \boxtimes K^+ \oplus f \boxtimes \bar{f} \boxtimes K^+ \oplus (\dots)_\varphi,$$

$$\mathcal{H}_{\text{bulk}}^{\text{NS,odd}}(F) = u \boxtimes \bar{v} \boxtimes K^- \oplus v \boxtimes \bar{u} \boxtimes K^- \oplus (\dots)_\varphi,$$

$$\mathcal{H}_{\text{bulk}}^{\text{R,even}}(F) = u \boxtimes \bar{u} \boxtimes K^+ \oplus v \boxtimes \bar{v} \boxtimes K^+ \oplus (\dots)_\varphi,$$

$$\mathcal{H}_{\text{bulk}}^{\text{R,odd}}(F) = w \boxtimes \bar{1} \boxtimes K^- \oplus 1 \boxtimes \bar{w} \boxtimes K^- \oplus f \boxtimes \bar{f} \boxtimes K^- \oplus (\dots)_\varphi, \qquad (222)$$

where $(\dots)_\varphi$ stands for the same summands again, but now all representation labels have the additional index $\varphi$, i.e. are taken from the third row of the Kac-table, rather than the first.

Note that restricting to the even subspace produces the state space of the $A$-type modular invariant at $c = \frac{4}{5}$, i.e. of the tetra-critical Ising model. This is the reason to call the theory obtained from $F$ the fermionic tetra-critical Ising model.

As in the Ising case, the precomposition with $N_F$ is equal to the identity on the multiplicity spaces $M_{\mathbf{1}}$ in (200) and to minus the identity on $M_G$. From (184) one obtains:

$$
\begin{aligned}
\text{NS,NS} \quad & \text{Corr}_F^{\text{spin}}(\mathbb{T}_{1,1}^2) = \chi_{\mathbf{1}} \otimes_{\mathbb{C}} \overline{\chi}_{\mathbf{1}} + \chi_w \otimes_{\mathbb{C}} \overline{\chi}_w + \chi_f \otimes_{\mathbb{C}} \overline{\chi}_f + \chi_u \otimes_{\mathbb{C}} \overline{\chi}_v + \chi_v \otimes_{\mathbb{C}} \overline{\chi}_u + (\dots)_\varphi \,, \\
\text{NS,R} \quad & \text{Corr}_F^{\text{spin}}(\mathbb{T}_{1,0}^2) = \chi_{\mathbf{1}} \otimes_{\mathbb{C}} \overline{\chi}_{\mathbf{1}} + \chi_w \otimes_{\mathbb{C}} \overline{\chi}_w + \chi_f \otimes_{\mathbb{C}} \overline{\chi}_f - \chi_u \otimes_{\mathbb{C}} \overline{\chi}_v - \chi_v \otimes_{\mathbb{C}} \overline{\chi}_u + (\dots)_\varphi \,, \\
\text{R,NS} \quad & \text{Corr}_F^{\text{spin}}(\mathbb{T}_{0,1}^2) = \chi_w \otimes_{\mathbb{C}} \overline{\chi}_{\mathbf{1}} + \chi_{\mathbf{1}} \otimes_{\mathbb{C}} \overline{\chi}_w + \chi_f \otimes_{\mathbb{C}} \overline{\chi}_f + \chi_u \otimes_{\mathbb{C}} \overline{\chi}_u + \chi_v \otimes_{\mathbb{C}} \overline{\chi}_v + (\dots)_\varphi \,, \\
\text{R,R} \quad & \text{Corr}_F^{\text{spin}}(\mathbb{T}_{0,0}^2) = -\chi_w \otimes_{\mathbb{C}} \overline{\chi}_{\mathbf{1}} - \chi_{\mathbf{1}} \otimes_{\mathbb{C}} \overline{\chi}_w - \chi_f \otimes_{\mathbb{C}} \overline{\chi}_f + \chi_u \otimes_{\mathbb{C}} \overline{\chi}_u + \chi_v \otimes_{\mathbb{C}} \overline{\chi}_v + (\dots)_\varphi \,.
\end{aligned}
\tag{223}
$$

Note that the R-R partition function is indeed modular invariant as it is the difference of the A-type and D-type modular invariant.

Finally, let us rewrite these expressions in terms of the characters (216) for the extended algebra $B$, as $\tau$-dependent partition functions:

$$
\begin{aligned}
Z_F^{\text{NS,NS}}(\tau) &= \tfrac{1}{2}\Big(|\chi_I|^2 + |\widetilde{\chi}_I|^2 + |\chi_{U_t}|^2 - |\widetilde{\chi}_{U_t}|^2\Big) + |\chi_S|^2 + (\dots)_\varphi \,, \\
Z_F^{\text{NS,R}}(\tau) &= \tfrac{1}{2}\Big(|\chi_I|^2 + |\widetilde{\chi}_I|^2 - |\chi_{U_t}|^2 + |\widetilde{\chi}_{U_t}|^2\Big) + |\chi_S|^2 + (\dots)_\varphi \,, \\
Z_F^{\text{R,NS}}(\tau) &= \tfrac{1}{2}\Big(|\chi_I|^2 - |\widetilde{\chi}_I|^2 + |\chi_{U_t}|^2 + |\widetilde{\chi}_{U_t}|^2\Big) + |\chi_S|^2 + (\dots)_\varphi \,, \\
Z_F^{\text{R,R}}(\tau) &= \tfrac{1}{2}\Big(-|\chi_I|^2 + |\widetilde{\chi}_I|^2 + |\chi_{U_t}|^2 + |\widetilde{\chi}_{U_t}|^2\Big) - |\chi_S|^2 + (\dots)_\varphi \,.
\end{aligned}
\tag{224}
$$

In writing these expressions, we did not distinguish $\chi_S$ and $\chi_{S^*}$, since they are the same as function of $\tau$, and since we did not investigate the action of $W(z)$ (which is now an odd field in the Ramond sector) on these fields in the spin model.

It is straightforward to check the modular properties (182) of spin tori using the explicit S-matrix in (217).

### 7.4.4 Comparison with other constructions

The approach in this paper and in [82] to the construction of theories on spin world sheets is to start from the category $\widehat{\mathcal{C}}$ labelling parity enriched topological defects, then to identify a suitable non-symmetric Frobenius algebra $F$ in $\widehat{\mathcal{C}}$. The spin CFT is constructed using this algebra.

An alternative construction, which is possibly more physically intuitive, is to introduce the coupling of an oriented CFT without parity to the spin structure by tensoring with a theory with parity which already sees the spin structure - the Arf theory from Section 7.1 - and then gauging a $\mathbb{Z}_2$ which couples the original theory to the spin structure, followed by secondary stacking with Arf to generate a (possibly different) spin CFT with parity. This approach is used for example in [57].

The construction in this paper can produce the same theories as the above approach, through a suitable choice of Frobenius algebra $F$. Let us illustrate this in the case of the $c = 4/5$ Virasoro minimal model by comparing our approach to that in [57].

There are four distinct CFTs with $c = 4/5$ discussed in [57], which they call **A**, **D**, **F** and $\widetilde{\mathbf{F}}$. **A** is the diagonal invariant for the Virasoro VOA and is a purely parity even CFT on oriented surfaces. **D** is the diagonal invariant for the $W_3$-algebra VOA which is also a purely parity even

CFT on oriented surfaces. $\mathbf{F}$ and $\widetilde{\mathbf{F}}$ are CFTs with parity on surfaces with spin structure which are related by stacking with the invertible Arf 2d spin TFT, which is the same as swapping the parity in the Ramond sector. The field contents are given in table II of [57].

In our construction, the theories $\mathbf{A}$ and $\mathbf{D}$ can be obtained from the Virasoro category $\mathcal{C}$ using $B_{\mathbf{A}} = \mathbf{1}$ and $B_{\mathbf{D}} = \mathbf{1} \oplus w$ in $\mathcal{C}$, respectively, while the spin theories $\widetilde{\mathbf{F}}$ and $\mathbf{F}$ are obtained from $\widehat{\mathcal{C}}$ using $F_{\widetilde{\mathbf{F}}} = \mathbf{1} \oplus \Pi w$ and $F_{\mathbf{F}} = (\mathbf{1} \oplus \Pi w) \otimes C\ell_1$, respectively. In this section we only treated the examples $B_{\mathbf{D}}$ and $F_{\widetilde{\mathbf{F}}}$ explicitly.

## 7.5 Bershadsky-Polyakov models: spin without parity and parity without spin

### 7.5.1 Chiral data

The Bershadsky-Polyakov models were introduced in [7,78] and have been investigated from the point of view of VOAs in [3, 5, 42]. We will consider the simplest model $W_3^{(2)}$ which contains a chiral algebra with 3 primary fields of weights $1, \frac{3}{2}, \frac{3}{2}$ denoted $J, G^{\pm}$, together with the Virasoro algebra. These field have modes $J_m, G_r^{\pm}, L_m$, where $m \in \mathbb{Z}$ but $r \in \mathbb{Z} + \frac{1}{2}$ (NS sector) or $r \in \mathbb{Z}$ (R sector). We take the commutation relations

$$
\begin{aligned}
&[L_m, L_n] = \tfrac{c}{12} m(m^2 - 1)\delta_{m+n,0} + (m-n)L_{m+n}\,, \\
&[L_m, G_r^{\pm}] = (m/2 - r)G_{m+r}^{\pm}\,, && [L_m, J_n] = -nJ_{m+n}\,, \\
&[J_m, G_r^{\pm}] = \pm G_{m+r}^{\pm}\,, && [J_m, J_n] = \tfrac{2k+3}{3}\, m\, \delta_{m+n}\,, \\
&[G_r^+, G_s^-] = \tfrac{(2k+3)(k+1)}{2}(r^2 - \tfrac{1}{4})\delta_{r+s,0} + \tfrac{3}{2}(k+1)(r-s)J_{r+s} + 3(JJ)_{r+s} - (k+3)L_{r+s}\,, \\
&[G_r^+, G_s^+] = [G_r^-, G_s^-] = 0\,.
\end{aligned}
\tag{225}
$$

The central charge is parametrised by $k \in \mathbb{C}$ as

$$
c = -\frac{(2k+3)(3k+1)}{k+3}\,.
\tag{226}
$$

The peculiar feature of these models is that the fields are all bosonic, i.e. of even parity, and despite having half-integer weights the modes of the fields satisfy commutation relations, rather than anti-commutation relations.

The irreducible VOA obtained from the $W_3^{(2)}$ algebra at central charge $c(k)$ is denoted by $BP_k$ in [42]. We shall take as an example the model with $k = -1/2$. The irreducible VOA $BP_{-1/2}$ is $C_2$-cofinite and rational [3] and has central charge is $c(k) = 2/5$. It is also called the "Bershadsky–Polyakov minimal model" $BP(5,2)$. Since the term "minimal model" usually refers to the full CFT and not just the VOA, we will use $BP_{-1/2}$.

$BP_{-1/2}$ has 12 (twisted) irreducible representations, 6 in the NS sector and in 6 in the R sector, see [42, Fig. 2] and Appendix A.5 for more details. The NS representations of $BP_{-1/2}$ are labelled by the highest-weight eigenvalues of $J_0$ and $L_0$ denoted by $j$ and $h$. The NS and R sectors are related by "spectral flow". We give details of this in Appendix A.5, but a summary is that if we define a highest weight state in the R sector to be annihilated by $G_0^-$ (as well as all positive modes) then there is a spectral flow $\sigma_{1/2}$ under which an NS representation with eigenvalues $(j, h)$ "flows" to an R representation with eigenvalues $(j - 1/3, h - j/3 + 1/12)$. We list these in Tables 3 and 4. We denote the R representation obtained from the NS representation $M$ by $M^s$.

The $W_3^{(2)}$ algebra has a $\mathbb{Z}_2$ automorphism, which we denote $(-1)^G$. It acts as $L \longmapsto L$, $J \longmapsto J$, $G^{\pm} \longmapsto -G^{\pm}$ and the VOA splits into $\pm 1$ eigenspaces under its action. This is analogous to the automorphism $W \longmapsto -W$ of the $W_3$ algebra in Section 7.4.1 under which the oriented algebra $B$ decomposes into $B = \mathbf{1} \oplus w$. We shall denote the representations appearing

Table 3: NS representations of $BP_{-1/2}$. We will also write $\mathbf{1} = \mathrm{id}_e$ and $G = \mathrm{id}_o$.

| name | id | $\phi$ | $\bar{\phi}$ | $u$ | $\psi$ | $\bar{\psi}$ |
|---|---|---|---|---|---|---|
| $(j,h)$ | $(0,0)$ | $(\frac{1}{3},\frac{1}{30})$ | $(-\frac{1}{3},\frac{1}{30})$ | $(0,\frac{1}{5})$ | $(\frac{2}{3},\frac{1}{3})$ | $(-\frac{2}{3},\frac{1}{3})$ |
| decomp. | $\mathbf{1} \oplus G$ | $\phi_e \oplus \phi_o$ | $\bar{\phi}_e \oplus \bar{\phi}_o$ | $u_e \oplus u_o$ | $\psi_e \oplus \psi_o$ | $\bar{\psi}_e \oplus \bar{\psi}_o$ |

Table 4: R representations of $BP_{-1/2}$.

| name | $\mathrm{id}^s$ | $\phi^s$ | $\bar{\phi}^s$ | $u^s$ | $\psi^s$ | $\bar{\psi}^s$ |
|---|---|---|---|---|---|---|
| $(j,h)$ | $(-\frac{1}{3},\frac{1}{12})$ | $(0,-\frac{1}{20})$ | $(-\frac{2}{3},\frac{17}{60})$ | $(-\frac{1}{3},\frac{17}{60})$ | $(\frac{1}{3},\frac{1}{12})$ | $(-1,\frac{3}{4})$ |
| decomp. | $\mathrm{id}^s_e \oplus \mathrm{id}^s_o$ | $\phi^s_e \oplus \phi^s_o$ | $\bar{\phi}^s_e \oplus \bar{\phi}^s_o$ | $u^s_e \oplus u^s_o$ | $\psi^s_e \oplus \psi^s_o$ | $\bar{\psi}^s_e \oplus \bar{\psi}^s_o$ |

in the decomposition of id as $\mathrm{id}_e \equiv \mathbf{1}$ and $\mathrm{id}_o \equiv G$. Since $G$ has weight $\frac{3}{2}$, its twist eigenvalue is

$$\theta_G = e^{2\pi i 3/2} = -1\,. \tag{227}$$

Each NS representation $M$ of the $W_3^{(2)}$ algebra can be decomposed into two subspaces, $M_e$ given by the action of $\mathbf{1} = \mathrm{id}_e$ on the highest weight state and $M_o$ given by the action of $G = \mathrm{id}_o$. In the NS sector, the highest weight state space is one-dimensional and if the NS highest weight state is defined to have $(-1)^G$ eigenvalue $+1$ then $M_e$ is the $(-1)^G$ eigenspace with eigenvalue $+1$ and $M_o$ the space with eigenvalue $-1$.

In the R sector, there is a non-trivial algebra with generators $G_0^{\pm}, J_0, L_0$ and so one has to make a choice for the definition of the highest weight space. We follow [3] and define the highest weight space to be annihilated by $G_0^-$ (this is the opposite convention to [42]) and define it to be an eigenvector of $(-1)^G$ of eigenvalue $+1$. As with NS representations, we define the subspace $M_e$ of a R representation $M$ to be given by the action of $\mathrm{id}_e$ on the highest weight space (and hence having $(-1)^G$ eigenvalue $+1$) and the subspace $M_o$ to be given by the action of $\mathrm{id}_o$ on the highest weight space (and hence having $(-1)^G$ eigenvalue $-1$).

The modular properties of the characters under $S$ are best expressed in terms of the traces of the full representations with or without including $(-1)^G$, just as are those of the Ising model. As in the previous examples, we will write $\chi_M$ for untwisted trace and $\widetilde{\chi}_M$ for the twisted trace:

$$\chi_M(\tau,z) = \mathrm{Tr}_M\big(z^{J_0} q^{L_0 - c/24}\big), \quad \widetilde{\chi}_M(\tau,z) = \mathrm{Tr}_M\big((-1)^G z^{J_0} q^{L_0-c/24}\big), \tag{228}$$

so that

$$\mathrm{Tr}_{M_e}\big(z^{J_0} q^{L_0-c/24}\big) = \frac{1}{2}(\chi_M + \widetilde{\chi}_M), \quad \mathrm{Tr}_{M_o}\big(z^{J_0} q^{L_0-c/24}\big) = \frac{1}{2}(\chi_M - \widetilde{\chi}_M). \tag{229}$$

We define the index set

$$\mathcal{J} := \{\mathrm{id}, \phi, \bar{\phi}, u, \psi, \bar{\psi}\}\,. \tag{230}$$

As in (216) we obtain four sets of characters, where in each case $x$ runs over all elements in $\mathcal{J}$:

$$\chi_x : \text{untwisted char. of NS-sector reps.}$$
$$\widetilde{\chi}_x : \text{twisted char. of NS-sector reps.}$$
$$\chi_{x^s} : \text{untwisted char. of R-sector reps.}$$
$$\widetilde{\chi}_{x^s} : \text{twisted char. of R-sector reps.} \tag{231}$$

The modular S-matrix S appears in the following relations,

$$\chi_i(\tau, 1) = \sum_j \mathsf{S}_{ij} \chi_j(-1/\tau, 1), \quad \frac{d}{dz}\chi_i(\tau, z)\Big|_{z=1} = -\frac{1}{\tau}\sum_j \mathsf{S}_{ij}\frac{d}{dz}\chi_i(-1/\tau, z)\Big|_{z=1}. \quad (232)$$

The S-matrix then has the following block decomposition (cf. Appendix A.5)

$$\mathsf{S} = \begin{pmatrix} & \chi & \widetilde{\chi} & \chi_s & \widetilde{\chi}_s \\ \hline \chi & 0 & 0 & 0 & C \\ \widetilde{\chi} & 0 & D & 0 & 0 \\ \chi_s & 0 & 0 & E & 0 \\ \widetilde{\chi}_s & C^T & 0 & 0 & 0 \end{pmatrix}, \quad (233)$$

with the individual blocks given by, in the order of vectors as stated in (230),

$$C = \gamma \begin{pmatrix} 1 & a & a & -a & 1 & 1 \\ -a\omega^2 & 1 & \omega & -\omega^2 & -a\omega & -a \\ -a\omega & 1 & \omega^2 & -\omega & -a\omega^2 & -a \\ -a & 1 & 1 & -1 & -a & -a \\ \omega & a & a\omega^2 & -a\omega & \omega^2 & 1 \\ \omega^2 & a & a\omega & -a\omega^2 & \omega & 1 \end{pmatrix}, \quad D = \gamma \begin{pmatrix} 1 & a & a & -a & 1 & 1 \\ a & -\omega & -\omega^2 & 1 & a\omega^2 & a\omega \\ a & -\omega^2 & -\omega & 1 & a\omega & a\omega^2 \\ -a & 1 & 1 & -1 & -a & -a \\ 1 & a\omega^2 & a\omega & -a & \omega & \omega^2 \\ 1 & a\omega & a\omega^2 & -a & \omega^2 & \omega \end{pmatrix},$$

$$E = \gamma \begin{pmatrix} -\omega & a & a\omega^2 & a\omega & -\omega^2 & -1 \\ a & 1 & 1 & 1 & a & a \\ a\omega^2 & 1 & \omega & \omega^2 & a\omega & a \\ a\omega & 1 & \omega^2 & \omega & a\omega^2 & a \\ -\omega^2 & a & a\omega & a\omega^2 & -\omega & -1 \\ -1 & a & a & a & -1 & -1 \end{pmatrix}, \qquad \begin{aligned} \gamma &= \frac{2\sin(2\pi/5)}{\sqrt{15}}, \\ \omega &= e^{2\pi i/3}, \\ a &= 2\cos(2\pi/5). \end{aligned} \quad (234)$$

The value of $\dim G$ can be deduced simply from the block structure (233), but explicitly we have, from the block $D$

$$\chi_1(\tfrac{-1}{\tau}) = \tfrac{1}{2}\Big(\chi_{\mathrm{id}}(\tfrac{-1}{\tau}) + \widetilde{\chi}_{\mathrm{id}}(\tfrac{-1}{\tau})\Big) = \tfrac{\gamma}{2}\widetilde{\chi}_{\mathrm{id}}(\tau) + \cdots = \tfrac{\gamma}{2}\chi_1(\tau) - \tfrac{\gamma}{2}\chi_G(\tau) + \cdots$$

$$\Rightarrow \mathsf{S}_{1,1} = -\mathsf{S}_{1,G} = \tfrac{\gamma}{2} \Rightarrow \dim(G) = \frac{\mathsf{S}_{1,G}}{\mathsf{S}_{1,1}} = -1. \quad (235)$$

We also have the value of $\theta_G$ from (227), so that altogether

$$\dim(G) = -1, \quad \theta_G = -1. \quad (236)$$

The monodromy charges (199) of the irreducible representations with respect to the subalgebra $\mathbf{1} = \mathrm{id}_e$ of the $BP_{-1/2}$ algebra can be read off from the decompositions given in Tables 3 and 4. Namely, for an NS-representation $M$, the $G$-modes change the $L_0$-weight by $\mathbb{Z} + \frac{1}{2}$, so that $\theta_{G\otimes M}/\theta_M = -1$. For an R-representation $N$, the shift is by $\mathbb{Z}$ and so $\theta_{G\otimes N}/\theta_N = 1$. Since $\theta_G = -1$, all $\mathrm{id}_e$-representations appearing as summands of NS representations have monodromy charge $q = 1$, while those appearing in R representations have $q = -1$:

$$q_{x_{e/o}} = 1, \quad q_{x^s_{e/o}} = -1, \quad \text{where} \quad x \in \mathcal{J}. \quad (237)$$

From Tables 3 and 4 one can also read off what the conjugate representation of each irreducible $\mathrm{id}_e$-representation is. For example, the conjugate of $\phi_e$ has $J_0$ charge $j = -\frac{1}{3}$ and $L_0$-weight $h = \frac{1}{30}$, and so has to be given by $\bar{\phi}_e$. The conjugate of $\psi^s_e$ has $j = -\frac{1}{3}$ and $h = \frac{1}{12}$

Table 5: The conjugates $x^*$ of irreducible $\mathrm{id}_e$-representations $x$. The grey columns contain the self-dual representations. Note that spectral flow does not preserve self-duality.

| $x$ | $\mathbf{1}$ | $G$ | $\phi_e$ | $\phi_o$ | $\bar{\phi}_e$ | $\bar{\phi}_o$ | $u_e$ | $u_o$ | $\psi_e$ | $\psi_o$ | $\bar{\psi}_e$ | $\bar{\psi}_o$ |
|---|---|---|---|---|---|---|---|---|---|---|---|---|
| $x^*$ | $\mathbf{1}$ | $G$ | $\bar{\phi}_e$ | $\bar{\phi}_o$ | $\phi_e$ | $\phi_o$ | $u_e$ | $u_o$ | $\bar{\psi}_e$ | $\bar{\psi}_o$ | $\psi_e$ | $\psi_o$ |

| $x$ | $\mathrm{id}^s_e$ | $\mathrm{id}^s_o$ | $\phi^s_e$ | $\phi^s_o$ | $\bar{\phi}^s_e$ | $\bar{\phi}^s_o$ | $u^s_e$ | $u^s_o$ | $\psi^s_e$ | $\psi^s_o$ | $\bar{\psi}^s_e$ | $\bar{\psi}^s_o$ |
|---|---|---|---|---|---|---|---|---|---|---|---|---|
| $x^*$ | $\psi^s_e$ | $\psi^s_o$ | $\phi^s_e$ | $\phi^s_o$ | $u^s_o$ | $u^s_e$ | $\bar{\phi}^s_o$ | $\bar{\phi}^s_e$ | $\mathrm{id}^s_e$ | $\mathrm{id}^s_o$ | $\bar{\psi}^s_e$ | $\bar{\psi}^s_o$ |

and so is given by $\mathrm{id}^s_e$. For $u^s_e$ the conjugate has $j = \frac{1}{3}$ and $h = \frac{17}{60}$. The highest weight state of $(u^s_e)^*$ is thus a $G^+_0$-descendent of that of $\bar{\phi}^s_e$, i.e. $(u^s_e)^* = \bar{\phi}^s_o$. (And indeed, the corresponding characters of $BP_{-1/2}$ in Table 8 show a two-fold degeneracy for the lowest $L_0$-weight in these representations.) Analogously one sees $(u^s_o)^* = \bar{\phi}^s_e$. These are the only two instances where the $e/o$-label is exchanged under conjugation. We list the full result in Table 5.

Next we give the state spaces and torus partition functions for the following two cases:

| type of CFT | no parity | parity |
|---|---|---|
| oriented | | $B = \mathbf{1} \oplus \Pi G$ |
| spin | $F = \mathbf{1} \oplus G$ | |

(238)

### 7.5.2 Results for $B$

Let $\mathcal{J}$ be as in (230). From (97) and (200) (with $\nu = -$) one finds[16]

$$\mathcal{H}^{\mathrm{even}}_{\mathrm{bulk}}(B) = \bigoplus_{x \in \mathcal{J}} \left( x_e \boxtimes (x_e)^* \boxtimes K^+ \ \oplus \ x_o \boxtimes (x_o)^* \boxtimes K^+ \right),$$

$$\mathcal{H}^{\mathrm{odd}}_{\mathrm{bulk}}(B) = \bigoplus_{x \in \mathcal{J}} \left( x^s_e \boxtimes (x^s_o)^* \boxtimes K^- \ \oplus \ x^s_o \boxtimes (x^s_e)^* \boxtimes K^- \right). \tag{239}$$

Thus

$$\mathrm{Corr}^{\mathrm{or}}_B(T^2) = \sum_{x \in \mathcal{J}} \left( \chi_{x_e} \otimes_{\mathbb{C}} \overline{\chi}_{(x_e)^*} + \chi_{x_o} \otimes_{\mathbb{C}} \overline{\chi}_{(x_o)^*} - \chi_{x^s_e} \otimes_{\mathbb{C}} \overline{\chi}_{(x^s_o)^*} - \chi_{x^s_o} \otimes_{\mathbb{C}} \overline{\chi}_{(x^s_e)^*} \right). \tag{240}$$

If we rewrite this as a $\tau$-dependent expression in terms of the untwisted and twisted characters (231) of the $BP_{-1/2}$ algebra, we obtain

$$Z_B(\tau) = \sum_{x \in \mathcal{J}} \tfrac{1}{2}\left( |\chi_x(\tau)|^2 + |\widetilde{\chi}_x(\tau)|^2 - |\chi_{x^s}(\tau)|^2 + |\widetilde{\chi}_{x^s}(\tau)|^2 \right). \tag{241}$$

This is indeed modular invariant since the matrices $C$, $D$ and $E$ in the block decomposition (233) are all unitary and the following combinations are separately modular invariant,

$$Z_1(\tau) = \sum_{x \in \mathcal{J}} \left( |\chi_x(\tau)|^2 + |\widetilde{\chi}_x(\tau)|^2 + |\widetilde{\chi}_{x^s}(\tau)|^2 \right), \qquad Z_2(\tau) = \sum_{x \in \mathcal{J}} |\chi_{x^s}(\tau)|^2. \tag{242}$$

The partition function (241) is the super-trace over the full space of states $\mathcal{H}^{\mathrm{even}}_{\mathrm{bulk}}(B) \oplus \mathcal{H}^{\mathrm{odd}}_{\mathrm{bulk}}(B)$ in (239) and so the minus sign in front of $\frac{1}{2} Z_2$ is fixed by the parity of the fields.

---

[16]Here and below we omit the bars over the representation label used to mark the antiholomorphic factor to avoid confusion with the naming of representations in Tables 3 and 4.

### 7.5.3  Results for $F$

In this example we have $F \in \mathcal{C}$, i.e. $F$ is purely even, and so the odd part of all bulk state spaces is zero. The even state spaces again follow from (200) (this time with $\nu = +$) and altogether one finds, unsurprisingly,

$$\mathcal{H}^{\text{NS}}_{\text{bulk}} = \bigoplus_{x \in \mathcal{J}} (x_e + x_o) \boxtimes (x_e + x_o)^*,$$
$$\mathcal{H}^{\text{R}}_{\text{bulk}} = \bigoplus_{x \in \mathcal{J}} (x_e^s + x_o^s) \boxtimes (x_e^s + x_o^s)^*. \tag{243}$$

As in the previous examples, precomposition with $N_F$ is equal to the identity on the multiplicity spaces $M_1$ and to minus the identity on $M_G$. From (184) one reads off the torus partition functions as

$$\text{NS,NS} \qquad \text{Corr}^{\text{spin}}_F(\mathbb{T}^2_{1,1}) = \sum_{x \in \mathcal{J}} (\chi_{x_e} - \chi_{x_o}) \otimes_{\mathbb{C}} (\overline{\chi}_{(x_e)^*} - \overline{\chi}_{(x_o)^*}),$$

$$\text{NS,R} \qquad \text{Corr}^{\text{spin}}_F(\mathbb{T}^2_{1,0}) = \sum_{x \in \mathcal{J}} (\chi_{x_e} + \chi_{x_o}) \otimes_{\mathbb{C}} (\overline{\chi}_{(x_e)^*} + \overline{\chi}_{(x_o)^*}),$$

$$\text{R,NS} \qquad \text{Corr}^{\text{spin}}_F(\mathbb{T}^2_{0,1}) = \sum_{x \in \mathcal{J}} (\chi_{x_e^s} - \chi_{x_o^s}) \otimes_{\mathbb{C}} (\overline{\chi}_{(x_e^s)^*} - \overline{\chi}_{(x_o^s)^*}),$$

$$\text{R,R} \qquad \text{Corr}^{\text{spin}}_F(\mathbb{T}^2_{0,0}) = \sum_{x \in \mathcal{J}} (\chi_{x_e^s} + \chi_{x_o^s}) \otimes_{\mathbb{C}} (\overline{\chi}_{(x_e^s)^*} + \overline{\chi}_{(x_o^s)^*}). \tag{244}$$

In terms of untwisted and twisted characters (231) of the $BP_{-1/2}$ algebra, this can be rewritten as a $\tau$-dependent function as follows:

$$Z^{\text{NS,NS}}_F(\tau) = \sum_{x \in \mathcal{J}} |\widetilde{\chi}_x(\tau)|^2, \qquad\qquad Z^{\text{NS,R}}_F(\tau) = \sum_{x \in \mathcal{J}} |\chi_x(\tau)|^2,$$
$$Z^{\text{R,NS}}_F(\tau) = \sum_{x \in \mathcal{J}} |\widetilde{\chi}_{x^s}(\tau)|^2, \qquad\qquad Z^{\text{R,R}}_F(\tau) = \sum_{x \in \mathcal{J}} |\chi_{x^s}(\tau)|^2. \tag{245}$$

It is easy to check from the block structure (233) and the fact that $C$, $D$ and $E$ are unitary, that $Z^{NS,NS}_F$ and $Z^{R,R}_F$ are invariant under S, and that $Z^{NS,R}_F(\tau) = Z^{R,NS}_F(-1/\tau)$ as required.

Notice that in contrast to the free fermion case in (212), the twisted characters now appear in the NS,NS and R,NS sectors, rather than in NS,R (and R,R, though that sector is zero for the free fermion). The reason is the minus sign in front of the second term in (184) which contributes in the free fermion example, while here there is no parity and only the first term contributes.

**Remark 7.2.**  1. In Table 1 also the product of a BP model with an Ising model is considered. Write $\mathcal{C}_{BP}$ for the modular fusion category spanned by the 24 irreducible representations of the integer weight subalgebra $\text{id}_e$ of $BP_{-1/2}$, and write $\mathcal{C}_{Is}$ for the Ising modular category treated in Section 7.3. We take $\widehat{\mathcal{C}} = \mathcal{C}_{BP} \boxtimes \mathcal{C}_{Is} \boxtimes \mathcal{SVect}$, and one finds that $B := \mathbf{1} \oplus \Pi(G \boxtimes \epsilon)$ is commutative and symmetric, and $F := \mathbf{1} \oplus (G \boxtimes \epsilon)$ is neither commutative nor symmetric but satisfies $N_F^2 = \text{id}_F$.

2. The formalism developed in this paper applies only to rational VOAs, i.e. to VOAs whose category of representations is a modular fusion category. However, if one looks beyond this class of models to include logarithmic examples, one finds that the symplectic fermions [60] are candidates to provide CFTs of types ② and ③, just as the BP model treated in this section.

Namely, the symplectic fermion VOA is of the form $\mathbf{1} \oplus \Pi G$ with $\dim(G) = -1$ and $\theta_G = 1$. Thus $B = \mathbf{1} \oplus \Pi G$ provides an oriented theory with parity, and $F = \mathbf{1} \oplus G$ a spin theory without parity. See [1] for the definition of the symplectic fermion VOA and [39, 81] for an explicit description of the corresponding non-semisimple modular tensor category.

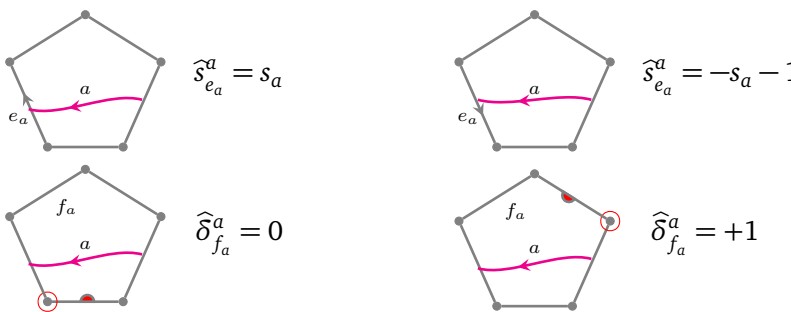

Figure 35: Arc $a \in A(\gamma)$, leaving the face $f_a$ at edge $e_a$. We define $\widehat{\delta}^a_{f_a} = 0$ if the clockwise vertex of $f_a$ is on the left side of $a$ and $\widehat{\delta}^a_{f_a} = 1$ otherwise. Furthermore $\widehat{s}^a_{e_a} = s_{e_a}$ or $-s_{e_a} - 1$ depending on the orientation of $e_a$.

## Acknowledgments

GW would like to thank Andreas Honecker for discussions on the Bershadsky-Polyakov algebra and its representations. We would like to thank Nils Carqueville and the anonymous referees for helpful comments on a draft of this article.

**Funding information** IR is grateful to King's College London for hospitality during the summer term 2022, and to the Aspen Center for Physics for a 2-week stay supported by National Science Foundation grant PHY-1607611. LS is grateful to King's College London for hospitality and support during the week of the Defects and Symmetry meeting in Summer 2022.

IR is partially supported by the Deutsche Forschungsgemeinschaft via the Cluster of Excellence EXC 2121 "Quantum Universe" - 390833306. LS is supported by the Walter Benjamin Fellowship of the Deutsche Forschungsgemeinschaft.

## A Appendix

### A.1 Details for the combinatorial model of $r$-spin structures

#### A.1.1 Proof of Theorem 2.2

By [85, Prop. 3.1.9] isomorphism classes of $r$-spin structures on an open-closed surface with parametrised boundary are in bijection with equivalence classes of admissible markings on a fixed polygonal decomposition of the surface. The equivalence relation is generated by the Moves (M2) and (M3). Recall that Move (M1) is an iterated application of Move (M3). Similarly as observed in [80, Rem. 2.14], the above moves commute, and when one fixes the edge orientations and the marked edges, the equivalence relation is generated by Move (M1).

In Theorem 2.2 we stated that the above holds without specifying the type of the boundary components, i.e. the holonomy along closed boundary components. The construction of [74] of an $r$-spin structure from the combinatorial data remains the same, therefore [85, Prop. 3.1.9] holds by dropping the admissibility condition for the chosen closed boundary vertices $(v_0)$, and by the above discussion, by fixing the edge orientations and markings.

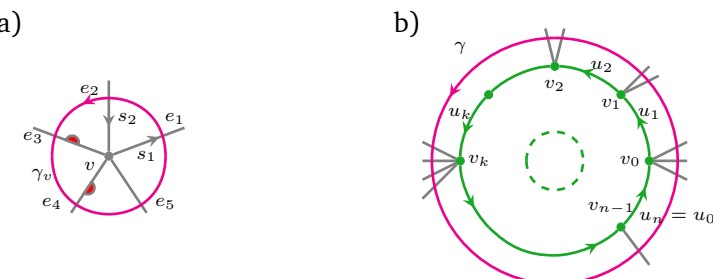

Figure 36: a) anticlockwise oriented loop $\gamma_v$ encircling an inner vertex $v$. b) Loop $\gamma$ parallel to the gluing boundary.

### A.1.2   Combinatorial model and holonomies

Let $\Sigma$ be a surface with parametrised boundary and $T_\Sigma(o, m, s)$ an admissible marked polygonal decomposition for fixed boundary types. Let $\gamma$ be an embedded loop in $\Sigma$ which intersects the edges transversally and does not intersect any vertices. The faces of the polygonal decomposition decompose $\gamma$ into a set $A(\gamma)$ of arcs. We may assume that each arc intersects exactly two edges: we can split edges via (M5). For $a \in A(\gamma)$ let $f_a$ denote the face containing $a$ and $e_a$ the edge where the arc leaves the face $f_a$. We define $\widehat{\delta}^a_{f_a}$ and $\widehat{s}^a_{e_a}$ in Figure 35, see [80, Sec. 2.4] for more details.

**Lemma A.1** (see [80, Prop. 2.15.3]). *The holonomy along $\gamma$ in the $r$-spin structure defined by $T_\Sigma(o, m, s)$ is*

$$\text{hol}(\gamma) = \sum_{a \in A(\gamma)} \left( \widehat{\delta}^a_{f_a} + \widehat{s}^a_{e_a} \right). \tag{A.1}$$

For an inner vertex $v$ write $\gamma_v$ for a small loop encircling $v$ anticlockwise as in Figure 36 a). Next consider a closed boundary component of type $y$ parametrised as in Section 2.2 with $n$ boundary edges $e_0, e_1, \ldots, e_n = e_0$ and $n$ boundary vertices $v_0, v_1, \ldots, v_n = v_0$. Let $\gamma$ be a loop running parallel to the closed gluing boundary, oriented in the same way as the edges on that boundary, see Figure 36 b). Recall that by definition the holonomy along $\gamma$ is $\text{hol}(\gamma) = 1 - y$.

**Lemma A.2.**   *1. The holonomy along $\gamma_v$ for an inner vertex $v$ is*

$$\text{hol}(\gamma_v) = |H_v| - |D_v| + \sum_{h \in H_v} \widehat{s}_h = +1. \tag{A.2}$$

*2. The holonomy along the loop $\gamma$ parallel to a closed gluing boundary of type $y$ is*

$$\text{hol}(\gamma) = \epsilon\Big( |H_{v_0}| - |D_{v_0}| - 1 + \sum_{h \in H_{v_0}} \widehat{s}_h \Big) = 1 - y. \tag{A.3}$$

*Here, $\epsilon = 1$ if the boundary is ingoing and $\epsilon = -1$ if it is outgoing.*

*Proof. Part 1:* We already explained that the holonomy is $+1$ as the $r$-spin structure extends over the vertex. We now show how to obtain the combinatorial expression in (A.2). Consider Figure 36 a) According to Lemma A.1 the contribution of an edge $e_i$ is exactly $\widehat{s}_i$, and the contribution of all faces is $|H_v| - |D_v|$: we count the faces for which the clockwise vertex from the marked edge is not $v$.

*Part 2:* We consider the case where the boundary component is ingoing as shown in Figure 36 b). For an outgoing boundary component the computation is similar and the resulting holonomy is $-(1-y)$. According to (26), the admissibility condition at $v_k$, $k = 1,\ldots,n-1$, is

$$\sum_{\substack{h \in H_{v_k} \\ h \text{ inner}}} \widehat{s}_h - u_k - 1 + u_{k+1} = |D_{v_k}| - |H_{v_k}| + 1 \quad \Leftrightarrow \quad |H_{v_k}| - 2 - |D_{v_k}| + \sum_{\substack{h \in H_{v_k} \\ h \text{ inner}}} \widehat{s}_h = u_k - u_{k+1}. \tag{A.4}$$

By Lemma A.1 the holonomy along $\gamma$ is

$$\sum_{k=0}^{n-1} \Big( |H_{v_k}| - 2 - |D_{v_k}| + \sum_{\substack{h \in H_{v_k} \\ h \text{ inner}}} \widehat{s}_h \Big) \overset{(A.4)}{=} |H_{v_0}| - 2 - |D_{v_0}| + \sum_{\substack{h \in H_{v_0} \\ h \text{ inner}}} \widehat{s}_h + \sum_{k=1}^{n-1}(u_k - u_{k+1})$$

$$= |H_{v_0}| - 2 - |D_{v_0}| + \sum_{\substack{h \in H_{v_0} \\ h \text{ inner}}} \widehat{s}_h + u_1 - u_n$$

$$= |H_{v_0}| - 1 - |D_{v_0}| + \sum_{h \in H_{v_0}} \widehat{s}_h, \tag{A.5}$$

where we used that the loop intersects $|H_{v_k}| - 2$ edges for each vertex $v_k$, and that $\widehat{u}_1 = u_1$ and $\widehat{u}_n = -u_n - 1$. □

### A.1.3 The gluing construction

Recall the gluing construction from Section 2 and the marked polygonal decompositions of the bordisms $\Sigma$ and $\Sigma'$, where the latter is obtained by gluing an in- and an outgoing boundary component along their boundary parametrisation. The edge indices of the glued boundary edges of $\Sigma'$ are

$$s_i = s_i^{\text{in}} + s_i^{\text{out}} \quad (i = 1,\ldots,n). \tag{A.6}$$

Let us first check that the assignment (A.6) results in an admissible marking. We distinguish two situations:

- *Constrained vertices on the gluing boundary:* Suppose $v \in T_{\Sigma'}$ arose from gluing two constrained vertices $v^{\text{in}}, v^{\text{out}} \in V^c$ on the gluing boundaries of $\Sigma$. To check the admissibility condition (26) at $v$, first note that

$$|H_v| = |H_{v^{\text{in}}}| + |H_{v^{\text{out}}}| - 2, \quad |D_v| = |D_{v^{\text{in}}}| + |D_{v^{\text{out}}}|. \tag{A.7}$$

Let $e_i \in H_v$ be the edge on $\Sigma'$ which resulted from gluing $e_i^{\text{in}}$ and $e_i^{\text{out}}$ and which points into $v$, and analogously for $e_{i+1}$ pointing out of $v$. Then the contribution to the admissibility condition at $v$ is

$$\widehat{s}_i + \widehat{s}_{i+1} = \widehat{s}_i^{\text{in}} + \widehat{s}_{i+1}^{\text{in}} + \widehat{s}_i^{\text{out}} + \widehat{s}_{i+1}^{\text{out}} + 1, \tag{A.8}$$

where we used (A.6). Substituting these relation shows that the admissibility condition at $v$ is obtained from adding those at $v^{\text{in}}$ and $v^{\text{out}}$.

- *The vertex $v_0$ on a closed gluing boundary:* Let $v \in T_{\Sigma'}$ arise from gluing $v^{\text{in}}$ and $v^{\text{out}}$ in $T_\Sigma$ as above, but now assume that $v^{\text{in}}, v^{\text{out}}$ are images of $v_0$ under the parametrisation of the closed gluing boundaries. At $v^{\text{in}}, v^{\text{out}}$ the condition on the edge indices is given in (30). Subtracting the two conditions from each other, the $1 - y$ term on the right hand side cancels and the remaining calculation is exactly as above.

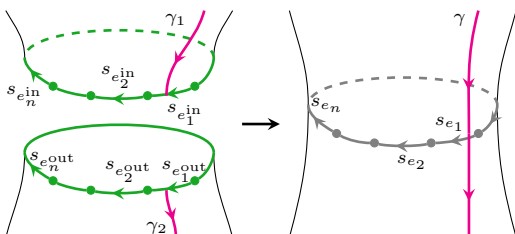

Figure 37: Arc segments before and after gluing along the boundary parametrisation.

Next we turn to comparing the geometric and combinatorial gluing procedure for spin structures. Let $\Sigma = (\Sigma, \sigma)$ and $\Sigma' = (\Sigma', \sigma')$ denote the $r$-spin bordisms defined by the corresponding marked polygonal decompositions $T_\Sigma$ and $T'_{\Sigma'}$ and $\Sigma^{\mathrm{glued}} = (\Sigma', \sigma^{\mathrm{glued}})$ the $r$-spin bordism, where the $r$-spin structure $\sigma^{\mathrm{glued}}$ is defined by gluing the $r$-spin structure $\sigma$ along the boundary parametrisation maps of $\Sigma$.

**Proposition A.3.** *The $r$-spin structures $\sigma'$ and $\sigma^{\mathrm{glued}}$ are isomorphic.*

*Proof.* In the case $n = 1$ this holds by [85, Prop. 3.1.14]. Note that our boundary edges are oriented oppositely to [85], so that $s_1^{\mathrm{StSz}} = -1 - s_1$. In [85] the index of the glued edge is $s_1^{\mathrm{StSz}} = s_1^{\mathrm{StSz,in}} + s_1^{\mathrm{StSz,out}} + 1$, which translates into (A.6) after substitution.

For arbitrary $n \geq 1$ the proof follows the same idea as the proof of [85, Prop. 3.1.14], which we sketch here. Recall from Section 2.1 that isomorphism classes of $r$-spin structures on $\Sigma$ (resp. $\Sigma'$) are in bijection with the set of holonomies assigned to a fixed set of loops and arcs in $\Sigma$ (resp. $\Sigma'$). Let us assume that these loops and arcs agree for $\Sigma$ and $\Sigma'$, except those starting and ending at the chosen gluing boundary components of $\Sigma$ and the one crossing the glued edge in $\Sigma'$. Consider the contribution $h'$ to the holonomy computed for an arc segment crossing the glued edge, and the contributions $h^{\mathrm{in}}$ and $h^{\mathrm{out}}$ for the two arc segments starting and ending on the in- and outgoing boundary components shown in Figure 37. The holonomy for the arc in $\Sigma'$ with $r$-spin structure $\sigma^{\mathrm{glued}}$ is $h^{\mathrm{in}} + h^{\mathrm{out}}$. The computation of holonomies from the combinatorial model in Lemma A.2 can be extended to arcs as in [80, Eq. 2.23], but we do not present the details here. The $r$-spin structures $\sigma'$ and $\sigma^{\mathrm{glued}}$ are isomorphic if and only if $h' = h^{\mathrm{in}} + h^{\mathrm{out}}$, which holds if (A.6) holds. $\qquad\square$

**Remark A.4.** Recall from Theorem 2.2 the bijection between isomorphism classes $\mathrm{Spin}_2^r(\Sigma)/\mathrm{iso}$ of $r$-spin structures on $\Sigma$ (with fixed holonomies along closed gluing boundary components) and the equivalence classes $\mathrm{Marking}(T_\Sigma)/\sim$ of admissible markings of a fixed polygonal decomposition $T_\Sigma$ of $\Sigma$ such that (A.3) holds at the images of $v_0$ on closed gluing boundaries. The edge index assignment (A.6) makes the following diagram of bijections commute:

$$
\begin{array}{ccc}
\mathrm{Spin}_2^r(\Sigma)/\mathrm{iso} & \xrightarrow{\ \text{glue}\ } & \mathrm{Spin}_2^r(\Sigma')/\mathrm{iso} \\
{\scriptstyle\mathrm{Thm.}\,2.2}\big\uparrow & & \big\uparrow{\scriptstyle\mathrm{Thm.}\,2.2} \\
\mathrm{Marking}(T_\Sigma)/\sim & \xrightarrow{\ (A.6)\ } & \mathrm{Marking}(T'_{\Sigma'})/\sim
\end{array}
\qquad
\begin{array}{ccc}
[\sigma] & \longmapsto & [\sigma^{\mathrm{glued}}] \ =\!=\ [\sigma'] \\
\big\uparrow & & \big\uparrow \\
[T_\Sigma(\sigma)] & \longmapsto & [T'_{\Sigma'}(\sigma')]
\end{array}
$$
$$(A.9)$$

This is the only edge index assignment that makes the above diagram commute for every surface.

## A.2 Invariance of correlators under local moves

We show invariance of correlators defined in Section 6 under the local moves (M1)-(M5) on marked polygonal decompositions. Recall that (M1) is a repeated application of (M2). These

are the moves shown in Figures 6, 7 and 11. The graphs of $F$-defects assigned to plaquettes and edges are shown in Figures 25, 29, and 30. The manipulation rules for networks of $F$-defects are summarised in Figure 28.

**Moves involving edges without free boundary or defect lines**

**Lemma A.5.** *The correlators are invariant under the move* (M2)*.*

*Proof.* We compare what is assigned to the two configurations:

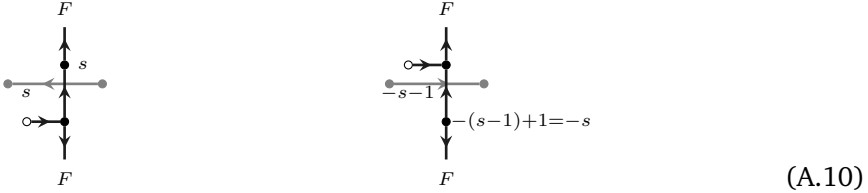

$$(A.10)$$

These are equal, as the Nakayama automorphism is a morphism of Frobenius algebras. □

**Lemma A.6.** *The correlators are invariant under the move* (M3)*.*

*Proof.* By Lemma A.5 the construction is invariant under changing the edge orientation, so we can fix a particular orientation.

We compare what is assigned to the two configurations:

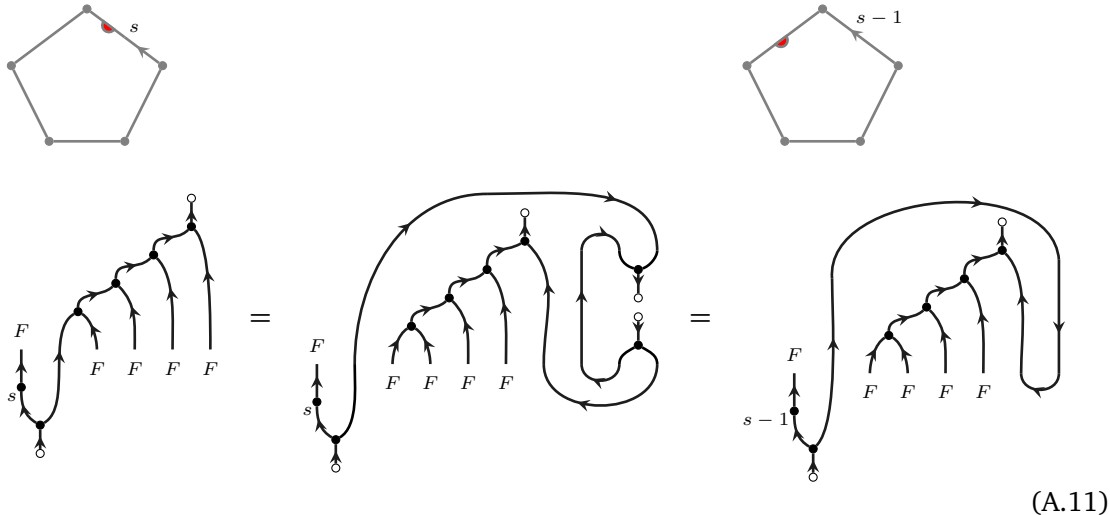

$$(A.11)$$

Here we used the definition of $N$ and that $N$ is a Frobenius algebra morphism. □

**Lemma A.7.** *The correlators are invariant under the move* (M4)*.*

*Proof.* By Lemma A.6 the construction is invariant under changing the marked edge of a face, so we can fix a particular edge of the face on the left. The two sides agree by the Frobenius

relation and (co)unitality:

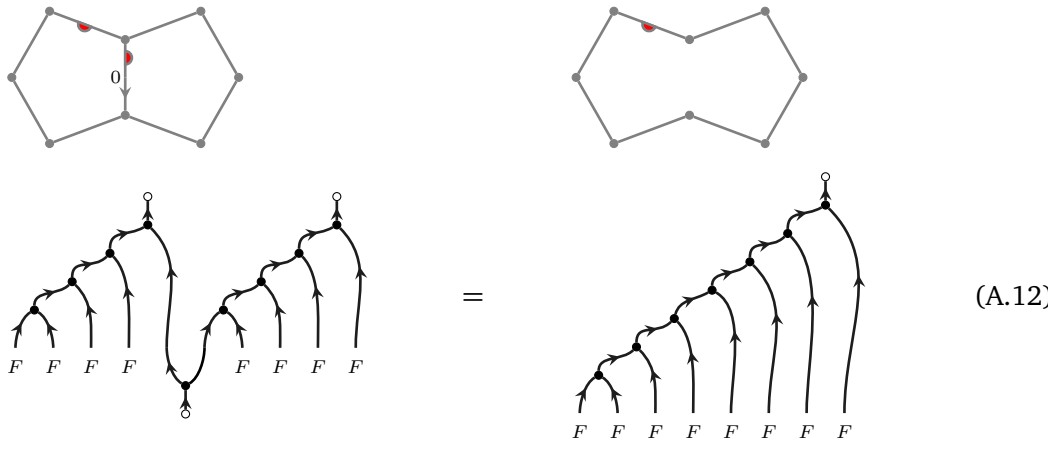

$$(A.12)$$

□

**Lemma A.8.** *The correlators are invariant under the move* (M5)*.*

*Proof.* By Lemma A.6 we have invariance under move (M3) and hence also under move (M1), which we use to set the index of the edge we want to split to zero. The construction assigns the left hand side of (A.12) to the two faces before splitting the edge. When we split the edge we get

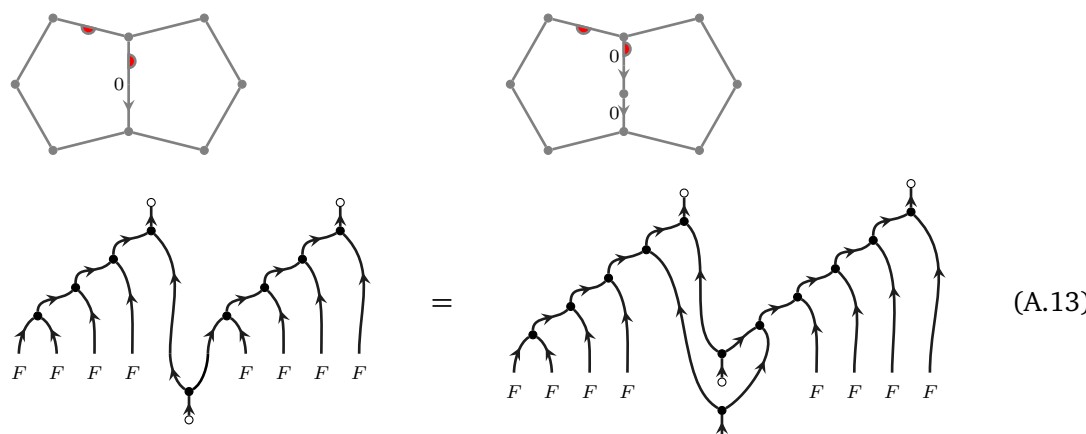

$$(A.13)$$

which is the same by applying associativity, the Frobenius relation and use that $F$ is special and unital.                                                                                                                □

**Moves involving the free boundary**

**Lemma A.9.** *The correlators are invariant under the move* (M2) *in the presence of boundary edges.*

*Proof.* This amounts to the following equality of defect networks:

$$(A.14)$$

□

By the previous lemma we can fix the orientations of inner edges so that the defect lines for the free boundary always cross them as shown on the left hand side of (A.14). Note that the orientation of edges on the gluing boundary is fixed.

Since we fixed the marked edge on every face touching the free boundary to the edge where the boundary leaves the face, we need to check invariance under moves which keep this choice of marked edges.

**Lemma A.10.** *The correlators are invariant under the move* (M1) *in the presence of boundary edges. .*

*Proof.* This holds by the $\mathbb{Z}_2$-equivariant property:

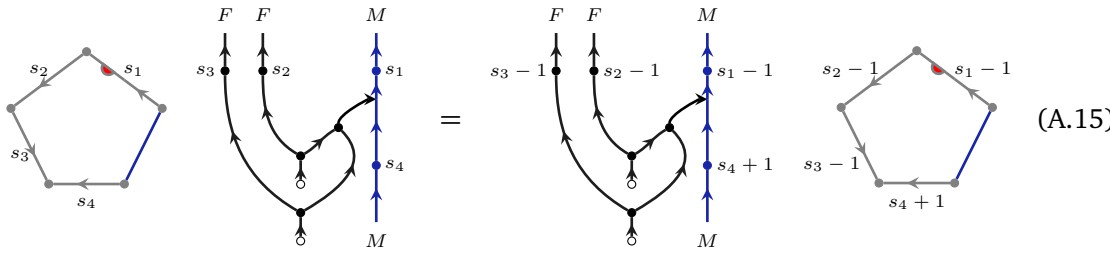

$$\tag{A.15}$$

$\square$

**Lemma A.11.** *The correlators are invariant under the move* (M4) *in the presence of boundary edges.*

*Proof.* Here we use again that $F$ is associative furthermore that $Y$ is a left $F$-module. Recall that bivalent vertices on the free boundary have no effect on the defect network.

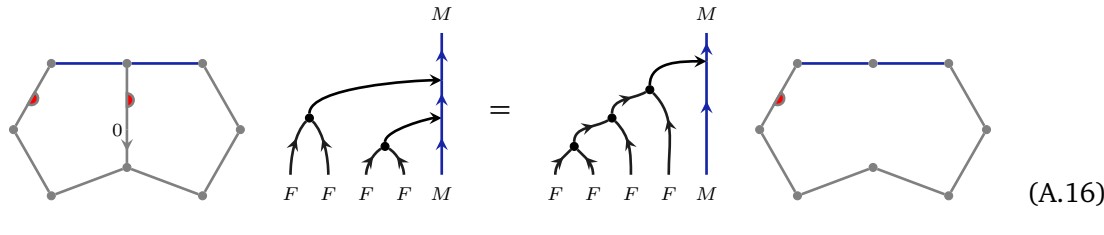

$$\tag{A.16}$$

$\square$

We furthermore have invariance under move (M5). The proof is analogous to that of Lemma A.15 below (in the same way that the proofs of Lemmas A.11 and A.13 are), and we skip it here.

**Moves involving defect lines**

The proof of the next lemma is the same as for Lemma A.9:

**Lemma A.12.** *The correlators are invariant under the move* (M2) *in the presence of defect lines.*

We can thus fix the orientations of inner edges so that the defect lines always cross them as shown on the left hand side of Figure 29 b).

**Lemma A.13.** *The correlators are invariant under the move* (M1) *in the presence of defect lines.*

*Proof.* This holds by the $\mathbb{Z}_2$-equivariant property:

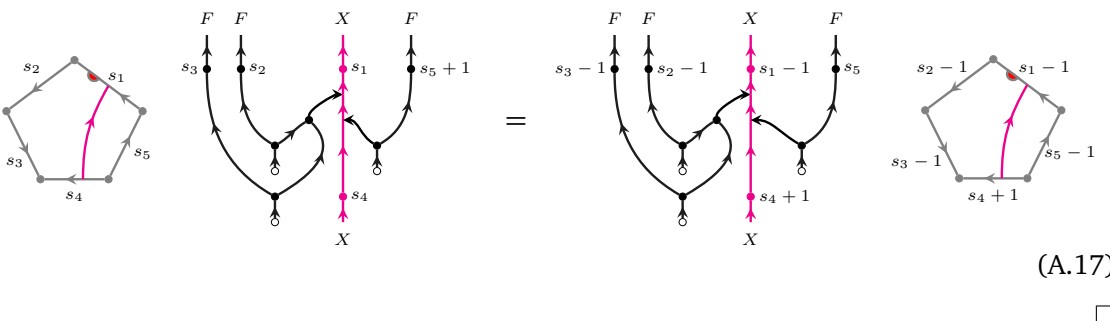

$$\tag{A.17}$$

$\square$

**Lemma A.14.** *The correlators are invariant under the move* (M4) *in the presence of defect lines.*

*Proof.* We use the bimodule property of $X$.

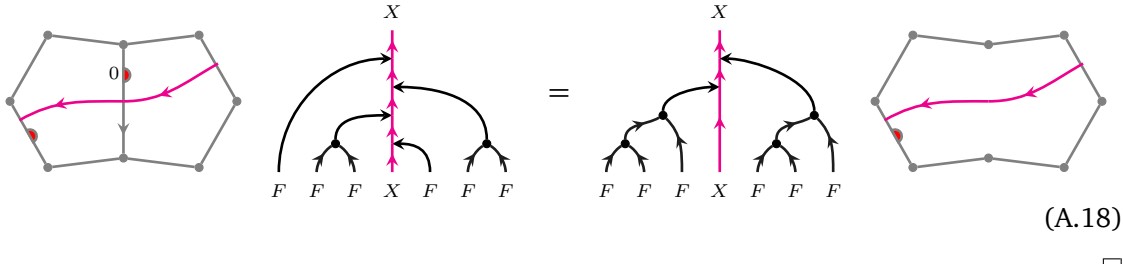

$$\tag{A.18}$$

$\square$

**Lemma A.15.** *The correlators are invariant under the move* (M5) *in the presence of defect lines.*

*Proof.* The left hand side is that of (A.18). We use that $X$ is a bimodule and $F$ is a special Frobenius algebra.

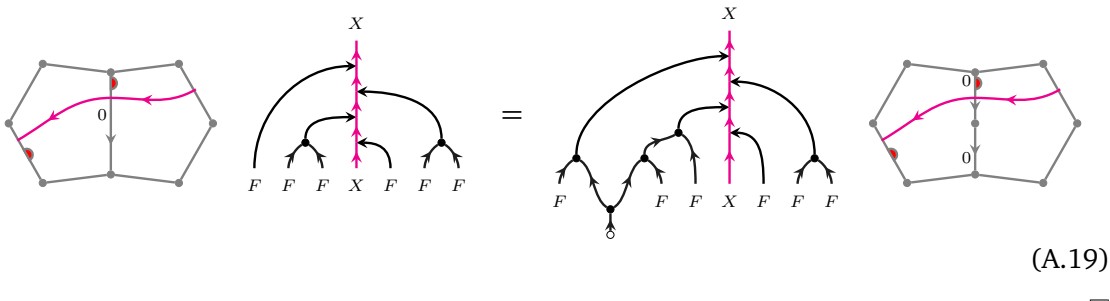

$$\tag{A.19}$$

$\square$

## A.3 Proof of factorisation

**Bulk factorisation with defects**

We follow the strategy of [37] and start with explaining how we can reduce the proof of factorisation to the presence of one defect line.

Consider a world sheet $\Sigma^{(n)}$ and an embedded circle $S$ crossed by $n$ defect lines labelled by bimodules $X_i$ ($i = 1, ..., n$) as in Figure 38 a). Furthermore consider the connecting manifold $M_{\Sigma^{(n)}}$ and the manifold $M'$ which agrees with $M_{\Sigma^{(n)}}$ except inside a ball containing all crossings of $S$ and the defect lines as shown in Figure 38 c). There $S$ is crossed by a single defect line labelled by the $B$-$B$-bimodule $X = X_n \otimes_{B_n} \cdots \otimes_{B_2} X_1$ ($B = B_1$). Note that we assumed that all defect lines point in the same direction, which we can do by relabelling by dual bimodules if needed. The next lemma follows from a similar discussion as in [37, Sec. 3.2.3]:

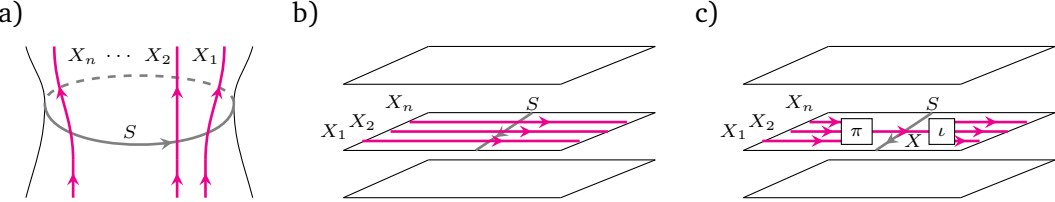

Figure 38: a) Embedded circle $S$ crossing $n$ defect lines in $\Sigma^{(n)}$. b) Detail of the connecting manifold of $\Sigma^{(n)}$ with embedded circle $S$ crossing $n$ defect lines. c) Modified 3-manifold, the morphisms $\iota$ and $\pi$ are the embedding and projection for the relative tensor product. The circle $S$ is now only crossed by one defect line labelled $X$.

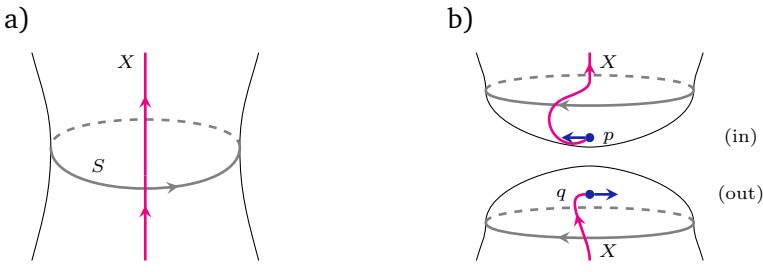

Figure 39: Cutting/gluing a world sheet along an embedded circle $S$.

**Lemma A.16.** *The correlator* $\mathrm{Corr}(\Sigma^{(n)})$ *can be computed as*

$$\mathrm{Corr}(\Sigma^{(n)}) \overset{\text{def.}}{=} \widehat{\mathcal{Z}}_{\mathcal{C}}(M_{\Sigma^{(n)}}) = \widehat{\mathcal{Z}}_{\mathcal{C}}(M'). \tag{A.20}$$

This lemma implies that there is no loss of generality if we assume that only one defect line is present where we glue. From now on we assume that the circle $S$ crosses a single defect line labelled by the $B$-$B$-bimodule $X$, and we will write $\Sigma' = \Sigma^{(1)}$ for the corresponding world sheet before cutting along $S$.

Let $U, \bar{V} \in \mathcal{C}$, $\epsilon \in \{\pm\}$ and let

$$\mathcal{J}_0, \mathcal{J}_1 \subset \widehat{\mathcal{C}}((U \otimes \bar{V}) \boxtimes K^\epsilon, X), \quad \text{and} \quad \bar{\mathcal{J}}_0, \bar{\mathcal{J}}_1 \subset \widehat{\mathcal{C}}(X, (U \otimes \bar{V}) \boxtimes K^\epsilon), \tag{A.21}$$

denote an eigenbasis of the idempotents $Q_0^{(\text{in})}$ and $Q_0^{(\text{out})}$ from (89) with eigenvalues 0 and 1 respectively which are dual with respect to the non-degenerate pairing (92):

$$\langle \bar{\beta}, \alpha \rangle_{\text{bulk}} = \delta_{\alpha,\beta}, \tag{A.22}$$

for $\alpha \in \mathcal{J}_i$ and $\bar{\beta} \in \bar{\mathcal{J}}_j$ with $i, j \in \{0, 1\}$. Note that by (93) this pairing is compatible with the idempotents $Q_0^{(\text{in/out})}$. By construction $\mathcal{J}_1$ is a basis of the multiplicity space $\mathcal{M}_{0,\epsilon}^{(\text{in})}(U, \bar{V}; X)$ and

$$\mathcal{J} := \mathcal{J}_0 \cup \mathcal{J}_1, \tag{A.23}$$

is a basis of $\widehat{\mathcal{C}}((U \otimes \bar{V}) \boxtimes K^\epsilon, X)$. Similarly we set $\bar{\mathcal{J}} := \bar{\mathcal{J}}_0 \cup \bar{\mathcal{J}}_1$.

Consider the world sheet $\Sigma(U, \bar{V}, \epsilon, \alpha, \bar{\beta})$ from (118) and the glued world sheet $\Sigma'$ with $\alpha \in \mathcal{J}_1$ and $\bar{\beta} \in \bar{\mathcal{J}}_1$, see Figure 39.

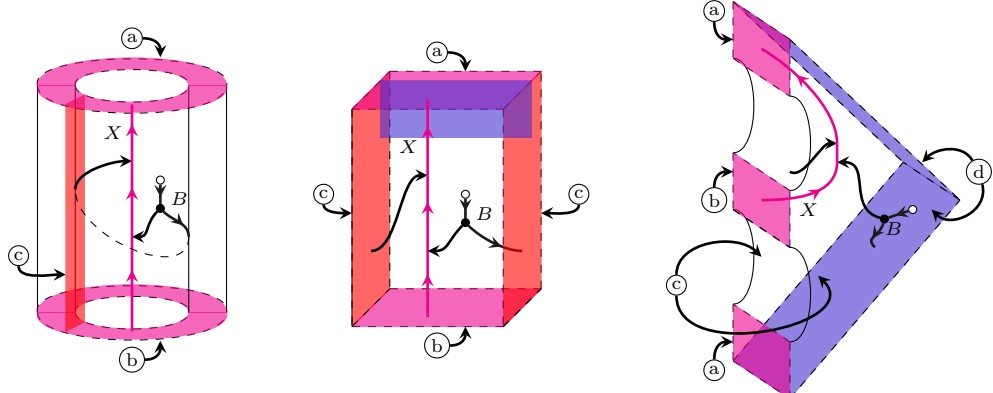

Figure 40: Wedge representation of a detail of the connecting manifold $M_{\Sigma'}$ of the glued surface.

Recall the gluing manifold $G_{U,\bar{V},\epsilon}$ from (122) and consider the connecting manifolds $M_{\Sigma(U,\bar{V},\epsilon,\alpha,\bar{\beta})}$, $M_{\Sigma'}$ and $G_{U,\bar{V},\epsilon} \circ M_{\Sigma(U,\bar{V},\epsilon,\alpha,\bar{\beta})}$ in Figures 40 and 41. We define the solid tori $\mathcal{T}_X, \mathcal{T}_{X;U,\bar{V},\epsilon}^{\alpha,\bar{\beta}} \colon \emptyset \longrightarrow T_X^2$ as in Figure 42 and define $M_{\Sigma',T^2} \colon T_X^2 \longrightarrow \Sigma'$ by cutting out the solid torus $\mathcal{T}_X$ from $M_{\Sigma'}$:

$$M_{\Sigma'} = M_{\Sigma',T^2} \circ \mathcal{T}_X. \tag{A.24}$$

By looking at Figure 41 we directly get:

**Lemma A.17.** *We have*

$$G_{U,\bar{V},\epsilon} \circ M_{\Sigma(U,\bar{V},\epsilon,\alpha,\bar{\beta})} = M_{\Sigma',T^2} \circ \mathcal{T}_{X;U,\bar{V},\epsilon}^{\alpha,\bar{\beta}}. \tag{A.25}$$

In order to be able to compare correlators on $\Sigma(U,\bar{V},\epsilon,\alpha,\bar{\beta})$ and $\Sigma'$ we need to consider how the invariants of the solid tori in Figure 42 are related. For this we need some preparation.

**Lemma A.18.** *The vectors*

$$v_{U,\bar{V},\epsilon}^{\alpha,\bar{\beta}} = \widehat{\mathcal{Z}}_{\mathcal{C}}(\mathcal{T}_{X;U,\bar{V},\epsilon}^{\alpha,\bar{\beta}}), \quad \text{and} \quad \bar{v}_{U,\bar{V},\epsilon}^{\bar{\alpha},\beta} = \frac{1}{(\widehat{\mathcal{Z}}_{\mathcal{C}}(S^3))^2 \epsilon \dim(U)\dim(\bar{V})} \widehat{\mathcal{Z}}_{\mathcal{C}}(\bar{\mathcal{T}}_{X;U,\bar{V},\epsilon}^{\bar{\alpha},\beta}), \tag{A.26}$$

*for $\alpha \in \mathcal{J}$ and $\bar{\beta} \in \bar{\mathcal{J}}$ and $U, \bar{V}$ irreducible objects in $\mathcal{C}$, form a dual basis pair of $\widehat{\mathcal{Z}}_{\mathcal{C}}(T_X^2)$ and $\widehat{\mathcal{Z}}_{\mathcal{C}}(T_X^2)^*$ respectively.*

*Proof.* Using (70) we can write

$$
\begin{aligned}
\widehat{\mathcal{Z}}_{\mathcal{C}}(T_X^2) &\cong \bigoplus_{\epsilon \in \{\pm\}} \widehat{\mathcal{C}}(X \otimes X^*, L \boxtimes K^\epsilon) \otimes K^\epsilon \\
&\cong \bigoplus_{\epsilon \in \{\pm\}} \widehat{\mathcal{C}}(\mathbf{1}, X^* \otimes (L \boxtimes K^\epsilon) \otimes X) \otimes K^\epsilon \\
&\cong \bigoplus_{\epsilon \in \{\pm\}} \bigoplus_i \widehat{\mathcal{C}}(\mathbf{1}, X^* \otimes (S_i \otimes S_i^*) \boxtimes K^\epsilon \otimes X) \otimes K^\epsilon \\
&\stackrel{(*)}{\cong} \bigoplus_{\epsilon \in \{\pm\}} \bigoplus_{i,j} \widehat{\mathcal{C}}(\mathbf{1}, X^* \otimes (S_i \otimes S_j) \boxtimes K^\epsilon) \otimes \widehat{\mathcal{C}}(\mathbf{1}, (S_j^* \otimes S_i^*) \boxtimes K^\epsilon \otimes X) \otimes K^\epsilon \\
&\cong \bigoplus_{\epsilon \in \{\pm\}} \bigoplus_{i,j} \widehat{\mathcal{C}}(X, (S_i \otimes S_j) \boxtimes K^\epsilon) \otimes \widehat{\mathcal{C}}((S_i \otimes S_j) \boxtimes K^\epsilon, X) \otimes K^\epsilon.
\end{aligned}
\tag{A.27}
$$

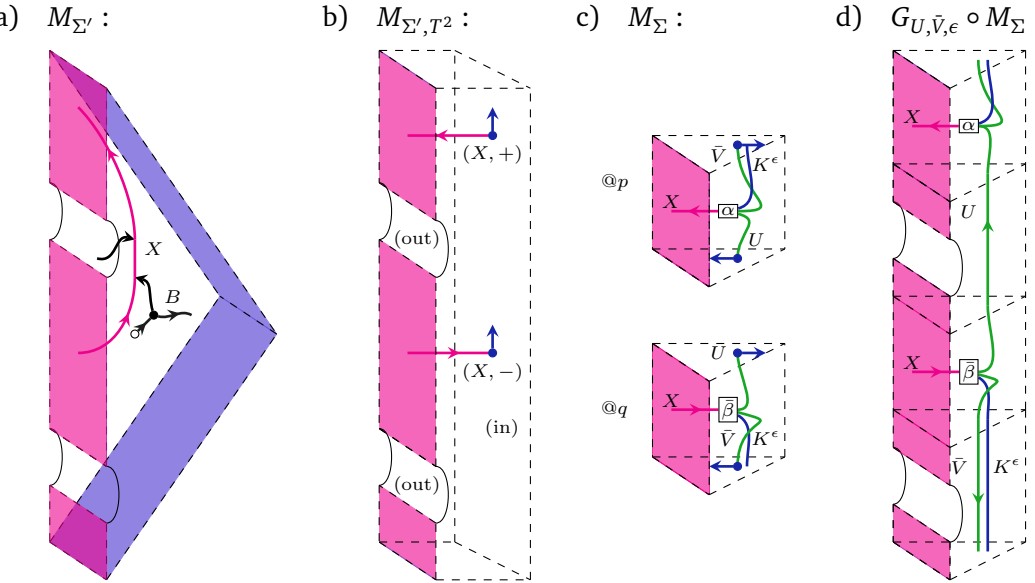

Figure 41: Details of connecting manifolds $M_{\Sigma'}$ (Part *a*), $M_{\Sigma',T^2}$ (Part *b*), $M_\Sigma = M_{\Sigma(U,\bar{V},\epsilon,\alpha,\bar{\beta})}$ (Part *c*) and $G_{U,\bar{V},\epsilon} \circ M_\Sigma = G_{U,\bar{V},\epsilon} \circ M_{\Sigma(U,\bar{V},\epsilon,\alpha,\bar{\beta})}$ (Part *d*). In c) we have rotated one of the tangent vectors in each of the pieces shown by 180° relative to the definition in Figure 18 in order to be consistent with the paper plane framing implicit when drawing ribbons just as lines.

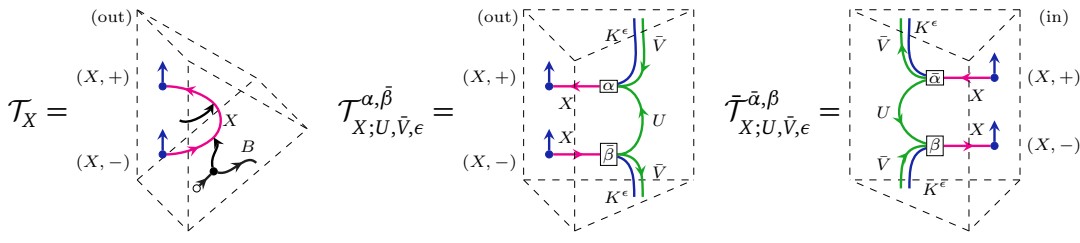

Figure 42: The solid tori $\mathcal{T}_X, \mathcal{T}_{X;U,\bar{V},\epsilon}^{\alpha,\bar{\beta}} : \emptyset \longrightarrow T_X^2$ and $\bar{\mathcal{T}}_{X;U,\bar{V},\epsilon}^{\bar{\alpha},\beta} : T_X^2 \longrightarrow \emptyset$.

In step (∗) we used [87, Lem. IV.2.2.2]. The inverse chain of isomorphisms sends $\alpha \otimes \bar{\beta} \longmapsto v_{S_i,S_j,\epsilon}^{\alpha,\bar{\beta}}$.

Now we check that we indeed have a dual basis. We have

$$\bar{v}_{U,\bar{V},\epsilon}^{\bar{\alpha},\beta} \circ v_{U',\bar{V}',\epsilon'}^{\alpha',\bar{\beta}'} = \frac{1}{(\widehat{\mathcal{Z}}_\mathcal{C}(S^3))^2 \epsilon \dim(U)\dim(\bar{V})} \widehat{\mathcal{Z}}_\mathcal{C}(\bar{\mathcal{T}}_{X;U,\bar{V},\epsilon}^{\bar{\alpha},\beta} \circ \mathcal{T}_{X;U',\bar{V}',\epsilon'}^{\alpha',\bar{\beta}'}). \tag{A.28}$$

The 3-manifold on the right hand side is $S^2 \times S^1$ with a ribbon graph:

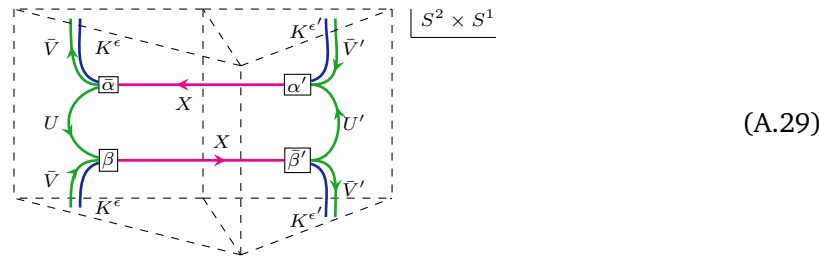

$$\tag{A.29}$$

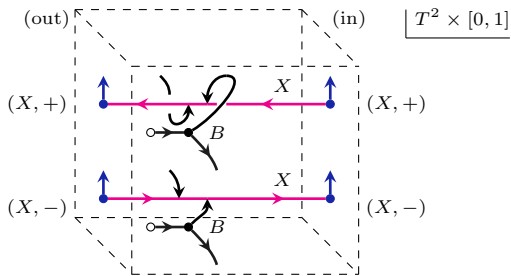

Figure 43: The bordism $\mathcal{P}_X : T_X^2 \longrightarrow T_X^2$.

Using the identity

$$
\left| \begin{array}{cccc} U & K^\epsilon & K^{\epsilon'} & (U')^* \\ & & & \end{array} \right| = \frac{\delta_{\epsilon,\epsilon'}\,\delta_{U,U'}}{\epsilon \dim(U)} \; \overset{U\ \ K^\epsilon\ K^\epsilon\ U^*}{\underset{U\ \ K^\epsilon\ K^\epsilon\ U^*}{\smile}} + \text{(other terms with intermediate simple object} \not\cong \mathbf{1}),
$$

$$(A.30)$$

and that the dimension of the state space on $S^2$ with a single insertion of the simple object $S_i$ is $\dim(\widehat{\mathcal{Z}}(S^2(i))) = \delta_{S_i,\mathbf{1}}$ we compute

$$
\bar{v}_{U,\bar{V},\epsilon}^{\bar{\alpha},\beta} \circ v_{U',\bar{V}',\epsilon'}^{\alpha',\bar{\beta}'} = \frac{1}{(\widehat{\mathcal{Z}}_{\mathcal{C}}(S^3))^2 \epsilon \dim(U)\dim(\bar{V})} \widehat{\mathcal{Z}}_{\mathcal{C}}\left( \vcenter{\hbox{\includegraphics{}}} \right)
$$

$$
= \frac{\delta_{\epsilon,\epsilon'}\,\delta_{U,U'}\,\delta_{\bar{V},\bar{V}'}}{(\widehat{\mathcal{Z}}_{\mathcal{C}}(S^3)\epsilon \dim(U)\dim(\bar{V}))^2} \widehat{\mathcal{Z}}_{\mathcal{C}}\left( \vcenter{\hbox{\includegraphics{}}} \right)
$$

$$(A.31)$$

$$
= \delta_{\epsilon,\epsilon'}\,\delta_{U,U'}\,\delta_{\bar{V},\bar{V}'} \langle \alpha, \bar{\alpha}' \rangle_{\text{bulk}} \langle \beta, \bar{\beta}' \rangle_{\text{bulk}} \widehat{\mathcal{Z}}_{\mathcal{C}}(S^2 \times S^1)
$$

$$
= \delta_{\epsilon,\epsilon'}\,\delta_{U,U'}\,\delta_{\bar{V},\bar{V}'}\,\delta_{\alpha,\alpha'}\,\delta_{\beta,\beta'} \underbrace{\dim(\widehat{\mathcal{Z}}_{\mathcal{C}}(S^2))}_{=1}.
$$

$\square$

**Lemma A.19.** *Consider the bordism* $\mathcal{P}_X : T_X^2 \longrightarrow T_X^2$ *in Figure 43.*

*1. $\widehat{\mathcal{Z}}_{\mathcal{C}}(\mathcal{P}_X)$ is idempotent and*

$$
\widehat{\mathcal{Z}}_{\mathcal{C}}(\mathcal{P}_X) \circ \widehat{\mathcal{Z}}_{\mathcal{C}}(\mathcal{T}_X) = \widehat{\mathcal{Z}}_{\mathcal{C}}(\mathcal{T}_X). \tag{A.32}
$$

2. For $\alpha \in \mathcal{J}$ and $\bar{\beta} \in \bar{\mathcal{J}}$ we have

$$\widehat{\mathcal{Z}}_{\mathcal{C}}(\mathcal{P}_X) \circ v^{\alpha,\bar{\beta}}_{X;U,\bar{V},\epsilon} = \delta_{\alpha \in \mathcal{J}_1} \delta_{\bar{\beta} \in \bar{\mathcal{J}}_1} v^{\alpha,\bar{\beta}}_{X;U,\bar{V},\epsilon} . \tag{A.33}$$

3. The vectors $\{v^{\alpha,\bar{\beta}}_{X;U,\bar{V},\epsilon}\}_{\alpha \in \mathcal{J}_1, \bar{\beta} \in \bar{\mathcal{J}}_1}$ span the image of $\widehat{\mathcal{Z}}_{\mathcal{C}}(\mathcal{P}_X)$.

*Proof. Part 1:* is a straightforward computation using that $B$ is special.

*Part 2:* We compute $\widehat{\mathcal{Z}}_{\mathcal{C}}(\mathcal{P}_X) \circ v^{\alpha,\bar{\beta}}_{X;U,\bar{V},\epsilon}$

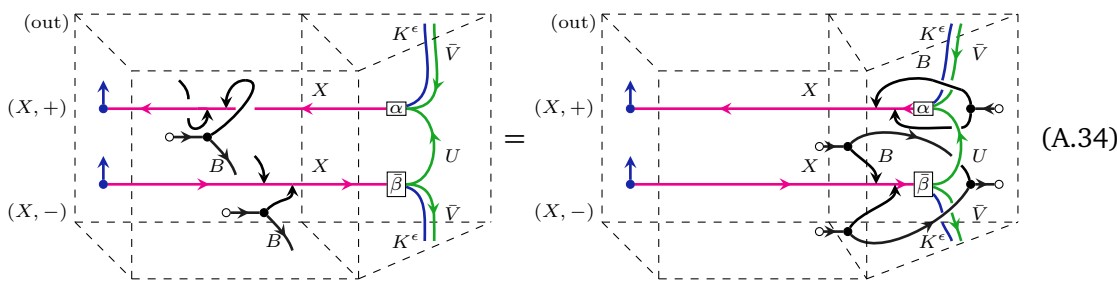

$$\tag{A.34}$$

which is $v^{\alpha,\bar{\beta}}_{X;U,\bar{V},\epsilon}$ if $\alpha \in \mathcal{J}_1$ and $\bar{\beta} \in \bar{\mathcal{J}}_1$, and which is 0 otherwise.

*Part 3:* Clear from Lemma A.18 and Part 2. $\qquad\square$

**Lemma A.20.** *We have*

$$\widehat{\mathcal{Z}}_{\mathcal{C}}(\mathcal{T}_X) = \sum_{U,\bar{V},\epsilon} \sum_{\alpha \in \mathcal{J}_1} \widehat{\mathcal{Z}}_{\mathcal{C}}(\mathcal{T}^{\alpha,\bar{\alpha}}_{X;U,\bar{V},\epsilon}) . \tag{A.35}$$

*Proof.* We can express $\widehat{\mathcal{Z}}_{\mathcal{C}}(\mathcal{T}_X)$ as a linear combination

$$\widehat{\mathcal{Z}}_{\mathcal{C}}(\mathcal{T}_X) \stackrel{(1)}{=} \sum_{U,\bar{V},\epsilon} \sum_{\substack{\alpha \in \mathcal{J}, \\ \bar{\beta} \in \bar{\mathcal{J}}}} K^{\alpha,\bar{\beta}}_{U,\bar{V},\epsilon} v^{\alpha,\bar{\beta}}_{U,\bar{V},\epsilon} \stackrel{(2)}{=} \sum_{U,\bar{V},\epsilon} \sum_{\substack{\alpha \in \mathcal{J}_1, \\ \bar{\beta} \in \bar{\mathcal{J}}_1}} K^{\alpha,\bar{\beta}}_{U,\bar{V},\epsilon} v^{\alpha,\bar{\beta}}_{U,\bar{V},\epsilon} . \tag{A.36}$$

Step 1 uses the basis from Lemma A.18. Step 2 follows from Lemma A.19, which states that $\widehat{\mathcal{Z}}_{\mathcal{C}}(\mathcal{T}_X)$ is in the image of the idempotent $\widehat{\mathcal{Z}}_{\mathcal{C}}(\mathcal{P}_X)$ and hence the sum over basis elements can be restricted from $\mathcal{J}$ to $\mathcal{J}_1$.

To determine the coefficients $K^{\alpha,\bar{\beta}}_{U,\bar{V},\epsilon}$ we evaluate a dual basis element in $\widehat{\mathcal{Z}}_{\mathcal{C}}(T^2_X)^*$ from Lemma A.18 on (A.36):

$$K^{\gamma,\bar{\delta}}_{R,\bar{S},\nu} = \bar{v}^{\bar{\gamma},\delta}_{X;R,\bar{S},\nu} \circ \widehat{\mathcal{Z}}_{\mathcal{C}}(\mathcal{T}_X) = \frac{1}{(\widehat{\mathcal{Z}}_{\mathcal{C}}(S^3))^2 \nu \dim(R) \dim(\bar{S})} \widehat{\mathcal{Z}}_{\mathcal{C}}(\bar{\mathcal{T}}^{\bar{\gamma},\delta}_{X;R,\bar{S},\nu}) \circ \widehat{\mathcal{Z}}_{\mathcal{C}}(\mathcal{T}_X) . \tag{A.37}$$

On the right hand side we have $\widehat{\mathcal{Z}}_{\mathcal{C}}$ evaluated on $S^3$ with a ribbon graph in it, which is

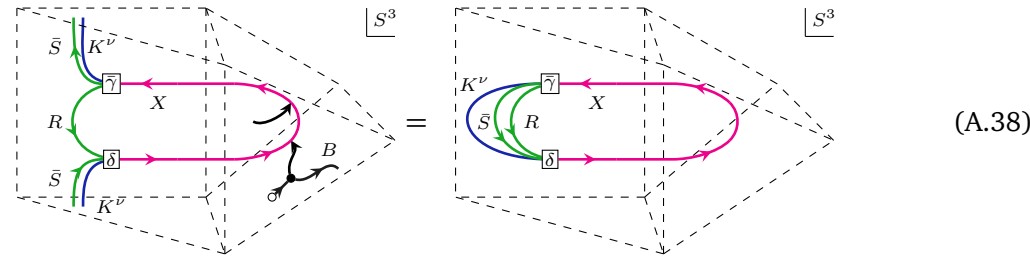

$$\tag{A.38}$$

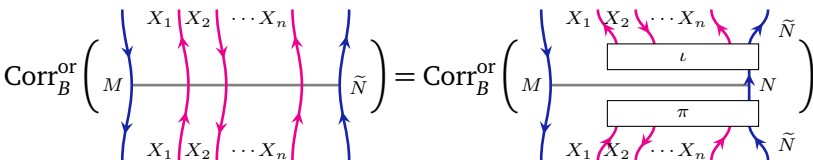

Figure 44: Reducing the boundary factorisation to the case without defect lines. Here, $N = X_1 \otimes_B \cdots \otimes_B X_n \otimes_B \widetilde{N}$.

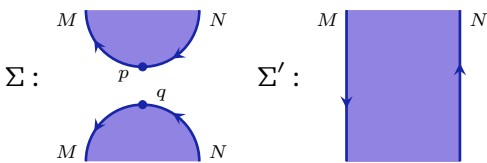

Figure 45: The cut world sheet $\Sigma = \Sigma(W, \epsilon, \alpha, \bar\beta)$ and the glued world sheet $\Sigma'$.

So altogether using (92) we have

$$K^{\gamma,\bar\delta}_{R,\bar S,\nu} = \frac{1}{\widehat{\mathcal{Z}}_{\mathcal{C}}(S^3)\nu \dim(R)\dim(\bar S)} \operatorname{tr}_{R \otimes \bar S \boxtimes K^\nu}(\bar\gamma \circ \delta) = \langle \bar\gamma, \delta \rangle_{\mathrm{bulk}} = \delta_{\gamma,\delta} \,. \tag{A.39}$$

$\square$

*Proof of Theorem 5.6 (closed case).* Combining Lemmas A.17 and A.20 we obtain

$$
\begin{aligned}
\mathrm{Corr}^{\mathrm{or}}_B(\Sigma') &\overset{\text{def.}}{=} \widehat{\mathcal{Z}}_{\mathcal{C}}(M_{\Sigma'}) \overset{(\mathrm{A.24})}{=} \widehat{\mathcal{Z}}_{\mathcal{C}}(M_{\Sigma',T^2}) \circ \widehat{\mathcal{Z}}_{\mathcal{C}}(\mathcal{T}_X) \\
&\overset{\text{Lem.\,A.20}}{=} \widehat{\mathcal{Z}}_{\mathcal{C}}(M_{\Sigma',T^2}) \circ \Big( \sum_{U,\bar V,\epsilon} \sum_{\alpha \in \mathcal{J}_1} \widehat{\mathcal{Z}}_{\mathcal{C}}(\mathcal{T}^{\alpha,\bar\alpha}_{X;U,\bar V,\epsilon}) \Big) \\
&\overset{\text{Lem.\,A.17}}{=} \sum_{U,\bar V,\epsilon} \sum_{\alpha \in \mathcal{J}_1} \widehat{\mathcal{Z}}_{\mathcal{C}}(G_{U,\bar V,\epsilon}) \circ \widehat{\mathcal{Z}}_{\mathcal{C}}(M_{\Sigma(U,\bar V,\epsilon,\alpha,\bar\alpha)}) \\
&\overset{\text{def.}}{=} \sum_{U,\bar V,\epsilon} \sum_{\alpha \in \mathcal{J}_1} \widehat{\mathcal{Z}}_{\mathcal{C}}(G_{U,\bar V,\epsilon})(\mathrm{Corr}^{\mathrm{or}}_B(\Sigma(U,\bar V,\epsilon,\alpha,\bar\alpha))) \,.
\end{aligned}
\tag{A.40}
$$

We did not discuss the gluing anomaly in terms of Maslov indices in detail here, but it is shown in [36, Thm. 3.9] and in [40, Sec. 5.1] that the relevant Maslov indices vanish in this computation. $\square$

**Boundary factorisation with defects**

Similarly as in the case of bulk factorisation, we can reduce the proof of boundary factorisation to the situation where we glue a world sheet $\Sigma$ at free boundary insertions where no defect lines meet, see Figure 44. From now on we assume this. Let

$$\mathcal{J} \subset \widehat{\mathcal{C}}(W \otimes K^\epsilon, M^* \otimes_B N), \quad \text{and} \quad \bar{\mathcal{J}} \subset \widehat{\mathcal{C}}(M^* \otimes_B N, W \otimes K^\epsilon), \tag{A.41}$$

denote bases of the in- and outgoing boundary multiplicity spaces, dual with respect to the pairing $\langle -,- \rangle_{\mathrm{bnd}}$ in (94). Consider the world sheet $\Sigma(W, \epsilon, \alpha, \bar\beta)$ for $\alpha \in \mathcal{J}$ and $\bar\beta \in \bar{\mathcal{J}}$ from (119) and its gluing $\Sigma'$ in Figure 45. The corresponding connecting manifolds and the gluing manifold are in Figure 46. We will need the following lemma.

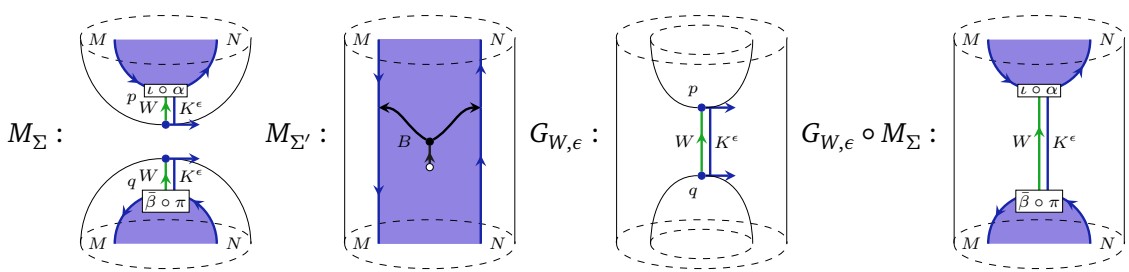

Figure 46: The connecting manifolds of $\Sigma = \Sigma(W, \epsilon, \alpha, \bar{\beta})$ and $\Sigma'$. The shaded areas show the world sheets.

**Lemma A.21.** *We have*

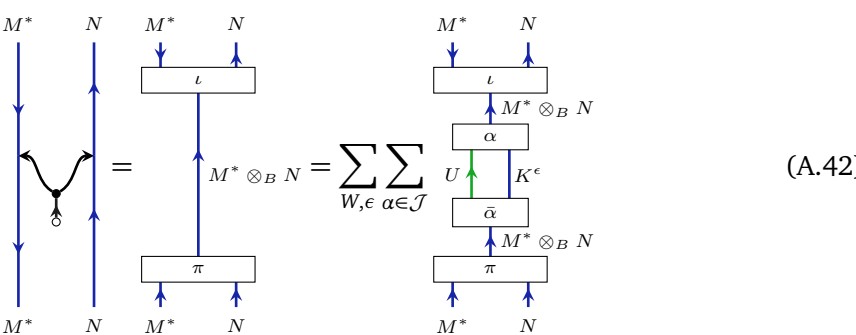

$$\tag{A.42}$$

*Proof.* The left hand side of (A.42) is the idempotent projecting onto $M^* \otimes_B N$, and we can split the idempotent as $\iota \circ \pi$. For the second equation we define

$$p := \sum_{W,\epsilon} \sum_{\alpha \in \mathcal{J}} \alpha \circ \bar{\alpha}, \tag{A.43}$$

and we show that $p = \mathrm{id}_{M^* \otimes_B N}$. This is equivalent to $p \circ \beta = \beta$ for any $\beta \in \mathcal{J}$ and for a fixed simple object $W' \boxtimes K^{\epsilon'}$. We compute

$$p \circ \beta = \sum_{W,\epsilon} \sum_{\alpha \in \mathcal{J}} \alpha \circ \bar{\alpha} \circ \beta \overset{(*)}{=} \sum_{W,\epsilon} \sum_{\alpha \in \mathcal{J}} \alpha \, \delta_{W,W'} \delta_{\epsilon,\epsilon'} \delta_{\alpha,\beta} = \beta \,. \tag{A.44}$$

In step $(*)$ we used that $\bar{\mathcal{J}}$ is a basis dual to $\mathcal{J}$ with respect to $\langle -, \rangle_{\mathrm{bnd}}$ as follows:

$$\bar{\alpha} \circ \beta = \frac{\mathrm{tr}_{W' \boxtimes K^{\epsilon'}}(\bar{\alpha} \circ \beta)}{\epsilon' \dim(W')} \, \mathrm{id}_{W' \boxtimes K^{\epsilon'}} = \langle \bar{\alpha}, \beta \rangle_{\mathrm{bnd}} \, \mathrm{id}_{W' \boxtimes K^{\epsilon'}} = \delta_{\alpha,\beta} \, \mathrm{id}_{W' \boxtimes K^{\epsilon'}} \,. \tag{A.45}$$

$\square$

*Proof of Theorem 5.6 (open case).* Using Lemma A.21 we can compute

$$\begin{aligned}
\mathrm{Corr}_B^{\mathrm{or}}(\Sigma') &\overset{\mathrm{def.}}{=} \widehat{\mathcal{Z}}_{\mathcal{C}}(M_{\Sigma'}) \overset{\mathrm{Lem.\,A.21}}{=} \sum_{W,\epsilon} \sum_{\alpha \in \mathcal{J}} \widehat{\mathcal{Z}}_{\mathcal{C}}(G_{W,\epsilon} \circ M_{\Sigma(W,\epsilon,\alpha,\bar{\alpha})}) \\
&= \sum_{W,\epsilon} \sum_{\alpha \in \mathcal{J}} \widehat{\mathcal{Z}}_{\mathcal{C}}(G_{W,\epsilon}) \widehat{\mathcal{Z}}_{\mathcal{C}}(M_{\Sigma(W,\epsilon,\alpha,\bar{\alpha})}) \\
&\overset{\mathrm{def.}}{=} \sum_{W,\epsilon} \sum_{\alpha \in \mathcal{J}} \widehat{\mathcal{Z}}_{\mathcal{C}}(G_{W,\epsilon})(\mathrm{Corr}_B^{\mathrm{or}}(\Sigma(W,\epsilon,\alpha,\bar{\alpha}))) \,.
\end{aligned} \tag{A.46}$$

As in the closed case, there is no gluing anomaly as relevant Maslov indices vanish, see [36, Thm. 3.10] and [40, Sec. 4.1].

$\square$

## A.4 Algebras and multiplicity spaces for a self-dual invertible object

Throughout this appendix we fix $G \in \mathcal{C}$ such that $G \otimes G \cong \mathbf{1}$. It follows that $G$ is simple, invertible, and self-dual. In this appendix we recall how to construct algebras from $G$ and compute the associated multiplicity and state spaces. We follow [46] (though there it is assumed that the quantum dimension is one), and also [18] where $\dim(G)$ is not constrained.

We write $\dim(G)$ for the quantum dimension of $G$ and $\theta_G$ for its twist eigenvalue. The quantum dimension is multiplicative, so $G \otimes G \cong \mathbf{1}$ implies that $\dim(G) \in \{\pm 1\}$. The self-braiding of $G$ is given by

$$\sigma_{G,G} = \dim(G)\, \theta_G \cdot \mathrm{id}_{G \otimes G}, \tag{A.47}$$

as can be seen by taking the partial trace over one factor of $G$ on both sides. The self-braiding defines a quadratic form on $\mathbb{Z}_2$, and hence can only take values in $\{\pm 1, \pm i\}$. This also constrains $\theta_G$ to be in $\{\pm 1, \pm i\}$. The quadratic form $\sigma_{G,G}$ determines the braided monoidal structure on the subcategory generated by $\{\mathbf{1}, G\}$ via direct sums, see e.g. [46, Sec. 2]. It thus also fixes the braided monoidal structure on the subcategory generated by $\{\mathbf{1}, G \boxtimes K^\nu\}$

### Algebras

Let $\nu \in \{\pm\}$ and denote with

$$A_\nu := \mathbf{1} \oplus G \boxtimes K^\nu \;=\; \begin{cases} \mathbf{1} \oplus G \in \mathcal{C}, & \text{if } \nu = +, \\ \mathbf{1} \oplus \Pi G \in \widehat{\mathcal{C}}, & \text{if } \nu = -. \end{cases} \tag{A.48}$$

We have:

**Lemma A.22.**     *1. There is a special Frobenius algebra structure on $A_\nu$ if and only if the twist on $G$ satisfies $\theta_G^2 = 1$. In this case, the special Frobenius algebra structure is unique up to isomorphism.*

    *Assume now that $\theta_G^2 = 1$, so that $A_\nu$ is a special Frobenius algebra. Abbreviate $\widehat{G} := G \boxtimes K^\nu$*

    *2. The restriction $\mu_{\widehat{G},\widehat{G}}$ of the product to $\widehat{G} \otimes \widehat{G}$ satisfies*

$$\mu_{\widehat{G},\widehat{G}} \circ \sigma_{\widehat{G},\widehat{G}} = \nu \dim(G)\, \theta_G \cdot \mu_{\widehat{G},\widehat{G}}. \tag{A.49}$$

    *In particular $A_\nu$ is commutative iff $\dim(G)\, \theta_G = \nu$.*

    *3. The Nakayama automorphism is given by*

$$N_{A_\nu} = \mathrm{id}_{\mathbf{1}} + \nu \dim(G) \cdot \mathrm{id}_{\widehat{G}}. \tag{A.50}$$

    *In particular $N_{A_\nu}^2 = \mathrm{id}_{A_\nu}$ and $A_\nu$ is symmetric iff $\dim(G) = \nu$.*

    *4. We have $\dim(A_\nu) = 1 + \nu \dim(G)$ and $D(A_\nu) = 2$.*

*Proof. Part 1:* From [46, Lem. 3.10]: The special Frobenius algebra structure exists iff the associator on the subcategory generated by $\{\mathbf{1}, \widehat{G}\}$ is trivialisable, which happens iff $\theta_G^2 = 1$, and so an associative product exists only in this case. Since $\mathrm{H}^2(\mathbb{Z}_2, \mathbb{C}^\times) = 0$, there is up to isomorphism only one product (which allows for a non-degenerate pairing) and hence also only one special Frobenius algebra structure.

    *Part 2* is immediate from the self-braiding (A.47).

    *Part 3:* Denote the components of the product by $\mu = \mu_{\mathbf{1},\mathbf{1}} + \mu_{\mathbf{1},\widehat{G}} + \mu_{\widehat{G},\mathbf{1}} + \mu_{\widehat{G},\widehat{G}}$. The explicit form of the coproduct and counit is

$$\Delta = \frac{1}{2} \cdot \left( \mu_{\mathbf{1},\mathbf{1}}^{-1} + \mu_{\mathbf{1},\widehat{G}}^{-1} + \mu_{\widehat{G},\mathbf{1}}^{-1} + \mu_{\widehat{G},\widehat{G}}^{-1} \right), \quad \varepsilon = 2 \cdot (\pi_{\mathbf{1}} \circ \eta)^{-1} \circ \pi_{\mathbf{1}}, \tag{A.51}$$

where $\pi_1 \colon A_v \longrightarrow \mathbf{1}$ is the projection onto the summand $\mathbf{1}$. From this we can compute the Nakayama automorphism restricted to $g \in \{\mathbf{1}, \widehat{G}\}$ as

$$ \text{(A.52)} $$

where we used that $\sigma_{g,g} = c_{g,g}\,\mathrm{id}_{g \otimes g}$ for some $c_{g,g} \in \mathbb{C}^{\times}$ and that

$$ = \epsilon \circ \mu_{g,g} \circ \frac{1}{2}\mu_{g,g}^{-1} \circ \eta = 1\,. \qquad \text{(A.53)} $$

*Part 4* is clear from the explicit expression for $N_{A_v}$. $\qquad\square$

**Multiplicity spaces**

Define the *monodromy charge* $q_U$ of a simple object $U \in \mathcal{C}$ with respect to $G$ to be [46, Sec. 3.5]

$$ \sigma_{U,G} = q_U \cdot \sigma_{G,U}^{-1} \quad \Longleftrightarrow \quad q_U = \frac{\theta_{G \otimes U}}{\theta_G\,\theta_U}\,. \qquad \text{(A.54)} $$

Recall the definition of the s-matrix from (36) in terms of invariants of the Hopf link.

**Lemma A.23.** *For $U \in \mathcal{C}$ simple we have*

$$ q_U = \frac{\mathsf{s}_{U,G}}{\dim(U)\dim(G)} \in \{\pm 1\}, \quad \text{and} \quad q_G = 1\,. \qquad \text{(A.55)} $$

*Proof.* The first equality is immediate from the definition of $\mathsf{s}_{U,V}$ as a trace over the double braiding. That $q_U \in \{\pm 1\}$ can be seen as follows. First note that

$$ = q_U \qquad = \dim(G)\,q_U \qquad \text{(A.56)} $$

This can be used to compute

$$ q_U^2 = \bigl(\dim(G)\,q_U\bigr)^2 \overset{\text{(A.56)}}{=} \qquad G = \qquad G \otimes G = \qquad \text{(A.57)} $$

That $q_G = 1$ follows directly from comparing (A.47) and (A.54). $\qquad\square$

**Remark A.24.** By fixing $G \in \mathcal{C}$ one obtains a $\mathbb{Z}_2$-grading on $\mathcal{C}$ via the monodromy charges $q_U$ in (A.54). When $\theta_G = -1$, a modular fusion category with this structure is an example of a *spin modular category* in the sense of [6, 11], see also [8], and can be used for the construction of invariants of 3-dimensional spin manifolds.

After these preparations, we can compute the multiplicity spaces $\mathcal{M}_{x,\epsilon}^{(\mathrm{in})}(U,\bar{V};A_\nu)$ from (90) for the algebra $A_\nu \in \widehat{\mathcal{C}}$.

**Lemma A.25.** *The multiplicity spaces for $A_\nu$ are given by $\mathcal{M}_{x,\epsilon}^{(\mathrm{in})}(S_i,\bar{S}_j;A_\nu) = M_{\mathbf{1}} \oplus M_G$, where*

$$M_{\mathbf{1}} = \begin{cases} \mathcal{C}(S_i \otimes \bar{S}_j, \mathbf{1}), & \text{if } \epsilon = + \text{ and } q_i = (\nu \dim(G))^x, \\ \{0\}, & \text{else}, \end{cases}$$

$$M_G = \begin{cases} \mathcal{C}(S_i \otimes \bar{S}_j, G), & \text{if } \epsilon = \nu \text{ and } q_i = (\nu \dim(G))^{x+1}\theta_G, \\ \{0\}, & \text{else}. \end{cases} \tag{A.58}$$

*Proof.* Let $\phi \in \widehat{\mathcal{C}}(S_i \otimes \bar{S}_j \otimes K^\epsilon, A_\nu)$. We compute $Q_x^{(\mathrm{in})}(\phi)$:

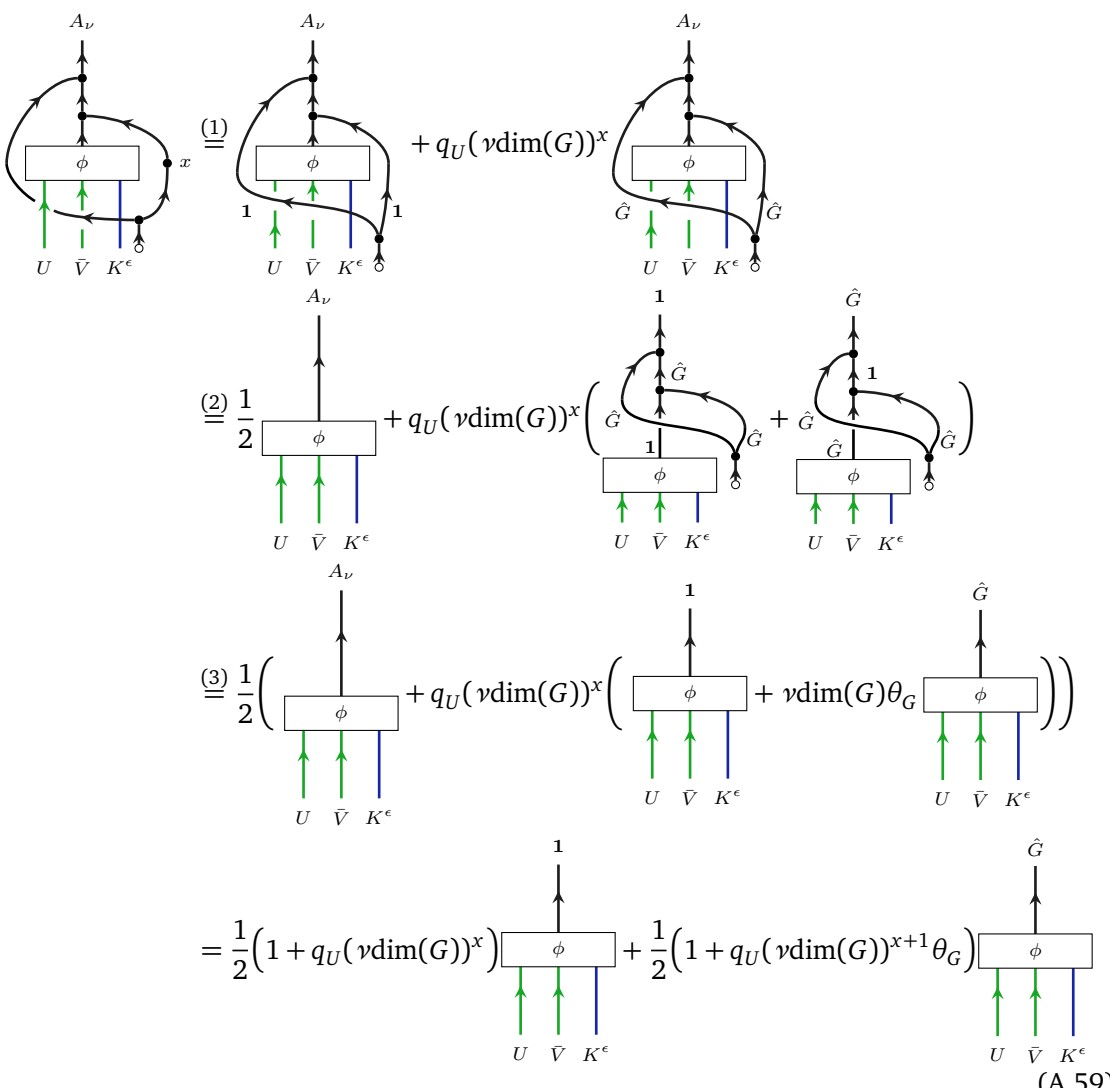

$$\tag{A.59}$$

where in (1) we used (A.50) and (A.54), in (2) and in (3) we used that the projector to $\mathbf{1}$ is $\frac{1}{2}\eta \circ \epsilon$, and in (3) we substituted (A.49) and used (A.53). From this we see that $Q_x^{(\mathrm{in})}$ is the projection onto the subspace given in the statement of the lemma. $\square$

### A.5 Some details on the Bershadsky-Polyakov algebra at $k = -1/2$

#### A.5.1 Spectral flow

"Spectral flow" is the name for a family of maps $\sigma_s$ between representations of the BP algebra labelled by $s \in \frac{1}{2}\mathbb{Z}$ which, in the case $k = -1/2$, acts on the generators as [42]

$$\sigma_s(L_m) = L_m - sJ_m + \frac{s^2}{3}\delta_{m,0}, \quad \sigma_s(J_m) = J_m - \frac{2s}{3}\delta_{m,0}, \quad \sigma_s(G_r^{\pm}) = G_{r \mp s}^{\pm}. \tag{A.60}$$

If $|j,h\rangle$ is a NS highest weight state with eigenvalues $(j,h)$ satisfying

$$L_0|j,h\rangle = h|j,h\rangle, \quad J_0|j,h\rangle = j|j,h\rangle, \quad G_{1/2}^+|j,h\rangle = G_{1/2}^-|j,h\rangle = 0, \tag{A.61}$$

then $|j,h\rangle$ is a R highest weight state under the action $\sigma_{1/2}$ with eigenvalues $(j - \frac{1}{3}, h - \frac{j}{2} + \frac{1}{12})$

$$\begin{aligned}
\sigma_{1/2}(L_0)|j,h\rangle &= (L_0 - \tfrac{1}{2}J_0 + \tfrac{1}{12})|j,h\rangle = (h - \frac{1}{2}j + \frac{1}{12})|j,h\rangle, \\
\sigma_{1/2}(J_0)|j,h\rangle &= (J_0 - \tfrac{1}{3})|j,h\rangle = (j - \tfrac{1}{3})|j,h\rangle, \\
\sigma_{1/2}(G_0^-)|j,h\rangle &= G_{1/2}^-|j,h\rangle = 0, \quad \sigma_{1/2}(G_1^+)|j,h\rangle = G_{1/2}^+|j,h\rangle = 0.
\end{aligned} \tag{A.62}$$

#### A.5.2 Change of grading

We presented the BP algebra in equations (225) in the form in which the generators $G^{\pm}$ have weight 3/2. It is, however, possible to make a change of basis to new generators in which $G^{\pm}$ have integer weights but $J$ is no longer a primary field. The new basis has generators $\widetilde{L}_m, \widetilde{J}_m, \widetilde{G}_m^{\pm}$ where (in the case $k = -1/2$) $\widetilde{L}_m$ satisfy the Virasoro algebra with central charge $\widetilde{c}$:

$$\widetilde{J}_m = J_m, \quad \widetilde{G}_m^{\pm} = G_{m \mp \frac{1}{2}}^{\pm}, \quad \widetilde{L}_m = L_m - \frac{1}{2}(m+1)J_m, \quad \widetilde{c} = c + 2 = \frac{8}{5}. \tag{A.63}$$

With this choice of Virasoro algebra, $\widetilde{G}^+$ is primary of weight 1, $\widetilde{G}^-$ is primary of weight 2 but $\widetilde{J}$ is no longer primary:

$$[\widetilde{L}_m, \widetilde{G}_r^+] = -r\,\widetilde{G}_{m+r}^+, \quad [\widetilde{L}_m, \widetilde{G}_r^-] = (m-r)\,\widetilde{G}_{m+r}^-, \quad [\widetilde{L}_m, \widetilde{J}_n] = -n\widetilde{J}_{m+n} - \frac{1}{3}m(m+1)\delta_{m+n,0}. \tag{A.64}$$

Since the fields $G^{\pm}$ have integer weight, the modes of $\widetilde{G}_m^{\pm}$ should have integer labels $m$, and hence the modes $G_m^{\pm}$ have half-integer modes. As a consequence, representations of the even-graded algebra correspond to NS representations of the "standard" algebra. When it comes to characters, however the difference between $c$ and $\widetilde{c}$, and the required shifts when mapping the zero modes of $\widetilde{T}$ and $\widetilde{J}$ from the cylinder to the plane (the shifts are the ubiquitous $\widetilde{L}_0 - \widetilde{c}/24$ and the perhaps unfamiliar $\widetilde{J}_0 - 1/3$) exactly replicate the effect of the spectral flow $\sigma_{1/2}$ and the characters of the even-graded algebra are identical to the Ramond sector characters. This is consistent with the fact that the R sector closes onto itself under modular transformations.

We note that the vacuum module in the even-graded theory is not self-conjugate and so falls outside the class of theories to which the TFT construction as we present it here is applicable.

#### A.5.3 Representations

The representations of the integer-graded algebra at $k = -1/2$ are given in [3, Prop. 2.4], in 1-1 correspondence with the representations of affine $sl(3)$ at level 2. They are indexed in [3]

Table 6: Representations of $BP_{-1/2}$. The first row is for the integer-graded algebra in the notation of [3], and the second and third rows are for the half-integer graded algebra as used in section 7.5 in the NS and R sectors respectively.

| $(ij),[mi]$ | $(13),[11]$ | $(12),[21]$ | $(11),[31]$ | $(22),[12]$ | $(21),[22]$ | $(31),[13]$ |
|---|---|---|---|---|---|---|
| $(\widetilde{j},\widetilde{h})=(\xi_{ij},\chi_{ij})$ | $(0,0)$ | $(\frac{1}{3},-\frac{2}{15})$ | $(\frac{2}{3},0)$ | $(-\frac{1}{3},\frac{1}{5})$ | $(0,\frac{1}{5})$ | $(-\frac{2}{3},\frac{2}{3})$ |
| $(j,h)_{NS}=(\widetilde{j},\widetilde{h}+\frac{1}{2}\widetilde{j})$ | $(0,0)$ | $(\frac{1}{3},\frac{1}{30})$ | $(\frac{2}{3},\frac{1}{3})$ | $(-\frac{1}{3},\frac{1}{30})$ | $(0,\frac{1}{5})$ | $(-\frac{2}{3},\frac{1}{3})$ |
| name | id | $\phi$ | $\psi$ | $\bar{\phi}$ | $u$ | $\bar{\psi}$ |
| $(j,h)_R=(\widetilde{j}-\frac{1}{3},\widetilde{h}+\frac{1}{12})$ | $(-\frac{1}{3},\frac{1}{12})$ | $(0,-\frac{1}{20})$ | $(\frac{1}{3},\frac{1}{12})$ | $(-\frac{2}{3},\frac{17}{60})$ | $(-\frac{1}{3},\frac{17}{60})$ | $(-1,\frac{3}{4})$ |
| name | $\mathrm{id}^s$ | $\phi^s$ | $\psi^s$ | $\bar{\phi}^s$ | $u^s$ | $\bar{\psi}^s$ |

by a pair of integers $(ij)$ with $i,j > 0$, $i + j < 5$, but many properties are more apparent if instead we use the labels $[mi]$ where $m = 5-i-j$. For NS representations of the half-integer graded algebra this agrees with the notation of [42] if $\lambda_1 = m$, $\lambda_2 = i$, and one finds e.g. that the NS representation $[im]$ is conjugate to the representation $[mi]$.

Relating the highest weight condition in [3] to that in the NS algebra, we find that the highest weights in [3] are indeed highest weights of the half-integer graded NS algebra:

$$\widetilde{G}_0^- \psi = G_{1/2}^- \psi = 0\,, \quad \widetilde{G}_1^+ \psi = G_{1/2}^+ \psi = 0\,. \tag{A.65}$$

This means that a HW state of the integer graded algebra satisfying (A.65) with eigenvalues $(\widetilde{j},\widetilde{h})$ corresponds to a HW state of the NS algebra with $(j = \widetilde{j}, h = \widetilde{h} + \frac{1}{2}\widetilde{j})$.

This also agrees with the list of NS and R representations given in [42, Fig. 2], provided one takes into account that our convention for the highest weight state in the R sector is opposite to theirs (we take it to be annihilated by $G_0^-$), so that $j$ differs by 1 for $\bar{\phi}^s$ and $u^s$ and 2 for $\bar{\psi}^s$.

In Table 6, we give the values of $(ij)$, $[mi]$, $(\widetilde{j},\widetilde{h})$ (denoted $(\xi_{ij},\chi_{ij})$ in [3]) and the corresponding $NS$ values $(j,h)_{NS}$ and the values of $(j,h)_R$ for the R representations obtained in turn from these by spectral flow.

The (full) characters $\mathrm{Tr}_M(z^{J_0}q^{L_0-c/24})$ could in principle be obtained from the Hamiltonian reduction using formulae in [5] or [42]; here, in tables 7 and 8, we give the leading terms from a direct calculation of the rank of the inner product matrix at each $L_0$-level.

Note that the modular transformation formulae given in [5] do not apply here as we are not taking the $W_3^{(2)}$ algebra in a "good even grading", which would imply all the fields have integral weights as discussed above. We obtained the explicit S-matrix given in (233) numerically from the approximate characters. The exact entries are confirmed by matching against entries of the S-matrix of $\widehat{su}(3)_2$.

Table 7: Characters of the NS representations of $BP_{-1/2}$.

| $M$ | $\text{Tr}_M\big((\pm)^G z^{J_0} q^{L_0-c/24}\big)$ |
|---|---|
| id | $q^{-\frac{1}{60}}\big(1+q\pm(z+\frac{1}{z})q^{\frac{3}{2}}+3q^2\pm2(z+\frac{1}{z})q^{\frac{5}{2}}+(5+z^2+\frac{1}{z^2})q^3\pm4(z+\frac{1}{z})q^{\frac{7}{2}}+\dots\big)$ |
| $\phi$ | $z^{\frac{1}{3}}q^{\frac{1}{60}}\big(1\pm\frac{1}{z}q^{\frac{1}{2}}+2q\pm(z+\frac{2}{z})q^{\frac{3}{2}}+(4+\frac{1}{z^2})q^2\pm(2z+\frac{5}{z})q^{\frac{5}{2}}+(8+\frac{2}{z^2})q^3\pm(5z+\frac{9}{z})q^{\frac{7}{2}}+\dots\big)$ |
| $\bar\phi$ | $z^{-\frac{1}{3}}q^{\frac{1}{60}}\big(1\pm zq^{\frac{1}{2}}+2q\pm(2z+\frac{1}{z})q^{\frac{3}{2}}+(4+z^2)q^2\pm(5z+\frac{2}{z})q^{\frac{5}{2}}+(8+2z^2)q^3\pm(9z+\frac{5}{z})q^{\frac{7}{2}}+\dots\big)$ |
| $u$ | $q^{\frac{11}{60}}\big(1\pm(z+\frac{1}{z})q^{\frac{1}{2}}+2q\pm2(z+\frac{1}{z})q^{\frac{3}{2}}+5q^2\pm4(z+\frac{1}{z})q^{\frac{5}{2}}+(z^2+9+\frac{1}{z^2})q^3\pm8(z+\frac{1}{z})q^{\frac{7}{2}}+\dots\big)$ |
| $\psi$ | $z^{\frac{2}{3}}q^{\frac{19}{60}}\big(1\pm\frac{1}{z}q^{\frac{1}{2}}+(1+\frac{1}{z^2})q\pm\frac{2}{z}q^{3/2}+(3+\frac{1}{z^2})q^2\pm(z+\frac{4}{z})q^{\frac{5}{2}}+(5+\frac{3}{z^2})q^3\pm(2z+\frac{8}{z})q^{\frac{7}{2}}+\dots\big)$ |
| $\bar\psi$ | $z^{-\frac{2}{3}}q^{\frac{19}{60}}\big(1\pm zq^{\frac{1}{2}}+(1+z^2)q\pm2zq^{3/2}+(3+z^2)q^2\pm(4z+\frac{1}{z})q^{\frac{5}{2}}+(5+3z^2)q^3\pm(8z+\frac{2}{z})q^{\frac{7}{2}}+\dots\big)$ |

Table 8: Characters of the R representations of $BP_{-1/2}$.

| $M$ | $\text{Tr}_M\big((\pm)^G z^{J_0} q^{L_0-c/24}\big)$ |
|---|---|
| $\text{id}^s$ | $z^{-\frac{1}{3}}q^{\frac{1}{15}}\big(1+(1\pm z)q+(3\pm2z+z^2\pm\frac{1}{z})q^2+(5\pm4z\pm\frac{2}{z}+z^2)q^3+\dots\big)$ |
| $\phi^s$ | $q^{-\frac{1}{15}}\big(1+(2\pm z\pm\frac{1}{z})q+(4\pm2z\pm\frac{2}{z})q^2+(8\pm5z\pm\frac{5}{z})q^3+\dots\big)$ |
| $\bar\phi^s$ | $z^{-\frac{2}{3}}q^{\frac{4}{15}}\big((1\pm z)+(2\pm2z+z^2)q+(4\pm\frac{1}{z}\pm5z+2z^2)q^2+(8\pm\frac{2}{z}\pm9z+4z^2)q^3+\dots\big)$ |
| $u^s$ | $z^{-\frac{1}{3}}q^{\frac{4}{15}}\big((1\pm z)+(2\pm\frac{1}{z}\pm2z)q+(5\pm\frac{2}{z}\pm4z+z^2)q^2+(9\pm\frac{4}{z}\pm8z+2z^2)q^3+\dots\big)$ |
| $\psi^s$ | $z^{\frac{1}{3}}q^{\frac{1}{15}}\big(1+(1\pm\frac{1}{z})q+(3\pm\frac{2}{z}\pm z+\frac{1}{z^2})q^2+(5\pm\frac{4}{z}\pm2z+\frac{1}{z^2})q^3+\dots\big)$ |
| $\bar\psi^s$ | $z^{-1}q^{\frac{11}{15}}\big((1\pm z+z^2)+(1\pm2z+z^2)q+(3\pm4z+3z^2)q^2+(5\pm\frac{1}{z}\pm8z+5z^2\pm z^3)q^3+\dots\big)$ |

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
