# Peer review of "Parity and Spin CFT with boundaries and defects"

_SciPost Physics, doi:SciPost Phys. 15, 207 (2023)_

## Round 1 · Referee Report · Anonymous (Referee 1) · 2023-2-12

Report

This work gives a detailed discussion of four types of 2d CFTs depending on whether they have spin structure and/or parity dependence on the local fields. Following the notations of eq.(1.2), the diagonal cases 1 and 4 are better understood, and the off-diagonal cases 2 and 3 are new to this work. The presentation are clear, rigorous, and contains enough details. This result is definitely worth to be published, and the referee recommends so.

It would be nice if the authors can clarify further on the following points:

In the case of Z2 symmetry, the authors clarified in Sec. 7.1 that the CFTs of off-diagonal types, i.e. 2 and 3, are non-unitary, while the diagonal types 1 and 4 are unitary. It is known, e.g. in [KTT] and [HNT] as cited in the current work, that starting with 1, gauging Z2 while introducing the spin structure dependence (via including an Arf term), leads to 4. This is known as the fermionization map and is indeed described in Sec.7.2. However, in the introduction Sec.1.5, the authors commented that starting with 2 and performing a (possibly different) gauging of topological symmetry, one leads to 4. It would be useful if the authors could clarify the difference between the normal fermionization map 1->4 and the “new gauging” 2->4. In particular, how to understand a non-unitary theory 2 is mapped to a unitary theory 4? This would help the readers to understand the relations between four types of CFTs better.
  • validity: -
  • significance: -
  • originality: -
  • clarity: -
  • formatting: -
  • grammar: -

Author:  Lóránt Szegedy  on 2023-03-01  [id 3415]

(in reply to Report 1 on 2023-02-12)
Category:
answer to question

Thank you very much for your report and suggestions. We would prefer to delay implementing any changes or additions until invited to do so by the editor and will detail these in a separate reply. Here we just wanted to comment quickly about unitarity: Theories of type 2 and 3 cannot be unitary as they violate spin-statistics. Theories of type 1 and 4 can be unitary, but need not be unitary, so there is no inherent contradiction in going from type 2 to type 4.

---

## Round 1 · Referee Report · Anonymous (Referee 2) · 2023-3-6

Report

The authors discuss the structures of RCFTs graded by parity and in the presence of spin structures. The paper builds upon previous work by the authors and incorporates the most general RCFT observables that involve defects and boundaries. The exposition is comprehensive, clear and skillfully illustrated. It clarifies various subtleties in the literature on such RCFTs. This will be a very useful reference to learn about these RCFTs and certainly deserves to be published.

Some minor comments:

  1. While the authors focus on RCFT in this paper, it maybe worthwhile to briefly comment on how some of the constructions can be generalized to general 2d CFT which may not be rational (e.g. gauging a special Frobenius algebra F which does not rely on starting with an underlying modular tensor category) .

  2. The authors mention in the abstract that the construction (gauging F) is like "as gauging a possibly non-invertible symmetry". This was not mentioned in the main text. It would be good to mention and clarify this point (e.g. when and how it becomes non-invertible).

  3. The main general construction of type 4 spin CFT comes from gauging F in a type 2 CFT as stated in Sec 1.5. This is related but different from the usual fermionization construction (which starts from type 1). Just for clarity, in discussing the Ising example (similarly for the other examples), it may be good to fill in the empty boxes in the table below (7.8).

  4. Usual fermionization map from bosonic to fermionic (spin) CFT comes with a choice (e.g. by stacking with fermionic SPT). It would be good to mention where that choice is contained in the construction here (e.g. in the F algebra).

  • validity: high
  • significance: high
  • originality: high
  • clarity: high
  • formatting: excellent
  • grammar: excellent

Author:  Lóránt Szegedy  on 2023-03-20  [id 3497]

(in reply to Report 2 on 2023-03-06)
Category:
answer to question

Thank you for your helpful comments. We will address your points in a revised version once invited to do so by the editor. One remark regarding the empty boxes in the table below (7.8) we can make already now: The table answers the question "Which types of CFT do we get starting from Ising, or from Ising with defects enhanced by parity, by taking the algebra 1 + eps, or 1 + Pi(eps)?" And the answer is the two filled places (type 1 and type 4), the empty spots cannot be reached in this way. We will try to make this more clear in a revised version.

---

## Round 2 · Referee Report · Anonymous (Referee 1) · 2023-8-15

Report

The comments are addressed appropriately, and I recommend for publication.

---

## Round 2 · Author Response

{\bf Report 1 and reply}

The authors discuss the structures of RCFTs graded by parity and in the presence of spin structures. The paper builds upon previous work by the authors and incorporates the most general RCFT observables that involve defects and boundaries. The exposition is comprehensive, clear and skillfully illustrated. It clarifies various subtleties in the literature on such RCFTs. This will be a very useful reference to learn about these RCFTs and certainly deserves to be published.

Some minor comments:

  1. While the authors focus on RCFT in this paper, it maybe worthwhile to briefly comment on how some of the constructions can be generalized to general 2d CFT which may not be rational (e.g. gauging a special Frobenius algebra F which does not rely on starting with an underlying modular tensor category) .

Reply:

We have added some comments at the end of section 1.5

  1. The authors mention in the abstract that the construction (gauging F) is like "as gauging a possibly non-invertible symmetry". This was not mentioned in the main text. It would be good to mention and clarify this point (e.g. when and how it becomes non-invertible).

Reply:

We have explained the relation to gauging symmetries and added an example of a non-invertible symmetry and its gauging in section 1.5. In any given CFT, a symmetry (in the sense of an indecomposable topological defect) is either invertible or non-invertible. A symmetry may become non-invertible after gauging a different symmetry, but we do not need this in our construction and would prefer not to comment on it.

  1. The main general construction of type 4 spin CFT comes from gauging F in a type 2 CFT as stated in Sec 1.5. This is related but different from the usual fermionization construction (which starts from type 1). Just for clarity, in discussing the Ising example (similarly for the other examples), it may be good to fill in the empty boxes in the table (7.12) below (7.11).

Reply:

There are two related constructions in this paper which both actually start with a type 1 CFT and which result in a theory without parity (bosonic theory'') using the algebra $A_+$, and a theory with parity (fermionic theory'') using the algebra $A_-$.

Depending on the details of the theory, using $A_+$ will result in a theory of type 3 or of type 1 (which is a type 3 theory which is actually insensitive to the spin structure), and using $A_-$ will result in a theory of type 4 or of type 2 (which is a type 4 theory which is actually insensitive to the spin structure). The examples in section 7 cover all four of these cases.

The construction of a theory of type 4 starts with a theory of type 1 which is extended to a theory of type 2 and then gauged to get the desired theory of type 4. This is indeed different to the ``usual" fermionisation procedure which starts with the a theory of 1, tensors with Arf to get a trivial tensor product theory of type 4 and then gauges to get the desired theory of type 4.

We have attempted to make this clearer by \begin{itemize} \item adding text on how the different types relate in more detail and how the spin theories of types 3 and 4 are constructed at the end of sections 1.2 and 1.3, \item adding a discussion of how the Arf 2d TFT fits into the our construction in section 7.1,
\item adding a discussion on how our results relate to these in [HNT] in the case of $c=4/5$ minimal models in section 7.4.4. \end{itemize}

In the Ising example, the table (7.11) is intended to show which two theories can be constructed by $A_\pm$, and since only two are possible, this is why only two boxes are filled. In the same way, the tables in sections 7.4.1 (now numbered (7.22)) and at the end of section 7.5.1 (now numbered (7.46)) also show only two boxes filled, as these are the two theories that can be constructed using $A_\pm$.

We think it would actually be confusing to include the intermediate stage CFTs in these boxes, when applicable.

  1. Usual fermionization map from bosonic to fermionic (spin) CFT comes with a choice (e.g. by stacking with fermionic SPT). It would be good to mention where that choice is contained in the construction here (e.g. in the F algebra).

Reply:

There are indeed choices made in this paper.

The first choice is in the construction of the fermionic CFT from the category $\hat\Cc$. We have to make a choice of how to construct a spin theory from $\hat\Cc$ and a Frobenius algebra $A$. Thereafter, all the choices lie in the choice of $A$. We have chosen what seemed the most natural construction. Other choices of construction are also possible, but are equivalent to changing the choice of algebra, so we have not elaborated on this in the paper.

The second choice is the choice of Frobenius algebra $A$ -- we give examples of algebras which lead to particular CFTs. Morita equivalent algebras lead to the same CFTs, but there can be more than one Morita equivalence class, and so more than one CFT that can be defined from a given $\hat\Cc$ -- indeed, when the Ramond parity-swap leads to an inequivalent theory, there will be two inequivalent Frobenius algebras which give the two theories related by parity-swapping the Ramond sector. Not surprisingly since the Arf theory is given by the choice of the Clifford algebra as the Frobenius algebra, the Ramond-parity swap operation is given by tensoring the Frobenius algebra with the Clifford algebra. We have included an explicit comparison to other constructions in the new section 7.4.4, although we defer the details of the Ramond parity-swap to a later paper.

{\bf Report 2 and reply}

This work gives a detailed discussion of four types of 2d CFTs depending on whether they have spin structure and/or parity dependence on the local fields. Following the notations of eq.(1.2), the diagonal cases 1 and 4 are better understood, and the off-diagonal cases 2 and 3 are new to this work. The presentation are clear, rigorous, and contains enough details. This result is definitely worth to be published, and the referee recommends so.

It would be nice if the authors can clarify further on the following points:

  1. In the case of Z2 symmetry, the authors clarified in Sec. 7.1 that the CFTs of off-diagonal types, i.e. 2 and 3, are non-unitary, while the diagonal types 1 and 4 are unitary.

Reply:

We have moved section 7.1 to 7.2 and added a comment on the unitarity of the various theories, to the effect that for the examples considered there types 2 and 3 are necessarily non-unitary but 1 and 4 can be either, depending on the details of the model.

  1. It is known, e.g. in [KTT] and [HNT] as cited in the current work, that starting with 1, gauging Z2 while introducing the spin structure dependence (via including an Arf term), leads to 4. This is known as the fermionisation map and is indeed described in Sec.7.2. However, in the introduction Sec.1.5, the authors commented that starting with 2 and performing a (possibly different) gauging of topological symmetry, one leads to 4. It would be useful if the authors could clarify the difference between the normal fermionisation map 1$\to$4 and the “new gauging” 2$\to$4. In particular, how to understand a non-unitary theory 2 is mapped to a unitary theory 4? This would help the readers to understand the relations between four types of CFTs better.

Reply:

Our procedure is to start with type 1, allow for defects with parity, resulting in a theory of type 2 (but the bulk theory is still type 1 -- so no contradiction with unitarity), and then gauging a defect to arrive at type 4. This is now explained in more detail at the end of section 1.3. We have added section 7.4.4 to explain in an example how our construction produces the different theories in [HNT].

---

## Round 2 · List of Changes

See in author comments.

---

## Editorial Decision

published